# Dynamic fibroblast–immune interactions shape recovery after brain injury

Nathan A. Ewing-Crystal[1,2,3], Nicholas M. Mroz[1,2], Amara Larpthaveesarp[4], Carlos O. Lizama[4,5], Remy Pennington[4,5], Pailin Chiaranunt[2,6], Jason I. Dennis[6], Anthony A. Chang[1,2], Eric Dean Merrill[2], Sofia E. Caryotakis[1,2], Nikhita Kirthivasan[1], Leon Teo[7], Tatsuya Tsukui[8], Aditya Katewa[4,5], Gabriel L. McKinsey[4,5], Sophia C. K. Nelson[1], Agnieszka Ciesielska[9], Nicole C. Lummis[10], Lucija Pintarić[10], Madelene W. Dahlgren[2], Amha Atakilit[8], Helena Paidassi[11], Saket Jain[12], Xiaodan Liu[13], Duan Xu[13], Manish K. Aghi[12], James A. Bourne[7], Jeanne T. Paz[9,14,15,16], Richard Daneman[10], Fernando F. Gonzalez[4], Dean Sheppard[8], Anna V. Molofsky[6,14,15], Thomas D. Arnold[4,5 ✉] & Ari B. Molofsky[1,2,17 ✉]

Fibroblasts and immune cells coordinate tissue regeneration and necessary scarring after injury. In the brain, fibroblasts are border-enriched cells whose dynamic molecular states and immune interactions after injury remain unclear[1]. Here we define the shared fibroblast–immune response to brain injury. Early profibrotic myofibroblasts develop from pre-existing brain fibroblasts and infiltrate brain lesions, orchestrated by fibroblast TGFβ signalling, profibrotic macrophages and microglia, and perilesional glia. Myofibroblasts transition into several late fibroblast states, including lymphocyte-interactive fibroblasts. Interruption of the early myofibroblast state exacerbated sub-acute brain injury, tissue loss and secondary neuroinflammation, with increased mortality in the transient middle cerebral artery occlusion stroke model. Disruption of late lymphocyte–fibroblast niches via selective loss of fibroblast chemokine CXCL12 led to late brain-specific innate inflammation and lymphocyte dispersal with increased IFNγ production. These data indicate the response to brain injury is coordinated by evolving temporal and spatial fibroblast states that limit functional tissue loss and chronic neuroinflammation.

Central nervous system (CNS) injuries, including stroke, traumatic brain injury (TBI) and spinal cord injury, are leading causes of death and disability[2–4]. Treatments for CNS injuries are limited, reflecting a critical knowledge gap concerning the mechanisms that dictate the dynamic phases of CNS injury and repair[2–5]. Previous studies have focused on astrocytic gliosis, whereas more recent studies have highlighted the roles of CNS stromal cells, including both mural cells (pericytes and smooth muscle cells) and fibroblasts, in injury[6–11] and disease[1,12–14]. CNS fibroblasts are enriched at CNS borders and maintain brain meningeal and vascular structure, promote immune surveillance[15] and may regulate exchange between cerebrospinal fluid and interstitial fluid[1]. Fibroblasts display both tissue-restricted subsets and cross-organ conserved states, including a 'universal' state that is enriched around natural tissue borders[16–18]. Cross-organ immune-interactive fibroblast states appear to be dynamic and critical mediators of local immune composition and function[17,19]. By contrast, a distinct profibrotic state emerges with tissue injury and disease, driving wound contraction,

matrix deposition and sometimes scarring[20], reinforced by the cytokine TGFβ[21,22]. Here we focus on how brain fibroblast states and their local immune partners sculpt the injured, healing and remodelled brain.

## Fibroblast response to brain injury

We used collagen 1 lineage tracing ($Col1a2^{creER}$; $Rosa26^{tdT}$, where $tdT$ is tdTomato) to locate brain fibroblasts, observing that they localized to CNS borders as previously shown[1,23] (Fig. 1a and Extended Data Fig. 1a). In a photothrombotic (PT) injury model of focal brain ischaemia, fibroblasts expanded into damaged regions, produced collagen 1 and other extracellular matrix (ECM) components, and formed a lesion distinct from but adjacent to parenchymal astrocytic gliosis by 14 days post-injury (dpi), with similar results in models of TBI and stroke with focal ischaemia–reperfusion injury (Fig. 1b–d and Extended Data Fig. 1b,c). In a model of non-human primate cortical stroke, perilesional fibrosis (COL6⁺) was detected at 7 dpi and persisted at 1 year post-injury

[1]Biomedical Sciences Graduate Program, University of California, San Francisco, San Francisco, CA, USA. [2]Department of Laboratory Medicine, University of California, San Francisco, San Francisco, CA, USA. [3]Medical Scientist Training Program, University of California, San Francisco, San Francisco, CA, USA. [4]Department of Pediatrics, University of California, San Francisco, San Francisco, CA, USA. [5]Newborn Brain Research Institute, University of California, San Francisco, San Francisco, CA, USA. [6]Department of Psychiatry, University of California, San Francisco, San Francisco, CA, USA. [7]Australian Regenerative Medicine Institute, Monash University, Clayton, Victoria, Australia. [8]Lung Biology Center, Department of Medicine, University of California, San Francisco, San Francisco, CA, USA. [9]Gladstone Institute of Neurological Disease, Gladstone Institutes, San Francisco, CA, USA. [10]Departments of Pharmacology and Neurosciences, University of California, San Diego, La Jolla, CA, USA. [11]Centre International de Recherche en Infectiologie (CIRI), Univ Lyon, Inserm, U1111 Université Claude Bernard Lyon 1, CNRS, UMR5308, ENS de Lyon, Lyon, France. [12]Department of Neurosurgery, University of California, San Francisco, San Francisco, CA, USA. [13]Department of Radiology and Biomedical Imaging, University of California, San Francisco, San Francisco, CA, USA. [14]Neurosciences Graduate Program, University of California, San Francisco, San Francisco, CA, USA. [15]Department of Neurology, University of California, San Francisco, San Francisco, CA, USA. [16]The Kavli Institute for Fundamental Neuroscience and The Weill Institute for Neurosciences, University of California, San Francisco, San Francisco, CA, USA. [17]Diabetes Center, University of California, San Francisco, San Francisco, CA, USA. ✉e-mail: thomas.arnold@ucsf.edu; ari.molofsky@ucsf.edu

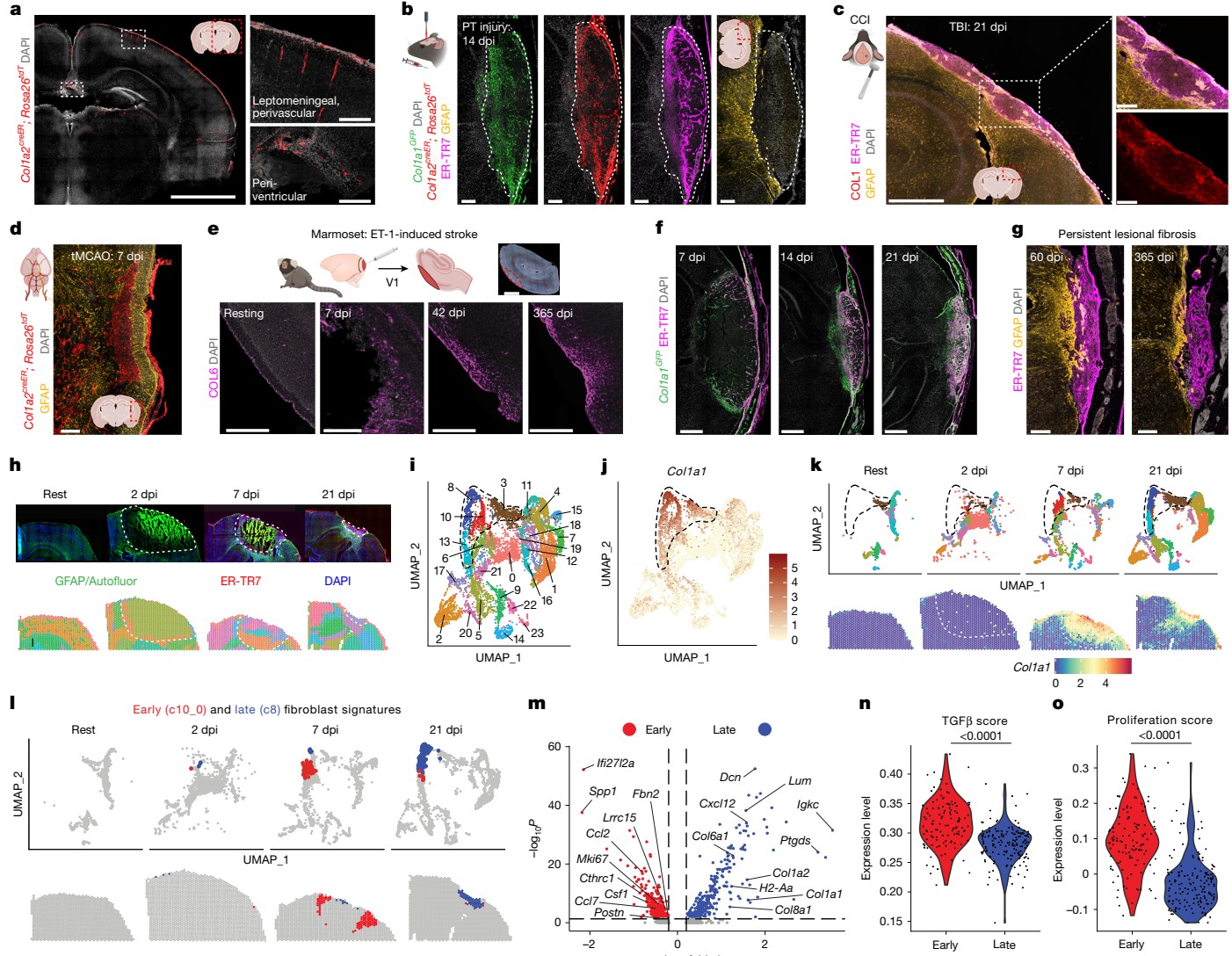

**Fig. 1 | Identification of a dynamic and persistent fibroblast response to brain injury. a**, Homeostatic brain fibroblasts are labelled in *Col1a2^creER^; Rosa26^tdT+^* mice. Pial/perivascular fibroblasts (top right), periventricular/choroid plexus fibroblasts (bottom right). Mice were treated with tamoxifen from day −9 to day −7 before mice were euthanized for imaging. Scale bars: 2,000 μm (left), 200 μm (right). **b**, Lesional fibroblasts are labelled in *Col1a1^GFP^; Col1a2^creER^; Rosa26^tdT+^* mice after PT cortical brain injury (image of GFAP⁺ gliosis (far right) is from a separate mouse). Mice were treated with tamoxifen from day −16 to day −14 before injury. Scale bars, 200 μm. **c**, A fibrotic lesion labelled for collagen 1 (COL1) and ER-TR7 (a fibroblast marker monoclonal antibody) within a glial scar (GFAP⁺) in the controlled cortical impact (CCI) model of TBI. Scale bars: 500 μm (left), 100 μm (right). **d**, Lesional fibroblasts after tMCAO stroke. Mice were treated with tamoxifen on days −7 to −5 before injury to track ontogeny. Scale bar, 250 μm. **e**, Time course showing collagen 6 (COL6) expression and persistence after endothelin-1-induced (ET-1) ischaemic stroke in adult marmosets. Scale bars, 250 μm. **f,g**, Time course in the PT injury model, showing fibroblasts (**f**; *Col1a1*-GFP⁺) and/or associated ECM (**g**; ER-TR7). Scale bars: 500 μm (**f**), 200 μm (**g**). **h**, Immunofluorescence showing astrocyte-dense (GFAP⁺) and fibroblast-dense regions (ER-TR7⁺) (top) and spatial plots (bottom, from same tissue) with 55-μm spot-based clusters. Early necrosis-associated autofluorescence is prominent; white dotted lines denote lesion borders (10X Genomics, Visium). **i,j**, Uniform manifold approximation and projection (UMAP) plots showing spot-based clusters (**i**) or fibroblast-enriched spots (**j**; *Col1a1⁺*). Four main fibroblast-enriched clusters are highlighted (black dashed line). **k**, Time course showing fibroblast-enriched spots in UMAP (top) or spatial (bottom) plots. **l,m**, UMAP and spatial plots (**l**) and volcano plot (**m**) comparing 7 dpi (early) fibroblast-enriched signature (cluster 10_0, red) and 21 dpi (late) fibroblast-enriched signature (cluster 8, blue). **n,o**, TGFβ score (**n**; genes upregulated in lung adventitial fibroblasts cultured with TGFβ) and proliferation score (**o**; 23-gene signature) across early and late fibroblast signatures. *n* = 118 (early) and *n* = 160 (late) spots. MAST (model-based analysis of single-cell transcriptomics) test (hurdle model with likelihood ratio test, false discovery rate (FDR)-adjusted; **m**); two-way Mann–Whitney test (**n,o**). Slice thickness: 200 μm (**a**), 14 μm (**b,f,g**), 30 μm (**c**), 50 μm (**d**), 40 μm (**e**) and 10 μm (**h**). Images represent two or more mice.

(Fig. 1e). Focusing on the spatiotemporally reproducible PT injury model, we detected fibroblasts near the leptomeningeal wound border by 4 dpi (Extended Data Fig. 1d), which expanded by 7 dpi, surrounded and infiltrated the contracting lesion by 14–21 dpi, and persisted at 1 year post-injury (Fig. 1f,g and Supplementary Video 1). Fibroblasts accounted for around 40% of all lesional nuclei at 14 dpi, a frequency similar to that seen in the dural meninges (Extended Data Fig. 1e,f).

Fibroblasts were present both at lesional borders and in association with intralesional vascular remodelling (Extended Data Fig. 1g,h and Supplementary Video 2).

Next, we used dual-reporter mice for *Col1* expression to simultaneously track current and previous fibroblast identity (*Col1a1^GFP^* with *Col1a2^creER^; Rosa26^tdT^*)[24]. *Col1a2^creER^* recombination was more than 95% specific for *Col1a1*-GFP⁺ brain fibroblasts, with around 50% sensitivity

(percentage of tdT⁺ cells) among GFP⁺ resting fibroblasts; we observed the same sensitivity among PT lesional fibroblasts that recombined before injury, consistent with a predominant fibroblast origin for lesional fibroblasts (Extended Data Fig. 1i–l). Lesional fibroblasts expressed canonical fibroblast markers (for example, DCN, COL6A1 and POSTN; *Gli1*-lineage⁺) but not the mural cell marker desmin (Extended Data Fig. 1m,n). We observed minimal contributions of pericyte (traced with *Ng2^creER*) or vascular smooth muscle cell (*Acta2^creER*) lineages to PT lesional fibroblasts (Extended Data Fig. 1o–s). Similar results were observed in the distal middle cerebral artery occlusion model of stroke (Extended Data Fig. 1t–x). To determine whether lesional fibroblasts emerged from the dural meninges, we induced sparse recombination in dural fibroblasts but not leptomeningeal or perivascular fibroblasts by local application of 4-OHT (Extended Data Fig. 1y–ac). After PT injury, levels of fibroblast recombination in lesions were similar to those detected in the dural meninges (8–10% tdT⁺ cells; Extended Data Fig. 1ad,ae).

Next, we performed spatial transcriptomics across pre-injury (rest) and post-injury time points: acute (2 dpi), sub-acute (7 dpi) and chronic (21 dpi) phases of brain injury and repair (Fig. 1h). Dimensionality reduction revealed 23 spot-based spatial clusters, with 1 cluster that formed further subclusters (Fig. 1i and Extended Data Fig. 2a,b). We identified four major fibroblast-containing spatial clusters (*Col1a1*-enriched) with distinct temporal, molecular and microanatomical patterns (Fig. 1j,k and Extended Data Fig. 2c–g). Two of these clusters were injury-associated: an early cluster abundant at 7 dpi in perilesional regions (red) and a late cluster found at 21 dpi in the lesion core (blue) (Fig. 1l). The early cluster was enriched for profibrotic genes, particularly including targets of TGFβ signalling[18,21], and for macrophage-related chemokines and growth factors, whereas the late cluster was enriched for ECM and adaptive immune-associated genes (Fig. 1m). TGFβ has intersecting roles in physiology and immunity and is critical to driving profibrotic ECM-depositing fibroblasts (referred to here as myofibroblasts) that often express α-smooth muscle actin (αSMA)[25]. The early cluster was enriched for a myofibroblast gene score (Fig. 1n, Extended Data Fig. 2h and Supplementary Table 1) and for genes related to proliferation, a feature of myofibroblasts[25] (Fig. 1o and Extended Data Fig. 2i).

Single-nucleus RNA sequencing (snRNA-seq) of control uninjured cortex and meninges, as well as lesional/perilesional cortex after injury, revealed seven PT lesion-enriched fibroblast clusters, five meningeal fibroblast clusters and one mural cell cluster (Fig. 2a–c and Extended Data Fig. 3a–g). PT lesional fibroblast states evolved across the injury timespan, mirroring the spatial transcriptomics data (Fig. 2d and Extended Data Fig. 3h). Myofibroblast states dominated sub-acute time points (7 dpi) and expressed canonical markers such as *Cthrc1*, *Acta2*, *Lrrc15*, *Postn* and *Fn1*, with enrichment for programmes associated with ECM organization (Fig. 2e,f). *Tgfb1* was identified as the ligand most likely to drive the myofibroblast state (Fig. 2g). In a stroke model in non-human primates, we also identified discrete populations of brain fibroblasts[26], with sub-acute fibroblast clusters (7 dpi) showing increased expression of myofibroblast genes and a myofibroblast-associated score (Extended Data Fig. 3i–n and Supplementary Table 1). Reanalysis of published single-cell RNA-sequencing (scRNA-seq) data from human patients with TBI[27] or glioblastoma multiforme (GBM)[12] similarly revealed small fibroblast clusters with evidence of a myofibroblast programme (Extended Data Fig. 3o–w and Supplementary Table 1).

Next, we orthogonally validated the sub-acute myofibroblast response in vivo. Lesional fibroblasts at 7 dpi were enriched for phosphorylated SMAD3 (pSMAD3), a hallmark of active TGFβ signalling, and immunofluorescence microscopy confirmed a transient lesional myofibroblast population marked by αSMA (Extended Data Fig. 4a–e). We also performed myofibroblast lineage tracing using *Cthrc1*, a gene that is selectively expressed in profibrotic myofibroblasts[28] (*Cthrc1^creER*; *Rosa26^tdT*; Fig. 2h,i). Continuous *Cthrc1*-lineage labelling throughout the PT injury time course revealed fibroblast expression in the lesion and

injured dural meninges (Fig. 2j, Extended Data Fig. 4f and Supplementary Video 3). As in the uninjured lung[28], *Cthrc1*-lineage⁺ myofibroblasts were sparse in resting adult meninges and brain, and lineage-tracing experiments indicated they did not give rise to lesional fibroblasts (Fig. 2k). Using timed tamoxifen induction, we found that fibroblast *Cthrc1* expression (indicated by tdT⁺ cells) peaked between 0 and 7 dpi and was sparse by 14 dpi, although lesional fibroblasts that were *Cthrc1* lineage-traced persisted for one year; similar myofibroblast kinetics were observed using αSMA (*Acta2^creER*) myofibroblast lineage tracers (Fig. 2l–n and Extended Data Fig. 4g–l). Proliferation (measured by EdU incorporation) was also increased in early myofibroblasts (Extended Data Fig. 4m,n). *Cthrc1*-lineage-derived fibroblasts were observed in both transient middle cerebral artery occlusion (tMCAO) and TBI models of brain injury (Extended Data Fig. 4o,p). Together, these data suggest a conserved myofibroblast state associated with sub-acute brain injury that ultimately wanes, giving rise to discrete, persistent late lesional fibroblast states.

## Early myofibroblasts and macrophages

Similar to fibroblasts, macrophages are present across organs, including resident microglia in the CNS, and functionally adapt to their local organ; however, they can also adopt cross-organ conserved states that regulate organ damage and fibrotic responses[22,29,30]. Concurrent with the fibroblast response, an injury-responsive population of myeloid cells formed a perilesional ring by 2–7 dpi, infiltrated the PT lesion core by 14 dpi, and persisted for weeks near fibroblasts (Fig. 2o and Extended Data Fig. 5a,b). Lineage tracing revealed that perilesional myeloid cells derived from resident microglia (*P2ry12^creER*)[31], infiltrating blood monocytes (*Ccr2^creER*)[32], and border-associated macrophages (BAMs) and/or perivascular macrophages (PVMs) (*Pf4^cre*)[30,33]. Microglia-derived and monocyte-derived cells spatially diverged by 14 dpi—with enrichment in the outer glial scar and inner lesional core, respectively—but overlapped in the lesional border, inhabiting similar regions of sub-acute fibroblastic ECM. By contrast, BAM- and/or PVM-derived cells were more diffuse across lesional areas and time points (Extended Data Fig. 5c–n).

We also used snRNA-seq to define myeloid cell heterogeneity, identifying microglia, disease-associated microglia (DAMs) and several macrophage subsets, including monocyte-derived scar-associated macrophages (SAMs) or lipid-associated macrophages (LAMs) with fibrosis-associated roles in other organs[22], here collectively referred to as SAMs (Fig. 2p and Extended Data Fig. 5o–q). A SAM score[22,34] was enriched in both SAM and DAM clusters, which were spatially enriched in early areas of lesional border fibrosis at 7 dpi (Fig. 2q and Extended Data Fig. 5r). The SAM marker FABP5 colocalized with both microglial and monocytic lineage cells near *Cthrc1*-lineage⁺ myofibroblasts (Extended Data Fig. 5s–u). A similar SAM score enrichment was observed among myeloid cells in non-human primate stroke and human TBI (Extended Data Fig. 5v,w). Ligand–receptor analysis predicted that SAMs or DAMs could signal to myofibroblasts via diverse signals, including *Tgfb1* (Extended Data Fig. 5x), and that lesional myofibroblasts could reciprocally influence SAMs and DAMs via signals such as M-CSF (encoded by *Csf1*)[31] (Fig. 2r). Indeed, in an ex vivo coculture assay, 7 dpi PT lesions—enriched in myofibroblasts—promoted increased microglial expression of DAM markers compared with 21 dpi lesions or microglial monoculture controls (Extended Data Fig. 5y). These data suggest the existence of a transient myeloid cell programme shared by microglia and recruited macrophages that is spatiotemporally and functionally associated with myofibroblasts, probably reflecting shared environmental damage-associated cues[22].

## Late lesional fibroblasts and T cells

Distinct fibroblast states emerged at late time points after injury (21 dpi). 'Altered dural' and leptomeningeal (pial and arachnoid)

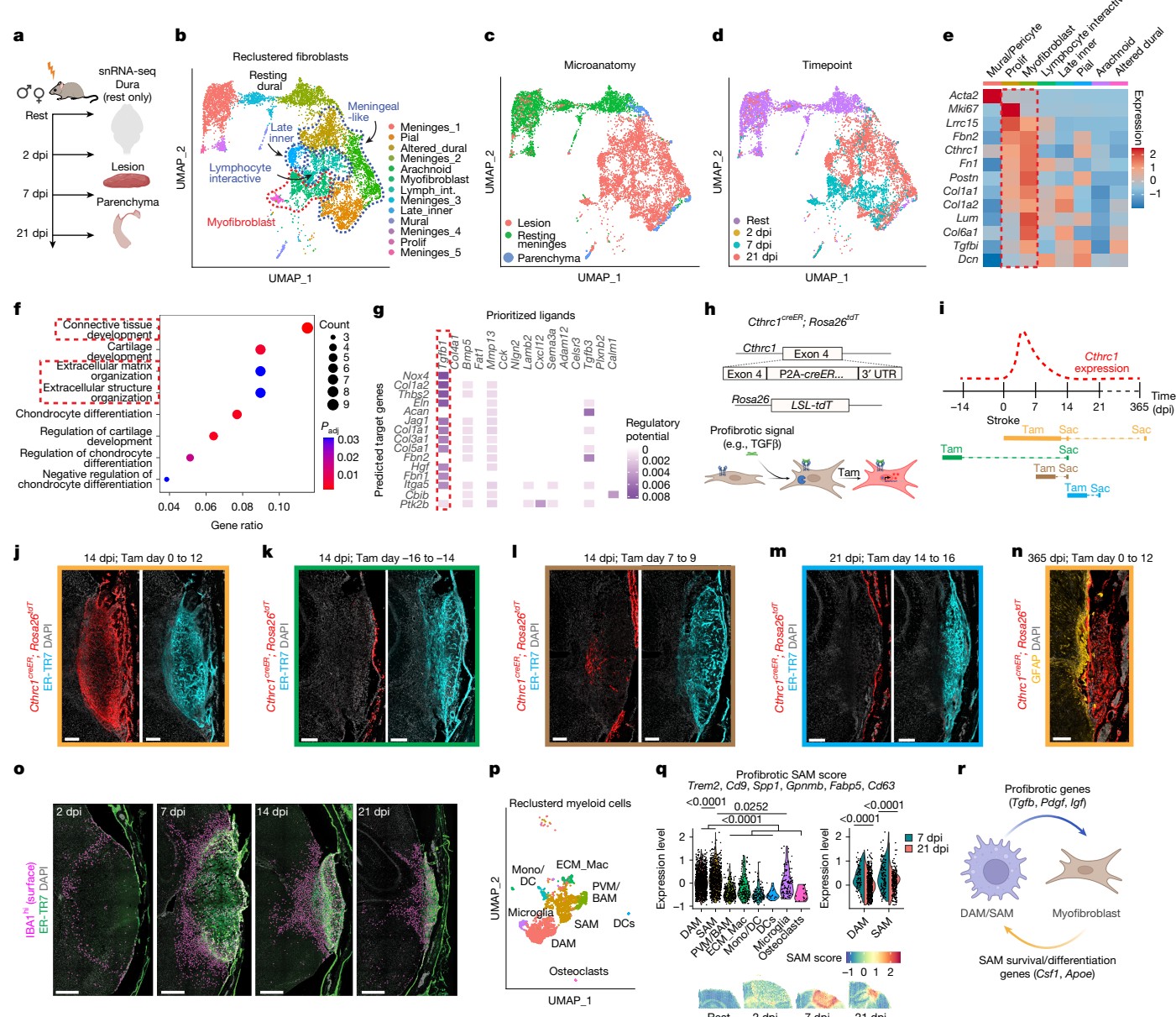

**Fig. 2 | Myofibroblasts spatiotemporally correlate with lesional profibrotic macrophages and disease-associated microglia. a**, Schematic of the snRNA-seq experiment showing sample time points and microanatomy resulting in a library of 28,187 nuclei. **b–d**, UMAP with 8,096 fibroblasts and 189 mural cells, showing fibroblast subclusters (**b**), microanatomy (**c**) or time point (**d**). Dotted lines (**b**) highlight early myofibroblast/proliferative clusters (red) and multiple late states (blue). **e**, Heat map of select fibrosis-related genes expressed in proliferative and myofibroblast clusters (red dashed line). **f,g**, Gene set enrichment analysis (**f**) and ligand–transcriptional-network analysis (**g**; NicheNet) of myofibroblasts. *Tgfb1* is highlighted as the top predicted driving ligand. $P_{adj}$, adjusted $P$ value. **h,i**, Schematics showing *Cthrc1^{creER}*; *Rosa26^{tdT}* lineage tracing[28] (**h**) and hypothesized *Cthrc1* expression trajectory (**i**; with tamoxifen (Tam) induction regimens for **j–n**). **j–n**, Confocal microscopy showing *Cthrc1*-lineage⁺ lesional fibroblasts after PT injury (tamoxifen induction and collection days indicated), with robust injury-induced *Cthrc1*–tdT expression (**j**) but lack of lineage-traced fibroblasts (**k**). Active *Cthrc1* expression is reduced but present at 7 dpi (**l**; shown at 14 dpi) and absent by 14 dpi

(**m**; shown at 21 dpi). Labelled fibroblasts persist to 365 dpi (**n**). Scale bars, 200 μm. **o**, Confocal microscopy showing time course of lesional accumulation of myeloid cells (IBA1^hi surface). Scale bars, 500 μm. **p,q**, Annotated UMAP of reclustered myeloid cells (**p**) and violin or spatial plots of profibrotic SAM score (**q**; *Trem2, Cd9, Spp1, Gpnmb, Fabp5* and *Cd63*). **q**, Left, score by cluster; DAM: n = 1,267, SAM: n = 816, PVM/BAM: n = 305, ECM macrophages (ECM_Macs): n = 173, monocyte/dendritic cell (Mono/DC): n = 116, dendritic cells (DCs): n = 20, microglia: n = 110, osteoclasts: n = 19 nuclei. Right, score by time point within DAM and SAM lesional clusters; 7 DPI DAM: n = 168, 21 dpi DAM: n = 1,080, 7 DPI SAM: n = 247, 21 dpi SAM: n = 506. Bottom, score mapped onto spatial transcriptomic Visium data. **r**, Schematic showing potential ligand–receptor interactions between SAM and DAM and myofibroblasts, derived from Extended Data Fig. 5x. Over-representation test (one-sided Fisher's exact test, FDR-adjusted) (**f**); Kruskall–Wallis test, Dunn's multiple comparisons correction (**q**, left; relevant comparisons shown); two-way Mann–Whitney test, Bonferroni correction (**q**, right). Slices thickness, 14 μm. Images represent two or more mice.

fibroblasts, expressing meningeal layer markers[35], showed a predicted border distribution at rest and after injury (Fig. 3a–c and Extended Data Fig. 6a). Quantitative microscopy confirmed this topography after injury, with leptomeningeal fibroblasts (ALDH1A2⁺LAMA1⁺) adjacent

to reactive astrocytes, whereas altered dural fibroblasts (ALPL⁺) were enriched at lesion-meningeal interfaces (Fig. 3d,e and Extended Data Fig. 6b–d). A 'late inner' fibroblast state was enriched in the lesion core (Fig. 3d,e, FGF13⁺), expressing ECM genes and markers associated with

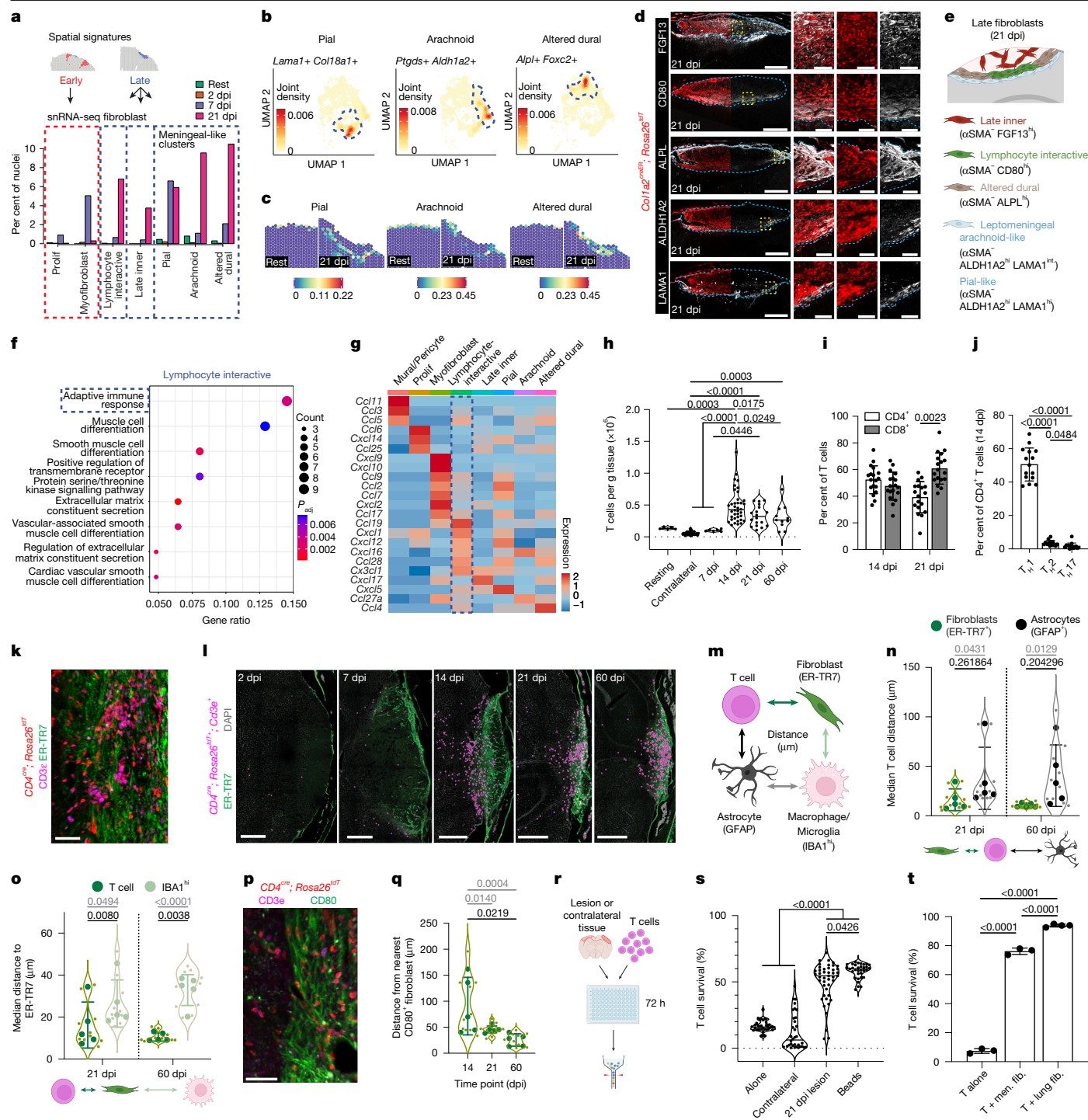

**Fig. 3 |** See next page for caption.

smooth muscle function (for example, *Cdh18* and *Sema3c*) but lacking markers of bona fide myofibroblasts (Fig. 2e and Extended Data Fig. 6e–i). 'Lymphocyte-interactive' fibroblasts (CD80⁺) were found in outer lesional border regions and phylogenetically resembled dural fibroblasts (Extended Data Fig. 6j). *Cthrc1*-lineage tracing suggested that most injury-associated fibroblasts pass through a transient myofibroblast state before acquiring their discrete late identities and positionings (Extended Data Fig. 6k). Analogous late fibroblast states were observed in the tMCAO stroke model (Extended Data Fig. 6l).

The lymphocyte-interactive fibroblast cluster was enriched for signals that recruit lymphocytes (for example, *Cxcl12*, *Ccl19* and *Cxcl16*[15,36];

Fig. 3f,g). Consistent with this, T lymphocytes were rare in the uninjured brain and early phases of injury but accumulated by 14 dpi (Fig. 3h, Extended Data Fig. 6m and Supplementary Fig. 1a). Infiltrating T cells were predominantly CD4⁺ and CD8⁺ T cells, with enriched CD8⁺ T cells at chronic time points; most CD4⁺ T cells expressed the transcription factor TBET, consistent with a T helper 1 (T_H1) IFNγ-expressing identity (Fig. 3i,j and Extended Data Fig. 6n). T cells were elevated in the lesioned hemisphere through at least 60 days, where they were enriched for a memory phenotype (CD44⁺CD69⁺; Fig. 3h and Extended Data Fig. 6o). To define the spatial relationships between T cells and lesional fibroblasts, we used a pan T cell reporter that labels both CD4⁺

**Fig. 3 | Distinct late fibroblast states include lymphocyte-interactive fibroblasts associated with T cell persistence. a**, Fibroblast cluster abundance over time, with corresponding early and late spatial signatures. **b**, Co-expression of pial, arachnoid and dural genes[35] among corresponding clusters. **c**, snRNA-seq signatures of meningeal fibroblast subsets mapped onto spatial transcriptomic data. **d,e**, Fluorescence microscopy (**d**) and cartoon (**e**) of late fibroblast subset topography and protein expression. Scale bars: 500 μm (main images), 100 μm (enlarged views). **f**, Gene set enrichment analysis among lymphocyte-interactive fibroblasts. **g**, Heat map of chemokines expressed in ≥0.5% of any cluster; lymphocyte-interactive fibroblasts are highlighted (blue). **h–j**, Total T cells (**h**; CD3ε⁺), CD4⁺ and CD8⁺ T cells (**i**) and or $T_H1$ (TBET⁺), T helper type 2 ($T_H2$; GATA3⁺) and T helper type 17 ($T_H17$; RORγt⁺) CD4⁺ T cells (**j**; 14 dpi) after PT injury (cortical flow cytometry). Rest: n = 5, contralateral: n = 46, 7 dpi: n = 6, 14 dpi: n = 38, 21 dpi: n = 18, 60 dpi: n = 11 mice (**h**); n = 18 mice per time point (**i**); n = 15 mice (**j**). **k,l**, Native microscopy (**k**; 21 dpi) and time course of T cell surfaces (**l**; $Cd4^{cre}$; $Cd3e^{+}$) near fibroblast-rich lesions (ER-TR7⁺). Scale bars: 50 μm (**k**), 500 μm (**l**). **m**, Schematic of proximity analysis between

T cells ($CD4^{cre}$; $Rosa26^{tdT+}$; $Cd3e^{+}$) or myeloid cells (IBA1ʰⁱ, macrophages and reactive microglia) and fibroblast ECM (ER-TR7) or astrocytes (GFAP). **n,o**, Median T cell distance from nearest fibroblast ECM or astrocyte surface (**n**) and T cell or myeloid cell distance from nearest fibroblast surface (**o**). 21 and 60 dpi. n = 5 mice per time point (2 slices per mouse; lighter green or grey dots and P values represent tissue slices; darker dots and P values are per mouse). **p,q**, Image (**p**; 21 dpi) and quantification (**q**) of T cell proximity to CD80⁺ lymphocyte-interactive fibroblasts at 14, 21 and 60 dpi. n = 5 mice per time point (2 slices per mouse). Scale bar, 50 μm. **r,s**, Schematic for ex vivo lesional coculture (**r**) and quantification of T cell survival (**s**; 21 dpi lesions). Alone: n = 36, contralateral: n = 34, 21 dpi lesion: n = 44, beads: n = 33 wells. **t**, T cell survival after coculture with dural or lung fibroblasts. T cells alone (T alone): n = 3, T cells plus meningeal fibroblasts (men. fib.): n = 3, lung: n = 4 wells. Over-representation test (one-sided Fisher's exact test, FDR-adjusted) (**f**); one-way ANOVA, Tukey post-test (**h,q,s,t**); multiple two-way t-tests, Holm–Sidak correction (**i**; paired per point, unpaired per slice); one-way repeated-measures ANOVA, Tukey post-test (**j**). Slice thickness, 14 μm. Images represent two or more mice.

---

and CD8⁺ T cells ($CD4^{cre}$; $R26^{tdT}$; Fig. 3k,l). By chronic time points after injury, persisting T cells increasingly localized near fibroblastic ECM (Fig. 3m,n and Extended Data Fig. 6p,q). T cells showed increased association with lesions compared with myeloid cells and were near lymphocyte-interactive fibroblasts (Fig. 3o–q and Extended Data Fig. 6r,s). In an ex vivo coculture assay, dissected 21 dpi PT lesions uniquely supported T cell survival over 72 h without affecting T cell proliferation or activation (Fig. 3s, Extended Data Fig. 6t–v and Supplementary Fig. 1b). Chronic lesions at 21 dpi promoted increased T cell survival compared with early 7 dpi lesions, and purified meningeal fibroblasts were also sufficient to support T cell survival in vitro (Fig. 3t and Extended Data Fig. 6w,x). Dendritic cells were enriched in the lesioned cortex to 60 dpi and may also contribute to lesional lymphocyte support in vivo (Extended Data Fig. 6y). These data suggest that late injury-associated fibroblasts can recruit and support brain lymphocytes.

## TGFβ coordinates brain myofibroblasts

Next, we functionally tested the roles of myeloid cells after brain injury. We confirmed the predicted beneficial role of lesional macrophages[32,37] using clodronate liposomes to preferentially deplete infiltrating monocytes, macrophages and perilesional myeloid cells (Extended Data Fig. 7a–e and Supplementary Fig. 1c,d). Treatment with clodronate liposomes reduced PT lesional fibroblasts and fibroblastic ECM and increased lesion size (Fig. 4a–c and Extended Data Fig. 7f–h). Ligand–receptor analysis highlighted *Tgfb1* as a macrophage ligand that could contribute to the support of myofibroblasts (Fig. 2r and Extended Data Fig. 5x). *Tgfb1* colocalized with perilesional myeloid cells by 2 dpi, and myeloid cells exhibited the highest *Tgfb1* expression after injury (Extended Data Fig. 7i,j). *TGFB1* was also highly expressed by myeloid cells after stroke in non-human primates and in human TBI (Extended Data Fig. 5v,w and Extended Data Fig. 7k,l). Conditional deletion of TGFβ1 from microglia and macrophages ($Cx3cr1^{creER}$; $Tgfb1^{flox/GFP-KO}$) resulted in decreased lesional fibroblastic ECM after PT injury, suggesting that myeloid-derived TGFβ1 contributes to the injury-induced fibroblast response (Extended Data Fig. 7m–o).

Next, we tested the functional contributions of myofibroblasts, which are canonically driven by TGFβ[25]. Given the diverse roles of TGFβ in development[38], we used a conditional-knockout (cKO) system to inducibly delete TGFβ signalling in fibroblasts ($Col1a2^{creER}$; $Tgfbr2^{flox}$; Fig. 4d). Adult *Tgfbr2*-cKO mice were viable and had normal fibroblast resting topography (Extended Data Fig. 8a), consistent with a minimal contribution of the myofibroblast state in adult physiology[18] and/or decreased efficiency of the $Col1a2^{creER}$ allele prior to injury. After injury, *Tgfbr2*-cKO mice had markedly reduced lesional fibroblasts and associated ECM and increased lesion size, effects requiring sustained tamoxifen induction

during injury (Fig. 4e–h and Extended Data Fig. 8b–i). Neutrophil and monocyte numbers were increased acutely in the PT-injured cortex but declined similarly by 4 dpi in both control and *Tgfbr2*-cKO mice; however, *Tgfbr2*-cKO mice showed secondary increases in neutrophils and monocytes to at least 21 dpi (Fig. 4i and Extended Data Fig. 8j,k). Late increases in neutrophil and monocyte numbers were specific to the lesioned cortex (Extended Data Fig. 8l–r). *Tgfbr2*-cKO mice had increased cleaved caspase-3⁺ puncta, a marker of white matter degeneration after TBI[39], in the corpus callosum (Extended Data Fig. 8s,t). Control and *Tgfbr2*-cKO mice had similar amounts of vascular leakage and haemorrhage after PT injury (Extended Data Fig. 8u,v).

To orthogonally test the role of brain myofibroblasts after injury, we used myofibroblast deleter mice ($Cthrc1^{creER}$; $Rosa26^{DTA}$), in which tamoxifen drives selective loss of myofibroblasts[28]. Similar to *Tgfbr2*-cKO mice, Cthrc1-deleter mice had increased brain lesion size and reduced lesional ECM (Fig. 4j–l); however, these effects were modest, consistent with inefficient *Cthrc1*-driven myofibroblast deletion[28]. We also examined the role of TGFβ-driven myofibroblast state in chronic stage wound healing ('late tamoxifen', beginning at 14 dpi). Late tamoxifen-induced *Tgfbr2*-cKO mice showed no differences in lesion size or neutrophil levels, with mild reductions in lesional ECM (Fig. 4m,n and Extended Data Fig. 8w). These data suggest a critical sub-acute window after brain injury, during which myofibroblasts limit perilesional damage and secondary neuroinflammation.

TGFβ is canonically secreted into the ECM as a latent cytokine that requires activation[38], often via $α_v$-paired integrins[40]. In the brain, co-expression of genes encoding $α_vβ_1$ or $α_vβ_6$ integrin was restricted to resting meningeal fibroblasts, whereas $α_vβ_8$ (*Itgav* and *Itgb8*) co-expression was high in astrocytes and oligodendrocytes and variable in lesional fibroblasts (Extended Data Fig. 8x). The $α_vβ_8$-blocking antibody ADWA11[41] partially phenocopied *Tgfbr2*-cKO mice, with reduced lesional fibroblasts and fibroblastic ECM coverage, although lesion sizes were unaltered (Fig. 4o,p and Extended Data Fig. 8y–ab). As in *Tgfbr2*-cKO mice, $α_vβ_8$ blockade drove chronic neutrophilia and trending monocytosis that were specific to the lesioned cortex (Fig. 4q and Extended Data Fig. 8ac–ae). *Itgb8* reporter mice labelled perilesional astrocytes but not lesional fibroblasts (Extended Data Fig. 8af). Accordingly, genetic deletion of *Itgb8* from the brain parenchyma, including both glial cells and neurons ($GFAP^{cre}$; $Itgb8^{flox}$), led to reduced lesional ECM with unchanged lesion size, phenocopying $α_vβ_8$-blockade (Fig. 4r,s and Extended Data Fig. 8ag). Similar results were observed using cortex-specific $Emx1^{cre}$ (Extended Data Fig. 8ah–aj). These data suggest that $α_vβ_8$ from lesion-adjacent glial cells licenses TGFβ-mediated myofibroblast expansion, although additional mechanisms of TGFβ activation are likely to also contribute[38].

Next, we performed snRNA-seq of PT lesional tissue and associated residual brain parenchyma in *Tgfbr2*-cKO, ADWA11-treated and control

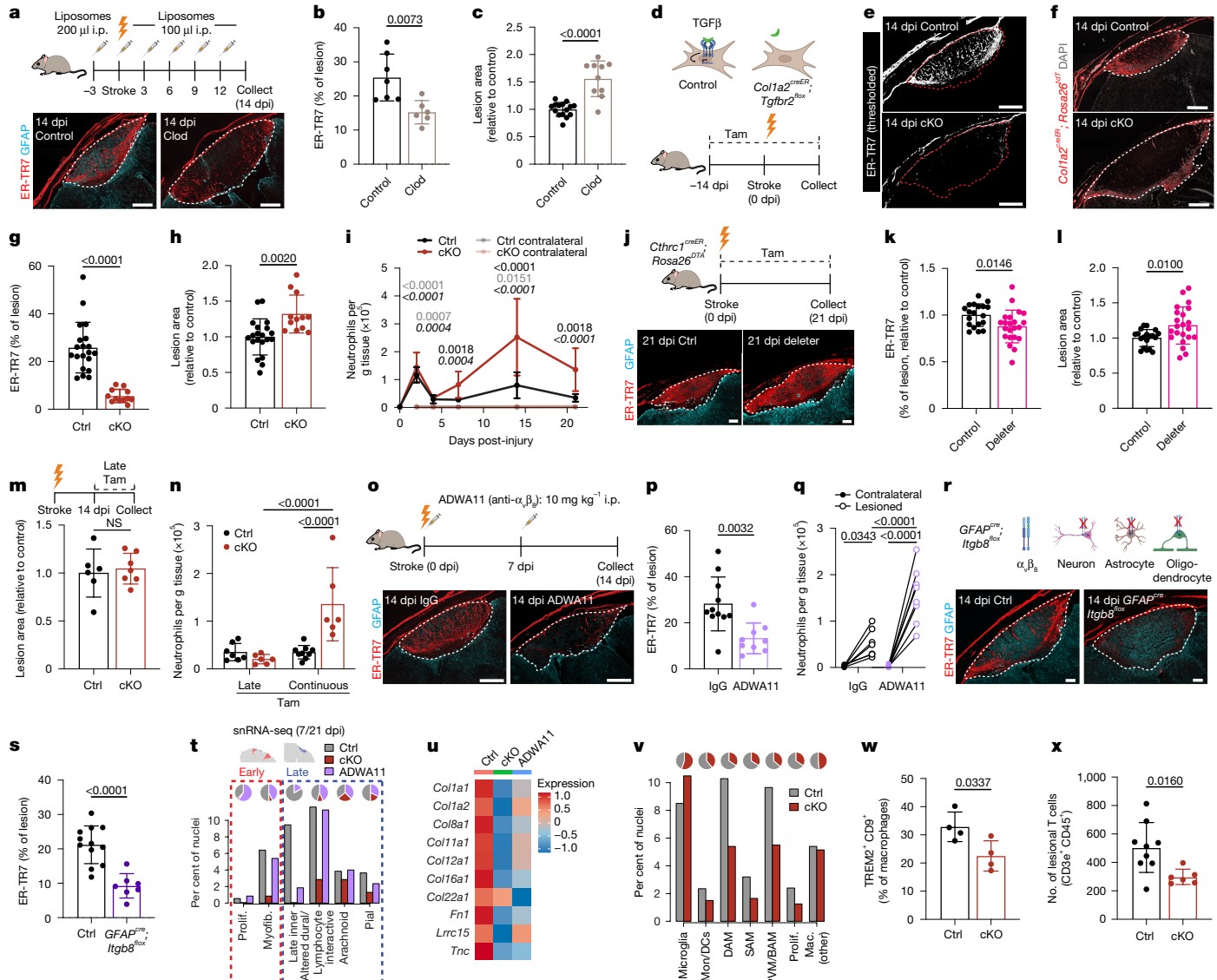

**Fig. 4 | Brain lesional myofibroblasts are coordinated by TGFβ signalling, profibrotic myeloid cells and perilesional glia to drive wound healing and limit chronic inflammation. a–c,** Treatments with control or clodronate (Clod) liposomes and images of fibroblasts and astrocytes in lesions (**a**), ECM (ER-TR7) coverage (**b**) and lesion size (**c**) at 14 dpi. **b,** Control: *n* = 7, clodronate: *n* = 6 mice. **c,** Control: *n* = 15, clodronate: *n* = 10 mice. Two slices per mouse. Scale bars, 500 μm. **d,** Schematic of ablated fibroblast TGFβ signalling (*Col1a2^creER^; Tgfbr2^flox^*; tamoxifen treatment on days −14 to −12, −7, 0 to 2, 5, 8 and every 3 days subsequently until sample collection). **e–h,** Images of control and *Tgfbr2*-cKO (cKO) lesions with thresholded ER-TR7 (**e**) or *Col1a2^creER^; Rosa26^tdT+^* fibroblasts (**f**), ER-TR7 coverage (**g**) and lesion size (**h**) at 14 dpi. Control (ctrl): *n* = 20, *Tgfbr2*-cKO: *n* = 12 mice; 2 slices per mouse. Scale bars, 500 μm. **i,** Cortical neutrophils in controls and *Tgfbr2*-cKO lesions. 0 dpi control (non-littermate-sham): *n* = 8, 2, 4 and 7 dpi control and 2, 7 and 21 dpi *Tgfbr2*-cKO: *n* = 6, 14 dpi control: *n* = 12, 21 dpi control: *n* = 9, 0 dpi *Tgfbr2*-cKO: *n* = 3, 4 dpi *Tgfbr2*-cKO: *n* = 5, 14 dpi *Tgfbr2*-cKO: *n* = 7 mice. **j–l,** Images of lesions in control (cre-negative or vehicle) and myofibroblast deleter (Cthrc1^creER^; Rosa26^DTA^) mice (**j**), and ER-TR7 coverage (**k**) and lesion size (**l**) at 21 dpi. Control: *n* = 19, deleter: *n* = 22 mice; 2 slices per mouse. Scale bars, 200 μm. **m,** Lesion size at 28 dpi in controls and *Tgfbr2*-cKO mice after late tamoxifen treatment (days 14–16, 19, 22 and 25). Control: *n* = 6, *Tgfbr2*-cKO: *n* = 7 mice per group; 2 slices per mouse. **n,** Cortical

neutrophils at 21 dpi after late or continuous tamoxifen treatment. Control late: *n* = 7, *Tgfbr2*-cKO late: *n* = 6, *Tgfbr2*-cKO continuous: *n* = 6, control continuous: *n* = 9 mice. **o–q,** Images of cortical lesions in control (IgG) mice or mice with α_vβ_8 blockade (ADWA11) at 14 dpi (**o**), ER-TR7 coverage (**p**) and cortical neutrophils (**q**). IgG: *n* = 11, ADWA11: *n* = 9 mice (**p**; 2 slices per mouse); IgG: *n* = 7, ADWA11: *n* = 8 mice (**q**). Scale bars, 500 μm. **r,s,** Cortical lesions in control and *GFAP^cre^; Itgb8^flox^* mice at 14 dpi (**r**) and ER-TR7 coverage (**s**). Control: *n* = 12 (control), *Tgfbr2*-cKO: *n* = 7 mice; 2 slices per mouse. Scale bars, 200 μm. **t,** Fibroblast snRNA-seq cluster abundance in control, *Tgfbr2*-cKO (*Col1a2^creER^; Tgfbr2^flox^*), and ADWA11-treated mice at 7 dpi (early) and 21 dpi (late). **u,** Expression of myofibroblast genes in myofibroblasts across time points. **v–x,** Myeloid cluster abundance (**v**), SAM frequency (flow cytometry) (**w**) and perilesional T cell numbers (**x**; 14 dpi). *n* = 4 mice per group (**w**); control: *n* = 9, *Tgfbr2*-cKO: *n* = 6 mice (**x**; 2 slices per mouse). Quantification normalized to controls in each experiment (**c,h,k,l**). Two-way Student's *t*-test (**b,c,g–i** (0 dpi), **k–m,p,s,w,x**); two-way repeated-measures ANOVA, Sidak's post-test (**i**) (7–21 dpi, per time point; bold: lesioned control versus lesioned *Tgfbr2*-cKO, grey: lesioned control versus contralateral control, italic: lesioned *Tgfbr2*-cKO versus contralateral *Tgfbr2*-cKO); two-way repeated-measures ANOVA, Sidak's post-test (**q**); two-way ANOVA, Sidak's post-test (**n**). Slice thickness: 14 μm. Dotted lines indicate lesion boundary (**a,e,f,j,o,r**); images represent two or more mice.

mice (Extended Data Fig. 9a,b). We focused on lesional stromal cells, which mapped onto our previously defined brain fibroblast states (Extended Data Fig. 9c–i). *Tgfbr2*-cKO mice exhibited a profound

loss of the sub-acute myofibroblast clusters at 7 dpi and a substantial reduction in multiple late fibroblast clusters at 21 dpi, consistent with a requirement for myofibroblasts in generating late lesional fibroblasts

(Fig. 4t and Extended Data Fig. 9j). Quantitative microscopy from *Tgfbr2*-cKO mice showed reductions in myofibroblasts at 7 dpi and several late fibroblast subsets at 21 dpi, including lymphocyte-interactive, dural-like and pial fibroblasts (Extended Data Fig. 9k–p). Mice with $\alpha_v\beta_8$-blockade had relatively preserved fibroblast states; however, myofibroblasts from both *Tgfbr2*-cKO and $\alpha_v\beta_8$-blocked mice expressed fewer ECM and profibrotic genes (Fig. 4u and Extended Data Fig. 9q,r), consistent with impairment of the myofibroblast programme.

snRNA-seq analysis showed that *Tgfbr2*-cKO mice also had fewer cells in DAM and SAM myeloid clusters, and trajectory analysis revealed impaired monocyte-to-SAM and microglia-to-DAM transitions; flow cytometry validated a relative decrease in SAM (Fig. 4v,w and Extended Data Fig. 9s–x). ADWA11 treatment ($\alpha_v\beta_8$-blockade) led to a pro-inflammatory and dysmature myeloid signature, consistent with roles for *Itgb8*-mediated TGFβ autocrine signalling in sustaining microglial identity[42,43] (Extended Data Fig. 9y,z). *Tgfbr2*-cKO mice also had reduced lesion-associated T cells, consistent with the loss of late lymphocyte-interactive fibroblasts, with residual T cells scattered at lesion boundaries (Fig. 4x and Extended Data Fig. 9aa). Lymphocytes from *Tgfbr2*-cKO mice brains, particularly CD8[+] and γδ T cells, had altered expression of genes such as *Itgae* (encoding CD103) by 21 dpi (Extended Data Fig. 9ab–ad). These data suggest that myofibroblast interactions after injury shape the brain immune landscape.

## Roles of brain fibroblasts after injury

Our PT damage model caused mild injury with minimal functional impairment[44]. Therefore, we turned to tMCAO, a severe ischaemia–reperfusion injury model that mirrors aspects of human stroke microanatomy and pathophysiology[45]. Control mice formed robust fibrotic lesions by 14 dpi, whereas *Tgfbr2*-cKO mice with TGFβ-blind fibroblasts displayed impaired fibrotic lesion formation and substantial cortical tissue loss despite intact glial scarring (Fig. 5a). *Tgfbr2*-cKO mice also had increased sub-acute mortality (approximately 90% versus 15% of controls), whereas uninjured cKO mice were grossly normal with no mortality (Fig. 5b and Extended Data Fig. 10a). The lesioned hemispheres from tMCAO-injured *Tgfbr2*-cKO mice at 3 dpi were enlarged relative to controls, with increased midline shift, elevated polyclonal IgM staining and trending increased IgG and Ter119 staining (Fig. 5c,d and Extended Data Fig. 10b–d). These data suggest the possibility of exacerbated vasogenic oedema, a common cause of mortality in human patients with stroke[46]. *Tgfbr2*-cKO brains also had decreased mature oligodendrocytes and relatively more degenerating or dying neurons (Fig. 5e,f). To investigate potential downstream effects of exacerbated CNS damage, we performed vital sign monitoring at early time points. tMCAO caused a decrease in heart rate and a trending decrease in blood pressure, both of which were significantly exacerbated in *Tgfbr2*-cKO mice (Fig. 5g,h, Extended Data Fig. 10e,f). Injured *Tgfbr2*-cKO mice also showed signs of liver damage (elevated alanine transaminase (ALT) and hepatocyte apoptosis), although kidney function (serum creatinine) and oxygen saturation were unaltered (Extended Data Fig. 10g–k). These data suggest that an intact myofibroblast response is required to limit early-to-sub-acute tMCAO ischaemic brain damage. In *Tgfbr2*-cKO mice, although the precise mechanism underlying cardiovascular pathology remains incompletely defined, we hypothesize that tissue loss above a critical threshold, or in critical brain regions, promotes cardiac dysfunction, systemic decompensation, end-organ damage and death[47,48].

Next, we focused on the lymphocyte-interactive fibroblast state that emerged at chronic time points after injury. The chemokine *Cxcl12* was highly expressed by late fibroblast subsets, particularly including the lymphocyte-interactive fibroblast state, but was minimally expressed by myofibroblasts (Extended Data Fig. 10l,m). We conditionally deleted *Cxcl12* from all fibroblasts in adult mice, bypassing developmental roles of CXCL12[49] (Fig. 5i). Chronic lesions from *Cxcl12*-cKO mice had

decreased lesional T cells without changes in PT lesion area, fibroblastic ECM deposition or lymphocyte-interactive fibroblasts (Fig. 5j–l and Extended Data Fig. 10n–q), as well as a trend towards decreased lesion-proximal MHCII[+] cells (Extended Data Fig. 10r), in accordance with reduced late lesional IFNγ-producing type 1 lymphocytes[50]. By contrast, total brain-infiltrating T cells were not altered, and effector cytokine expression (IFNγ and IL-17A) in brain CD8[+] T cells, CD4[+] conventional T cells and γδ T cells was increased (Fig. 5m–p, Extended Data Fig. 10s–y and Supplementary Fig. 1a). Neutrophil numbers were modestly increased in injured hemispheres of *Cxcl12*-cKO mice (Fig. 5q). No differences were found in the production of the inhibitory cytokine IL-10 or in meningeal or circulating immune cells (Extended Data Fig. 10z–af). These data suggest that loss of fibroblast-derived CXCL12 causes T cell mislocalization into the brain with immune dysregulation and elevated cytokine expression.

We also performed snRNA-seq on PT lesional tissue and perilesional cortex in *Cxcl12*-cKO mice and controls at 21 dpi (Fig. 5r,s). We identified all expected late fibroblast states, and *Cxcl12*-cKO mice showed minimal changes in fibroblast state abundance (Extended Data Fig. 10ag,ah). We identified sparse lesional type 1 T cells that were diminished in number in *Cxcl12*-cKO mice, consistent with microscopy data (Fig. 5t and Extended Data Fig. 10ai–ak). After identifying all expected myeloid subtypes (Extended Data Fig. 10al,am), we generated a myeloid IFNγ-response signature comprising genes that were upregulated in microglia after in vivo IFNγ injection; in line with impaired lesional localization of type 1 T cells in *Cxcl12*-cKO mice, lesional myeloid cells showed a decreased IFNγ response signature (Fig. 5u and Supplementary Table 2), although parenchymal myeloid cell yield was insufficient for further analysis. DAM and SAM lesional myeloid states were enriched in *Cxcl12*-cKO mice, particularly including a SAM subset with increased expression of pro-inflammatory/M1-like genes (Extended Data Fig. 10an,ao). After identifying excitatory and inhibitory neuronal subtypes (Fig. 5v), we generated a neuronal IFNγ-response signature comprising genes that were upregulated after IFNγ injection. *Cxcl12*-cKO mice showed increased expression of IFNγ-induced genes among both excitatory and inhibitory neurons (Fig. 5w). Control neurons (compared with *Cxcl12*-cKO neurons) were enriched for several programmes involving synaptic organization and transmission, consistent with evidence that IFNγ fine-tunes neuronal function and can modulate excitatory–inhibitory balance[51–54] (Extended Data Fig. 10ap). These data suggest that late fibroblast subset(s) maintain lesional immune niches, at least in part via the chemokine CXCL12, to support long-term lymphocyte accumulation with dampened cytokine expression (Fig. 5x). Together, fibroblasts and their interactions with immune cells appear to be critical regulators of early brain injury and repair as well as long-term CNS neuroimmune balance (Supplementary Fig. 2).

## Discussion

Here we characterize fibroblast–immune interactions after brain injury, which are conserved across several injury models and species, involving fibroblast activation, proliferation, migration and ultimate persistence in multiple spatially and functionally discrete states. We identify two discrete adaptive functions of brain fibroblasts after injury that act at distinct temporal phases. Although stromal cells are recognized as participants in CNS physiology and pathology, their ontogeny remains controversial. Distinct studies arguing for pericyte[6,7,9] or fibroblast[11,55] origins for damage-associated CNS stromal cells have relied on shared markers[1] or qualitative lineage tracing[8]. For example, spinal cord injury-responsive stromal cells were first described as pericytes on the basis of expression of *Slc1a3* (also known as *Glast*); however, *Slc1a3* is also expressed by CNS and injury-responsive fibroblasts[11,56], potentially consistent with an *Slc1a3*-lineage[+] fibroblast origin. We use stromal cell lineage tracing to define resting CNS fibroblasts as a major source for PT injury-responsive fibroblasts, though we do not rule out

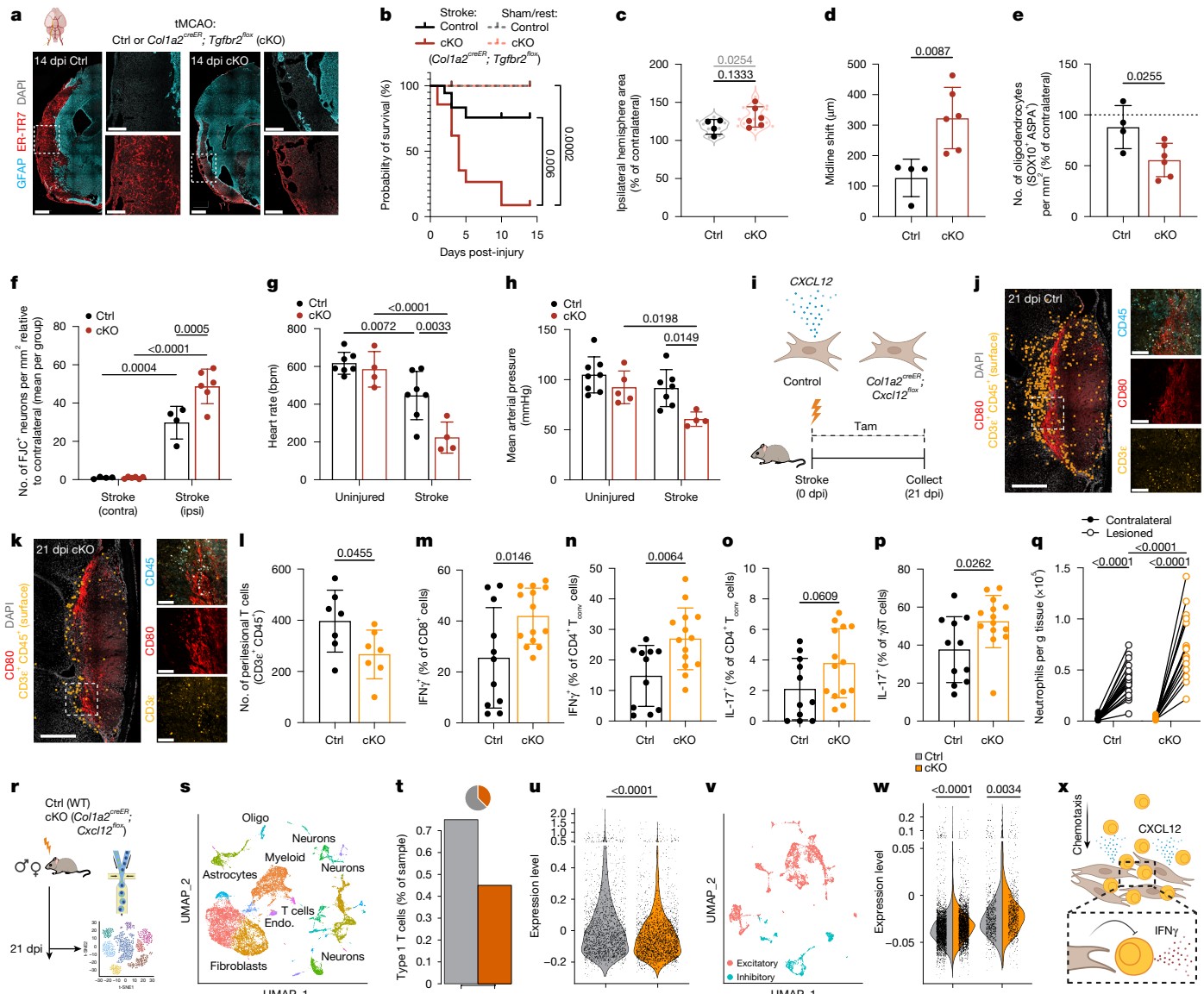

**Fig. 5 | Discrete beneficial functions of early myofibroblasts and late lymphocyte-interactive fibroblasts. a,b,** tMCAO-induced fibrotic lesion at 14 dpi (**a**) and mouse survival (**b**). Control stroke: *n* = 18, *Tgfbr2*-cKO stroke: *n* = 21, control sham/rest: *n* = 10, *Tgfbr2*-cKO sham/rest: *n* = 15 mice. Mice were treated with tamoxifen at −3 to −1, 2, 5, 8 and 11 dpi. Scale bars: 500 μm (main images), 200 μm (enlarged views). **c–f,** Ipsilateral hemisphere area (**c**), brain midline shift (**d**), oligodendrocyte density (**e**) and FluoroJade-C⁺ (degenerating) neuron density (**f**) at 3 days after tMCAO. Control, *n* = 4, *Tgfbr2*-cKO, *n* = 6 mice. Two slices per mouse (**c,d**); light grey or red dots and text represent values and *P* values per tissue slice, dark colors per mouse (**c**); normalized to contralateral (**c,e,f,** per group). **g,h,** Heart rate (**g**) and mean arterial pressure (**h**) 1 to 3 days after tMCAO. Control groups: *n* = 7, *Tgfbr2*-cKO groups: *n* = 4 mice (**g**); control uninjured: *n* = 8, *Tgfbr2*-cKO uninjured: *n* = 5, control stroke: *n* = 7, *Tgfbr2*-cKO stroke: *n* = 4 mice (**h**). *n* = 1 *Tgfbr2*-cKO stroke recorded at 1 dpi, *n* = 1 *Tgfbr2*-cKO stroke recorded at 2 dpi; remaining mice recorded at 3 dpi. **i,** Schematic showing loss of CXCL12 production in *Col1a2^creER^; Cxcl12^flox^* fibroblasts with tamoxifen treatment at 0 to 2, 7 to 9 and 14–16 dpi. **j–l,** Surfaced T cells (CD3ε⁺CD45⁺, processed via Imaris software 'surface' function) near lymphocyte-interactive fibroblasts (CD80⁺) in control (**j**) or *Cxcl12*-cKO mice at 21 dpi (**k**) with

quantification (**l**). *n* = 7 mice per group; 2 slices per mouse. Scale bars: 500 μm (left), 100 μm (right). **m–p,** IFNγ expression in CD8⁺ T cells (**m**), IFNγ (**n**) and IL-17A (**o**) expression in CD4⁺ conventional T (T_conv) cells, and IL-17A expression in γδ T cells (**p**) at 21 dpi. Control: *n* = 11, *Cxcl12*-cKO: *n* = 14 mice. **q,** Cortical neutrophils in control and *Cxcl12*-cKO mice at 21 dpi. Control, *n* = 21, *Cxcl12*-cKO: *n* = 16 mice. **r,** Schematic of snRNA-seq experiment. Control (wild-type (WT)) and *Cxcl12*-cKO mice were collected at 21 dpi (2 mice per genotype), lesions and parenchyma were micro-dissected and multiplexed, and nuclei were sorted and sequenced (19,668 nuclei). **s,** Global UMAP analysis of samples depicted in **r. t,** Type 1 T cell abundance (180 nuclei). **u,** IFNγ response score among lesional myeloid cells. Control: *n* = 1,659, *Cxcl12*-cKO: *n* = 2,056 nuclei. **v,** UMAP of neuronal cells (5,862 nuclei). **w,** IFNγ response score of excitatory and inhibitory neurons. Wild-type excitatory: *n* = 2,392, wild-type inhibitory: *n* = 576, *Cxcl12*-cKO excitatory: *n* = 2,229, *Cxcl12*-cKO inhibitory: *n* = 665 nuclei. **x,** Schematic showing a model of T cell regulation via late lesional fibroblast-derived CXCL12. log-rank (Mantel–Cox) test, Bonferroni correction (*k* = 4) (**b**); two-way Student's *t*-test (**c–e,l–p**); two-way repeated-measures ANOVA, Sidak's post-test (**f,q** (repeated-measures),**g,h**); two-way Mann–Whitney test (**u,w** (Bonferroni correction)). Slice thickness: 14 μm. Images represent two or more mice.

additional contributions. However, the origin of injury-associated fibroblasts may depend on the CNS injury site and type[11], similar to the varied ontogenies of injury-driven myofibroblasts in peripheral organs[28].

Independent of ontogeny, we show that the myofibroblast state has a critical and temporally restricted role in brain injury. Profibrotic myofibroblasts proliferate, migrate and reciprocally interact with fibrosis-associated macrophages and microglial states during

sub-acute injury time points. This convergent fibrotic state—resembling both peripheral SAMs[22,29,30] and DAMs[31]—suggests that the signals driving PT injury-induced fibrosis have broader roles across CNS pathology. Whereas fibrosis-associated myeloid programming transcends ontogeny and traditional macrophage polarization states[57], heterogeneous expression of pro-inflammatory 'M1-like' markers and reparative 'M2-like' markers (for example, *Arg1*) within myeloid subsets suggests opportunities for further macrophage state and positional parsing[58]. Disruption of myeloid cells that provide TGFβ1, perilesional brain cells expressing integrin $\alpha_v\beta_8$ that liberate TGFβ, or fibroblast TGFβ signalling each attenuated the sub-acute brain fibroblast response. In mice with inducible loss of the myofibroblast state, brain lesions were larger and associated with increased loss of neuronal tissue and delayed innate inflammation surpassing the post-injury peak. The mechanisms driving late perilesional inflammation, which also occurred after blockade of $\alpha_v\beta_8$-mediated TGFβ activation, remain to be elucidated; however, they are likely to involve feedback loops with secondary damage and further inflammatory cell recruitment. We propose that myofibroblasts prevent this cycle by rapidly forming a new emergency border between lesioned tissue and salvageable brain, limiting damage-induced neutrophil migration between ischaemic tissue with disrupted vasculature and susceptible parenchyma[23,59]. Loss of the myofibroblast state in the tMCAO model of severe stroke caused exacerbated vasogenic oedema with midline shift, oligodendrocyte loss and neuronal degeneration, probably involving interplay between fibroblasts, astrocytes, neurons and immune cells. In association with cardiovascular dysfunction and higher mortality, these data suggest that the myofibroblast state can be clinically beneficial in sub-acute time frames after CNS injury.

We also characterized the spatiotemporal evolution of the brain fibroblast response to injury. Sub-acute myofibroblasts transitioned to multiple late states, including lymphocyte-interactive fibroblasts that persisted long term. These transitions likely reflect evolving tissue requirements after disruption of brain anatomic and immunologic protection[60], with early prioritization of physical boundary reformation and later prioritization of meningeal border homeostasis and immune protection. Fibroblast CXCL12 loss resulted in increased brain lymphocyte pro-inflammatory cytokines and neutrophilia, suggesting a critical immunomodulatory role for late lesional fibroblasts. Moreover, disrupting this fibroblast–lymphocyte axis led to transcriptional changes in neurons associated with increased IFNγ signalling, highlighting the potential role of post-injury lymphocytes and their positioning in modulating neuronal activity. As brain injuries are associated with long-term sequelae including seizures, psychiatric disease and neurodegeneration, we speculate that brain fibroblasts modulate chronic neuroimmune positioning and tone that affect these disease susceptibilities. Fibroblasts therefore display dynamic functions as coordinators of brain wound repair, and represent intriguing therapeutic targets for CNS disease.

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

# Methods

## Mice

For lineage tracing of resting fibroblasts and tracking of injury-responsive fibroblasts, we crossed *Col1a2*[creERT2] mice (MGI 6721050, a gift from B. Zhou)[61] with *Rosa26*[tdT-Ai14] mice (R26-CAG-RFP, Jackson 007914) containing a *flox*-stop-*flox* sequence upstream of a CAG-RFP-WPRE cassette in the constitutively expressed *ROSA26* locus. Tamoxifen induces RFP expression in *Col1a2*-lineage[+] cells. *Col1a2*[creERT2] mice were also crossed to *Rosa26*[Sun1GFP] mice (Jackson 030952), enabling fibroblast nuclear identification. For dMCAO experiments, a distinct *Col1a2*[creERT] allele was used (Jackson 029567). Additional stromal reporters used include *Col1a1*[GFP] mice (a gift from D. Brenner) to mark active *Col1a1*-expressing cells[62], *Pdgfra*[GFP] (PDGFRa-H2B-eGFP nuclear-localized GFP, Jackson 007669) to track fibroblasts and oligodendrocyte lineage cells[63,64], *Rosa26*[Tdt-Ai14] mice crossed to *Gli1*[creERT2] mice (Jackson 007913) to track adventitial fibroblasts[16], *Twist2*[cre] mice (Jackson 008712) to track fibroblasts and mural cells[65], *Acta2*[creERT2] mice to track myofibroblasts and smooth muscle cells[66], *Ng2*[creER] mice (Jackson 008538) to mark pericytes and oligodendrocyte precursor cells[67], *Atp13a5*[creERT2] mice to mark pericytes[68], and *Cthrc1*[creER] mice (a gift from D. Sheppard) to track myofibroblasts[28].

To track immune cell subsets, we crossed *Rosa26*[tdT-Ai14] mice to *CD4*[cre] mice (Jackson 022071) to track CD4[+] and CD8[+] T cells[69], *Ccr2*[creERT2] mice (a gift from B. Becher) to track monocyte-derived cells, *P2ry12*[creERT2] mice (Jackson 034727) to track microglia-derived cells, *Cx3cr1*[creER] (Jackson 020940) to track macrophages and *Pf4*[cre] (Jackson 008535) to track BAMs. We also used TBET (*Tbx21*)-zsGreen transgenic mice (provided by J. Zhu)[70] to track type 1 lymphocytes. *Cx3cr1*[creER] mice were additionally crossed with *Tgfb1*[GFP] mice (MGI 3719583)[71] and *Tgfb1*[flox] mice (Jackson 033001) to drive deletion of *Tgfb1* in macrophages (in *Cx3cr1*[creER]; *Tgfb1*[GFP/flox] mice). Controls were tamoxifen-induced, littermate *Cx3cr1*[creER]; *Tgfb1*[flox/+] mice.

To conditionally delete fibroblast *Tgfbr2*, we crossed *Col1a2*[creERT2] mice to *Tgfbr2*[flox] mice (both *Tgfbr2-exon2*[flox], MGI 2384513[72] and *Tgfbr2-exon4*[flox], Jackson 012603). To conditionally delete fibroblast *Cxcl12*, we crossed *Col1a2*[creERT2] mice to *Cxcl12*[flox] mice (Jackson 021773). We used *Itgb8*[tdT] mice (Itgb8-IRES-tdT, provided by H. Paidassi)[73] to visualize *Itgb8* expression. We used *Itgb8*[flox] mice (MGI 3608910)[74] crossed to *Emx1*[cre] mice (Jackson 005628) or hGfap[cre] mice (Jackson 004600), to delete *Itgb8* from neurons and glial cells[42,75,76]; some hGfap[cre] mice were crossed to iSure[cre] (MGI 6361135, [77]) to optimize Cre efficiency. For the above conditional-knockout strains, controls were tamoxifen-induced (when relevant), littermate Cre-negative or *flox*-heterozygous mice, co-housed non-littermate lineage tracer mice (for select experiments involving *Col1a2*-lineage[+] fibroblast quantification), or age-matched, uninduced non-littermate controls (a portion of immunophenotyped resting mice and select tMCAO experiments). Additionally, to enable depletion of myofibroblasts, we crossed Cthrc1[creER] mice to *Rosa26*[DTA] mice; controls were littermate cre-positive vehicle-treated (saline or corn oil) or cre-negative tamoxifen-induced mice.

Mice of both sexes were backcrossed on C57BL/6 for at least ten generations, or on a mixed genetic background (*Gli1*[creERT2], *Cx3cr1*[creER], *Emx1*[cre]). If not otherwise stated, all experiments were performed with 7–21 week old male and female mice. All mice were bred and maintained in specific-pathogen-free conditions, at 25 °C and ambient humidity under a 12 h:12 h day:night cycle, at the animal facilities of University of California, San Franciso (UCSF) or University of California, San Diego (UCSD). Sample sizes were estimated based on standard power calculations ($a = 0.05$, 80% power) performed for similar published experiments. Mice were used in accordance with institutional guidelines and under study protocols approved by the UCSF or UCSD Institutional Animal Care and Use Committee (protocols AN193180-01J, AN195716-01B (UCSF) and s14044 (UCSD)).

## Marmosets

Eleven outbred middle-aged marmoset monkeys (*Callithrix jacchus*; aged >5 years; median age ~7 years) were used in this study. No siblings were used. Animals were housed in family groups (12 h:12 h light:dark cycle, temperature 31 °C, humidity 65%). Experiments were conducted according to the Australian Code of Practice for the Care and Use of Animals for Scientific Purposes and were approved by the Monash University Animal Ethics Committee. Marmosets were obtained from the National Nonhuman Primate Breeding and Research Facility (Monash University, Australia).

## Tamoxifen-induced Cre recombination

Mice were injected intraperitoneally with 200 µl of tamoxifen (Sigma-Aldrich) dissolved in corn oil at 10 mg ml[−1]. For transcranial activation, 4-OH-tamoxifen (Sigma-Aldrich), the active metabolite of tamoxifen[78], was dissolved in acetone at 100 µg ml[−1]. One-hundred microlitres of 4-OHT was applied to a specific cranial location, determined stereotactically as described in 'Photothrombotic injury'. Tamoxifen was applied via micropipette (approximately 5 µl at a time) and allowed to evaporate in between applications.

## Photothrombotic injury

For PT injury surgeries[79–81], mice were anaesthetized via inhaled isoflurane, shaved on the scalp, and stereotaxically fixed. After sterilization with iodine and 70% ethanol, 0.5% lidocaine was administered subcutaneously to the scalp. The cranium was surgically exposed, and a fibre optic white light is placed over the S1 cortex (3.0 mm, −0.5 mm ($x$, $y$) from bregma, with coordinates determined via stereotax). Mice were injected intraperitoneally with 8 mg kg[−1] Rose Bengal dye; after 1 min, the cranium was exposed to high intensity white light (150 W) for 2 min. The scalp was sutured using nylon sutures and surgical glue, and buprenorphine was administered.

## TBI

For TBI surgeries[82], mice were anaesthetized and stereotaxically fixed, as above. A 3-mm craniotomy was performed over the right S1 centred at −1 mm posterior from bregma, +3 mm lateral from the midline. TBI was performed with a CCI device (Impact One Stereotaxic Impactor for CCI, Leica Microsystems) equipped with a metal piston using the following parameters: 3 mm tip diameter, 15° angle, 0.8 mm depth from the dura, 3 m s[−1] velocity, and 100 ms dwell time. Sutures were administered as above.

## tMCAO

Mice (age postnatal day (P)25–45) underwent focal ischaemia–reperfusion with tMCAO for 3 h, or sham surgery, as detailed previously[33,83,84]. In brief, the right internal carotid artery (ICA) was dissected, and a temporary ligature was tied using a strand of 6-0 suture at its origin. This ligature was retracted laterally and posteriorly to prevent retrograde blood flow. A second suture strand was looped around the ICA above the pterygopalatine artery and an arteriotomy was made proximal to the isolated ICA. A silicone coated 6-0 nylon filament from Doccol Corporation was inserted 6.5–7 mm to occlude the MCA and the second suture strand was tied off to secure the filament for the duration of occlusion. Injury was confirmed by severe left frontal/hindlimb paresis resulting in circling movements during the occlusion period. For reperfusion, each animal was anaesthetized and all suture ties and the occluding filament were removed. Avitene Microfibrillar Collagen Hemostat was placed over the arteriotomy and the skin incision was closed. According to pre-established criteria, cohorts displaying excessive bleeding at time of reperfusion were excluded. Cohorts with significantly underweight mice (75% or less of expected weight at time of surgery) were excluded from survival analysis. Mice that died during surgery were also excluded. Sham animals were anaesthetized for 15 min, equivalent to

surgery procedure time; at the time of reperfusion, the sham animals were once again anaesthetized for 5 min, equivalent to the reperfusion procedure time for tMCAO animals.

## dMCAO

dMCAO surgeries were performed as previously described[85]. In brief, mice were sedated with isoflurane and analgesic administered before surgery. An incision was made between the eye and ear, the temporal muscle retracted, and the skull removed above the distal middle cerebral artery (dMCA). The dMCA was then occluded with a bipolar coagulator which clots the artery, reducing blood flow to the ipsilateral cortex. Following this procedure, the incision was closed and mice monitored daily.

## Marmoset stroke

Induction of focal stroke to the marmoset primary visual cortex (V1) was performed by vasoconstrictor-mediated vascular occlusion of the calcarine branch of the posterior cerebral artery (PCAca), as detailed previously[86,87]. In brief, following anaesthesia (Alfaxalone 5 mg kg$^{-1}$; maintained with inspired isoflurane 0.5–4%), a craniotomy and dural thinning was performed, followed by intracortical injections of endothelin-1 (ET-1; 0.1 μl per 30 s pulse at 30 s intervals, totalling ~0.7 μl over 7 sites) proximal to the PCAca, which supplies operculum V1. The craniotomy was replaced, secured with tissue adhesive (Vetbond; 3 M) and the skin sutured closed. Monkeys recovered for 7 days ($n = 2$), 6 weeks ($n = 2$) and 1 year ($n = 2$; equal numbers of each sex).

## Vitals monitoring

The MouseOx Pulse Oximeter system (STARR Life Sciences) was used to measure arterial oxygen saturation from awake mice. Mice were shaved at time of injury or prior to measurement. Mice were measured at 5 measurements per second for at least 5 min and at least 10 successful readings. All successful measurements (error code = 0) were averaged for each mouse.

For blood pressure and heart rate measurements, mice were restrained and placed on a warming platform. Measurements were taken using the CODA-HT4 Noninvasive Blood Pressure System (Kent Scientific), using default settings for sets, cycles, deflation time and failed cycle exclusion. Accepted cycles were averaged per mouse. Mean arterial pressure (MPA) was calculated as MAP = (systolic pressure + 2 × (diastolic pressure))/3.

Vitals were recorded on day 3 unless mice appeared to be decompensating, in which case measurements were taken daily, beginning days 1–2, to maximize data collection; the latest available (pre-death) measurement was taken for each mouse.

## Liposome injection

Clodronate liposomes or empty control liposomes (Encapsula Nanoscience) were randomly assigned and administered intraperitoneally, beginning with injection 3 days prior to injury (200 μl) followed by injection on the day of injury (100 μl) and every 3 days subsequently (100 μl) until collection (14 dpi).

## Antibody injection

ADWA11[88,89] (generously provided by D. Sheppard) or IgG1 isotype control (InVivoMab) was randomly assigned, diluted in sterile DPBS to 10 mg kg$^{-1}$, and injected intraperitoneally on the day of injury (0 dpi) and every week subsequently until collection (7 dpi, etc.).

## EdU injection

To measure proliferation, 5-ethynyl-2′-deoxyuridine (EdU, Thermo Scientific) was reconstituted at 5 mg ml$^{-1}$ in DPBS and injected at 50 mg kg$^{-1}$ intraperitoneally. Mice were injected either every other day throughout the injury time course or 2 h prior to euthanasia.

## Mouse tissue processing for imaging

Following $CO_2$ euthanasia, mice were transcardially perfused with 10 ml DPBS and 10 ml of 4% paraformaldehyde (PFA) (Thermo Scientific). Skullcaps and/or brains were removed from bases. Skullcaps (for meningeal imaging) or skullcaps and brains were fixed overnight (4% PFA, 4 °C). Skullcaps/brains were washed (DPBS) and decalcified in 0.3 M EDTA (VWR) (1 week, 4 °C) followed by cryoprotection (30% sucrose). When removed from skullcaps, brains were cryoprotected directly after fixation. Brains were frozen in O.C.T. (Thermo Scientific) on dry ice and sliced to indicated thickness on a cryostat (Leica). Spinal cords processed similarly (decalcifying intact vertebra). For spatial transcriptomics, tissue processing was performed as above, without PFA perfusion and fixation. Brains were removed from skullcaps and directly frozen. For quantitative imaging of (blinded) PT lesions, 2× 14-μm sections were collected per 100 μm sliced, and sections representing a lesion's maximal cross-sectional area were stained, imaged and quantified. Lesion size outliers (diameter <25% of normal, representing technical errors during injury) were excluded prior to unblinding, resulting in one exclusion from Fig. 4g,h, and one exclusion from Fig. 4p and Extended Data Fig. 8ab. Mice with hydrocephalus were also excluded.

## Marmoset tissue processing for imaging

For immunofluorescence, naive controls ($n = 2$; equal numbers of each sex) and post-stroke marmosets were administered an overdose of pentobarbitone sodium (100 mg kg$^{-1}$; intraperitoneal injection). Following apnoea, animals were transcardially perfused with 0.1 M heparinized PBS, followed by 4% PFA in PBS (0.1 M). Brains were dissected, post-fixed and cryoprotected, as outlined previously[26,86,87]. Following separation of the hemispheres, each hemisphere was bisected coronally at the start of the caudal pole of the diencephalon and frozen in liquid nitrogen at −40 °C. Tissue was cryosectioned in the parasagittal plane at −20 °C to obtain 40 μm for free-floating sections stored in cyroprotectant solution (50% PBS 0.1 M, 20% ethylene glycol, 30% glycerol).

## Immunohistochemistry

Slide-mounted thin sections were thawed, washed (DPBS) and blocked (1 h, DPBS/0.4% Triton X-100/5% secondary host serum). For select antibodies, antigen retrieval was performed prior to blocking, involving: (1) incubation for up to 15 min in Liberate Antibody Binding Solution (Polysciences) followed by a 5 min wash in PBS (ASPA); or (2) incubation for 3 (ALDH1A2 (thin section only), ALPL, αSMA (thin section only), FGF13, LAMA1, SEMA3C, CDH18) or 5 min (cleaved caspase-3 (cCasp3)) in 0.01 M $Na_3C_6H_5O_7$ (heated to 95 °C in a water bath), followed by cooling to room temperature (20 min) and 3 washes in PBS (5 min each). Samples were then incubated in primary antibodies diluted in blocking solution (room temperature, 1 h, or 4 °C, overnight). Samples were washed (DPBS/0.05% Triton X-100, 5 min, 3 times) and incubated in secondary antibodies diluted 1:1,000 in blocking solution (room temperature, 45–60 min). Samples were washed, mounted in DAPI Fluoromount-G (Thermo Scientific), and imaged. Proliferation was measured using the Click-iT EdU Alexa Fluor 647 Imaging Kit (Thermo Scientific), according to the manufacturer's instructions (in between primary and secondary staining steps). FluoroJade C was stained using the FluoroJade C Ready-to-Dilute Staining Kit (VWR), according to the manufacturer's instructions.

Medium-thickness sections (30–50 μm) were blocked in 250 μl DPBS/0.25% Triton X-100/5% secondary host serum (1 h, room temperature). Samples were subsequently incubated in primary antibody diluted in blocking solution (4 °C, overnight), washed (DPBS/0.05% Triton X-100, 5 min, 4 times), and incubated in secondary antibodies diluted 1:500 in blocking solution (room temperature, 45–60 min). Samples were washed three times, mounted in DAPI Fluoromount-G, and imaged.

Meninges were blocked and stained before removal from skullcaps (block: 4 °C, overnight, in 2 ml DPBS/0.3% Triton X-100/5% FBS/0.5% BSA/0.05% NaN₃). Samples were incubated in primary antibody diluted in 600 µl of staining solution (DPBS/0.15% Triton X-100/7.5% FBS/0.75% BSA/0.075% NaN₃; 4 °C, 72 h), washed (DPBS/0.15% Triton X-100, 4 °C, 30 min, 3 times), and incubated in secondary antibodies diluted 1:400 in staining solution solution (4 °C, 24 h). Samples were washed and incubated in 10 µg ml$^{-1}$ DAPI in PBS (room temperature, 1 h). Dural meninges were subsequently micro-dissected from skullcap, mounted in 50–100 µl of Refractive Index Matching Solution (RIMS, DPBS/Histodenz (133.33 g per 100 ml)/0.017% Tween-20/0.17% NaN₃), and imaged.

Thick sections (100–200 µm) were stained using the iDISCO protocol[90] with 1–2 days of permeabilization, 1–2 days of blocking, 3 days of primary antibody and 3 days of secondary antibody.

For marmoset imaging, free-floating sections comprising the infarct and peri-infarct regions were selected and washed in PBS (0.1 M) and pre-blocked in a solution of 10% normal goat serum in PBS + 0.3% Triton X-100 (TX; Sigma) before incubation with primary antibodies overnight at 4 °C. Sections were rinsed in 0.1% PBS-Tween and incubated with secondary antibodies (1 h). After washes in PBS, sections were treated with DAPI, mounted in Fluoromount-G, and imaged.

For haematoxylin and eosin (H&E) imaging and liver cCasp3 immunohistochemistry, histology was performed by HistoWiz (https://histowiz.com) using a standard operating procedure and fully automated workflow. Samples were processed, embedded in OCT, and sectioned at 4 µm for H&E staining or cCasp3 immunohistochemistry. Immunohistochemistry was performed on a Bond Rx autostainer (Leica Biosystems) with enzyme treatment (1:1,000) using standard protocols. Bond Polymer Refine Detection (Leica Biosystems) was used according to the manufacturer's protocol. After staining, sections were dehydrated and film coverslipped using a TissueTek-Prisma and Coverslipper (Sakura). Whole slide scanning (40×) was performed on an Aperio AT2 (Leica Biosystems).

## Imaging antibodies

Primary antibodies used for mouse imaging include chicken anti-GFP (Aves Labs GFP-1020, 1:200), rabbit anti-dsRed (Takara 632496, 1:300), chicken anti-GFAP (Invitrogen PA1-10004, 1:200 or 1:500), rat anti-GFAP (2.2B10, Invitrogen 13-0300, 1:200), rat anti-ER-TR7 (Novus Biologicals NB100-64932, 1:200), rabbit anti-αSMA (Abcam ab5694, 1:300), rat anti-CD31 (MEC13.3, Biolegend 102514, 1:200), goat anti-Desmin (GenWay Biotech GWB-EV0472, 1:200), rat anti-PDGFRβ (APB5, Invitrogen 14-1402-82, 1:500), rabbit anti-NG2 (Millipore Sigma ab5320, 1:500), goat anti-Decorin (Novus Biologicals AF1060, 1:200), goat anti-collagen 1 (Southern Biotech 1310-01, 1:200 or 1:500), rabbit anti-collagen 6α1 (Novus Biologicals NB120-6588, 1:200), rat anti-periostin (345613, Novus Biologicals MAB3548, 1:200), rat anti-ICAM1 (YN1/1.7.4, Biolegend 116110, 1:200), Syrian hamster-anti-CD3ε (500A2, BD Biosciences 553238, 1:200), goat anti-S100A8 (R&D Systems AF3059, 1:200), chicken anti-NeuN (Millipore Sigma ABN91, 1:200), rabbit anti-IBA1 (Aif3, Fujifilm Wako 019-19741, 1:200–1:1,000), mouse anti-FGF13 (N235/22, Invitrogen MA5-27705, 1:100), goat anti-CD80 (R&D Systems AF740, 1:200), goat anti-ALPL (Novus Biologicals AF2910, 1:50), rabbit anti-LAMA1 (EPR27258-37, Abcam ab307542, 1:200), rabbit anti-SEMA3C (Invitrogen PA5-103168, 1:100), rabbit anti-CDH18 (Invitrogen PA5-112902, 1:50), rabbit anti-ALDH1A2 (Novus Biologicals NBP2-92915, 1:200), goat anti-SOX10 (R&D Systems AF2864, 1:300), rabbit anti-ASPA (Genetex GTX113389, 1:1,000), mouse anti-E-cadherin (Clone 36, BD Biosciences 610181), rat anti-I-A/I-E (MHCII, M5/114.15.2, eBioscience 14-5321-82), mouse anti-Ly76 (TER119, Biolegend 116232, 1:200), goat anti-mouse IgM (Invitrogen 31172, 1:200), and rabbit anti-cCasp3 (Cell Signaling Technology 9661T, 1:400; Cell Signaling Technology 9991, Histowiz). For marmoset imaging, rabbit anti-COL6 (Abcam ab6588, 1:500) was used.

Secondary antibodies were used at 1:500 (for thin sections) and 1:1,000 (for thicker sections), as specified in relevant Methods sections. Secondary antibodies used include donkey anti-rat IgG AF488 (Thermo Scientific A21208), donkey anti-rat IgG AF555 (Thermo Scientific A78945), donkey anti-rat IgG AF647 (Abcam ab150155), donkey anti-rabbit IgG AF488 (Thermo Scientific A21206), donkey anti-rabbit IgG AF555 (Thermo Scientific A31572), donkey anti-rabbit IgG AF647 (Thermo Scientific A31573), donkey anti-goat IgG AF488 (Thermo Scientific A11055), donkey anti-goat IgG AF555 (Thermo Scientific A21432), donkey anti-goat IgG AF647 (Thermo Scientific A21447), donkey anti-chicken IgG AF488 (Sigma, SAB4600031-250UL), donkey anti-chicken IgG AF647 (Thermo Scientific A78952), goat anti-rat IgG AF488 (Thermo Scientific A11006), goat anti-rabbit IgG AF555 (Thermo Scientific A21429), goat anti-rabbit IgG AF647 (Thermo Scientific A21245), goat anti-hamster IgG AF647 (Thermo Scientific A21451) and donkey anti-mouse IgG AF647 (Thermo Scientific A31571).

## Confocal and wide-field microscopy

Confocal images (for thick sections, immune cell quantification and some thin sections) were imaged using a Nikon A1R laser scanning confocal including 405, 488, 561 and 650 laser lines for excitation and imaging with 16×/0.8 NA Plan Apo long working distance water immersion, 20×/0.95 NA XLUM PlanFl long working distance water immersion, or 60×/1.2 NA Plan Apo VC water immersion objectives. z-steps were acquired every 4 µm. Wide-field images (for thin sections, fibrosis quantification and lesion size quantification) were imaged using a Zeiss Axio Imager.M2 wide-field fluorescent microscope with a 10×/0.3 or 20×/0.8 air objective, or on a Leica Aperio Versa 8 Slide Scanner with a HC PL APO 20X/0.75 CS2 air objective. Marmoset images were acquired using the VS200 slide scanner (Olympus).

## Image analysis and quantification

z-stacks were rendered in 3D and quantitatively analysed using Bitplane Imaris v9.8 software package (Andor Technology). Individual cells (for example, lymphocytes, fibroblasts, etc.) or fibrotic and glial scars were annotated using the Imaris surface function, thresholding on fluorescent signal (based on antibody staining or reporters when available), along with additional co-stains (for example, CD45, IBA1 and CD3ε), DAPI staining and size or morphological characteristics; when helpful, background signal in unrelated channels was excluded. Colocalization was determined using 'intensity mean', 'intensity min' (for DAPI) or 'intensity max' (for select secreted factors). Parameter values were chosen for optimal signal-to-background balance. 3D distances between lymphocytes and stromal/glial cells were calculated using the Imaris Distance Transform Matlab extension. Proportions of immune or stromal cells within given cortical or lesional regions were determined by manually tracing regional borders (for example, GFAP–ER-TR7) and filtering cell surfaces based on inclusion. Additional surface statistics were calculated using Imaris. Lesion sizes were calculated in Fiji (ImageJ version 1) by tracing the fibroblast–astrocyte border (using ER-TR7 and/or GFAP). Fibroblast or myeloid cell coverage was determined by thresholding the relevant channel (for example, ER-TR7, IBA1 and tdT), using a consistent threshold for each slice except for cases of exceptionally high tissue background. Fibroblast or pericyte coverage after dMCAO (Extended Data Fig. 1) was determined using particle analysis, adjusting thresholds based on image background. In both cases, coverage was calculated as thresholded area normalized to lesion area, and data were subsequently unblinded. For pooled data, equivalent thresholds were applied across experiments when possible, with exceptions made for varying tissue background. Lineage-traced pericytes were manually counted. Border thickness was calculated in Fiji, using a macro to calculate the distance between each (manually traced) border edge at points 100 µm apart, followed by averaging these distances. Midline shift was calculated from coronal images by tracing: (1) a line connecting dorsal and ventral brain midpoints; (2) a curve tracking observed

midline structures; and (3) a perpendicular line between lines 1 and 2. For analysis of cCasp3 puncta in liver immunohistochemistry images (Histowiz), five representative fields of view (10×) were captured per slide; for each, immunohistochemistry signal was isolated in ImageJ via colour deconvolution, cCasp3 was thresholded, and puncta were counted using the ImageJ particle analysis function.

## Serum chemistries

Serum analysis was performed by the Unit for Laboratory Animal Medicine Pathology Core (University of Michigan). Whole blood was collected into serum separator tubes, allowed to clot, and separated into serum by centrifugation. Serum chemistries were run on an AU480 Chemistry Analyzer (Beckman Coulter) using the manufacturer's provided reagents. For relevant analytes (for example, alanine transaminase), severely haemolysed samples were excluded according to Unit for Laboratory Animal Medicine and manufacturer guidelines.

## Tissue processing for Visium

Tissue was collected from 1× resting mouse, 1× 2 dpi mice, 2× 7 dpi mice and 4× 21 dpi mice (including 2 $Cthrc1^{creER}$; $Rosa26^{DTA}$ mice, discarded after initial clustering due to undetectable deletion; PT injury). Tissue was collected as above. Ten-micrometre slices were prepared via cryostat and directly mounted onto Tissue Optimization and Spatial Gene Expression slides (10X Genomics). Tissue optimization was carried out as per manufacturer's recommendations, resulting in an optimal tissue permeabilization time of 12 min. Tissue mounted on the Spatial Gene Expression slide was processed per the manufacturer's recommendations, imaged on a Leica Aperio Versa slide scanner at 20×, and transferred to the Gladstone Genomics Core for library preparation according to the manufacturer's protocol. Samples were subsequently transferred to the UCSF Center for Advanced Technology for sequencing on the NovaSeq 6000 system.

## Tissue processing for mouse nuclear isolation

Nuclear isolation was employed to overcome limitations of traditional flow cytometry and/or scRNA-seq, including ECM interference with fibroblast isolation and dissociation signatures that disproportionately affect stromal and immune cells[91]. For nuclear flow cytometry, tissue was collected from resting, 7 or 14 dpi (PT injury) $Pdgfra$-GFP mice. For snRNA-seq experiment 1 (wild-type time course), tissue was collected from 2 wild-type mice per time point (1 male and 1 female) at rest, 2 dpi, 7 dpi and 21 dpi. Two mice per time point with impaired TGFβ signalling ($Col1a2^{creER}$; $Tgfbr2^{flox}$ and $Cdh5^{creER}$; $Tgfbr2^{flox}$) were collected at 7 and 21 dpi but were discarded after initial clustering and not analysed separately due to insufficient yield. For snRNA-seq experiment 2 (wild-type, $Tgfbr2$-cKO and ADWA11-treated mice), tissue was collected from 2 wild-type mice, 2 $Col1a2^{creER}$; $Tgfbr2^{flox}$ mice, and 2 ADWA11-treated mice at 7 and 21 dpi (1 male and 1 female per time point or condition). For snRNA-seq experiment 3 (wild-type and $Cxcl12$-cKO), tissue was collected from 2 wild-type and 2 $Col1a2^{creER}$; $Cxcl12^{flox}$ mice at 21 dpi.

Following $CO_2$ euthanasia, mice were transcardially perfused (10 ml DPBS) and decapitated. Brains were removed from skullcaps and placed in iMED+ (15 mM HEPES (Fisher) and 0.6% glucose in HBSS with phenol red)[92]. For nuclear flow cytometry and snRNA-seq experiments 1 (time course) and 3 ($Cxcl12$-cKO), dura, lesion and perilesional cortex (with or without contralateral cortex) were micro-dissected and processed separately. For snRNA-seq experiment 2 (wild-type, $Tgfbr2$-cKO and ADWA11), only lesions were dissected. Microdissection involved meningeal/skullcap separation, removal of subcortical structures, and separation of lesions from skullcaps (lesions often separate from cortex during initial dissection but can be micro-dissected as necessary). For snRNA-seq, tissue from male and female mice within experimental conditions was combined.

Tissue was processed using ST-based buffer protocol[91], with the following modifications: initial centrifugation was performed at 500$g$ for 10 min. After lysis and initial centrifugation, nuclei were resuspended in 1 ml ST buffer (nuclear flow cytometry, RNA-sequencing experiment 2) or PBS/1% BSA/0.2 U μl$^{-1}$ Protector RNase inhibitor (Roche) (RNA-sequencing experiments 1 and 3), filtered through 35-μm cell strainers, and subsequently processed as below.

For nuclear flow cytometry and snRNA-seq experiment 2, nuclei were centrifuged for 5 min at 500 g, resuspended in FANS buffer (DPBS/1% BSA/0.1 mM EDTA)[93] with 2 μg μl$^{-1}$ DAPI and 0.2 U μl$^{-1}$ RNase inhibitor (snRNA-seq), and stained (nuclear flow) or sorted (snRNA-seq).

For snRNA-seq experiments 1 and 3, cell counts were performed after initial centrifugation (NucleoCounter, Chemometic), and a maximum of 2 × 10$^6$ nuclei were multiplexed using CellPlex Multiplexing technology (10X Genomics) according to the manufacturer's instructions (using protocol 1 for nuclear multiplexing, with only one wash after multiplexing to increase yield). Nuclei were resuspended in FANS buffer with 0.2 U μl$^{-1}$ RNase inhibitor and 2 μg μl$^{-1}$ DAPI and nuclear concentrations were determined. Immediately before sorting, multiplexed microanatomical regions (including lesion, parenchyma and dural meninges) from individual mice were combined at desired ratios (75% lesion, 25% parenchyma). Nuclei were sorted (forward and side scatter and DAPI) into pre-coated tubes containing 200 μl PBS/1% BSA/0.2 U μl$^{-1}$ RNase inhibitor (BDFACSAria II sorting system, 100 μm nozzle size, 4 way-purity sort mode). Resting dural fibroblasts were enriched using $Col1a2^{creER}$; $Rosa26^{Sun1GFP}$ and resting nuclei were combined after sorting (50% GFP$^+$ meninges, 25% GFP$^-$ meninges, 25% parenchyma).

After sorting and pooling (snRNA-seq), samples were centrifuged (500$g$, 15 min), and supernatant was removed to leave a minimum final volume of 45 μl with a maximum of 16,500 nuclei. Final nuclear concentrations were acquired, and samples were transferred to the UCSF Genomics CoLab for library preparation. Up to 1.6 × 10$^4$ nuclei were loaded onto the Chromium Controller (10X Genomics). Chromium Single Cell 3′ v3.1 reagents were used for library preparation according to the manufacturer's protocol. Libraries were transferred to the UCSF Institute for Human Genetics for sequencing on the NovaSeq 6000 system.

## Marmoset tissue processing for nuclear isolation

For single-nuclei RNA sequencing (snRNA-seq), naive control marmosets ($n$ = 3; 1 female, 2 male; median age 4 years) were administered an overdose of pentobarbitone sodium (100 mg kg$^{-1}$; intraperitoneal). Following apnoea, frontal lobes were recovered and dissected under aseptic conditions in sterile ice-cold phosphate buffered saline (PBS; 0.1 M; pH 7.2). Tissues were and snap frozen in isopentane chilled in liquid nitrogen. The procedures and dissections were performed in chilled RNAase-free PBS with RNase-free sterilized instruments under RNase-free conditions. Approximate time from apnoea to snap-freezing ranged from 20–30 min. All six samples passed quality control. Nuclear isolation was performed as described previously[26], involving pulverization in liquid nitrogen, lysis in lysis buffer, dounce homogenization, and gradient purification. After isolation, nuclei were counted and diluted to 1 million per ml with sample-run buffer (0.1% BSA, RNase inhibitor (80 U ml$^{-1}$), 1 mM DTT in DPBS). snRNA-seq was performed on the 10x Genomics Chromium System. Cellranger commercial software was utilized to conduct initial data processes including sequence alignment to the marmoset genome (CalJac3).

## Tissue processing for mouse single-cell isolation

Single-cell suspensions were prepared from tissues including brain, spinal cord, meninges, blood and spleen. Immediately following $CO_2$ euthanasia, spleens were removed into RPMI/10% FBS and peripheral blood was collected through the right ventricle into heparin tubes. Mice were subsequently transcardially perfused through the left ventricle with 10 ml DPBS, decapitated, and brains were carefully removed from skullcaps and placed in iMED+ as above[92]. For select experiments, spinal cords were carefully dissected from vertebra. Cortex, lesion and

meninges were dissected as above. Brain was weighed and subsequently homogenized in iMED+ homogenized using a 2-ml glass tissue grinder (VWR; 6 plunges, followed by filtration through a 70-μm filter, addition of 2 ml iMED+ and 6 more plunges). Filtered suspensions were centrifuged at 220$g$ for 10 min and resuspended in 5 ml of 22% Percoll (GE Healthcare) in Myelin Gradient Buffer (5.6 mM $NaH_2PO_4 \cdot H_2O$, 20 mM $Na_2HPO_4 \cdot 2H_2O$, 140 mM NaCl, 5.4 mM KCl, 11 mM glucose in $H_2O$)[92]. PBS (1 ml) was layered on top of Percoll. Samples were centrifuged at 950$g$ for 20 min at 4 °C with no break to separate myelin and resuspended in fluorescence activated cell sorting (FACS) buffer. Dissected meninges were incubated in digestion medium (RPMI/10% FBS/80 μg ml$^{-1}$ DNase I/40 μg ml$^{-1}$ Liberase TM (Roche)). Tissue was subsequently mashed through 70-μm filters, followed by centrifugation and resuspension in FACS buffer. Spleens were prepared by mashing tissue through 70-μm filters without tissue digestion, followed by centrifugation. Red blood cells were lysed for 2 min using 1× Pharm-Lyse and the remaining cell pellet were resuspended in FACS buffer. Blood samples were centrifuged for 5 min at 500$g$. Pellets were resuspended in 1× Pharm-Lyse 5 min at room temperature, followed by centrifugation and suspension in FACS buffer. For cytokine restimulation assays, samples were transferred to U-bottom plates and incubated in stimulation medium (RPMI supplemented with 10% FBS, 1% penicillin/streptomycin, 1× Glutamax (Thermo Scientific), 1× HEPES buffer (Fisher), 1× non-essential Amino Acids (Thermo Scientific), 1 mM $NaC_3H_3O_3$ (Thermo Scientific), 55 μM b-mercaptoethanol, 1× Cell Stimulation Cocktail (Tonbo), and 1× Brefeldin A (Thermo Scientific)) at 37 °C for 3 h, followed by centrifugation and transfer to a V-bottom plate.

## Flow cytometry

Resuspended samples were stained in 96-well V-bottom plates. Surface staining was performed at 4 °C for 45 min in 50 μl staining volume. For experiments involving intracellular staining (including transcription factor and cytokine staining), cells were fixed and permeabilized using Foxp3 Transcription Factor Staining Buffer Set (eBioscience) followed by staining at 4 °C for 1 h in 50 μl staining volume. All samples were acquired on a BD LSRII Fortessa Dual or a BD FACSAria II for cell sorting. Live cells or nuclei were gated based on their forward and side scatter followed by Zombie NIR fixable (Biolegend 423106), Fixable Viability Dye eF780 (eBioScience 65086514), Draq7 (Biolegend 424001) or DAPI (40,6-diamidine-20-phenylindole dihydrochloride; Millipore Sigma D9542-10MG) exclusion (or inclusion for nuclei). Lineages were subsequently identified as follows:

Oligodendrocyte-lineage nuclei were identified as DAPI$^+$ *Pdgfra*-GFP$^{hi}$Olig2$^+$. Fibroblast nuclei were identified as DAPI$^+$*Pdgfra*-GFP$^{int}$ or DAPI$^+$ *Col1a2$^{creER}$; Rosa26$^{Sun1GFP+}$* (snRNA-seq experiment 1, resting dural meninges). Bulk nuclei were identified as DAPI$^+$. Global lymphocytes were defined as CD45$^+$Thy1$^+$. T cells were identified as CD45$^+$CD11b$^-$CD19$^-$NK1.1$^-$CD3ε$^+$CD4$^+$ (CD4 T cells; further subset as FOXP3$^+$ (regulatory T cells) or FOXP3$^-$ (conventional CD4 T cells)) CD8α$^+$ (CD8 T cells) or TCRγδ$^+$ (γδ T cells) and were further defined as CD44$^+$CD69$^+$ (resident memory T cells (T$_{RM}$)), CD62L$^+$ (naive T cells), CD62L$^-$CD44$^+$ (activated T cells) or CTV$^{diluted}$ (proliferating T cells). Additionally, CD4 T cells were defined as TBET$^+$ (T$_H$1 cells), GATA3$^+$ (T$_H$2 T cells) or RORγt$^+$ (T$_H$17 T cells), and various T cell subsets were defined as cytokine-positive or negative (IFNγ, IL-17A or IL-10). Neutrophils were defined as CD45$^+$CD11b$^+$Ly6G$^+$ (and optionally Thy1$^-$CD19$^-$NK1.1$^-$). Monocytes were defined as CD45$^+$CD11b$^+$Ly6G$^-$Ly6C$^+$ (and optionally Thy1$^-$CD19$^-$NK1.1$^-$Siglec F$^-$). Microglia were defined as CD45$^{int}$CD11b$^+$. Macrophages were defined as CD45$^+$Ly6G$^-$Ly6G$^-$CD64$^+$ (optionally MERTK$^+$). Microglia/macrophages were further defined as DAM/SAM (CD9$^+$ and CD63$^+$/TREM2$^+$). cDCs were identified as CD45$^+$Ly6G$^-$Ly6 C$^-$CD64$^-$MHCII$^+$CD11c$^+$, and were further defined as cDC1s (CD11b$^{lo}$, optionally SIRPα$^-$) or cDC2s (CD11b$^{hi}$, optionally SIRPα$^+$). B cells were defined as CD45$^+$Thy1$^-$CD19$^+$. Eosinophils were defined as CD45$^+$Thy1$^-$CD19$^-$NK1.1$^-$Ly6G$^-$CD11b$^+$Siglec F$^+$. Populations were backgated to verify purity and gating.

Data were analysed using FlowJo software (TreeStar) and compiled using Prism (Graphpad Software). Cell counts were performed using flow cytometry counting beads (CountBright Absolute; Life Technologies) per manufacturer's instructions.

## Flow cytometry antibodies

Antibodies used for flow cytometry include rabbit anti-OLIG2 (Thermo Scientific P21954, 1:100), anti-CD45 (30-F11, BD Biosciences 564279 or Biolegend 103132 or 103104, 1:400), anti-CD90.2 (Thy1, 53-2.1, Biolegend 140327, BD Biosciences 553004, 1:200), anti-CD11b (M1/70, Biolegend 101224 or BD Biosciences 563015, 1:400), anti-CD19 (6D5, Biolegend 115554, 1:400), anti-NK1.1 (PK136, Biolegend 108736, 1:200), anti-CD3ε (17A2, Biolegend 100216, 1:200), anti-CD4 (RM4-5, Biolegend 100557 or GK1.5, BD Biosciences 563050, 1:200), anti-CD8α (53-6.7, Biolegend 100750, 1:200), anti-CD44 (IM7, Biolegend 103030, 1:200), anti-CD69 (H1.2F3, Biolegend 104505, 1:200), anti-CD62L (MEL-14, Biolegend 104407, 1:200), anti-TBET (4B10, Biolegend 25-5825-80, 1:100), anti-GATA3 (TWAJ, eBioscience 12-9966-41, 1:100), anti-RORγt (B2D, eBioscience 17-6981-82, 1:100), anti-Ly6G (1A8, Biolegend 127624, 1:200), anti-IFNγ (XMG1.2, Biolegend 505810, 1:100), anti-IL-17A (TC11-18H10.1, Biolegend 506922, 1:100), anti-IL-10 (JES5−16E3, eBioscience 12-7101-81, 1:100), anti-TCRγδ (Biolegend 118118, 1:200 (extracellular) or 1:400 (intracellular)), anti-FOXP3 (eBioscience 53-5773-82, 1:100), anti-Ly6C (HK1.4, Biolegend 128011 or 128035, 1:400), anti-CD64 (X-54-5/7.1, Biolegend 139323 or BD Biosciences 558539, 1:200), anti-MERTK (DS5MMER, eBioscience 46-5751-80, 1:200), anti-CD9 (KMC8, BD Biosciences 564235, 1:200), anti-TREM2 (237920, R&D systems FAB17291A, 1:200), anti-CD63 (NVG-2, Biolegend 143904, 1:200), anti-I-A/I-E (MHCII, M5/114.15.2, BD Biosciences 748845, 1:400), anti-CD11c (N418, Biolegend 117339 or 117318, 1:200), anti-CD172a (SIRPα, P84, eBioscience 12-1721-80, 1:200), anti-Siglec-F (E50-2440, BD Biosciences 740956, 1:200), anti-podoplanin (gp38, 8.1.1, Biolegend 127412, 1:200), anti-CD31 (390, Biolegend 102404 or 102408, 1:200), anti-EpCAM (G8.8, Biolegend 118230, 1:200), anti-PDGFRα (APA5, Biolegend 135908, 1:200), anti-Sca-1 (Ly-6A/E, D7, Biolegend 108131, 1:200), anti-phospho-SMAD3 (EP823Y, Abcam ab52903, 1:50) and anti-CD16/32 (2.4G2, BD Biosciences 553142, 1:100 or 1:250).

## Ex vivo coculture

For ex vivo coculture experiments, lesions and contralateral cortex were dissected as described above and divided into halves or equivalently sized sections (contralateral cortex). Tissue was added to 96-well round bottom plates in 100 μl of R10 (RPMI supplemented with 10% FBS, 1% penicillin/streptomycin, 1× Glutamax (Thermo Scientific), and 55 μM β-mercaptoethanol).

For myeloid cell cocultures, single-cell suspensions from perilesional (ipsilateral) cortex were generated and stained as above. Homeostatic microglia (negative for DAM markers (CD9 and CD63)) were sorted into pre-coated tubes containing 3 ml R10 (BDFACSAria II sorting system, 100 μm nozzle size, 4 way-purity sort mode) and checked for purity post-sort. Cells were subsequently counted, and a maximum of $1 × 10^7$ cells were labelled with 5 mM CellTrace CFSE in PBS (Thermo Scientific) for 10 min, followed by washing with R10, centrifugation, and resuspension at $4.5 × 10^4$ cells per ml. One-hundred microlitres of suspension (containing 45,000 microglia) was subsequently plated with lesions or alone.

For T cell cocultures, T cells were isolated from spleens and cervical/inguinal lymph nodes using EasySep Magnetic Bead negative selection (Stem Cell), according to the manufacturer's instructions (using 31.5 μl of vortexed selection beads). A maximum of $1 × 10^7$ T cells were labelled with 5 mM CellTrace Violet in PBS (Thermo Scientific) for 10 min, followed by washing with R10, centrifugation, and resuspension at $10^6$ cells per ml. One-hundred microlitres of suspension (containing 100,000 T cells) was subsequently plated with lesions, control tissue, or alone. As a positive control, anti-CD3/CD28 T cell activating

DynaBeads (Thermo Scientific) were magnetically washed in 1 ml DPBS and added to T cells at a 1:1 ratio.

For purified fibroblast cocultures, meninges were processed as above. Lungs were perfused, dissected, and digested in PBS with Dispase II (15 U ml$^{-1}$), collagenase 1 (22,500 U ml$^{-1}$), and DNAse 1 (10 mg ml$^{-1}$) for 30 min at 37 °C, followed by centrifugation, filtration, and RBC lysis as above. Endothelial and hematopoietic cells were negatively selected using magnetic beads per manufacturers' instructions. After staining as above, fibroblasts were sorted as Lin$^-$PDGFRα$^+$gp38$^+$ (meningeal) or PDGFRα$^+$gp38$^+$Sca-1$^+$ (lung adventitial), plated at up to 15,000 per well in a flat bottom 96-well plate, and cultured for 7 days prior to initiation of lymphocyte coculture, as above.

Plates were incubated at 37 °C, 5% CO$_2$ for 72 h. After coculture completion, wells were mixed by pipetting and cells were transferred to a V-bottom plate, and blocked, stained, and analysed as above. For cocultured wells, Cell Trace Violet (CTV) or carboxyfluorescein diacetate succinimidyl ester (CFSE) staining was used to identify plated (versus lesion-resident) T cells or microglia; given the low rates of proliferation observed, gates were gates chosen to maximize exclusion of unlabelled cells while still including proliferating or CTV-diluting cells (up to at least three rounds of division). Figure 3s and Extended Data Fig. 6t–v include pooled control data from experiments where wild-type lesions were cultured with diphtheria toxin (100 ng ml$^{-1}$; any lesions expressing *Rosa26$^{DTR}$* were excluded), empty liposomes, or IgG1 isotype control (BioXCell). For direct comparisons between 7 dpi and 21 dpi lesions, media refeeding was performed daily (replacing 100 μl, 2×) to mitigate rapid media acidification by 7 dpi lesions.

## Vascular permeability and haemorrhage analysis
Vascular permeability was measured using Evans Blue extravasation[94]. In brief, Evans Blue (1% in saline) was injected intraperitoneally (8 ml kg$^{-1}$) 3 h prior to euthanasia. Following euthanasia, mice were transcardially perfused (10 ml DPBS), and mice that did not show appropriate liver clearing were excluded. Brains were dissected as above, whole hemispheres were weighed and added to 250 or 500 μl formamide, and tissue was incubated for 44–48 h to extract Evans Blue. Evans Blue fluorescence was measured on a SpectraMax microplate reader (excitation 620 nm, emission 680 nm). A standard curve (four-point logistic regression) was used to calculate extravasated mass, which was subsequently normalized to tissue weight.

To quantify haemorrhage (bleeding), photographs were taken of frozen brains during slicing (1 photograph per 100–150 μm). Representative photographs were chosen from across the lesion volume for quantification (3 photographs per brain). Regions of overtly visible blood (corresponding to erythroid cell accumulation, as visible by microscopy) were traced in Fiji, followed by normalization to lesion area (traced via tissue discoloration) and averaging per animal.

## Visium data processing
Sequencing data were aligned to mouse genome mm10 with SpaceRanger version 2.0.0 (10x Genomics). Data were processed using the Seurat R package, version 4.2.1[95]. Individual capture areas were processed using Seurat's SCTransform function to normalize data, select variable features for dimensionality reduction, and scale data. Capture areas were subsequently merged for further analysis. Principle components were calculated using Seurat's RunPCA function, followed by graph-based clustering using Seurat's FindNeighbors (dims = 1:30) and FindClusters (res = 0.8) functions and 2D visualization using Seurat's RunUMAP function (dims = 1:30). Feature and spatial feature plots, violin plots, and UMAP plots were generated using Seurat. We used FindAllMarkers to identify markers for each cluster (test.use = MAST, min.pct = 0.05, logfc.threshold = 0.2) and generated resulting dot plots using Seurat. Fibroblast-containing clusters were identified via expression of *Col1a1* and further investigated using feature plots and spatial feature plots. One large cluster (cluster 10) was subclustered

using Seurat's FindSubCluster function (res = 0.5); subcluster 10_0 was identified as a 7 dpi fibroblast-enriched cluster and selected for further analysis based on expression of *Pdgfra*. Cluster 8 was identified as a 21 dpi fibroblast-enriched cluster.

Analysing wild-type tissue, we used Seurat's FindMarkers function (min.pct = 0.05, logfc.threshold = 0.2, test.use = MAST) to determine markers for cluster 10_0 and 8 and used the EnhancedVolcano R package to visualize relevant differentially expressed genes (DEGs) (EnhancedVolcano function, minfc = 1.15, alpha = 0.05)[96]. To calculate fibroblast TGFβ scores, we utilized a previously generated bulk sequencing dataset of primary lung fibroblasts treated with TGFβ (1 ng ml$^{-1}$) or PBS[97] for 48 h. TGFβ-upregulated genes were identified from normalized data (log$_2$(fold change (FC)) > 0, $q$ < 0.05). Module scores were generated using Seurat's AddModuleScore function. To calculate proliferation scores, we utilized a proliferation signature composed of 23 genes (*Ccnb1, Ccne1, Ccnd1, E2f1, Tfdp1, Cdkn2b, Cdkn1a, Plk4, Wee1, Aurkb, Bub1, Chek1, Prim1, Top2a, Cks1, Rad51l1, Shc, Racgap1, Cbx1, Mki67, Mybl2, Bub1* and *Plk1*)[98,99].

## Mouse snRNA-seq data processing
Sequencing data were aligned to mouse genome mm10 (snRNA-seq experiments 1 (wild-type time course) and 2 (wild-type, *Tgfbr2*-cKO and ADWA11)) or Grcm39 (snRNA-seq experiment 3 (wild-type and *Cxcl12*-cKO)) with CellRanger version 7.1.0 (experiment 1), version 7.2.0 (experiment 2), or version 9.0.0 (experiment 3) (10x Genomics). Data were processed using the Seurat R package, version 4.2.1 (experiment 1), 5.0.1 (experiment 2), or 5.2.1 (experiment 3). We excluded cells with high mitochondrial gene expression and low or high unique molecular identifier (UMI) and feature counts, using the bottom and top 2.5 percentiles as our cutoff. We used Seurat's SCTransform function to normalize data, select variable features for dimensionality reduction (with the removal of certain sex-related and mitochondrial genes), and scale data. Principle components were calculated using Seurat's RunPCA function, followed by graph-based clustering using Seurat's FindNeighbors (dims = 1:30) and FindClusters (res = 0.5 (experiments 1 and 2) or 0.25 (experiment 3)) functions and 2D visualization using Seurat's RunUMAP function (dims = 1:30). We used FindAllMarkers (test.use = MAST, experiment 1) or RunPrestoAll[100] (experiments 2 and 3) to identify markers for each cluster (min.pct = 0.05, logfc.threshold = 0.2) and annotated clusters using select common and well-validated lineage marker genes (for example, *Col1a1, Col1a2, Pgdfra* (fibroblasts); *Cspg4* (mural); *Itgam* (myeloid); *P2ry12, Sall1* (microglia); *Cd3e, Cd4, Cd8* (T cells); *Rbfox3* (neurons); *Dcx, Prom1* (neural progenitors); *Gfap, Aldh1l1* (astrocytes); *Mbp, Olig2, Sox10* (oligodendrocytes); *Olig2, Sox10, Pdgfra* (oligodendrocyte precursor cells); and *Pecam1* (endothelial cells)). We identified and excluded 2 clusters made up of likely doublets based on gene expression (experiments 1 and 2) and excluded 1 sample due to insufficient nuclear yield (experiment 1), visualizing resultant clusters on UMAPs and dot plots using Seurat. Final datasets comprised 28,187 cells including 8,096 fibroblasts; 189 mural cells; 4,568 myeloid cells/microglia; 548 T cells; 11,216 neurons; 2,026 astrocytes; 537 oligodendrocytes; 94 oligodendrocyte precursor cells; 470 endothelial cells; 259 neural progenitor cells; and 184 unassigned nuclei (experiment 1); 60,070 cells including 18,455 fibroblasts; 496 mural cells; 24,302 myeloid cells/microglia; 2,271 T cells; 5,994 neurons; 2,781 astrocytes; 1,669 oligodendrocytes; 2,399 oligodendrocyte precursor cells; and 1,703 endothelial cells (experiment 2); and 19,668 cells including 7,954 fibroblasts; 3,792 myeloid cells/microglia; 180 T cells; 5,862 neurons; 1,213 astrocytes; 390 oligodendrocytes; 218 endothelial cells; and 59 unassigned nuclei (experiment 3). We subsequently subset the data to wild-type cells for downstream analysis (experiment 1). FindMarkers was used to identify differentially expressed genes between time points or genotypes/conditions (test.use = wilcox). Additional feature plots, UMAP plots, dot plots and heat maps were generated using Seurat. Joint densities for gene combinations (including published meningeal

layer signatures)[35] were visualized using the NebulosaPlot R package (plot_density function)[101] and modified using the ScCustomize R package (Plot_Density_Custom function)[102].

Fibroblasts and immune cells were subset and reclustered as above (immediately prior to wild-type subsetting (experiment 1); res = 0.5 (experiment 1), 0.2 (experiment 2) or 0.3 (experiment 3) for fibroblasts, res = 0.15 for immune cells (experiment 1), res = 0.25 for myeloid cells (experiments 2 and 3), res = 0.5 for T cells (experiments 2 and 3), res = 0.25 for neurons (experiment 3)) and DEGs were recalculated. Fibroblast clusters expressing high levels of myeloid or neuronal genes were excluded as likely contaminants and removed from global UMAP and bar plots. For experiment 1, resting dural meningeal fibroblasts were removed (by microanatomical metadata and cluster) for downstream analysis. For experiments 2 and 3, clusters were annotated by expression of previously defined CNS fibroblast or myeloid cell 'signatures' generated from experiment 1 DEG marker lists (visualized using AddModuleScore and violin plots). Dural fibroblasts were subclustered (FindSubCluster function) to identify lymphocyte-interactive and dural subsets (experiment 2). Inhibitory and excitatory neurons were identified by expression of *Gad1/Gad2* and *Slc17a6/Slc17a7*, respectively. SAM were subclustered to identify subsets that changed across genotypes (experiment 3). Relative abundance across time was calculated for each fibroblast, myeloid, or T cell subcluster after normalizing for time point or condition sample size (total number of nuclei). Gene ontology analysis was performed using the ClusterProfiler R package for gene set testing (enrichGO function)[103]. Ligand–receptor and ligand–signalling network interactions underlying myofibroblast emergence were interrogated using the NicheNetR R package[104] (experiment 1). Beginning with the global Seurat object, fibroblasts were treated as 'receiver' cells, with resting fibroblasts as the 'reference condition' and 7 dpi myofibroblasts as the 'condition of interest'. Potential 'sender' cells were any cells present at rest, 2 or 7 dpi. Integration of single nuclear and Visium data was performed using the SpaceXR package[105] (to deconvolute individual spots) and using Seurat's AddModuleScore function (to score each Visium cluster with marker sets for single nuclear fibroblast clusters from experiment 1). Multi-gene scores were calculated as follows and generated using AddModuleScore. Myofibroblast (fibroblast TGFβ) scores were calculated as above. Profibrotic SAM scores were generated as published previously, using a combination of six genes (*Trem2, Cd9, Spp1, Gpnmb, Fabp5* and *Cd63*)[22]. DAM scores were generated using the top 30 published markers specific to DAM ($P < 0.001$, ranked by logFC)[31]. Dysmaturity scores were generated using genes significantly upregulated in microglia from *Emx1^cre; Itgb8^flox* mice (relative to controls; bulk sequencing data accessed via the GEO and analysed with originally described parameters)[41]. IFNγ response scores for myeloid cells were generated using genes significantly upregulated ($q < 0.05$, $\log_2 FC > 1.5$) in microglia that were sort-purified 22 h after intraventricular injection of IFNγ (100 ng) into juvenile (P9) mice[54]. IFNγ response scores for neurons were generated using genes significantly upregulated ($q < 0.05$, $\log_2 FC > 1.5$) in neurons that were sort-purified 72 h after intraparenchymal injection of IFNγ (40 ng) into the ventral midbrain of adult mice[106]. Macrophage-fibroblast ligand–receptor interactions were interrogated and visualized using CellPhoneDB (version 4, statistical method; experiment 1)[107]. Pseudotime trajectories for myeloid cells were generated using the monocle3 R package v1.3.4 (learn_graph and order_cells functions), with homeostatic microglia and monocytes chosen as 2 possible roots (experiment 2). Phylogenetic trees were calculated using Seurat's BuildClusterTree function.

## Marmoset snRNA-seq data processing

Processed data from 7 dpi marmosets (ET-1-induced stroke)[26] and resting marmosets (unpublished) were generously provided by the J. Bourne laboratory. Fibroblasts from both datasets were identified via *COL1A1* expression, subset and integrated using the Seurat FindIntegrationAnchors and IntegrateData functions (dims = 1:30). The integrated data were subsequently rescaled, principal components were calculated, clusters were identified, and marker genes were selected as above. UMAP plots, bar plots, and heat maps of selected fibrosis-related genes were generated using Seurat. Myofibroblast (fibroblast TGFβ) scores were calculated as above.

## Human TBI snRNA-seq data processing

Human TBI snRNA-seq data were accessed via the Gene Expression Omnibus (GEO)[27]. Datasets representing individual patients were merged, and quality control was performed on mitochondrial expression, UMI counts, and gene counts, as above. We used Seurat's SCTransform function to normalize data, select variable features for dimensionality reduction (with the removal of certain sex-related and mitochondrial genes), and scale data. Principle components were calculated using Seurat's RunPCA function, followed by batch correction using Harmony[108], graph-based clustering, and cluster identification as above (res = 0.1). After cluster annotation, fibroblasts were identified via COL1A1 expression and subset. Heat maps and violin plots were generated using Seurat. Myofibroblast (fibroblast TGFβ) scores were calculated as above.

## Human GBM scRNA-seq data processing

Human GBM data were generously provided by the M. Aghi lab[12]. QC was performed in accordance with the original publication (excluding cells with mt.percent > 20% and fewer than 200 or more than 20,000 UMIs). We used Seurat's SCTransform function to normalize data, select variable features for dimensionality reduction (with the removal of certain sex-related and mitochondrial genes), and scale data. Principle components were calculated using Seurat's RunPCA function, followed by graph-based clustering and cluster identification as above (res = 0.1). After cluster annotation, fibroblasts were identified via *COL1A1* expression, subset, and reclustered as above (res = 0.6). UMAP plots, feature plots, and heat maps were generated using Seurat. Myofibroblast (fibroblast TGFβ) scores were calculated as above.

## Other software

Python coding (using Python v3.11.0) was performed in Jupyter Notebook. R coding (using R version 4.3.2) was performed in RStudio. Additional R packages used include Presto, DESeq2, dplyr, ply, ape, cowplot, Matrix, variancePartition, MAST, HGNChelper, openxlsx, RColorBrewer, gridExtra, ggpubr, ComplexHeatmap, tidyverse, tibble, biomaRt, data.table, glmGamPoi, SeuratWrappers, patchwork, magrittr, s2, gplots, stringr, ggnewscale, ggbreak, coin and dunn.test.

## Illustrations

Illustrations were created using BioRender as follows. Molofsky, A. (2025) https://BioRender.com/z04g601 (Fig. 1a–d and Extended Data Fig. 8s); https://BioRender.com/hnt0g2e (Fig. 1b); https://BioRender.com/cxeipj8 (Figs. 1c,d, 5a and Extended Data Fig. 1t,u); https://BioRender.com/uw0fsmi (Fig. 1e); https://BioRender.com/9vqfw5o (Fig. 2a); https://BioRender.com/a2wgpw9 (Fig. 2r and Extended Data Fig. 5x); https://BioRender.com/y29m122 (Fig. 2h); https://BioRender.com/0ke2wow (Fig. 3e); https://BioRender.com/r5vzeln (Fig. 3r); https://BioRender.com/1nz9lls (Fig. 3m–o); https://BioRender.com/tppusn7 (Figs. 4a,d,j,o and 5i and Extended Data Figs. 1k,m,o,r,t,u, 5d,g and 7n); https://BioRender.com/6ciab3h (Fig. 4d); https://BioRender.com/448atyq (Fig. 4r and Extended Data Fig. 8ah); https://BioRender.com/e38i116 (Fig. 5i); https://BioRender.com/6cliny1 (Fig. 5r and Extended Data Fig. 9a); https://BioRender.com/w33x285 (Fig. 5x); https://BioRender.com/by786tm (Extended Data Figs. 1a,z and 4f); https://BioRender.com/95e6eac (Extended Data Fig. 1y); https://BioRender.com/8dp80qj (Extended Data Fig. 5c); https://BioRender.com/d5bl8qu (Extended Data Fig. 6l); https://BioRender.com/eglk5p9 (Extended Data Fig. 7n); https://BioRender.com/soxo827 (Extended Data Fig. 8g); https://BioRender.com/ezibz8f (Extended Data Fig. 9x); and https://BioRender.com/g41o383 (Supplementary Fig. 2).

## Statistical analysis

All data were analysed by comparison of means using unpaired (unless otherwise noted) two-tailed Student's *t*-tests; for multiple comparisons, one-way ANOVA (with Tukey post hoc test) or two-way ANOVA (with Sidak's post hoc test, applied over within-subject and between-subject comparisons) were used as appropriate (Prism, GraphPad Software). Graphs display mean ± s.d. unless otherwise noted. When possible, results from independent experiments were pooled. All data points reflect individual biological mouse replicates, unless otherwise noted. Select experiments were performed once, for reasons including breeding, cost, and technical constraints, including the following: tMCAO in *Cthrc1^creER^; R26^tdT^* mice; analysis of spinal cords of *Tgfbr2*-cKO mice; analysis of cortical scar-associated macrophages in *Tgfbr2*-cKO mice; meningeal fibroblast coculture (using fibroblasts sorted from 18 mice); quantification of recombination in *Ng2^creER^* mice; and quantification of *Col1a1^GFP+^* fibroblasts after ADWA11 treatment. RNA-sequencing experiments contained two pooled mice per sample. All other experiments were performed at least 2 times with successful reproduction.

## Reporting summary

Further information on research design is available in the Nature Portfolio Reporting Summary linked to this article.

## Data availability

Mouse spatial and snRNA-seq data generated in this paper are deposited at the Gene Expression Omnibus (GEO) under the accession number GSE254164. Mouse genomes were downloaded via 10X Genomics, including Mm10 (https://cf.10xgenomics.com/supp/cell-exp/refdata-gex-mm10-2020-A.tar.gz) and Grcm39 (https://cf.10xgenomics.com/supp/cell-exp/refdata-gex-GRCm39-2024-A.tar.gz). The raw data from Boghdadi et al.[26] (marmoset stroke) are available at GSE179141. The raw data from Garza et al.[27] (human TBI) are available at GSE209552. The raw data from Jain et al.[12] (human GBM) are available at GSE132825. The raw data from Keren-Shaul et al.[31], used to generate DAM scores, are available at GSE98971. The raw data from Yin et al.[41], used to generate dysmaturity scores, are available at GSE239603. The data used to generate IFNγ scores in neurons were obtained from Hobson et al.[106] (see Supplementary Table 1). The genes from Sbierski-Kind et al.[97], used to generate TGFβ/myofibroblast scores, are presented in Supplementary Table 1. The genes from Mroz et al.[54], used to generate IFNγ scores in myeloid cells, are presented in Supplementary Table 2. Source data are provided with this paper.

## Code availability

All original code generated to analyse RNA-sequencing data is available at GitHub (https://github.com/newingcrystal/CNS_Fibroblasts).

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

**Acknowledgements** The authors thank R. Locksley for his constructive feedback on this manuscript as well as the UCSF Parnassus Flow Core RRID:SCR_018206 and DRC Center Grant NIH P30 DK063720, the UCSF Parnassus Advanced Light Microscopy CoLab, the UCSF Genomics CoLab, the UCSF Institute for Human Genetics, the Gladstone Histology and Light Microscopy Core (HLMC), the Gladstone Genomics Core, the Monash Histology Platform, the University of Michigan University for Laboratory Animal Medicine Pathology Core, and Histowiz for instruments and services. N.A.E.-C. is supported by the NIGMS (NIH T32GM007618). P.C. is supported by the Schmidt Science Fellows in partnership with the Rhodes Trust. T.T. is supported by the NHLBI (NIH K99/R00HL155786). L.T. and J.A.B. are supported by the National Health and Medical Research Council (NHMRC, Australia; APP20140228) and The Yulgilbar Foundation Fund to J.A.B. The Australian Regenerative Medicine Institute is supported by the State Government of Victoria and the Australian Government. J.T.P. is supported by the NINDS (NIH R01NS096369). F.F.G. is supported by the NINDS (NIH R01NS107039). D.S. is supported by the NHLBI (NIH R01HL142568). T.D.A. is supported by the NINDS (NIH R01NS119615-01). J.T.P., A.B.M. and A.V.M. are supported by the NINDS (NIH R01NS126765) and the UCSF PBBR grant; A.B.M. is additionally supported by UCSF Department of Laboratory Medicine discretionary funds and by the NIAID (NIH R01AI162806).

**Author contributions** N.A.E.-C., T.D.A., A.V.M. and A.B.M. conceptualized and designed the study. N.A.E.-C. and N.M.M. performed PT injuries. A.L. performed tMCAO injuries. N.C.L. and L.P. performed dMCAO injuries. A.C. performed CCI injuries. L.T. performed marmoset stroke injuries. N.A.E.-C., N.M.M., R.P., P.C., J.I.D., A.A.C., E.D.M., N.K., A.K., S.C.K.N., A.C., N.C.L. and L.P. performed mouse handling and injections. C.O.L. and R.P. performed vital sign monitoring. N.A.E.-C., N.M.M., C.O.L., R.P., P.C., A.A.C., E.D.M., S.E.C., N.K., L.T., S.C.K.N., N.C.L. and L.P. processed tissues. N.A.E.-C., N.M.M., P.C., N.K., L.T., N.C.L. and L.P. performed immunofluorescence staining and imaging. N.A.E.-C. and N.K. performed quantitative wide-field microscopy analysis. N.A.E.-C. performed quantitative confocal microscopy analysis. N.A.E.-C. and S.C.K.N. performed ex vivo coculture experiments and analysis. N.A.E.-C. performed extravasation analysis. N.A.E.-C., N.M.M., N.K. and S.C.K.N. performed flow cytometry experiments and analysis. N.A.E.-C. performed spatial transcriptomics experiments and analysis. N.A.E.-C. and N.M.M. performed snRNA-seq experiments. N.A.E.-C. performed snRNA-seq analysis. T.T., G.L.M. and H.P. generated animal models. A.A. generated ADWA11. M.W.D., S.J. and M.K.A. generated and analysed additional RNA-sequencing data. X.L. and D.X. performed MRI analysis. N.A.E.-C., T.D.A. and A.B.M. generated the figures. N.A.E.-C. and A.B.M. wrote the manuscript. N.A.E.-C., T.D.A., A.V.M. and A.B.M. edited the manuscript with collective input from all authors. J.A.B., J.T.P., R.D., T.D.A., F.F.G., D.S., A.V.M., T.D.A. and A.B.M. supervised experiments, provided resources and contributed to methodology. A.B.M. supervised the entire study.

**Competing interests** D.S. and UCSF hold patents on the uses of antibodies that block integrin $\alpha_v\beta_8$. D.S. is a founder of Pliant Therapeutics and has received research funding from Abbvie, Pfizer, and Pliant Therapeutics. D.S. serves on the Scientific Review Board for Genentech, and on the Inflammation Scientific Advisory Board for Amgen. The remaining authors declare no competing interests.

**Additional information**
**Correspondence and requests for materials** should be addressed to Thomas D. Arnold or Ari B. Molofsky.

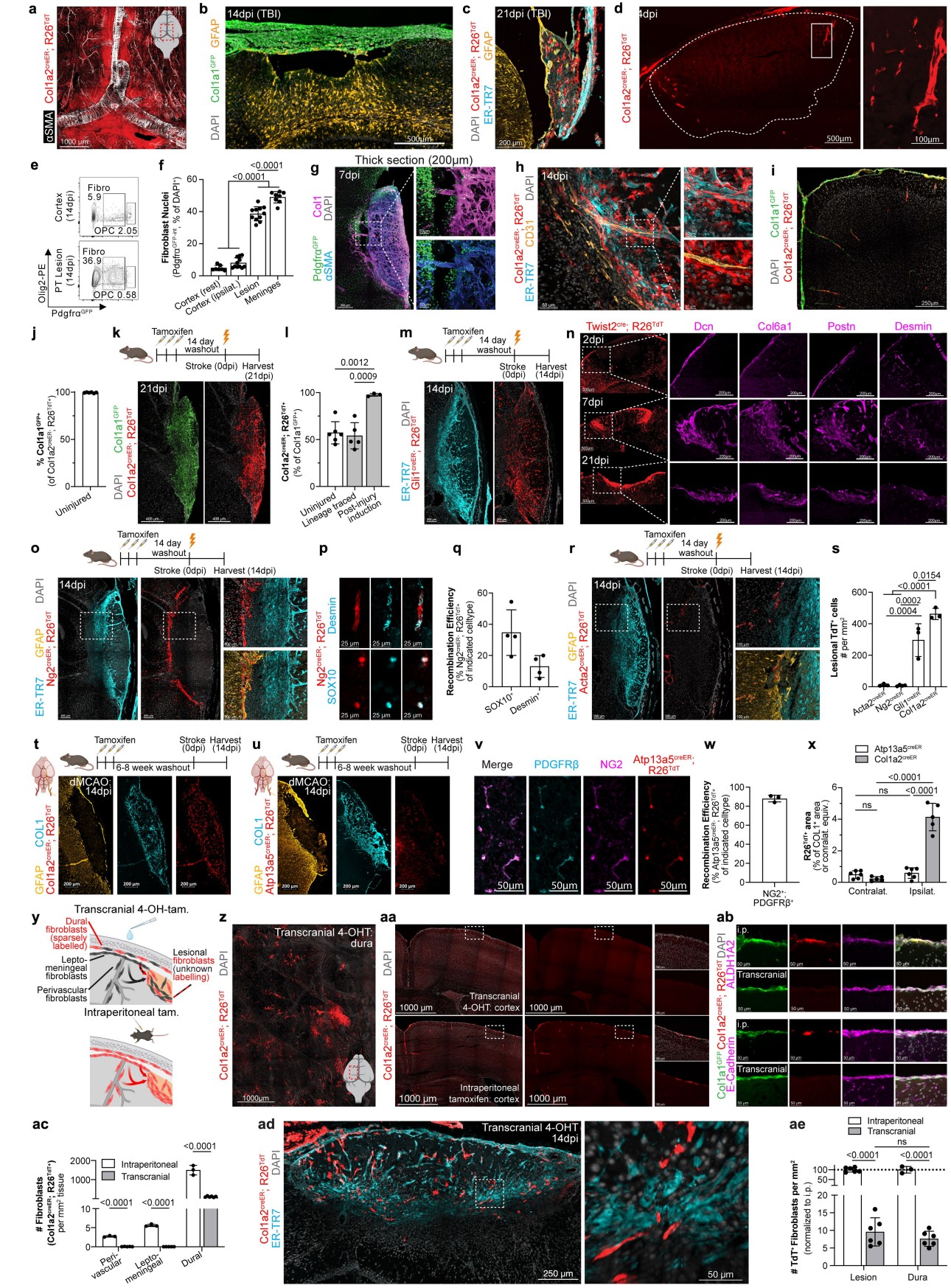

**Extended Data Fig. 1** | See next page for caption.

**Extended Data Fig. 1 | Additional characterization of the CNS fibroblast response to PT injury, related to Fig. 1. a**, Homeostatic dural meningeal fibroblasts (Col1a2$^{creER}$; Rosa26$^{TdT+}$; whole-mounted dura, tamoxifen days ⁻11–⁻7 before harvest). **b-c**, fibroblast infiltration (Col1a1$^{GFP+}$, **b**, or Col1a2$^{creER}$; Rosa26$^{TdT+}$, **c**) after CCI model of TBI (tamoxifen days ⁻18–⁻16). **d**, Fibroblast expansion near lesion borders (white dotted line) by 4dpi (tamoxifen days ⁻14–⁻12,⁻7,0–2dpi). **e-f**, Flow plot (**e**) and quantification (**f**) of fibroblast nuclei in Pdgfrα$^{GFP}$ mice (marking fibroblasts and oligodendrocyte precursor cells [OPC]). n = 7(resting cortex), n = 11(ipsilateral cortex/lesion, 14dpi), or n = 8(dura, resting or 14dpi) mice. **g-h**, Expanded perilesional vasculature and associated ECM after PT injury, with Pdgfrα$^{GFP+}$ nuclei (**g**) or fibroblasts (**h**) visualized in lesional perivascular spaces. **i-l**, Resting fibroblasts (Col1a1$^{GFP+}$) partially recombined by Col1a2$^{creER}$ (i.e., TdT$^+$; **i**), quantification of *GFP* expression within recombined cells (**j**), 21dpi lesional fibroblasts (Col1a1$^{GFP+}$) with a fibroblast origin (TdT$^+$) (**k**), and quantification of recombination efficiency before and after injury (**l**). Tamoxifen days ⁻16–⁻14 (before injury, "uninjured"/"lineage-traced") or 7–9dpi ("post-injury induction"). n = 6(uninjured), n = 5(lineage traced), or n = 3(post-injury) mice (1 40μm slice [uninjured] or 2 14μm slices/mouse). **m**, Lineage-traced fibroblasts in Gli1$^{creER}$; Rosa26$^{TdT}$ mice (preferentially recombining fibroblast subsets in multiple tissues). Tamoxifen days ⁻16–⁻14 before injury. **n**, Time course showing stromal/fibroblast (Twist2$^{cre}$; Rosa26$^{TdT+}$) expression of fibroblast markers (Dcn, Col6a1, Postn) and pericyte marker (Desmin). Serial sections; images representative of n = 2–3 (2dpi), n = 2–6 (7dpi), or n = 2–7 (21dpi) mice. **o-q**, Lack of lineage-traced lesional cells (with extra-lesional accumulation) in Ng2$^{creER}$; Rosa26$^{TdT}$ mice (**o**), images of *Ng2*-lineage$^+$ pericytes (Desmin$^+$) or oligodendrocyte-lineage cells (SOX10$^+$; **p**), and quantification of recombination

efficiency within each lineage (**q**). n = 4 mice. Tamoxifen days ⁻16–⁻14 before injury. **r**, Lack of lineage-traced lesional cells (with sparse smooth muscle visible) in Acta2$^{creER}$; Rosa26$^{TdT}$ mice. Tamoxifen days ⁻16–⁻14 before injury. **s**, Quantification of lesional recombined cells in Acta2$^{creER}$, Ng2$^{creER}$, Gli1$^{creER}$, and Col1a2$^{creER}$ mice. n = 3(Acta2$^{creER}$, Gli1$^{creER}$, Col1a2$^{creER}$) or n = 4(Ng2$^{creER}$) mice. **t-u**, Lineage traced fibroblasts (**t**, Col1a2$^{creER}$; Rosa26$^{TdT}$) but not pericytes (**u**, Atp13a5$^{creER}$; Rosa26$^{TdT}$) accumulating within fibrotic lesions (Col1$^+$) after distal Middle Cerebral Artery Occlusion (dMCAO), 14dpi. Tamoxifen given for 3 days, 6–8 weeks prior to injury. **v-w**, Images of *Atp13a5*-lineage$^+$ pericytes (**v**, PDGFRβ$^+$, NG2$^+$) and quantification of recombination efficiency (**w**). n = 3 mice (3 slices/mouse). **x**, Quantification of lesional or contralateral recombined pericytes or fibroblasts (TdT$^+$ area in Atp13a5$^{creER}$ or Col1a2$^{creER}$; Rosa26$^{TdT}$ mice) after dMCAO, 14dpi. n = 6(Atp13a5$^{creER}$) or n = 5(Col1a2$^{creER}$) mice (2–3 slices/ mouse). **y-ac**, Schematic (**y**) showing transcranial 4-hydroxy-tamoxifen (4-OHT) induction (top) and intraperitoneal tamoxifen induction (bottom). Transcranial 4-OHT sparsely recombines dural (**z**) but not perivascular or leptomeningeal (pial or arachnoid) fibroblasts (**aa**), highlighted with arachnoid markers ALDH1A2 and E-Cadherin (**ab**, top and bottom) and quantified in **ac**. Shown 2–10 days after induction; n = 3(intraperitoneal), n = 5(transcranial-perivascular/ leptomeningeal), or n = 6(transcranial-dural) mice. **ad-ae**, Image (**ad**) and quantification (**ae**) of lesional fibroblasts recombined via transcranial 4-OHT. n = 6(lesion [2 slices/mouse] and dura-transcranial) or n = 3(dura-intraperitoneal) mice. One-way ANOVA, Tukey post-test (**f,l,s**); two-way ANOVA, Sidak's post-test (**x** [repeated measures], **ae**); multiple two-way T-tests, Holm-Sidak correction (**ac**). 14μm (**b-d,h,k,m-p,r,ad**), 200μm (**g**), 40μm (**i,n** [desmin],**aa,ab**), or 10μm (**t,u,v**) slices; images represent two or more mice.

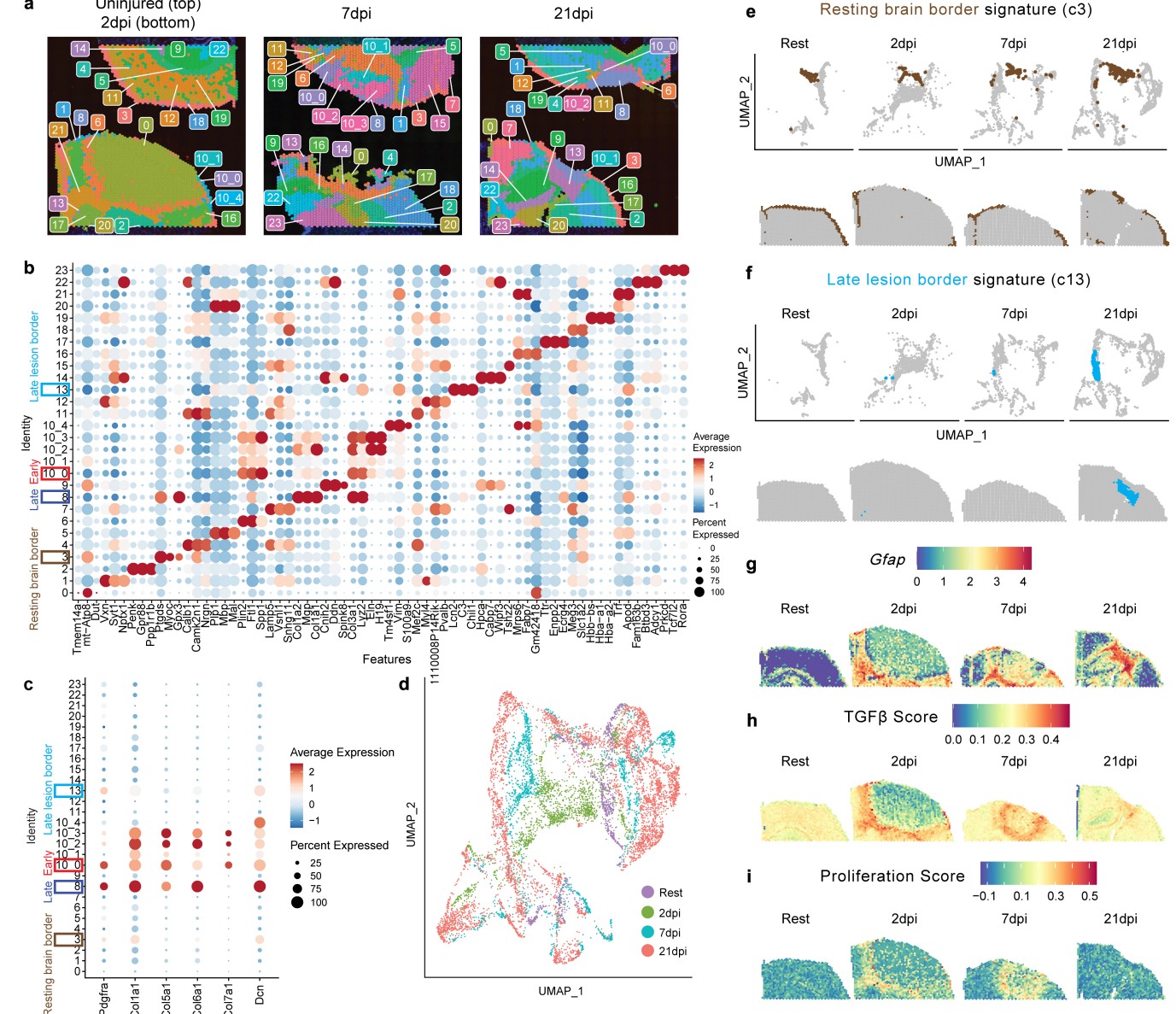

**Extended Data Fig. 2 | Spatial transcriptomics characterization of PT injury, related to Fig. 1. a**, Spatial transcriptomics plot showing all spot-based clusters. Slices shown in original Visium slide configuration with two slices/capture area: rest (top left), 2dpi (bottom left), 7dpi (middle, 2 biological replicates), and 21dpi (right, 2 biological replicates). **b-c**, Dot plots showing marker genes (**b**) or fibroblast-associated genes (**c**) for all spot-based clusters. c3 ("resting brain border"), c8 ("late fibroblast"), c10_0 ("early fibroblast"), and c13 ("late lesion border") are highlighted (colored boxes). c10_0 was selected from c10 for further analysis based on high *Pdgfra* expression. **d**, UMAP showing spot-based clusters colored by timepoint, highlighting divergence among fibroblast-containing clusters. **e-g**, UMAP and/or spatial plots showing the "resting brain border" signature near the cortical boundary (**e**, cluster 3, brown) and the "late lesion border" signature (**f**, cluster 13, cyan) overlying astrocyte-containing spots (**g**, *Gfap*⁺). **h-i**, Spatial plots showing distribution of "fibroblast TGFβ score" (**h**) or "proliferation score" (**i**).

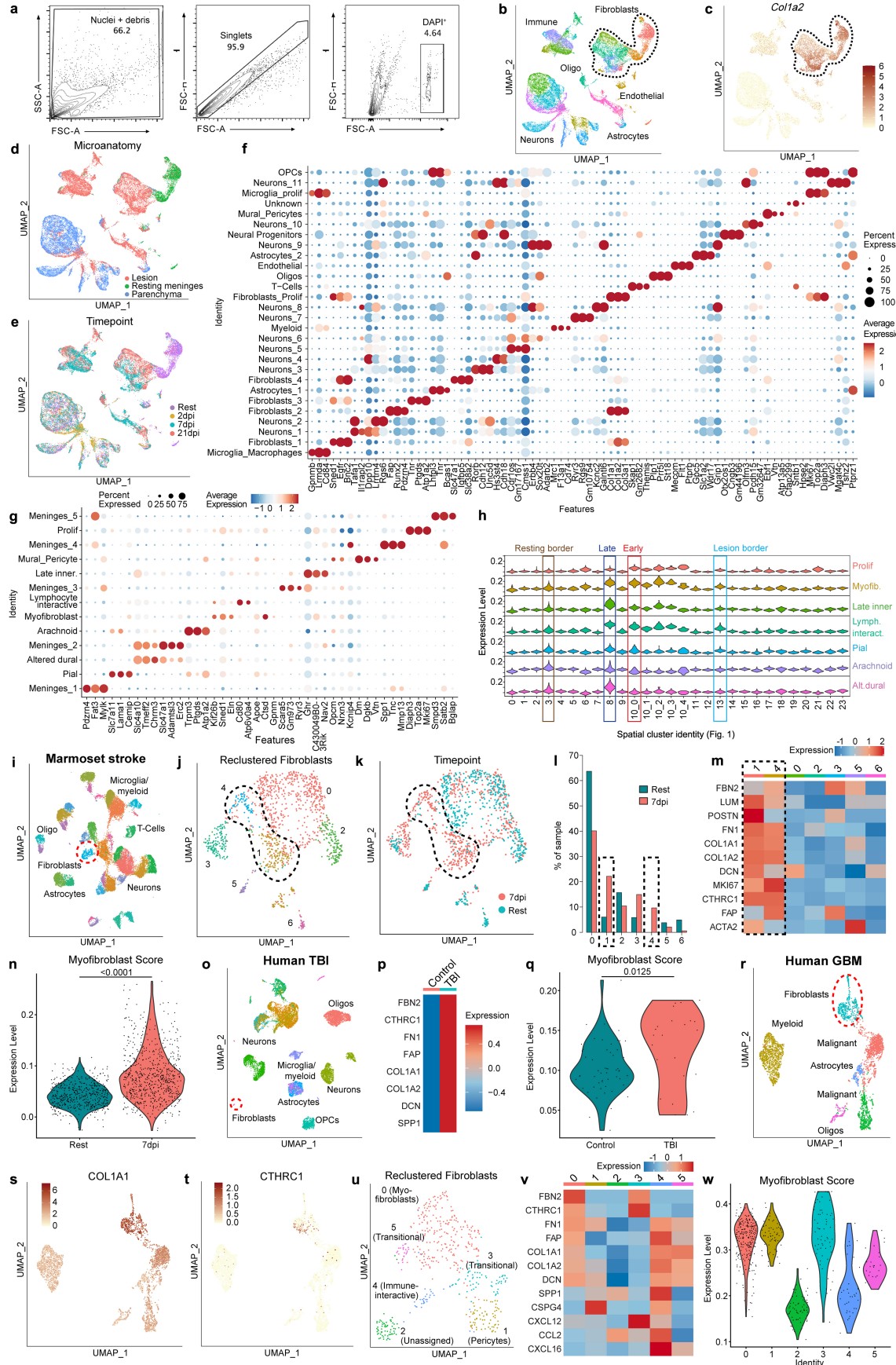

**Extended Data Fig. 3** | See next page for caption.

**Extended Data Fig. 3 | Additional transcriptomic characterization of fibroblasts across species and injury models, related to Fig. 2. a**, Gating strategy for sorting DAPI⁺ nuclei for snRNAseq. **b-c**, Global UMAP (**b**) and *Col1a2* feature plot (**c**) showing fibroblast clusters (black dotted line) along with immune, oligodendrocyte lineage, endothelial, astrocyte, and neuron clusters. **d-e**, Global UMAPs showing barcode-based microanatomical metadata (**d**) or timepoint metadata (**e**). Neurons show parenchymal origin and broad temporal distribution, dural fibroblasts (dissected only at rest) show meningeal origin and resting distribution, and lesional fibroblasts, immune cells, and glial cell subsets show lesional origin and temporal heterogeneity. **f-g**, Dot plots showing marker genes for all cellular clusters (**f**) or all fibroblast subclusters (**g**). **h**, Violin plots mapping each lesional fibroblast cluster identity (via marker-DEGs) onto spatial transcriptomic clusters (Fig. 1), with spatial "fibroblast-enriched signatures" highlighted. **i**, Adult marmoset annotated snRNAseq data (Boghdadi et al.[26]) from endothelin-1-induced stroke (cortex, 7dpi). Fibroblasts highlighted (red dotted line) based on COL1A2 expression. **j-k**, UMAPs showing integrated fibroblasts from 7dpi (described in **i**) and resting adult marmoset dataset, showing annotated fibroblast subclusters (**j**) and timepoint (**k**). Black dotted lines highlight 7dpi-enriched clusters 1 and 4.

**l-m**, Bar plot showing relative cluster abundance (**l**, as proportion of fibroblast nuclei), and heatmap of select fibroblast/myofibroblast-related genes across fibroblast clusters (**m**). Black dotted lines highlight 7dpi-enriched clusters 1 and 4. **n**, Violin plot showing myofibroblast-associated TGFβ score. n = 529(rest) or n = 663(7dpi) fibroblasts. **o**, Human TBI annotated snRNAseq data (Garza et al.[27]), including control tissues (varying timepoints; controls from subjects with non-neurological causes of death). Fibroblasts highlighted (red dotted line) based on COL1A2 expression. **p-q**, Heatmap and violin plot showing elevated expression of selected fibroblast/myofibroblast genes (**p**) or myofibroblast-associated TGFβ score (**q**) in post-TBI fibroblasts. n = 54(control) or n = 23(post-TBI) fibroblasts. **r-t**, Human glioblastoma multiforme (GBM) tissue snRNAseq data (Jain et al.[12]), shown as UMAP (**r**) and feature plots of COL1A1 (**s**) or CTHRC1 (**t**). Cancer-associated fibroblasts are highlighted (**r**, red dotted line). **u-w**, UMAP showing reclustered fibroblasts (**u**), and heatmap or violin plot showing fibroblast expression of select fibroblast/myofibroblast genes (**v**) or myofibroblast-associated TGFβ score (**w**), enriched in cluster 0. n = 235(c0), n = 67(c1), n = 66(c2), n = 63(c3), n = 38(c4), or n = 25(c5) fibroblasts. Two-way Mann-Whitney test (**n,q**).

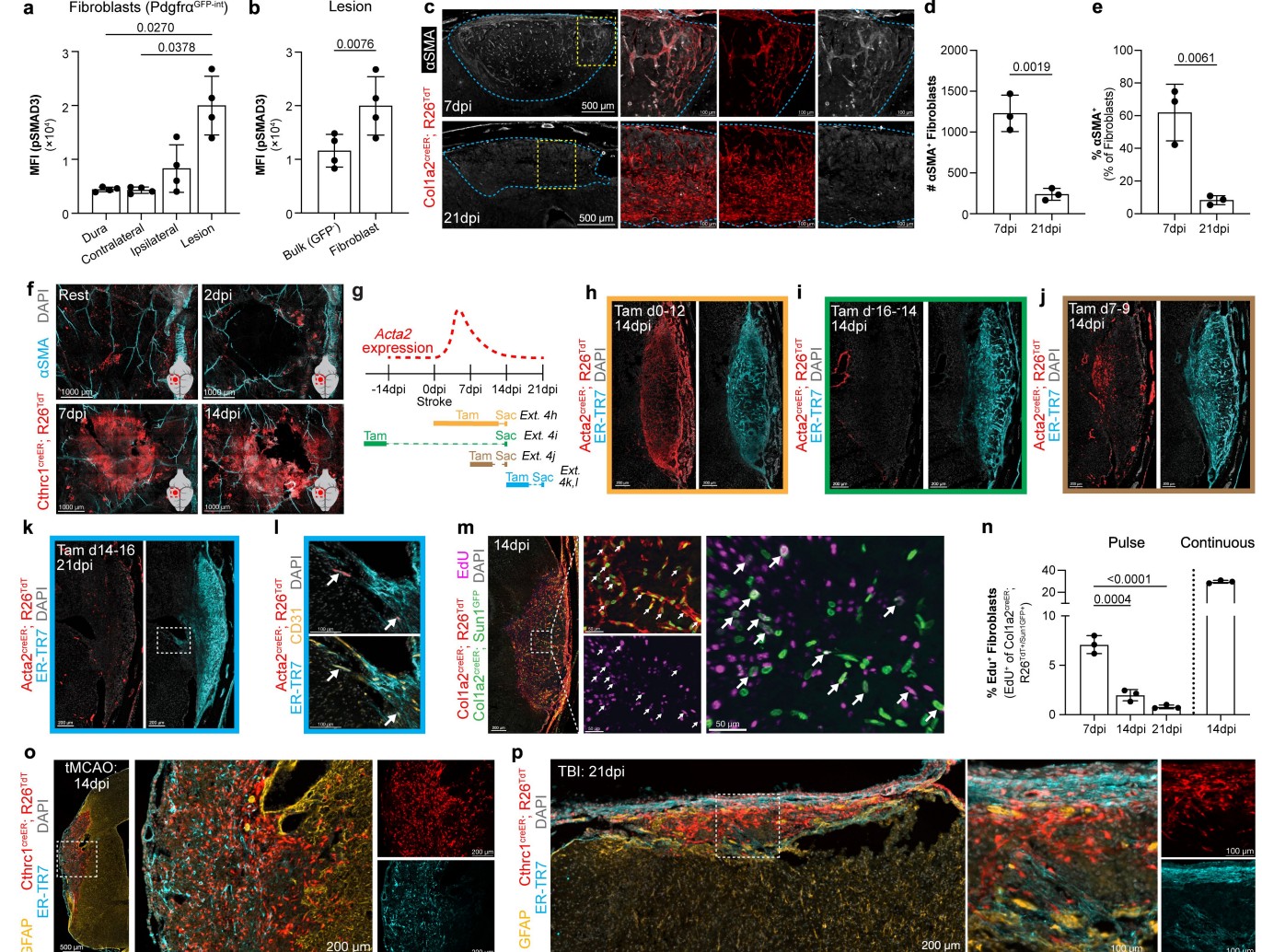

**Extended Data Fig. 4 | Additional in vivo validation of early TGFβ-associated myofibroblasts, related to Fig. 2. a-b**, Nuclear flow cytometry showing elevated pSMAD3 expression within lesional fibroblasts (Pdgfrα$^{GFP-int}$) relative to dural/parenchymal fibroblasts (**a**) or other lesional cells (**b**), 7dpi. n = 4 mice. **c-e**, Image of αSMA$^+$ expression among fibroblasts (Col1a2$^{creER}$; Rosa26$^{TdT+}$) at 7/21dpi (**c**), and quantification of the number (**d**) and percentage (**e**) of αSMA$^+$ fibroblasts. n = 3 mice/group. **f**, Confocal microscopy showing Cthrc1$^{creER}$; Rosa26$^{TdT+}$ fibroblasts in whole-mounted dural meninges at indicated timepoints. Tamoxifen days ‾16–‾14 prior to injury (rest); days ‾1–1 (2dpi); days 0–2,5 (7dpi); or days 0–2,5,12 (14dpi). **g**, Schematic showing tamoxifen induction for (**h-l**) and hypothesized trajectory for *Acta2* (gene for αSMA) expression. **h-l**, Confocal microscopy showing *Acta2*-lineage$^+$ lesional fibroblasts after PT injury (tamoxifen induction and harvest days indicated), with robust injury-induced

*Acta2* (TdT) expression (**h**) but lack of lineage-traced fibroblasts (**i**). Active *Acta2* expression is reduced but present at 7dpi (**j**, shown at 14dpi) and absent by 14dpi (**k**, shown at 21dpi). **l** (inset from **k**) shows expected Acta2$^{creER}$; Rosa26$^{TdT+}$ vascular smooth muscle cells (white arrows) near CD31$^+$ vessels. **m-n**, Image (**m**) and quantification (**n**) of fibroblast proliferation (EdU incorporation) 2 h after EdU pulse ("Pulse"; 7dpi, 14dpi, or 21dpi) or after EdU injection every other day ("Continuous", and shown in **m**; 14dpi). In insets (**m**), arrows mark proliferating fibroblasts. Fibroblasts defined as Col1a2$^{creER}$; Rosa26$^{TdT+/Sun1GFP+}$. Tamoxifen days 0,1,2,5,12. n = 3 mice/timepoint (2 slices/mouse). **o-p**, *Cthrc1*-lineage$^+$ fibroblasts within fibrotic scar after tMCAO (**o**) or TBI (**p**). Tamoxifen days 0,1,2,5,12 (**o**) or 0,1,2,5 (**p**). One-way repeated-measures ANOVA, Tukey post-test (**a**); paired two-way Student's T-test (**b**); two-way Student's T-test (**d,e**); one-way ANOVA, Tukey post-test (**n**). 14μm slices.

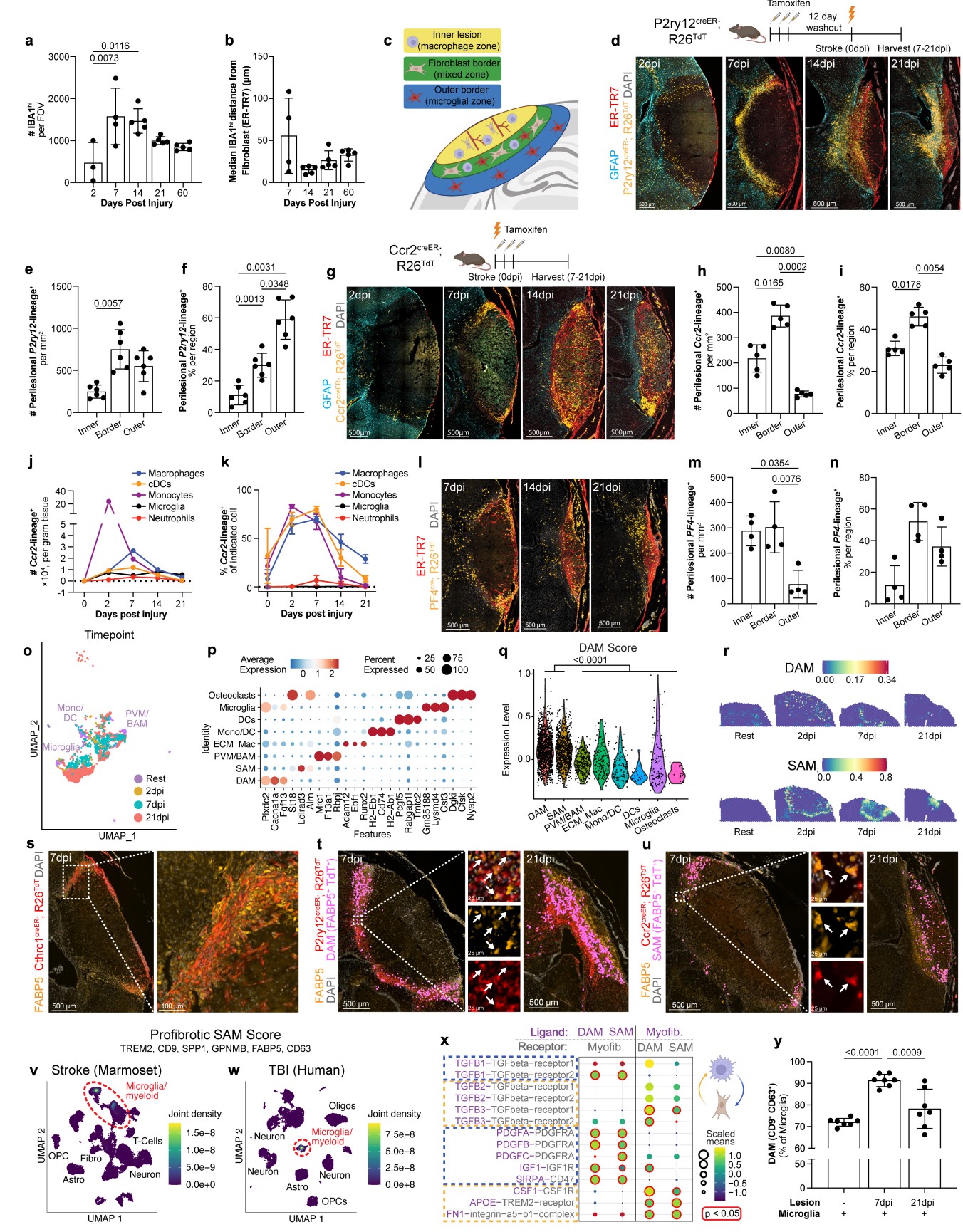

**Extended Data Fig. 5** | See next page for caption.

**Extended Data Fig. 5 | Additional characterization of injury-responsive profibrotic CNS myeloid cells, related to Fig. 2. a-b**, Quantification of Fig. 2o showing number of myeloid cells (IBA1[hi], macrophages and reactive microglia) per slice (**a**) and median myeloid cell distance from the nearest fibroblast-associated ECM (ER-TR7[+]) surface (**b**). n = 3(2dpi), n = 4(7dpi), or n = 5(14/21/60dpi) mice (2–4 slices/mouse). **c**, Schematic showing myeloid cell distribution by ontogeny. **d-f**, Immunofluorescent imaging from microglia lineage tracer mice (**d**, P2ry12[creER];Rosa26[TdT]; tamoxifen days −14−−12 before injury), with quantification of lineage traced cell density (**e**) and proportion (**f**) in inner core, fibrotic border, or outer glial regions at late timepoints (14/21dpi). n = 6 mice. **g-i**, Imaging from monocyte lineage tracer mice (**g**, Ccr2[creER];Rosa26[TdT]; tamoxifen 0–2dpi), with quantification of density (**h**) and proportion (**i**) by region, as above. n = 5 mice. **j-k**, Time course showing *Ccr2*-lineage traced cells differentiating into both lesional macrophages and dendritic cells, shown as normalized counts (**j**) or as a proportion of the indicated population (**k**). Tamoxifen 0–2dpi. n = 4(0/2dpi), n = 3(7/21dpi), or n = 5(14dpi) mice. **l-n**, Imaging from BAM/PVM lineage tracer mice (**l**, PF4[cre];Rosa26[TdT]), with quantification of density (**m**) and proportion (**n**) by region, as above. n = 4 mice. **o-q**, UMAP of reclustered myeloid cells (**o**, colored by timepoint; major resting populations labelled), dot plot showing myeloid subcluster marker genes (**p**), and violin plot of Disease Associated Microglia ("DAM") score (**q**, an aggregate of 30 published DAM-specific markers). n = 1267(DAM), n = 816(SAM), n = 305(PVM/BAM), n = 173(ECM_Macs), n = 116(Mono/DC), n = 20(DCs), n = 110(Microglia), or n = 19(Osteoclasts) nuclei. **r**, DAM/SAM snRNAseq identities mapped onto spatial transcriptomic data. **s-u**, Imaging of FABP5[+] cells near Cthrc1[creER]; R26[TdT+] myofibroblasts at 7dpi (**s**), and surfaced images of FABP5[+] DAM (**t**, microglia-derived) or SAM (**u**, monocyte-derived) in perilesional/border regions at 7dpi (with inset) or 21dpi. **v-w**, Density plots showing expression of SAM score among myeloid cells after stroke in marmosets (**v**) or TBI in humans (**w**). **x**, Ligand-receptor interactions between SAM or DAM and myofibroblasts (CellPhoneDB). Left half: ligand (first/purple molecule in labelled pair) expressed by SAM/DAM, receptor (grey) expressed by myofibroblast. Right half: ligand expressed by myofibroblast, receptor expressed by SAM/DAM. Dotted lines (left) and schematic (right) highlight modules with macrophage-expressed ligands (blue) or myofibroblast-expressed ligands (orange). **y**, Expression of DAM markers (CD9/CD63) on sorted homeostatic microglia cultured for 72 h alone or with 7/21dpi lesions. n = 7 wells/condition. One-way ANOVA, Tukey post-test (**a,b,y**); one-way repeated-measures ANOVA, Tukey post-test (**e,f,h,i,m,n**); Kruskall-Wallis test, Dunn's multiple comparisons correction (**q**, relevant comparisons shown). 14μm slices; images represent 2 or more mice.

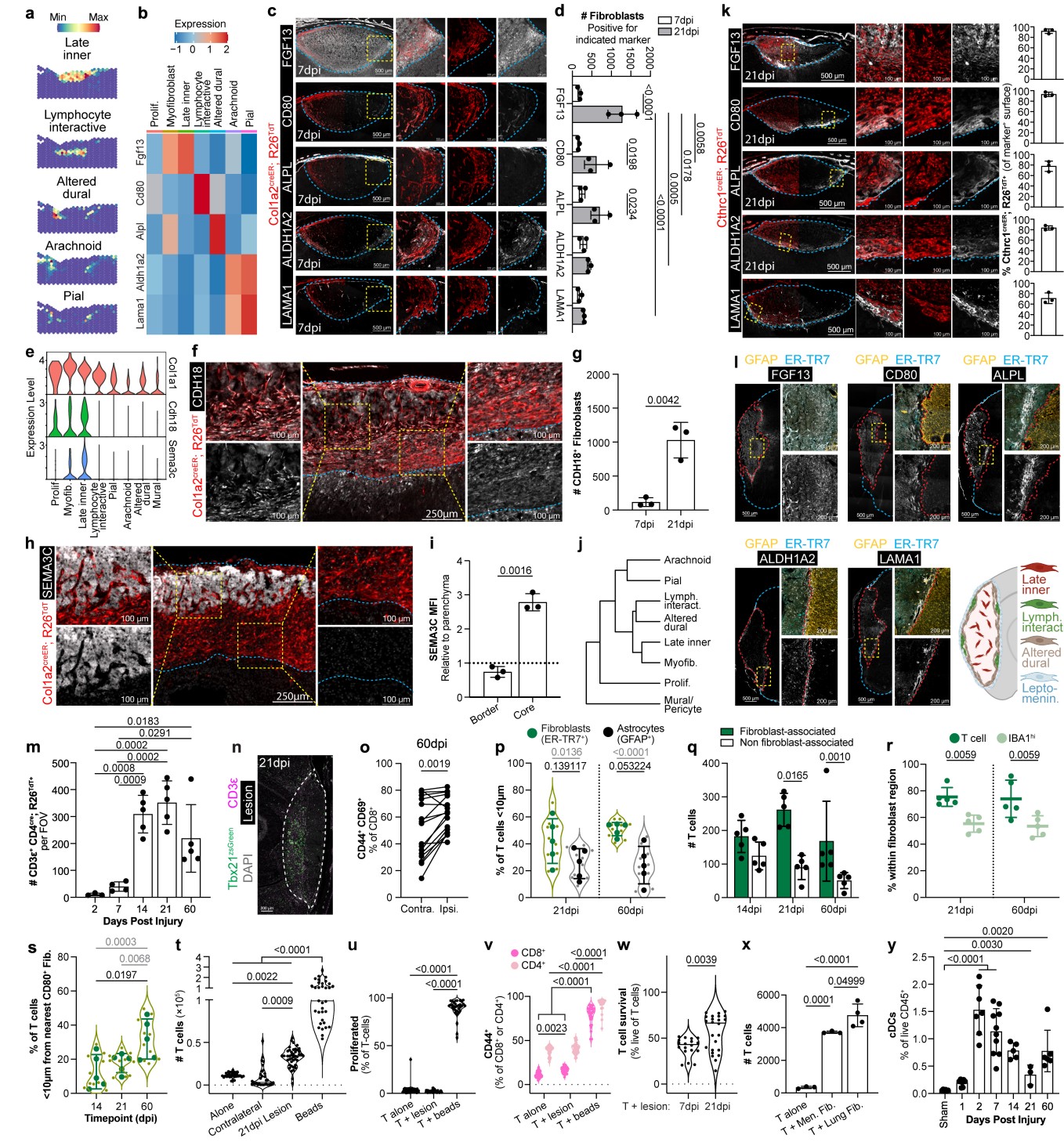

**Extended Data Fig. 6** | See next page for caption.

**Extended Data Fig. 6 | Additional characterization of late lesional fibroblasts, related to Fig. 3. a**, Single nuclear signatures of late lesional fibroblast states mapped onto spatial transcriptomic data, showing expected localization. **b**, Heatmap of selected late fibroblast state markers used for imaging. **c**, Late fibroblast state markers shown at 7dpi. **d**, Quantification of **c** and Fig. 3d, showing the number of subset marker-positive fibroblasts at 7/21dpi. n = 3 mice/timepoint. **e-i**, Violin plots (**e**) showing ECM (*Col1a1*) and smooth muscle-related[109,110] (*Cdh18, Sema3c*) genes (enriched among late inner fibroblasts), with imaging and quantification of CDH18+ fibroblasts (**f,g**) and SEMA3C within the inner core (**h,i**, 21dpi). n = 3 mice/timepoint. **j**, Phylogenetic tree showing lesional stromal cell cluster relationships. **k**, Imaging (left) and quantification (right) of subset marker expression among *Cthrc1*-lineage+ fibroblasts (tamoxifen continuously, i.e. days 0–2,5,8,11, etc.; fibroblast- vs. non-fibroblast-adjacent FGF13/LAMA1 quantified). n = 3 mice. **l**, Fibroblast subset marker staining after tMCAO (14 dpi) and cartoon of fibroblast subset topography. **m**, Perilesional T cell time course (CD4cre; Rosa26TdT+; CD3ε+). n = 3(2dpi), n = 4(7dpi), or n = 5(14/21/60dpi) mice (2–5 slices/mouse). **n**, Lesional type 1 lymphocytes (TbetzsGreen+, 21dpi). **o**, Tissue resident memory CD8+ T cell distribution (CD44+ CD69+; 60dpi). n = 17 mice. **p-r**, T cell proportion <10μm from nearest fibroblast-associated ECM (ER-TR7+) or astrocyte (GFAP+) surface (**p**), T cells within or outside fibroblast-dense lesion (**q**), and T or myeloid cell (IBA1hi) proportion within lesion (**r**). n = 5 mice/timepoint (2–4 slices/mouse; slice values/statistics shown with lighter colors). **s**, T cell proportion <10μm from nearest CD80+ fibroblast. n = 5 mice/timepoint (2 slices/mouse; slice values/statistics shown with lighter colors). **t-v**, T cell counts (**t**), proliferation (**u**, CTV dilution), and activation (**v**, CD44+) after coculture (21dpi lesions). n = 36(**t,u**, alone), n = 33(**v**, alone), n = 31(contralateral), n = 43(**t**, 21dpi lesion), n = 44(**u,v**, 21dpi lesion), n = 33(**t,u**, beads), or n = 30(**v**, beads) wells. **w**, T cell survival after coculture with 7dpi or 21dpi lesion. n = 18(7dpi) or n = 24(21dpi) wells. **x**, T cell survival counts after coculture with dural meningeal or lung fibroblasts. n = 3(alone, meningeal) or n = 4(lung) wells. **y**, Conventional dendritic cell (cDC) infiltration and persistence after PT injury (cortical flow cytometry). n = 6(sham/60dpi), n = 7(1/2dpi), n = 10(7dpi), n = 5(14 dpi), or n = 3(21dpi) mice. Two-way repeated-measures ANOVA, Sidak's post-test (**d,q,v**); two-way Student's T-test (**g,w**); paired two-way Student's T-test (**i,o**); one-way ANOVA, Tukey post-test (**m,s-u,x,y**); multiple two-way T-tests, Holm-Sidak correction (**p** [per slice]); multiple paired T-tests, Holm-Sidak correction (**p** [per point], **r**). 14μm slices; images represent two or more mice.

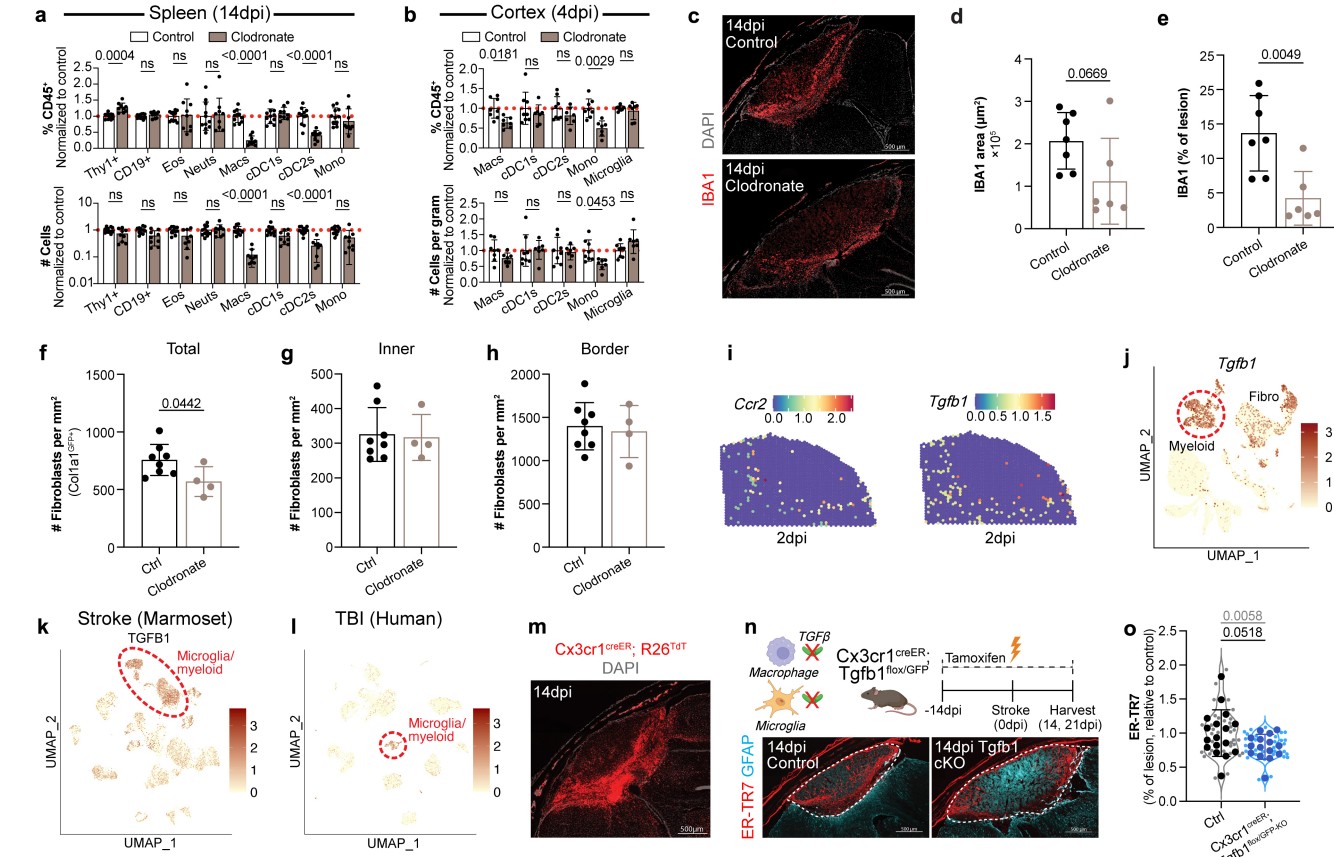

**Extended Data Fig. 7 | Profibrotic macrophages promote CNS fibroblast expansion, related to Fig. 4. a-b,** 14 dpi splenic immune cells (**a**) or 4 dpi cortical phagocytic myeloid cells (**b**) after treatment with control or clodronate liposomes, as frequency (top) or counts (bottom). n = 11/8 (control, **a/b**) or n = 9/7 (clodronate, **a/b**) mice. Within each experiment, clodronate liposome-treated mice were normalized to control liposome-treated mice. **c-e,** Perilesional myeloid cell (IBA1) staining at 14 dpi, with images (**c**) and quantification via thresholding (**d**) and normalization to reactive astrocyte (GFAP)-traced lesion area (**e**). n = 7 (control) or n = 6 (clodronate) mice (2 slices/mouse). **f-h,** Quantification of lesional Col1a1^GFP+ fibroblast total density (**f**) and density within inner core (**g**) or fibrotic border (**h**) in control or clodronate-treated mice. 14 dpi; n = 8 (control) or n = 4 (clodronate) mice (2 slices/mouse). **i,** Spatial transcriptomic feature plots from Visium dataset (Fig. 1) showing *Ccr2* and

*Tgfb1* transcripts in perilesional distribution at 2 dpi. **j-l,** Feature plot of *Tgfb1*, plotted in order of expression, from mouse PT injury snRNAseq (**j**), marmoset stroke snRNAseq (**k**), or human TBI snRNAseq (**l**); dotted lines highlight myeloid cells. **m,** Representative image showing recombined macrophages and microglia (TdT⁺) in a Cx3cr1^creER; Rosa26^TdT mouse (tamoxifen as in Fig. 4d; 14 dpi). **n-o,** Lesional images (**n**; dotted line shows lesion boundary) and ER-TR7 coverage (**o**, normalized per experiment) following genetic targeting of macrophage and microglial *Tgfb1* in Cx3cr1^creER; Tgfb1^flox/GFP-KO mice, 14 dpi. Controls were littermate Cx3cr1^creER; Tgfb1^flox/+ mice; tamoxifen as in Fig. 4d. n = 17 (control) or n = 15 (*Tgfb1* cKO) slices (2 slices/mouse; slice values/statistics shown with lighter colors). Multiple T-tests, Holm-Sidak correction (**a,b**); two-way Student's T-test (**d-h, o**). 14 μm slices; images represent two or more mice.

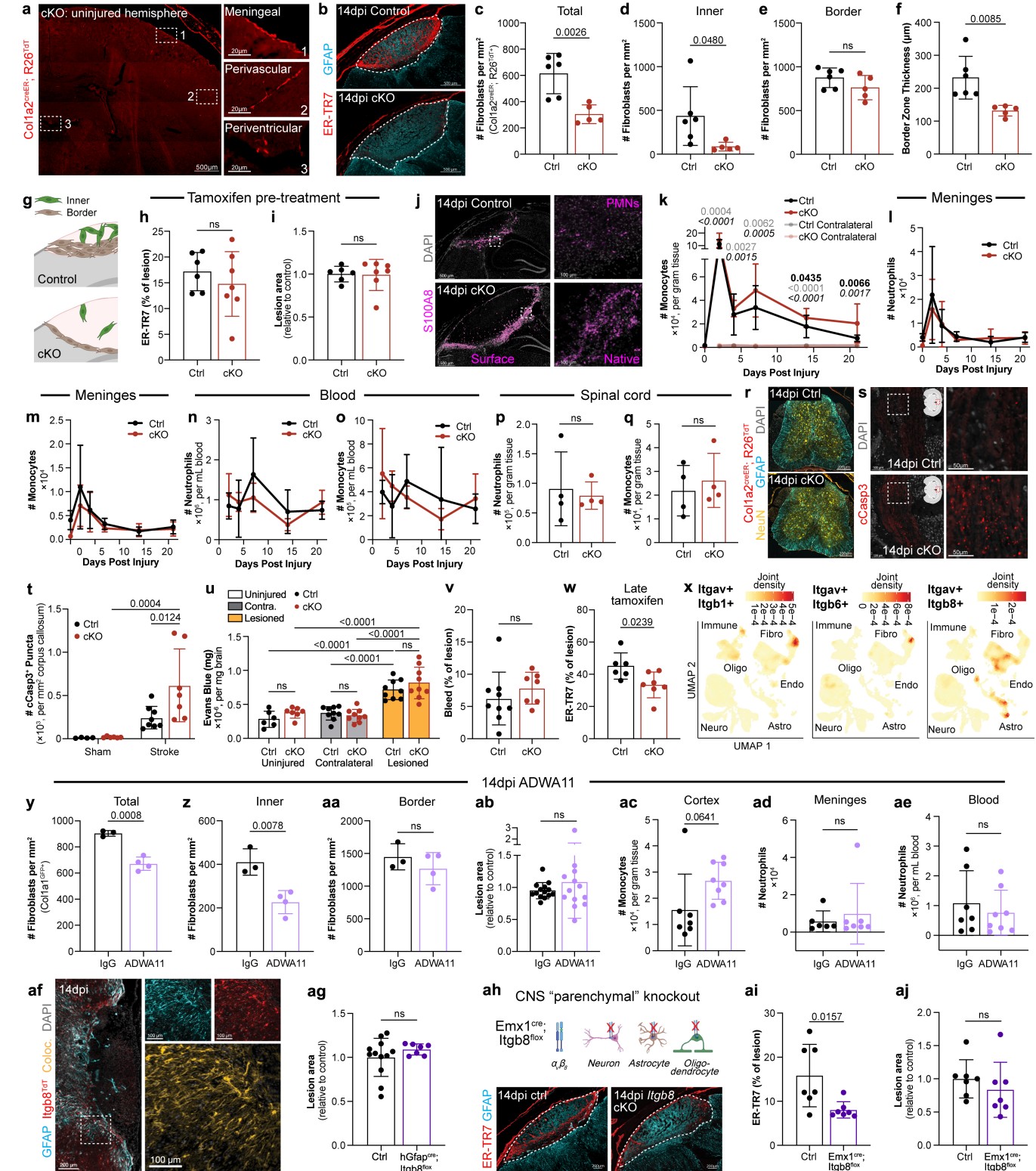

**Extended Data Fig. 8** | See next page for caption.

**Extended Data Fig. 8 | TGFβ drives CNS wound healing and resolution of inflammation, related to Fig. 4. a**, Intact fibroblast topography in uninjured cKO (Col1a2$^{creER}$; Tgfbr2$^{flox}$) parenchyma (insets: leptomeningeal/perivascular/periventricular fibroblasts). **b**, Lesional images showing ER-TR7 in controls or cKOs, 14dpi (tamoxifen as in Fig. 4d; dotted line shows lesion boundary). **c-g**, Quantification of 14dpi lesional Col1a2$^{creER}$; Rosa26$^{TdT+}$ fibroblast total density (**c**) and density within in inner core (**d**) or fibrotic border (**e**), with cKOs showing overall and inner fibroblast reduction with an intact but thinner lesional border (**f**), summarized via schematic (**g**), 14dpi. n = 6(control) or n = 5(cKO) slices/group (2 slices/mouse). **h-i**, ER-TR7 coverage (**h**) and lesion size (**i**, normalized per experiment) in control or cKO mice after tamoxifen pre-treatment only (tamoxifen days $^{-}16–^{-}14$) without follow-up induction, 14dpi; n = 6(control) or n = 7(cKO) mice. **j**, S100A8$^+$ neutrophils (PMN; left, surfaced; right, native) in controls (top) or cKOs (bottom), 14dpi (tamoxifen as in Fig. 4d). **k**, Time course showing cortical monocytes in controls/cKOs. n = 8(0dpi-control [non-littermate sham]), n = 6(2/4/7dpi-control, 2/7/21dpi-cKO), n = 12(14dpi-control), n = 9(21dpi-control), n = 3(0dpi-cKO), n = 5(4dpi-cKO), or n = 7(14dpi cKO) mice. **l-q**, Neutrophil or monocyte counts in meninges (**l-m**), blood (**n-o**), or 14dpi spinal cord (**p-q**) of controls/cKOs. n = 5(0dpi control [non-littermate sham], 7dpi cKO), n = 3(0dpi cKO), n = 6(2/4/7dpi control, 2/21dpi cKO), n = 12(14dpi control), n = 9(21dpi control), n = 5(**l-m**, 4dpi cKO), n = 6(**n-o**, 4dpi cKO), or n = 7(14dpi cKO) mice (**l-o**); n = 4 mice/group (**p-q**). **r**, Control/cKO spinal cords showing intact neurons without fibrosis/gliosis, 14dpi. **s-t**, Image (**s**) and quantification (**t**) of cCasp3$^+$ puncta within corpus callosum in control or cKO mice, 14dpi. n = 4(sham-control), n = 6(sham-cKO), n = 8(stroke-control),

or n = 7(stroke-cKO) mice (2–4 slices/mouse). **u**, Evans Blue extravasation in uninjured (resting or sham) or contralateral/lesioned hemispheres of controls/cKOs, 4dpi. n = 6(uninjured-control), n = 7(uninjured-cKO), or n = 9(contralateral/lesioned-control and cKO) mice. **v**, Quantification of overt bleeding in controls/cKOs, 4dpi. n = 9(control) or n = 7(cKO) mice. **w**, ER-TR7 coverage in control or cKO mice induced with late tamoxifen (beginning 14dpi), 28dpi. n = 6(control) or n = 7(cKO) mice. **x**, Co-expression of TGFβ-activating integrin pairs *Itgav* and *Itgb1*, *Itgb6*, or *Itgb8* (snRNAseq). **y-aa**, Quantification of lesional Col1a1$^{GFP+}$ fibroblast total density (**y**) and density within in inner core (**z**) or fibrotic border (**aa**) in IgG- or ADWA11-treated mice, 14dpi. n = 3(IgG) or n = 4(ADWA11) mice (2 slices/mouse). **ab-ae**, Lesion size (**ab**, normalized per experiment), cortical monocytes (**ac**), and meningeal (**ad**) or blood neutrophils (**ae**) in IgG- and ADWA11-treated mice, 14dpi. n = 14(IgG) or n = 13(ADWA11) mice (**ab**, 2 slices/mouse); n = 7(IgG) or n = 8(ADWA11) mice (**ac**,**ae**); n = 6(IgG) or n = 7(ADWA11) mice (**ad**). **af**, Itgb8$^{TdT}$ mouse, with perilesional/astrocytic *TdT*, 14dpi (insets: GFAP/TdT colocalization, yellow). **ag**, Lesion size in hGfap$^{cre}$; Itgb8$^{flox}$ mice and controls (normalized per experiment), 14dpi. n = 12(control) or n = 7(cKO) mice (2 slices/mouse). **ah-aj**, Control or Emx1$^{cre}$; Itgb8$^{flox}$ lesional images (**ah**), ER-TR7 coverage (**ai**), and lesion size (**aj**, normalized per experiment), 14dpi. n = 7 mice/group (2 slices/mouse). Two-way Student's T-test (**c-f**,**h**,**i**,**k** [0dpi],**p**,**q**,**v**,**w**,**y-ae**,**ag**,**ai**,**aj**); two-way ANOVA, Sidak's post-test (**k** [repeated measures, 7–21dpi, per timepoint; bold=lesioned-control/lesioned-cKO, grey=lesioned-control/contralateral-control, italics=lesioned-cKO/contralateral-cKO],**t**,**u**); multiple two-way T-tests, Holm-Sidak correction (**l-o**). 14μm slices; images represent two or more mice.

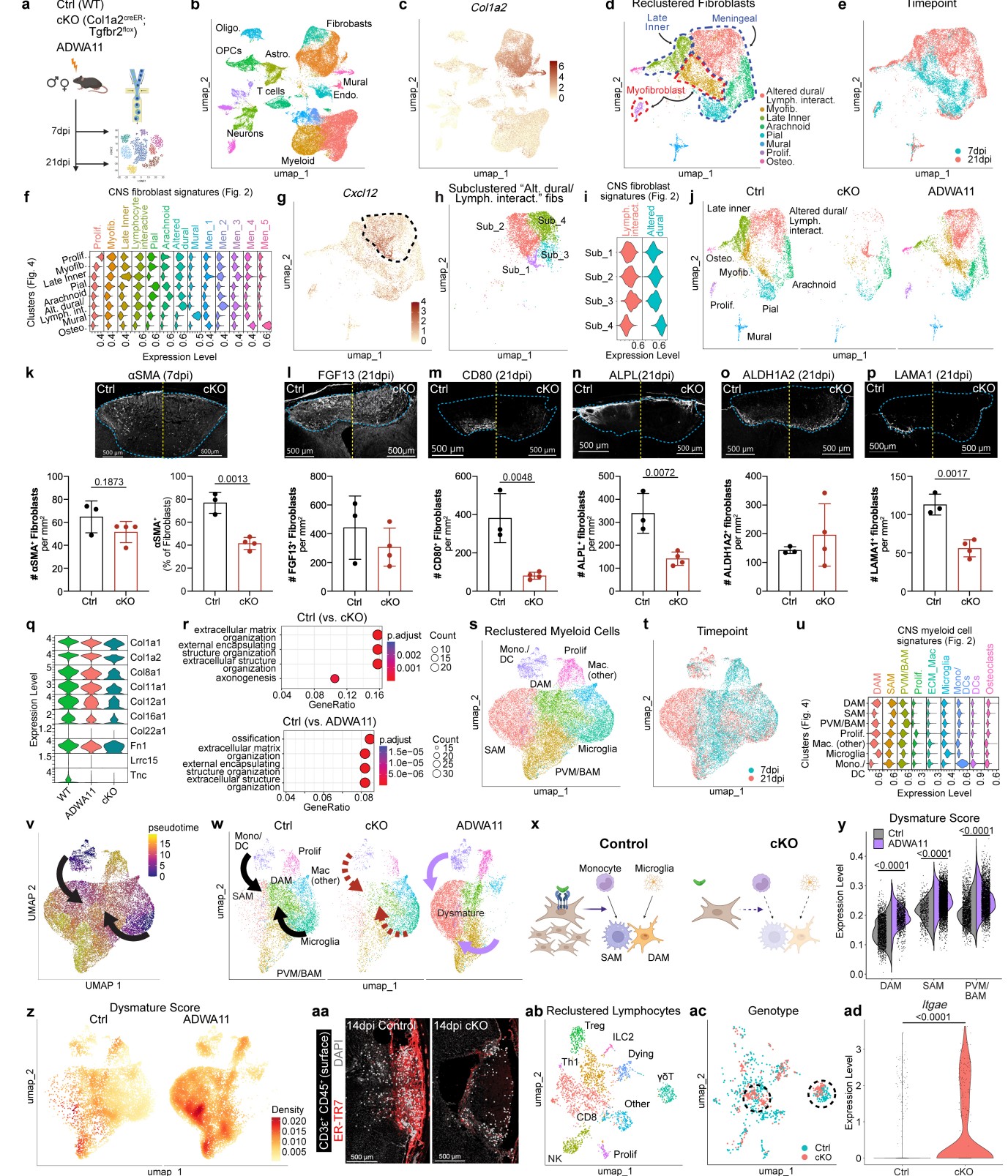

**Extended Data Fig. 9 | See next page for caption.**

**Extended Data Fig. 9 | Additional molecular characterization of injury response after fibroblast impairment, related to Fig. 4. a**, snRNAseq schematic. Control (WT), cKO (Col1a2$^{creER}$; Tgfbr2$^{flox}$), and $\alpha_v\beta_8$-blocking antibody (ADWA11)-treated mice were harvested at 7 and 21dpi (2 mice per timepoint/condition), lesions were micro-dissected, and nuclei were sorted (DAPI$^+$) and sequenced (final library 60,070 nuclei). **b-c**, Global UMAPs from snRNAseq data showing indicated cell types (**b**) or *Col1a2* expression (**c**). **d-e**, UMAPs (18,455 fibroblasts and 496 mural cells) with fibroblast subclusters (**d**) or timepoint (**e**). **f**, Violin plots mapping "CNS fibroblast signatures" (derived from Fig. 2, x-axis) onto clusters identified in **d** (y-axis). **g-i**, Characterization of "Altered dural/lymphocyte-interactive" fibroblasts, shown as highlighted cluster in *Cxcl12* feature plot (**g**), UMAP with subclusters (**h**), and violin plots showing subcluster expression of "Lymphocyte-interactive" and "Altered-dural" signatures (**i**, derived from Fig. 2), highlighting 3 lymphocyte-interactive-like and 1 altered-dural-like subcluster(s). **j**, Fibroblast UMAP separated by condition, with pan-fibroblast reduction in cKO mice and late-inner fibroblast reduction in $\alpha_v\beta_8$-blocked (ADWA11) mice. **k-p**, Images (top) and quantification (bottom) of fibroblast subset marker expression among Col1a2$^{creER}$ R26$^{TdT}$ fibroblasts in control or cKO mice. 7dpi cKOs show a decrease in $\alpha$SMA$^+$ myofibroblasts (**k**, quantified as myofibroblast density [left] or proportion [right]). 21dpi cKOs show a trending decrease in FGF13$^+$ late inner fibroblasts (**l**) and decreases in CD80$^+$ lymphocyte interactive fibroblasts (**m**), ALPL$^+$ altered dural fibroblasts (**n**),

and LAMA1$^+$ pial fibroblasts (**p**), though ALDH1A2$^+$ leptomeningeal/arachnoid fibroblasts are preserved (**o**). Blue dotted lines show lesion border. n = 3(control) or n = 4(cKO) mice. **q**, Violin plots showing expression of myofibroblast-related genes among the myofibroblast cluster. **r**, Gene set enrichment among the myofibroblast cluster, with controls showing ECM-process enrichment. **s-t**, UMAPs showing myeloid cell subclusters (**s**) or timepoint (**t**). **u**, Violin plots mapping expression of "CNS myeloid cell signatures" (derived from Fig. 2, x-axis) onto clusters identified in **s** (y-axis). **v-w**, Pseudotime (**v**, Monocle3) and UMAP plots separated by condition (**w**) showing potential myeloid differentiation trajectories, including monocyte-to-SAM (top) and microglia-to-DAM (bottom), with monocytes and microglia as root states. **x**, Schematic showing effect of TGFβ-activated myofibroblasts on SAM/DAM formation. **y-z**, Violin plots (**y**) or feature plots (**z**) showing "dysmature" transcriptional signature across conditions. n = 1841/1281(Ctrl/ADWA11 DAM), n = 573/5551(Ctrl/ADWA11 SAM), or n = 1724/3181(Ctrl/ADWA11 PVM/BAM) nuclei (**y**). **aa**, Surfaced T cells (CD3ε$^+$; CD45$^+$) in control and cKO mice, with fibroblast-rich (ER-TR7$^+$) lesion. **ab-ac**, UMAP showing lymphocyte subclusters (**ab**) or genotype (**ac**, Ctrl and cKO). Dotted black lines highlight CD8 and γδT cells. **ad**, Violin plot of lymphocyte *Itgae* expression. n = 495(WT) or n = 336(cKO) nuclei. Two-way Student's T-test (**k-p**); over-representation test (one-sided Fisher's exact test, FDR-adjusted, **r**); two-way Mann-Whitney test (**y** [Bonferroni correction], **ad**). 14μm slices; images represent two or more mice.

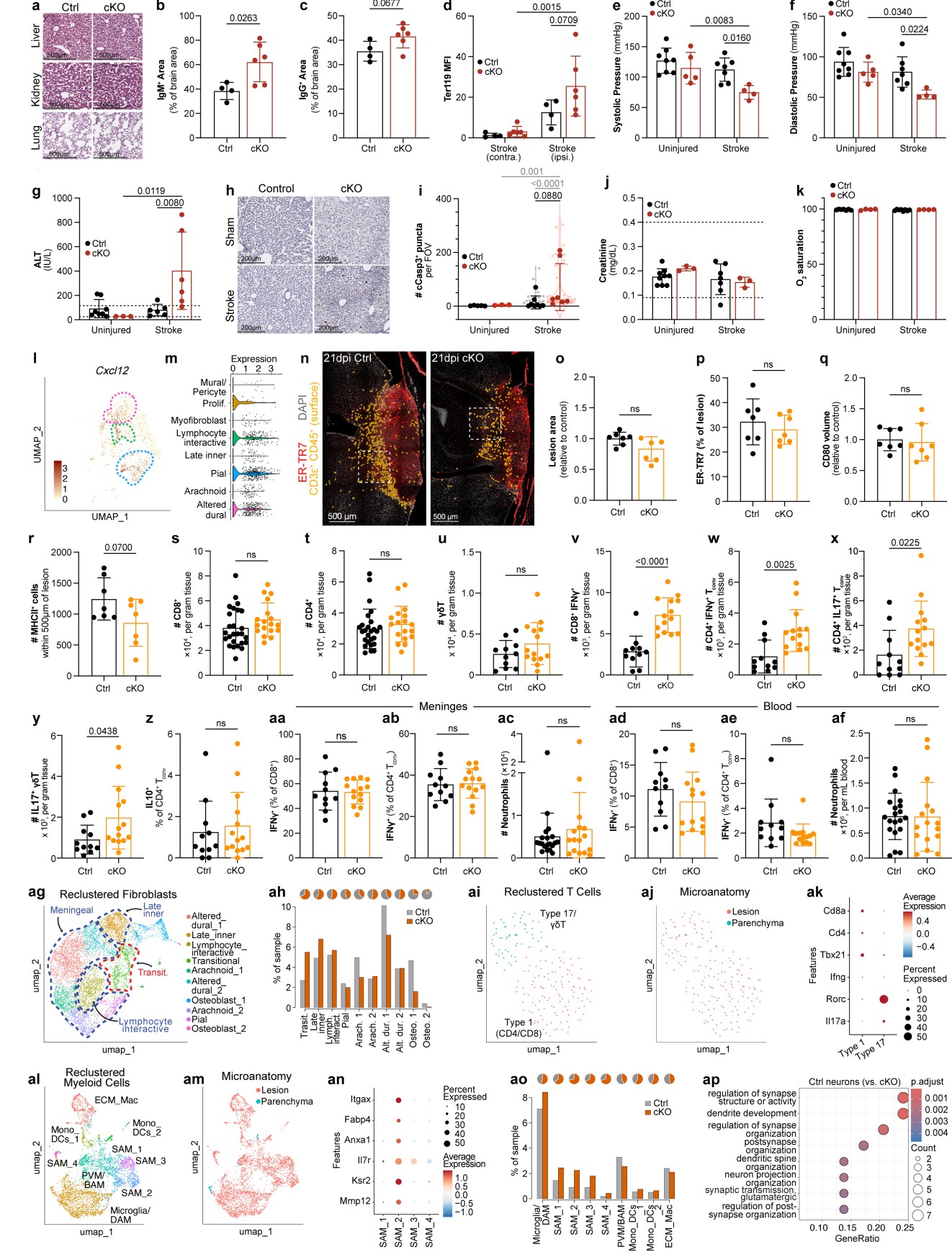

**Extended Data Fig. 10** | See next page for caption.

**Extended Data Fig. 10 | Additional characterization of *Tgfbr2* and *Cxcl12* cKO mice, related to Fig. 5. a**, Hematoxylin and eosin staining of liver, kidney, and lung tissue in control and cKO (Col1a2$^{creER}$; Tgfbr2$^{flox}$) mice without brain injury. Images representative of n = 2 mice/group. **b-d**, Area positive for mouse IgM (**b**) or IgG (**c**), or MFI of Ter119 (**d**) in control or cKO mice (3dpi, tMCAO). n = 4(control) or n = 6(cKO) mice. **e-f**, Systolic (**e**) or diastolic (**f**) blood pressure in control or cKO mice (1–3dpi, tMCAO). n = 8(control-uninjured), n = 5(cKO-uninjured), n = 7(control-stroke), or n = 4(cKO-stroke) mice; 1 cKO-stroke recorded 1dpi, 1 cKO-stroke recorded 2dpi, remaining mice recorded 3dpi. **g**, Serum levels of ALT in control or cKO mice (3dpi, tMCAO). Dotted lines indicate bounds of normal range. n = 8(control-uninjured), n = 3(cKO-uninjured), or n = 6(control-stroke, cKO-stroke) mice. **h-i**, Representative images (**h**) and quantification (**i**) of cCasp3$^+$ liver cells/puncta in control or cKO mice (3dpi, tMCAO). n = 5(control-uninjured), n = 3(cKO-uninjured), n = 8(control-stroke), or n = 7(cKO-stroke) mice (5 FOVs averaged per mouse; FOV values/statistics shown with lighter colors). **j**, Serum levels of creatinine in control or cKO mice (3dpi, tMCAO). Dotted lines indicate bounds of normal range. n = 9(control-uninjured), n = 3(cKO-uninjured), n = 7(control-stroke), or n = 3(cKO-stroke) mice. **k**, O$_2$ saturation (pulse oximetry) in control or cKO mice (1–3dpi, tMCAO). n = 7(control-uninjured), n = 4(cKO-uninjured, cKO-stroke), or n = 8(control-stroke) mice. 1 cKO-stroke recorded 1dpi, 1 cKO-stroke recorded 2dpi, remaining mice recorded 3dpi. **l-m**, Feature plot (**l**) or violin plot (**m**) showing *Cxcl12* expression among late fibroblast subsets including lymphocyte interactive, pial, and altered dural fibroblasts. **n-p**, Surfaced images showing T cells (CD3ε$^+$; CD45$^+$) near ER-TR7$^+$ lesion in control or *Cxcl12* cKO mice (**n**), with quantification

of lesion size (**o**, normalized per experiment) and ER-TR7 coverage (**p**), 21dpi. n = 7 mice/group (2 slices/mouse, **o**). **q**, Volume of surfaced CD80$^+$ cells in control or *Cxcl12* cKO mice, 21dpi. n = 7 mice/group. **r**, MHCII$^+$ cells within 500 μm of lesion, 21dpi. n = 7 mice/group. **s-z**, Quantification of total cortical CD8$^+$ T cells (**s**), CD4$^+$ T cells (**t**), or γδ T cells (**u**); IFNγ$^+$ CD8$^+$ T cells (**v**); IFNγ$^+$ CD4$^+$ conventional T (T$_{conv}$) cells (**w**) or IL17$^+$ CD4$^+$ T$_{conv}$ cells (**x**); IL17$^+$ γδ T cells (**y**); and IL10 + CD4$^+$ T$_{conv}$ cells (**z**), 21dpi. n = 26(control) or n = 18(*Cxcl12* cKO) mice (**s,t**); n = 11(control) or n = 14(*Cxcl12* cKO) mice (**u-z**). **aa-af**, Expression of IFNγ among CD8$^+$ or CD4$^+$ T cells and neutrophil counts in meninges (**aa-ac**) or blood (**ad-af**). n = 11(control) or n = 14(*Cxcl12* cKO) mice (**aa,ab,ad,ae**); n = 20(control) or n = 16(*Cxcl12* cKO) mice (**ac,af**). **ag**, Fibroblast UMAP (7,954 nuclei) showing subclusters annotated according to fibroblast identity mapping (e.g., Extended Data Fig. 9f). **ah**, Fibroblast cluster abundance in control and *Cxcl12* cKO mice. **ai-ak**, T cell UMAP plots showing type 1 and 17 clusters (**ai**) or microanatomy (**aj**), and dotplot showing expression of relevant distinguishing marker genes (**ak**). **al-am**, Myeloid UMAP plots (3,792 nuclei), annotated according to myeloid identity mapping (e.g., Extended Data Fig. 9u) with subclustered SAM (**al**), or annotated by microanatomy (**am**). **an**, Dotplot showing expression of inflammatory/M1-like genes across SAM subsets, with higher expression in SAM_2. **ao**, Myeloid cluster abundance, including SAM subclusters, in control and *Cxcl12* cKO mice. **ap**, Gene set enrichment among control (vs. *Cxcl12* cKO) neurons. Two-way Student's T-test (**b,c,o-af**); two-way ANOVA, Sidak's post-test (**d** [repeated measures]; **e-g,i-k**); over-representation test (one-sided Fisher's exact test, FDR-adjusted, **ap**). 14μm slices; images represent two or more mice.

Ari B. Molofsky

# Reporting Summary

## Statistics

For all statistical analyses, confirm that the following items are present in the figure legend, table legend, main text, or Methods section.

| n/a | Confirmed | |
|---|---|---|
| ☐ | ☒ | The exact sample size (*n*) for each experimental group/condition, given as a discrete number and unit of measurement |
| ☐ | ☒ | A statement on whether measurements were taken from distinct samples or whether the same sample was measured repeatedly |
| ☐ | ☒ | The statistical test(s) used AND whether they are one- or two-sided *Only common tests should be described solely by name; describe more complex techniques in the Methods section.* |
| ☒ | ☐ | A description of all covariates tested |
| ☐ | ☒ | A description of any assumptions or corrections, such as tests of normality and adjustment for multiple comparisons |
| ☐ | ☒ | A full description of the statistical parameters including central tendency (e.g. means) or other basic estimates (e.g. regression coefficient) AND variation (e.g. standard deviation) or associated estimates of uncertainty (e.g. confidence intervals) |
| ☐ | ☒ | For null hypothesis testing, the test statistic (e.g. *F*, *t*, *r*) with confidence intervals, effect sizes, degrees of freedom and *P* value noted *Give P values as exact values whenever suitable.* |
| ☒ | ☐ | For Bayesian analysis, information on the choice of priors and Markov chain Monte Carlo settings |
| ☒ | ☐ | For hierarchical and complex designs, identification of the appropriate level for tests and full reporting of outcomes |
| ☒ | ☐ | Estimates of effect sizes (e.g. Cohen's *d*, Pearson's *r*), indicating how they were calculated |

*Our web collection on statistics for biologists contains articles on many of the points above.*

## Software and code

Policy information about availability of computer code

| Data collection | Imaging data was collected using NIS-Elements v5.11.03 (Nikon), Zen Pro 2 v2.0.0.0 (Zeiss), or VS200 ASW v3.4.1 (Olympus). Flow cytometry data was collected using FACSDiva v9.0 (BD Biosciences). Pulse oximetry data was collected using the MouseOx Pulse Oximeter system (STARR Life Sciences). Blood pressure data was collected using the CODA-HT4 Noninvasive Blood Pressure System (Kent Scientific). |
|---|---|
| Data analysis | Images were analyzed with Imaris v9.8.0 (Oxford Instruments) or ImageJ v2.1.0/1.53c (NIH). Flow cytometry data was analyzed with FlowJo v10.7.2 (BD Biosciences). Graphs were created using Prism v10.1.1 (GraphPad Software, Inc). |

Single cell RNA sequencing data was aligned to the mouse genome mm10 or Grcm39 using SpaceRanger v2.0.0 and CellRanger v7.1.0, v7.2.0, or v9.0.0 (10X Genomics). Further RNAseq analysis was performed using R version 4.3.2 in RStudio v2023.03.1, with Seurat version 4.2.1, 5.0.1, or 5.2.1. Additional packages used include Presto (v1.0.0), EnhancedVolcano (v1.20.0), Nebulosa (v1.12.0), ScCustomize (v2.0.1), clusterProfiler (v4.10.0), nichenetr (v2.0.5), spacexr (v2.2.1), monocle3 (v1.3.4), DESeq2 (v1.42.0), dplyr (v1.1.4), ply (v1.8.9), ape (v5.7-1), cowplot (v1.1.2), Matrix (v1.6-4), variancePartition (v1.32.2), MAST (v1.28.0), HGNChelper (v0.8.1), openxlsx (v4.2.5.2), RColorBrewer (v1.1-3), gridExtra (v2.3), ggpubr (v0.6.0), ComplexHeatmap (v2.18.0), tidyverse (v2.0.0), tibble (v3.2.1), biomaRt (v2.58.0), data.table (v1.14.10), glmGamPoi (v1.14.0), SeuratWrappers (v0.3.2), patchwork (v1.1.3), magrittr (v2.0.3), s2 (v1.1.6), gplots (v3.1.3), stringr (v1.5.1), ggnewscale (v0.4.9), ggbreak (v0.1.4), coin (v1.4-3), and dunn.test (v1.3.6). Custom code used for single cell RNA sequencing analysis is available at GitHub ("https://github.com/newingcrystal/CNS_Fibroblasts"). CellPhoneDB (cellphonedb package v4.1.0) utilized Python v3.11.0 in Jupyter Notebook.

For manuscripts utilizing custom algorithms or software that are central to the research but not yet described in published literature, software must be made available to editors and reviewers. We strongly encourage code deposition in a community repository (e.g. GitHub). See the Nature Portfolio guidelines for submitting code & software for further information.

## Data

Murine spatial and single nuclear transcriptomic data generated in this paper are deposited in Gene Expression Omnibus (GEO) under the accession number GSE254164. Mouse genomes were downloaded via 10X Genomics, including Mm10 (https://cf.10xgenomics.com/supp/cell-exp/refdata-gex-mm10-2020-A.tar.gz) and Grcm39 (https://cf.10xgenomics.com/supp/cell-exp/refdata-gex-GRCm39-2024-A.tar.gz). The raw data from Boghdadi et al. (marmoset stroke) is available at GSE179141 (https://www.ncbi.nlm.nih.gov/geo/query/acc.cgi?acc=GSM5410579, Series 1). The raw data from Garza et al. (human TBI) is available at GSE209552 (https://www.ncbi.nlm.nih.gov/geo/query/acc.cgi?acc=GSE209552). The raw data from Jain et al. (human GBM) is available at GSE132825 (https://www.ncbi.nlm.nih.gov/geo/query/acc.cgi?acc=GSE132825). The raw data from Keren-Shaul et al., used to generate DAM scores, is available at GSE98971 (https://www.ncbi.nlm.nih.gov/geo/query/acc.cgi?acc=GSE98971, SubSeries GSE98969). The raw data from Yin et al., used to generate dysmaturity scores, is available at GSE239603 (https://www.ncbi.nlm.nih.gov/geo/query/acc.cgi?acc=GSE239603, SubSeries GSE234496). The data from Hobson et al., used to generate IFNg scores in neurons, is available at doi:10.1016/j.bbi.2023.04.008 (see Supplementary Table 1). The genes from Sbierski-Kind et al., used to generate TGFb/Myofibroblast scores, are supplied as Supplementary Table 1. The genes from Mroz et al., used to generate IFNg scores in myeloid cells, are supplied as Supplementary Table 2.

## Research involving human participants, their data, or biological material

| | |
|---|---|
| Reporting on sex and gender | Not applicable |
| Reporting on race, ethnicity, or other socially relevant groupings | Not applicable |
| Population characteristics | Not applicable |
| Recruitment | Not applicable |
| Ethics oversight | Not applicable |

Note that full information on the approval of the study protocol must also be provided in the manuscript.

# Field-specific reporting

Please select the one below that is the best fit for your research. If you are not sure, read the appropriate sections before making your selection.

☒ Life sciences  ☐ Behavioural & social sciences  ☐ Ecological, evolutionary & environmental sciences

For a reference copy of the document with all sections, see nature.com/documents/nr-reporting-summary-flat.pdf

# Life sciences study design

All studies must disclose on these points even when the disclosure is negative.

| | |
|---|---|
| Sample size | Sample sizes were estimated based on standard power calculations (a=0.05, 80% power) performed for similar published experiments. In general, statistical calculations were not used to predetermine sample sizes. |
| Data exclusions | All exclusions are noted in Methods and are based on pre--established criteria unless otherwise noted. For lesion size quantification, outliers were excluded if lesion diameter was visualized to be <25% of the average, representing technical errors during injury, resulting in a single exclusion (see Methods; established prior to analysis/unblinding). Mice with hydrocephalus were also excluded. Exclusion criteria for tMCAO survival analysis include death during surgery, cohorts with mice showing weight <=75% of expected at time of surgery (established based on pilot data), or cohorts with excessive bleeding. For serum analysis, severely hemolyzed samples were excluded according to pre-established ULAM and manufacturer guidelines. |
| Replication | Select experiments were performed once, for reasons including breeding, cost, and technical constraints, including the following: tMCAO in Cthrc1(creER); R26(TdT) mice; analysis of spinal cords of Tgfbr2 cKO mice; analysis of cortical scar-associated macrophages in Tgfbr2 cKO mice; meningeal fibroblast coculture (using fibroblasts sorted from 18 mice); quantification of recombination in Ng2creER mice; and quantification of Col1a1GFP+ fibroblasts after ADWA11 treatment. RNA sequencing experiments contained two pooled mice per sample. All other experiments were performed at least 2 times with successful reproduction. |
| Randomization | Littermate mice were randomized between experimental groups by sex and cage; for experiments involving more than one litter, each litter |

| Randomization | was equivalently randomized between experimental groups. Experimental groups were age and sex matched as possible. Littermate controls were used for genetic experiments. |
|---|---|
| Blinding | Mice remained unblinded during experiments for the purpose of cohort assignment and experimental treatment. Mice were assigned a numeric ID which was used throughout data collection/analysis and matched to experimental group after data analysis. For imaging studies, brains were blinded by an independent researcher at the time of tissue freezing; slicing, staining, imaging, and image quantification were performed prior to unblinding. |

# Reporting for specific materials, systems and methods

We require information from authors about some types of materials, experimental systems and methods used in many studies. Here, indicate whether each material, system or method listed is relevant to your study. If you are not sure if a list item applies to your research, read the appropriate section before selecting a response.

## Materials & experimental systems

| n/a | Involved in the study |
|---|---|
| ☐ | ☒ Antibodies |
| ☒ | ☐ Eukaryotic cell lines |
| ☒ | ☐ Palaeontology and archaeology |
| ☐ | ☒ Animals and other organisms |
| ☒ | ☐ Clinical data |
| ☒ | ☐ Dual use research of concern |
| ☒ | ☐ Plants |

## Methods

| n/a | Involved in the study |
|---|---|
| ☒ | ☐ ChIP-seq |
| ☐ | ☒ Flow cytometry |
| ☒ | ☐ MRI-based neuroimaging |

## Antibodies

| Antibodies used | Primary antibodies used for murine imaging include chicken anti-GFP (Aves Labs GFP-1020, 1:200), rabbit anti-dsRed (Takara 632496, 1:300), chicken anti-GFAP (Invitrogen PA1-10004, 1:200 or 1:500), rat anti-GFAP (2.2B10, Invitrogen 13-0300, 1:200), rat anti-ER-TR7 (Novus Biologicals NB100-64932, 1:200), rabbit anti-aSMA (Abcam ab5694, 1:300), rat anti-CD31 (MEC13.3, Biolegend 102514, 1:200), goat anti-Desmin (GenWay Biotech GWB-EV0472, 1:200), rat anti-PDGFRb (APB5, Invitrogen 14-1402-82, 1:500), rabbit anti-NG2 (Millipore Sigma ab5320, 1:200), goat anti-Decorin (Novus Biologicals AF1060, 1:200), goat anti-Collagen 1 (Southern Biotech 1310-01, 1:500), rabbit anti-Collagen 6a1 (Novus Biologicals NB120-6588, 1:200 or 1:500), rat anti-Periostin (345613, Novus Biologicals MAB3548, 1:200), rat anti-ICAM1 (YN1/1.7.4, Biolegend 116110, 1:200), syrian hamster-anti-CD3e (500A2, BD Biosciences 553238, 1:200), goat anti-S100A8 (R&D Systems AF3059, 1:200), chicken anti-NeuN (Millipore Sigma ABN91, 1:200), rabbit anti-Iba1 (Aif3, Fujifilm Wako 019-19741, 1:200-1:1000), mouse anti-FGF13 (N235/22, Invitrogen MA5-27705, 1:100), goat anti-CD80 (R&D Systems AF740, 1:200), goat anti-ALPL (Novus Biologicals AF2910, 1:50), rabbit anti-LAMA1 (EPR27258-37, Abcam ab307542, 1:200), rabbit anti-SEMA3C (Invitrogen PA5-103168, 1:100), rabbit anti-CDH18 (Invitrogen PA5-112902, 1:50), rabbit anti-ALDH1A2 (Novus Biologicals NBP2-92915, 1:200), goat anti-SOX10 (R&D Systems AF2864, 1:300), rabbit anti-ASPA (Genetex GTX113389, 1:1000), mouse anti-E-Cadherin (Clone 36, BD Biosciences 610181), rat anti-I-A/I-E (MHCII, M5/114.15.2, eBioscience 14-5321-82), mouse anti-Ly76 (TER119, Biolegend 116232, 1:200), goat anti-mouse IgM (Invitrogen 31172, 1:200), and rabbit anti cleaved Caspase 3 (Cell Signaling Technology 9661T, 1:400; Cell Signaling Technology 9991, Histowiz). For marmoset imaging, rabbit anti-COL6 (Abcam ab6588, 1:500) was used.

Secondary antibodies were used at 1:500 (for thin sections) and 1:1000 (for thicker sections), as specified in relevant Methods sections. Secondary antibodies used include Donkey anti-rat IgG AF488 (Thermo Scientific A21208), Donkey anti-rat IgG AF555 (Thermo Scientific A78945), Donkey anti-rat IgG AF647 (Abcam ab150155), Donkey anti-rabbit IgG AF488 (Thermo Scientific A21206), Donkey anti-rabbit IgG AF555 (Thermo Scientific A31572), Donkey anti-rabbit IgG AF647 (Thermo Scientific A31573), Donkey anti-goat IgG AF488 (Thermo Scientific A11055), Donkey anti-goat IgG AF555 (Thermo Scientific A21432), Donkey anti-goat IgG AF647 (Thermo Scientific A21447), Donkey anti-chicken IgG AF488 (Sigma, SAB4600031-250UL), Donkey anti-chicken IgG AF647 (Thermo Scientific A78952), Goat anti-rat IgG AF488 (Thermo Scientific A11006), Goat anti-rabbit IgG AF555 (Thermo Scientific A21429), Goat anti-rabbit IgG AF647 (Thermo Scientific A21245), Goat anti-hamster IgG AF647 (Thermo Scientific A21451), Donkey anti-mouse IgG AF647 (Thermo Scientific A31571).

Antibodies used for flow cytometry include rabbit anti-Olig2 (Thermo Scientific P21954, 1:100), anti CD45 (30-F11, BD Biosciences 564279 or Biolegend 103132 or 103104, 1:400), anti-CD90.2 (Thy1, 53-2.1, Biolegend 140327, BD Biosciences 553004, 1:200), anti-CD11b (M1/70, Biolegend 101224 or BD Biosciences 563015, 1:400), anti-CD19 (6D5, Biolegend 115554, 1:400), anti-NK1.1 (PK136, Biolegend 108736, 1:200), anti-CD3e (17A2, Biolegend 100216, 1:200), anti-CD4 (RM4-5, Biolegend 100557, or GK1.5, BD Biosciences 563050, 1:200), anti-CD8⍺ (53-6.7, Biolegend 100750, 1:200), anti-CD44 (IM7, Biolegend 103030, 1:200), anti-CD69 (H1.2F3, Biolegend 104505, 1:200), anti-CD62L (MEL-14, Biolegend 104407, 1:200), anti-Tbet (4B10, Biolegend 25-5825-80, 1:100), anti-Gata3 (TWAJ, eBioscience 12-9966-41, 1:100), anti-RORgt (B2D, eBioscience 17-6981-82, 1:100), anti-Ly6G (1A8, Biolegend 127624, 1:200), anti-IFNg (XMG1.2, Biolegend 505810, 1:100), anti-IL17A (TC11-18H10.1, Biolegend 506922, 1:100), anti-IL10 (JES5-16E3, eBioscience 12-7101-81, 1:100), anti-TCRgd (Biolegend 118118, 1:200 [extracellular] or 1:400 [intracellular]), anti-FoxP3 (eBioscience 53-5773-82, 1:100), anti-Ly6C (HK1.4, Biolegend 128011 or 128035, 1:400), anti-CD64 (X-54-5/7.1, Biolegend 139323 or BD Biosciences 558539, 1:200), anti-MERTK (DS5MMER, eBioscience 46-5751-80, 1:200), anti-CD9 (KMC8, BD Biosciences 564235, 1:200), anti-TREM2 (237920, R&D systems FAB17291A, 1:200), anti-CD63 (NVG-2, Biolegend 143904, 1:200), anti-I-A/I-E (MHCII, M5/114.15.2, BD Biosciences 748845, 1:400), anti-CD11c (N418, Biolegend 117339 or 117318, 1:200), anti-CD172a (SIRPa, P84, eBioscience 12-1721-80, 1:200), anti-Siglec-F (E50-2440, BD Biosciences 740956, 1:200), anti-Podoplanin (gp38, 8.1.1, Biolegend 127412, 1:200), anti-CD31 (390, Biolegend 102404 or 102408, 1:200), anti-EpCAM (G8.8, Biolegend 118230, 1:200), anti-PDGFRa |
|---|---|

(APA5, Biolegend 135908, 1:200), anti-Sca-1 (Ly-6A/E, D7, Biolegend 108131, 1:200), anti-phosphoSMAD3 (EP823Y, Abcam ab52903, 1:50), and anti-CD16/32 (2.4G2, BD Biosciences 553142, 1:100 or 1:250).

| Validation | All antibodies were validated by the manufacturer per species (mouse or marmoset) and per application (immunofluorescent imaging, flow cytometry). |

# Animals and other research organisms

Policy information about studies involving animals; ARRIVE guidelines recommended for reporting animal research, and Sex and Gender in Research

| Laboratory animals | Mice (Mus musculus):

Fibroblast lineage tracing was performed using Col1a2creERT2 mice (MGI 6721050, from Bin Zhou, Institute of Biochemistry and Cell Biology, Shanghai Institutes for Biological Sciences) crossed with Rosa26TdT-Ai14 mice (R26-CAG-RFP, Jackson 007914) or with Rosa26Sun1GFP mice (Jackson 030952). For dMCAO experiments, a distinct Col1a2creERT allele was used (Jackson 029567). Additional stromal reporters used include Col1a1GFP mice (from David Brenner, University of California, San Diego); PdgfraGFP (PDGFRa-H2B-eGFP nuclear-localized GFP, Jackson 007669); and Rosa26Tdt-Ai14 mice crossed to Gli1creERT2 mice (Jackson 007913); Twist2cre mice (Jackson 008712); Acta2creERT2 mice; Ng2creER mice (Ng2creERTM, Jackson 008538); Atp13a5creERT2 mice;  and Cthrc1creER mice (generously provided by Dean Sheppard).

Immune cell lineage tracing was performed using Rosa26TdT-Ai14 mice crossed to CD4cre mice (Jackson 022071); Ccr2creERT2 mice (from Burkhard Becher, University of Zurich, Switzerland); P2ry12creERT2 (Jackson 034727); Cx3cr1creER (Jackson 020940); and PF4cre (Jackson 008535). We also used T-bet (Tbx21)-zsGreen transgenic mice (generously provided by Jinfang Zhu, Lab of Immune System Biology, NIH). Cx3cr1creER mice were additionally crossed with Tgfb1GFP mice (MGI 3719583) and Tgfb1flox mice (Jackson 033001) to create in Cx3cr1creER; Tgfb1GFP/flox mice.

To generate Tgfbr2 conditional knockout mice, Col1a2creERT2 mice were crossed to Tgfbr2flox mice (both Tgfbr2-exon2flox, MGI 238451373, and Tgfbr2-exon4flox, Jackson 012603). To generate Cxcl12 conditional knockout mice, Col1a2creERT2 mice were crossed to Cxcl12flox mice (Jackson 021773). To generate Itgb8 conditional knockout mice, Itgb8flox mice (MGI 3608910) were crossed to Emx1cre mice (Jackson 005628) or hGfapcre mice (Jackson 004600); some hGfapcre mice were crossed to iSurecre (MGI 6361135) to optimize Cre efficiency. We also used Itgb8TdT mice (Itgb8-IRES-TdT, generously provided by Helena Paidassi). To generate myofibroblast deleter mice,  Cthrc1creER mice were crossed to Rosa26DTA mice.

Mice were mixed gender animals backcrossed on C57BL/6 for at least 10 generations, or on a mixed genetic background (Gli1creERT2, Cx3cr1creER, Emx1cre). If not otherwise stated, all experiments were performed with 7-21 week old male and female mice. Controls are defined in relevant figure legends and in Methods. All mice were bred and maintained in specific-pathogen-free conditions,  at 25°C and ambient humidity under a 12-hour day/night cycle, at the animal facilities of UCSF or UCSD.

Marmosets (Callithrix jacchus):

Outbred middle-aged marmoset monkeys (>5 years; median age ~7 years) were used in this study. No siblings were used. Animals were housed in family groups (12:12 hrs light/dark cycle, temperature 31°C, humidity 65%). Marmosets were obtained from the National Nonhuman Primate Breeding and Research Facility (Monash University, Australia). |

| Wild animals | No wild mice were used in this study. |

| Reporting on sex | Male and female mice were used for all experiments and data was analyzed by sex; as no sex-specific trends emerged, data is not reported separately by sex. |

| Field-collected samples | No field collected samples were used in this study. |

| Ethics oversight | All mice were bred and maintained in specific-pathogen-free conditions at the animal facilities of UCSF or UCSD and were used in accordance with institutional guidelines and under study protocols approved by the UCSF or UCSD Institutional Animal Care and Use Committee (protocols AN193180-01J, AN195716-01B [UCSF], s14044 [UCSD]). Marmoset experiments were conducted according to the Australian Code of Practice for the Care and Use of Animals for Scientific Purposes and were approved by the Monash University Animal Ethics Committee. |

Note that full information on the approval of the study protocol must also be provided in the manuscript.

# Plants

| | |
|---|---|
| Seed stocks | Not applicable |
| Novel plant genotypes | Not applicable |
| Authentication | Not applicable |

# Flow Cytometry

## Plots

Confirm that:

☒ The axis labels state the marker and fluorochrome used (e.g. CD4-FITC).

☒ The axis scales are clearly visible. Include numbers along axes only for bottom left plot of group (a 'group' is an analysis of identical markers).

☒ All plots are contour plots with outliers or pseudocolor plots.

☒ A numerical value for number of cells or percentage (with statistics) is provided.

## Methodology

**Sample preparation**

Flow cytometry preparation:

Single cell suspensions were prepared from tissues including brain, spinal cord, meninges, blood, and spleen. Immediately following CO2 euthanasia, spleens were removed into RPMI/10% FBS and peripheral blood was collected through the right ventricle into heparin tubes. Mice were subsequently transcardially perfused through the left ventricle with 10mL of DPBS, decapitated, and brains were carefully removed from skullcaps and placed in iMED+ as above. For select experiments, spinal cords were carefully dissected from vertebra. Cortex, lesion, and meninges were dissected as above. Brain was weighed and subsequently homogenized in iMED+ homogenized using a 2mL glass tissue grinder (VWR; 6 plunges, followed by filtration through a 70um filter, addition of 2mL iMED+, and 6 more plunges). Filtered suspensions were centrifuged at 220g for 10 minutes and resuspended in 5mL of 22% Percoll (GE Healthcare) in Myelin Gradient Buffer (5.6mM NaH2PO4•H2O, 20mM Na2HPO4•2H2O, 140mM NaCl, 5.4mM KCl, 11mM glucose in H2O). 1mL of PBS was layered on top of Percoll. Samples were centrifuged at 950g for 20 minutes at 4C with no break to separate myelin and resuspended in FACS buffer. Dissected meninges were incubated in digestion medium (RPMI/10% FBS/80ug/mL DNase I/40ug/mL Liberase TM (Roche)). Tissue was subsequently mashed through 70um filters, followed by centrifugation and resuspension in FACS buffer. Spleens were prepared by mashing tissue through 70um filters without tissue digestion, followed by centrifugation. Red blood cells were lysed for 2 minutes using 1X Pharm-Lyse and the remaining cell pellet were resuspended in FACS buffer.

Blood samples were centrifuged for 5 minutes at 500g. Pellets were resuspended in 1X Pharm-Lyse 5 minutes at room temperature, followed by centrifugation and suspension in FACS buffer. For cytokine restimulation assays, samples were transferred to U-bottom plates and incubated in stimulation medium (RPMI supplemented with 10% FBS, 1% penicillin/streptomycin, 1X Glutamax (Thermo Scientific), 1X HEPES buffer (Fisher), 1X non-essential Amino Acids (Thermo Scientific), 1mM NaC3H3O3 (Thermo Scientific), 55⊠M b-mercaptoethanol, 1X Cell Stimulation Cocktail (Tonbo), and 1X Brefeldin A (Thermo Scientific)) at 37⊠C for 3 hours, followed by centrifugation and transfer to a V-bottom plate.

Resuspended samples were stained in 96-well V-bottom plates. Surface staining was performed at 4C for 45 minutes in 50uL staining volume. For experiments involving intra-cellular staining, cells were fixed and permeabilized using Foxp3 Transcription Factor Staining Buffer Set (eBioscience) followed by staining at 4C for 1 hour in 50uL staining volume.

Nuclear preparation (mouse):

For nuclear flow cytometry and single nuclear RNAseq experiment 1 (timecourse), dura, lesion, and perilesional cortex were microdissected and processed separately. For RNAseq experiment 2 (WT, cKO, and ADWA11), only lesions were dissected. Microdissection involved meningeal/skullcap separation, removal of subcortical structures, and separation of lesions from skullcaps (lesions often separate from cortex during initial dissection but can be micro-dissected as necessary). For snRNAseq, tissue from male and female mice within experimental conditions was combined. Tissue was processed using ST-based buffer protocol, with the following modifications: initial centrifugation was performed at 500g for 10 minutes. After lysis/initial centrifugation, nuclei were resuspended in 1mL ST buffer (nuclear flow cytometry, RNAseq experiment 2) or PBS/1% BSA/0.2UuL Protector RNAse inhibitor (Roche) (RNAseq experiment 1), filtered through 35um cell strainers, and subsequently processed as below.

For nuclear flow cytometry and single nuclear RNAseq experiment 2, nuclei were centrifuged for 5 minutes at 500g, resuspended in FANS buffer (DPBS/1% BSA/0.1mM EDTA80) with 2ug/uL DAPI and 0.2U/uL RNase inhibitor (snRNAseq), and stained (nuclear flow) or sorted (snRNAseq). For single nuclear RNAseq experiment 1, cell counts were performed after initial

centrifugation (NucleoCounter, Chemometic), and a maximum of 2x106 nuclei were multiplexed using CellPlex Multiplexing technology (10X Genomics) according to the manufacturer's instructions (using protocol 1 for nuclear multiplexing, with only one wash after multiplexing to increase yield). Nuclei were resuspended in FANS buffer with 0.2U/uL RNase inhibitor and 2ug/uL DAPI and nuclear concentrations were determined. Immediately before sorting, multiplexed microanatomical regions (including lesion, parenchyma, and dural meninges) from individual mice were combined at desired ratios (75% lesion, 25% parenchyma).

Nuclear preparation (marmoset):

For single-nuclei RNA sequencing (snRNAseq), naïve control marmosets (n=3; 1 female, 2 male; median age 4Y) were administered an overdose of pentobarbitone sodium (100 mg.kg-1; intraperitoneal). Following apnea, frontal lobes were recovered and dissected under aseptic conditions in sterile ice-cold phosphate buffered saline (PBS; 0.1M; pH 7.2). Tissues were and snap frozen in isopentane chilled in liquid nitrogen. The procedures/ dissections were performed in chilled RNAase-free PBS with RNase-free sterilized instruments under RNase-free conditions. Approximate time from apnea to snap-freezing ranged from 20-30 minutes. All 6 samples passed QC. For nuclear isolation, frozen cerebral tissues from the injured cohort were finely pulverized to powder in liquid nitrogen, and 50 mg of pulverized tissue was added into 5 mL of ice-cold lysis buffer (320mM sucrose, 5mM CaCl2, 3 mM Mg(Ace)2, 10mM Tris-HCl [pH 8], protease inhibitors without EDTA, 0.1 mM EDTA , RNAse inhibitor [80U/mL], 1mM DTT, 0.1% TX-100 [v/v]). The suspension was homogenized in a Dounce tissue grinder (15mL, RNAse free, ice-cold) using loose and tight pestles, 30 cycles each, with constant pressure and without introduction of air. The homogenate was strained (40μm) and the strainer was washed with isolation buffer (1800 mM sucrose, 3mM Mg(Ace)2, 10mM Tris-HCl [pH 8], protease inhibitors without EDTA, RNAse inhibitor [80U/mL], 1mM DTT). The suspension was mixed via tube inversion and then gently pipetted onto the isolation buffer cushion (5 mL) without disrupting the phases and centrifuged at 30000g for 60 min at 4C. The resultant supernatant was removed and 100µl of resuspension buffer (250mM sucrose, 25mM KCl, 5mM MgCl2, 20mM Tris-HCl [pH 7.5], protease inhibitors without EDTA, RNAse inhibitor [80U/mL], 1mM DTT) was added dropwise on the pellet and incubated on ice for 15 minutes. Pellets were resuspended, pooled, and filtered through 40um cell strainers. Finally, nuclei were counted and diluted to 1 million/mL with sample-run buffer (0.1% BSA, RNAse inhibitor [80U/mL], 1mM DTT in DPBS).

| | |
|---|---|
| Instrument | Flow cytometry was performed on a BD LSRII Fortessa Dual; flow cytometric sorting was performed on a BD FACSAria. |
| Software | Flow cytometric data was acquired using FACSDiva v9.0 (BD Biosciences) and analyzed using FlowJo v10.7.2 (BD Biosciences) |
| Cell population abundance | For each experiment, single-cell suspensions were incubated with viability dyes, with live cells gated as viability-dye-negative. Cell counts were performed using flow cytometry counting beads (CountBright Absolute; Life Technologies) per manufacturer's instructions. Populations were given as percentage of live CD45+ cells or as backcalculated cell numbers. |
| Gating strategy | Oligodendrocyte-lineage nuclei were identified as DAPI+, Pdgfr GFP-hi, Olig2+. Fibroblast nuclei were identified as DAPI+, PdgfraGFP-int or DAPI+, Col1a2creER; Rosa26Sun1GFP+ (snRNAseq experiment 1, resting dural meninges). Bulk nuclei were identified as DAPI+. Global lymphocytes were defined as CD45+, Thy1+. T cells were identified as CD45+, CD11b-, CD19-, NK1.1-, CD3e+, CD4+ (CD4 T cells; further subset as FoxP3+ [regulatory T cells] or FoxP3- [conventional CD4 T cells]), CD8a+ (CD8 T cells), or TCRgd+ (gdT cells) and were further defined as CD44+, CD69+ (resident memory T cells, TRM), CD62L+ (naïve T cells), CD62L-, CD44+ (activated T cells), or CTVdiluted (proliferating T cells). Additionally, CD4 T cells were defined as T-bet+ (Th1 T cells), Gata3+ (Th2 T cells), or RORgt+ (Th17 T cells), and various T cell subsets were defined as cytokine-positive or negative (IFNg, IL17A, or IL10). Neutrophils were defined as CD45+, CD11b+, Ly6G+ (and optionally Thy1-, CD19-, NK1.1-). Monocytes were defined as CD45+, CD11b+, Ly6G-, Ly6C+ (and optionally Thy1-, CD19-, NK1.1-, Siglec F-). Microglia were defined as CD45int, CD11b+. Macrophages were defined as CD45+, Ly6G-, Ly6G-, CD64+ (optionally MERTK+). Microglia/macrophages were further defined as Damage Associated Microglia/Scar Associated Macrophages (DAM/SAM, CD9+ and CD63+/TREM2+). cDCs were identified as CD45+, Ly6G-, Ly6C-, CD64-, MHCII+, CD11c+, and were further defined as cDC1s (CD11blo, optionally SIRPa-) or cDC2s (CD11bhi, optionally SIRPa+). B cells were defined as CD45+, Thy1-, CD19+. Eosinophils were defined as CD45+, Thy1-, CD19-, NK1.1-, Ly6G-, CD11b+, Siglec F+. Populations were back-gated to verify purity and gating. |

☒ Tick this box to confirm that a figure exemplifying the gating strategy is provided in the Supplementary Information.

