## [Peer Review file · Nature]

Dynamic fibroblast-immune interactions shape recovery after brain injury

Corresponding Author: Dr Ari Molofsky

Version 0:

Reviewer comments:

Referee #1

(Remarks to the Author)

This paper investigates the interaction of brain resident fibroblast with other non-neuronal cells in the setting of focal brain injury produced by photothrombosis, trauma, or transient occlusion of the middle cerebral artery (tMCAO) in mice, and endothelin-induced focal cortical strokes in non-human primates. It was found that fibroblasts infiltrate the lesions in all models. Transcriptomic analysis identified early (transient) and late (more persistent) fibroblast clusters within the lesion. The early cluster (myofibroblasts) was associated with evidence of TGFbeta signaling and profibrotic markers, and was inferred to be mechanistically linked to the innate immune response induced by the lesion (microglia and macrophages). The "late" fibroblasts were transcriptomically diverse and included fibroblasts thought to interact with lesional lymphocytes, suggesting a transition from innate to adaptive immune responses. Additional clusters included "late myofibroblasts", possibly involved in vascular remodeling, and arachnoid and dural fibroblasts. Interaction of fibroblasts with macrophages was required for extracellular matrix coverage of the lesion, a process in part related to TGFbeta signaling. Suppression of TGFbeta signaling led to an increase in inflammatory cells in the lesion, and pharmacological experiments suggested involvement of the integrin alphaV-Beta8. In mice with suppression of fibroblast TGFbeta signaling the mortality after tMCAO at 14 days was increased. It is concluded that the brain fibroblasts coordinate reparative responses after brain injury.

This study addresses the role of brain fibroblast, a cell about which little is known, in the transcriptomic and cellular processes underlying brain injury and repair. The study makes extensive use of transcriptomic approaches, including spatial transcriptomics, reporter mice to track the fibroblast, as well Cre-lox based methods to investigate the role of select signaling pathways. While the experiments provide an extensive description of the fate and diversity of brain fibroblasts and associated cells, the impact of the study is reduced by methodological shortcomings and lack of functional evidence for a role of these cells in the outcome of brain injury.

The impact of the fibroblast on the functional and neuropathological outcome of the lesion remains unclear. Was the impact of the lesion on brain structure and function affected by the manipulations of brain fibroblasts? Was there an effect on neurons? How were the sensory-motor deficits induced by the lesion (*J Cereb Blood Flow Metab.* 2021;41: 2439) modified by changes in the fibroblasts? Was there less recovery of neuronal function? Was the neuronal connectivity that contributes to functional repair processes affected by the genetic manipulations of the fibroblasts?

Col1A2 is expressed widely in the body and the paper makes extensive use of Col1a2-Cre-ERT mice. While this approach is appropriate for cell-tracking experiments, the functional impact of brain fibroblast cannot be assessed by using Col1a2-Cre mice since the gene of interest will also be deleted in peripheral organs. This is a problem for the survival studies in which Col1a2creER; Tgfb2flox mice were used. In these and other similar studies, effects on peripheral organs need to be ruled out.

These survival experiments also raise the question: why did the mice die from deletion of TGFbeta receptor 2 in fibroblasts? How did changes in the extracellular matrix, cellular composition, etc. in the brain lesion lead to their demise? Were there effects on other vital organs?

Emx1-Cre is expressed in predominantly in excitatory neurons (J Neurosci 22:6309, 2002). The results Emx1cre; Itgb8flox mice do not necessarily implicate glial cells. Furthermore, peripheral effects on autonomic ganglia with a potential impact on the immune system need to be excluded (Y. Ning et al eNeuro, 2022).

Late fibroblasts are said to support lymphocyte survival *ex vivo*. In these studies, the whole brain lesion was added to the lymphocytes. It is unclear if the critical supporting cells are fibroblasts or other cells in the lesion.

TGFbeta signaling within the lesion is extensively examined using transcriptomics and KO approaches. Is there biochemical evidence of TGFbeta in the lesion?

(Remarks on code availability)

Referee #2

(Remarks to the Author)

The study by Ewing-Crystal et al. investigates the formation of fibrotic scar tissue after ischemic lesions to the brain. The authors address the origin and dynamic interaction of lesional fibroblasts with macrophages, lymphocytes, and perilesional glial cells and identify TGFb signaling as a key pathway coordinating CNS fibrosis. While the topic is highly relevant, the study rather confirms the known with additional expression analysis but does not significantly advance the conceptual understanding of CNS fibrosis. Fibrotic scarring in response to ischemic stroke lesions has been reported for mice and humans. It was also previously established that scar-forming fibroblasts originate from perivascular cells (fibroblasts/pericytes) that transition through a proliferative myofibroblast state in response to different kinds of CNS lesions, including cortical stroke (PMID: 33526922; 34535655; 23966707). The presented data on the cellular origin of scar-forming fibroblasts are rather confusing than addressing current limitations. The different experiments are not well aligned, and some data presentations and interpretations have flaws. Several critical points are listed below:

1. Col1a2+ cells contribute to about 50% of the lesional (myo)fibroblasts. An undefined part of this contribution comes from Col1a2+ cells in the dura/meninges. What is the cellular origin of the remaining fibroblasts?
2. Using lineage tracing, a previous study (PMID: 34535655) reported that Slc1a3/Glast+ perivascular cells contributed ~80% to all scar-forming PDGFRβ+ stromal cells after cortical ischemic lesions. Considering the reported 50+% derived from the Col1a2-CreERT2 line, there needs to be at least a substantial overlap regarding the cellular origin. This should be addressed or at least discussed.
3. Along the same line, the authors show results from lineage tracing of Gli1+ cells using Gli1-CreERT2xR26-TdTomato mice. Judging from the pictures shown in Extended Data Fig. 1i, it seems Gli1+ cells contribute substantially to scar-forming fibroblasts. However, quantification is missing. Do Gli1+ cells contribute more fibroblasts to the scar? In this case, why did the authors not use this line instead? Is the contribution similar to the Col1a2 line? This would further substantiate the evidence for another source besides fibroblasts.
4. Col1a2-CreERT2;tdTom mice label perivascular fibroblasts and all meningeal fibroblasts (dura, arachnoid, and pia). It is unclear from their lineage tracing experiments using the Col1a2-CreERT2 line (Ext Data Fig. 1h, l-q) what is the source of lesional (myo)fibroblasts. Do all these different fibroblast populations contribute to lesional myo(fibroblasts)?
5. The authors perform sparse labeling of dural fibroblasts and find sparse labeling of lesional fibroblasts. Once more, the interpretation of these results is unclear. What happens if dural meningeal fibroblast are efficiently recombined? Do 50% of all lesion fibroblasts derive from the dural fibroblasts? These questions need to be clarified.
6. Ext. Data Fig. 1: Scheme in (l) should contain arachnoid represented. It is unclear whether the arachnoid meningeal layer is labeled or not by transcranial application of 4-OH-tamoxifen, as this meningeal layer is omitted from images (n) and analyses (o). Images in (n) should be labeled with DAPI and provide evidence that the arachnoid meningeal layer is not recombined by transcranial application of 4-OH-tamoxifen (higher resolution required). In addition to perivascular and pial fibroblasts, quantifications in (o) should include arachnoid and dura meningeal fibroblast numbers, as intraperitoneal tamoxifen injection results in recombination of all Col1a2+ fibroblasts. From images in (p), it does not seem Col1a2-tdTom+ cells are positive for ER-TR7. Have these cells lost ER-TR7 expression? Or are these cells not expressing ER-TR7 (reticular fibroblast ECM marker)? Or are these Col1a2+ cells from the skull that are not fibroblasts (e.g., Osteoblasts / Bone cells)?
7. Are "meninges" quantifications in Ext Data Fig. 1e referring to dura meningeal fibroblasts only or fibroblasts from the dura and leptomeninges? The description of this figure in the main text implies that quantifications refer to the dura mater only. Why are not all meningeal fibroblasts quantified? Also, bear in mind that not all fibroblasts in the meninges express Pdgfra (arachnoid layer fibroblasts do not express Pdgfra. See <https://doi.org/10.1016/j.neuron.2023.09.002>). Therefore, Pdgfra-GFP mice might not be the best option for this quantification. Why are Col1a2-CreERT2;Rosa26Sun1GFP mice not used instead?
8. The recombination induction protocols are complicated to interpret for experiments involving Col1a2-CreERT2;tdTom mice. For some experiments (tMCAO, Fig. 1d; PT stroke Ext Data Fig. 1h), recombination is induced before injury, and results show lineage tracing of Col1a2+ cells. For some other experiments, recombination is induced before and after injury (PT stroke, Ext Data Fig. 1b), or recombination is induced after injury (5-7 days after injury, PT stroke, Fig. 1b). It is unclear why all these experiments use different approaches for recombination, as results are difficult to interpret. If the aim is to visualize fibroblasts at different stages after injury and across different types of lesions, then the authors should use Col1a1-GFP mice consistently across all models and time points as in Fig. 1f. In addition, for CCI experiments, the authors show the presence of Col1 and ER-TR7 (recognizing collagen type VI) ECM deposition in the lesion. This does not provide direct

- evidence that fibroblasts populate these lesions. Once more, Col1a1-GFP mice should be employed for CCI experiments.
9. Lines 100-103 “Fibroblasts were first detected near the leptomeninges by 4dpi (Extended Data Fig. 1b) and had surrounded and infiltrated the lesion by 14-21dpi (Fig. 1f, Supplementary Video 1), where they persisted for at least one year post-injury (Fig. 1g, Extended Data Fig. 1c).” – This description needs to be refined as fibroblasts are clearly visible in the lesion already at 7 days after PT injury (see Fig. 1f).
 10. Legends need refinement to describe what is shown in the images. For example, Ext Data Fig 1f-g, “Vascular remodeling after PT injury, with PdgfraGFP+ nuclei (f) or fibroblasts (g) visualized in perivascular spaces”. These images show the lesion and not perivascular spaces. Ext. Data Fig. 1f shows no vascular remodeling.
 11. It is unclear how many fibroblasts and other cell types are included in the snRNAseq analyses. Only the number of total nuclei is stated. Please state the number of analyzed nuclei instead.
 12. As specified in the materials and methods section, the authors excluded pericytes from their murine snRNAseq analysis (for example in Fig. 2e and Fig. 3b). Given that these cells are a potential source of lesional (myo)fibroblasts, the analyses would be substantially enhanced if these cells were not removed. The authors should include these cells in the analyses.
 13. Refrain from using the term “late myofibroblasts” for 21dpi. Fibroblasts at this stage are not myofibroblasts. As the authors show in fig. 2j-m, the myofibroblast stage is transient and very few cells exhibit myofibroblast features from 7-9 days after injury, with no lesional fibroblasts being labeled with Cthrc1;tdTomato at 21dpi (only a few cells in the meninges are tdTom+).
 14. Ext Data Fig. 5a shows that Cthrc1;tdTom+ dural fibroblasts react to PT injury in the meninges but seemingly do not contribute to lesional (myo)fibroblasts (contrary to Col1a2+ lineage traced dural fibroblasts – Ext Data Fig. 1 l-q). Specify in the figure legend and/or text when recombination was induced for this experiment and provide an interpretation of why you think Cthrc1;tdTom+ dural fibroblasts and Col1a2-tdTom+ dural fibroblasts respond differently to PT injury.
 15. Legend is missing for Ext Data Fig 5h.
 16. In Ext Data Fig. 5c tamoxifen induction should be from -16 until -14 days (and not from 16 to 14 days). The same in Fig. 2i.
 17. Fig 3: it would be interesting to visualize lymphocytes on the spatial data and cell-to-cell communication with the late-fibroblast population to understand how these populations interact.
 18. In Fig. 4a, ER-TR7 EMC deposition and lesion size are altered between control and clodronate groups. Does the total number of lesional fibroblasts (Col1a2-CreERT2;tdTom+ cells) change as well? Additionally, are only lesional fibroblasts in the inner lesion core (macrophage-contacting fibroblasts) affected, and are microglia-contacting lesional fibroblasts unaffected/less affected? Provide quantifications as this would enhance the analyses and provide relevant information.
 19. Similar quantifications should be provided in Fig. 4f (control versus cKO) and Fig. 4j (IgG versus ADWA11), does the total number of Col1a2-CreERT2;tdTom+ cells change between conditions? In addition, are only lesional fibroblasts in the inner lesion core (macrophage-contacting fibroblasts) affected, and are microglia-contacting lesional fibroblasts unaffected/less affected? Provide quantifications.
 20. Across Figs. 2n, 3j, 3k, 4b, 4c, 4g, 4h, 4k, 4l, 4n, 4s, and 4t, what do individual data points in the plots represent? It is unclear if each dot represents an individual animal or the number of slices analyzed. For transparency and statistical purposes, each dot in the plots should represent one animal, and this should be used for statistical tests. The same applies to all Ext Data Fig. plots.
 21. For experiments presented in Fig. 4d-h using Col1a2-CreERT2;tdTom;Tgfr2flox cKO mice, recombination is induced before injury and after injury. This essentially means that any cell that upregulates Col1a2 after injury (e.g., pericytes, smooth muscle cells, etc.) is also blinded to TGFbeta signaling. For this experiment to be restricted to fibroblasts, recombination needs to be induced before injury followed by a washout period, and then the injury should be performed.
 22. Fig 4p: the heatmap should be replaced by violin plots to determine if the number of cells expressing the mentioned genes is affected.
 23. Why use Emx1-Cre to delete Itgb8 in injury-induced reactive glia? In Emx-Cre;Itgb8flox mice, Itgb8 is deleted in glia and neurons across embryonic and postnatal development, which can greatly affect the interpretation of the results. To provide solid results without confounders, the authors should use an inducible CreERT2 mouse line with a glia-specific promoter instead.
 24. In the discussion, put your findings into context with previous literature classifying microglia/macrophages into previously described polarization states.

(Remarks on code availability)

Referee #3

(Remarks to the Author)

This manuscript by Ewing-Crystal et al. uses multiple approaches (for example, single-nuclei RNAseq, spatial transcriptomics and transgenic mouse models) to investigate the role of fibroblast subpopulations as regulators of CNS wound healing and neuroinflammation following brain injury.

This is a well-written manuscript, with clear figures presented throughout, however I think a significant amount of further work is required to fully leverage the datasets presented. For example, to add significant conceptual advance, can the authors manipulate a specific fibroblast subpopulation (identified and fully validated from their single nuclei RNAseq datasets) to demonstrate that these new datasets facilitate precision medicine approaches (via specific targeting of the critical fibroblast subpopulation(s)) in the context of brain injury and wound healing? My specific comments are listed below:

1. Figure 1, line 111: the authors use a Col1a1GFP;Col1a2creER;R26TdT transgenic mouse approach. How efficient was

the initial labelling (ie baseline recombination rate / TdTomato report in Col1a2+ cells in Col1a2CreER mice) in uninjured tissue? Furthermore, has the Col1a2creER mouse previously been shown to be specific in the CNS for fibroblasts? Does it only label fibroblasts? What proportion of total CNS fibroblasts does it label? Data should be presented regarding this.

And given that the knock-in reporter was Col1a1-GFP, why did the authors use a Col1a2creER rather than a Col1a1CreER?

Lines 91-93, Fig 1b: Col1a1 GFP knock-in and Col1a2CreER TdTom report appears in some cases to label different populations and no quantitation of overlap is presented here. Please show data regarding this and discuss why the two lines were chosen.

-Line 105 Ext Fig 1e: the authors use a nuclei PDGFRalpha-GFP knock-in line to help quantitate fibroblast numbers – how did the reporting / marking of CNS fibroblast populations in this PDGFRa-GFP line compare with col1a1-GFP and the col1a2CreER lines also used – this is not clear. Again data regarding degree of overlap, and also whether PDGFRa-GFP marks a subpopulation of CNS fibroblasts should be presented and discussed.

- the authors state that 'By 21dpi, over half of lesional fibroblasts (GFP+) were derived from pre-existing Col1-expressing fibroblasts (TdT+; Extended Data Fig. 1h)'. Where did the remaining lesional fibroblasts derive from? The authors then describe lineage tracing of meningeal fibroblasts with 'an equal frequency of TdT+ fibroblasts in the dura and lesion after PT injury (~10%)', but it remains unclear to the reader as to where the remaining 50% of lesional fibroblasts arise from.

2. Figure 2, Ext Data Fig 3, line 158: 'Early myofibroblasts spatiotemporally correlate with lesional profibrotic macrophages and damage-associated microglia'.

The data described here (first half of fig 2 and Ext fig 3) are in silico based, descriptive data. The authors should validate and annotate the proposed key fibroblast subpopulations in this section at the protein level with immunofluorescence staining using markers derived from the single-nuclei seq datasets, as this is a key dataset (with proposed fibroblast subpopulations) that underpins the rest of the manuscript. It would be important to show these protein-level data longitudinally in the PT injury model to delineate and confirm when and where these fibroblast subpopulations are manifest post-injury.

Also when the authors draw cross-species comparisons (lines 177-186), it would be more compelling for the reader if they could show (at least in some of the other species) immunofluorescence staining confirming at the protein level corollary subpopulations in the other species / injury states.

3. The authors use ER-TR7 throughout to label fibroblasts, however in the literature this appears to be a marker of an ECM protein (collagen VI). Can the authors also use specific fibroblast-marking antibodies throughout the manuscript to mark (and quantitate) the fibroblast subpopulations of interest.

4. Line 230: 'To investigate potential mechanisms underlying this profibrotic state, we performed ligand-receptor analysis, which predicted that SAM/DAM myeloid cells could signal to myofibroblasts via profibrotic signals...'

The proposed ligand-receptor analyses should be functionally interrogated using co-cultures of SAM/DAM myeloid cells and myofibroblasts with modulation of the proposed key ligand-receptor interactions.

Immunofluorescence staining-based assessment of SAM/DAM topography relative to the key myofibroblast subpopulation(s) should also be performed.

5. The authors use Visium-based spatial transcriptomics, however they should also mention the limited resolution of this method and how each spot (which is 55um in size) will comprise multiple cell types. For example, in Ext Fig 2i the authors show the enriched fibroblast spots are also enriched for proliferative genes, however in Ext Fig 3h one can see many other cell types have an increased proliferative signature.

Furthermore, do the authors have supplementary tables listing all the signatures used to identify lineages and subpopulations and cell identity scores?

6. Lines 230-238, ligand-receptor analysis. The authors show in Ext Fig 6m and Fig 2q the spatial location of the cells of interest using Visium data, however as discussed above this is not at single cell level (spots are 55um in size and 100um apart). Can the authors validate their findings using immunofluorescence staining to confirm whether the subpopulations of interest co-locate in the regions mentioned? This would add further strength to Fig 2r.

7. Line 282: 'late myofibroblasts expressed markers associated with smooth muscle differentiation...' – the late myofibroblasts should be further validated in tissue with multiplex immunofluorescence staining using markers derived from the in silico data.

Similarly, regarding the statement 'The final three late fibroblast (21dpi) clusters were identified as pial, arachnoid, and altered dural fibroblast states, based on resting leptomeningeal fibroblast distribution and meningeal layer signatures' – these 3 fibroblast subpopulations should also be validated in tissue with multiplex immunofluorescence staining (in addition to the integration with spatial sequencing already presented).

8. Figure 4f – in addition to showing the TdTom micrographs, the number of brain lesional fibroblasts should be quantitated

using specific immunofluorescence markers and counting.

9. Line 371: 'cKO mice had a profound loss of early myofibroblast clusters at 7dpi and a substantial reduction in late states at 21dpi, including late myofibroblasts, meningeal-like, and lymphocyte-interactive fibroblasts, supporting our model of sequential state transitions (Fig. 4o, Extended Data Fig. 10k)'.

The authors should validate (and quantitate) these findings at the protein level with multiplex immunofluorescence staining and counting of the various myofibroblast subpopulations.

10. Line 132: the authors state the 4 fibroblast clusters had distinct temporal, molecular and microanatomical patterns – however the authors only show Col1a1 gene expression in the Visium images (which marks all 4 clusters) – please add data demonstrating the distinct microanatomical location of these fibroblast populations.

11. Substantial amounts of data based on transgenic mouse approaches are presented, however to further enhance the manuscript can the authors show evidence of corollary myofibroblast subpopulation / immune cell subset interactions in diseases where wound healing occurs in human brain injury.

12. Line 250 Fig 3d: In addition to the flow cytometry data presented can the authors also show immunofluorescence staining to delineate the spatial location of the T cell population relative to the lymphocyte-interactive fibroblasts (not simply show ER-TR7 staining).

13. Line 150: please provide details of the genes used for the TGFb score and the data and experimental information used to generate this score. What were the primary cells used here, where/how were they obtained?

14. Figure 4u,v: functional interrogation experiment using conditional KO, Col1a2creER; Tgfb2flox mice in the context of the tMCAO model.

It is perhaps somewhat unsurprising that KO of Tgfb2 on Collagen1a2-expressing myofibroblasts results in impaired fibrotic scar formation in the CNS. To fully leverage the authors' interesting findings (using single-nuclei RNA seq to generate precise annotation of fibroblast subpopulations during CNS injury and wound healing), and to also add conceptual advance, can the authors use a Cre-driver approach to either delete a gene within, or deplete (eg using a Cre-DTR system) a specific fibroblast subpopulation which alters wound healing to demonstrate that their new datasets facilitate precision medicine approaches (via specific targeting of the critical fibroblast subpopulation(s)) in the context of brain injury and wound healing?

15. Line 114: the authors use dual labelling with Col1a1-GFP and Col1a2-TdT mice, however then call them 'Col1-expressing (TdT+)' – please be specific and consistent in the text – eg call these 'Col1a2-expressing (TdT+)'...

16. Ext Fig 1h has confusing axis labelling which does not match the schematic. In the figure legend it states 7-9dpi but the schematic and IHC shows 21dpi – please clarify.

17. Line 115: the authors use Gli1-TdT along with some I.F. stains to show that lesional fibroblasts express fibroblast markers. Please justify the use of the Gli1 strain in the text. Please also state 'n' of mice used at each timepoint. There are no scale bars presented. Please also show an overlay image if these are all the same mice for each IF micrograph.

18. Ext. Fig 1k, please show quantification of this and higher mag images.

19. Fig 1h: it is unclear in the methods and figure legend if the upper I.F. images were derived from the same sections as the lower spatial, or were these consecutive sections? Please clarify this.

20. Line 154: can the authors please validate this increased pro-fibrotic fibroblast response using I.F. eg using Ki67 and a marker of the pro-fibrotic fibroblasts.

21. Line 177: adult marmoset data. The authors reference a previous paper in the Ext Fig 4a legend, however also cite methods in the manuscript regarding generation of marmoset snRNAseq data but this doesn't appear to be referenced in the manuscript text. Please clarify this in the methods and main text.

22. Line 180: there is no reference provided for human paper cited. Also, the fibroblast cluster is very small – please state how many cells were in this. The number of spots in Ext Fig 4f doesn't seem to align with the violin plot presented in Ext Fig 4h – ie the violin plot looks like it contains far more cells – please clarify this by also presenting the number of cells (in addition to the gene expression) in the violin plots. Also please clarify what the control tissue was in these datasets.

23. Line 182: As GBM is a cancer, the authors should state that the fibroblasts in these datasets are cancer associated fibroblasts (unless I have misunderstood their reasoning here). The links presented here are somewhat tenuous - only some genes are shared between the datasets - it would potentially be more powerful to show a signature of myofibroblasts.

24. Line 206: the authors discuss subsets of microglia and macrophages and utilise multiple reporter strains – can the authors please clarify further which strains are being used to label which subpopulations of macrophages and microglia.

25. Ext Fig 6c-f: The authors describe the spatial changes in microglia, infiltrating monocytes and BAM/PVM, summarising this in panel 'f' – it is a nice spatial interpretation of these changes, however there is no mention of 'n' used to generate these results, nor any quantification to back this up. Please add analyses of these images.

26. Fig 3m, Ext Fig 7m+q: how are the authors controlling for the lesion area at 21dpi already having more (lesional) T cells at time of isolation and therefore culture? Similarly for 7dpi, which would have fewer (lesional) T cells also. Please clarify this.

(Remarks on code availability)

Version 1:

Reviewer comments:

Referee #1

(Remarks to the Author)

The revised paper provides further support to the conclusions of the paper related to the fate and contribution of brain fibroblasts to the ECM and fibrotic changes occurring after brain injury, and interaction with immune cells. The work is very detailed and comprehensive and provides new cellular, and signaling insights into the formation of the tissue scar.

However, the role of the fibroblast in the origin and evolution of the damage remains less clear. The massive early mortality after MCA occlusion remains largely unexplained. The new data presented do not provide sufficient evidence to conclude that the mice die as a consequence of enhanced brain injury. Similarly, the pathophysiological implications of the late interactions of fibroblasts with lymphocytes are also unclear.

Line 66: As stated in the cited review, the role of fibroblasts in CSF dynamics has been suggested but not demonstrated

Line 213: How was the PF4cre mouse characterized with respect to BAM specificity? Was this mouse used in the Drieu et al paper cited?

Line 545: How does elevated IGG staining support increased bleeding and edema? Did you document brain bleeding?

Figure 5c,d,e: How can these small differences in hemispheric volume, oligodendrocytes and fluoro jade-stained neuron explain 90% mortality?

Was there early systemic hypoxia, hypotension, hypothermia which could explain a life ending catastrophe? Did the mice had epileptiform activity which can also be a cause of cerebral death?

(Remarks on code availability)

Referee #2

(Remarks to the Author)

I was pleased to see that the authors took the criticism of their manuscript seriously. I acknowledge the effort they have made, and the revised version shows substantial improvement. However, the conceptual advancement and clinical relevance remain somewhat limited. While the authors attempted to explain the increased mortality rate following *Tgfr2* deletion in fibroblasts, the reported brain swelling/edema formation—likely causing neuronal and oligodendrocyte loss—is not clearly linked to the reported roles of myofibroblasts. It is more likely that the edema formation is related to off-target effects on astrocytes and their water channels.

1. Critical: I could not find any information regarding the treatment of controls in relation to tamoxifen-induced gene deletions. Tamoxifen has significant effects on T-cell recruitment (PMID: 19689284, 19689284). Since tamoxifen was administered continuously throughout the post-lesion stages, it could contribute to some of the observed phenotypes if the controls were not treated in a similar manner. Please include this information in the methods and relevant figure legends.

2. Line 64-65 "... recent studies have highlighted the roles of CNS stromal cells, including both mural cells and fibroblasts, in injury^{9–13} and disease^{14–18}.

Goritz et al., 2011, Science PMID: 21737741 AND Holl et al., 2024, Nat Neurosci. PMID: 38849523 should be cited here as well.

3. As in Ext. Data Fig. 1q authors are required to quantify the recombination efficiency of the *Atp13a5-CreER* line under uninjured conditions. What is the percentage of pericytes labelled with tdTomato over the total number of pericytes at rest in

the experiments conducted for this study?

4. Cell composition (blood vessel and fibroblasts numbers) and lesion size are not comparable in *Atp13a5creER*; R26TdT (Ext. Data Fig. 1u) and *Col1a2creER*; R26TdT mice (Ext. Data Fig. 1t). Replace image in Ext. Data Fig. 1u with a lesion of similar size and cellular composition. Also provide quantitative data showing the contribution of *Atp13a5*-lineage traced pericytes to the total number of lesion fibroblasts in comparison to the contribution of *Col1a2*-lineage traced fibroblasts after dMCAO.

5. The following text from the rebuttal letter needs to be stated in the discussion:

“These results could be interpreted to contrast with the reported contribution of a *Glast*+ pericyte population after spinal cord injury (referenced above). However, our own sequencing data (Fig. R2.3, below) reveals substantial expression of *Glast* (i.e., *Slc1a3*) within PT lesional fibroblasts at all time points, confirming previous findings^{19,21}. Subsequent analysis from the cited group found that 98.3% of fibroblasts (lineage-traced with a *Col1a1creER* allele) expressed *Slc1a3*, and that these fibroblasts significantly contributed to the stromal scar after spinal cord injury¹², potentially consistent with a *Slc1a3*-lineage+ fibroblast origin for stromal cells after PT injury.”

6. In figure 3d, laminin signal seems to localize to basement membrane of blood vessels and does not clearly overlap with a subset of lesion fibroblasts. This may be due to the use of a pan-laminin antibody and not a Laminin alpha 1 specific antibody. Recommended to revise the imaging with a *Lama1* specific antibody and/or ISH probe.

7. In Ext data fig.1v legend, change left and right by top and bottom.

8. Fig. R2.12: Tamoxifen pre-treatment is insufficient to prevent fibroblast expansion
Include these data in the article.

(Remarks on code availability)

Referee #3

(Remarks to the Author)

I have read through this revised manuscript and the authors' rebuttal comments and I would like to congratulate the authors on an outstanding, rich and exciting body of work. I have no further comments.

(Remarks on code availability)

Version 2:

Reviewer comments:

Referee #1

(Remarks to the Author)

The authors have continued to improve the paper by changes on the text and adding new data on cardiovascular parameters and organ damage in the mice with tMCAO, and transcriptomic effects of IFN γ on brain cells.

The pathological significance of the new data showing a 150 micron midline shift is questionable (fig. 5c,d). It is unclear how such small hemispheric deviation can cause the hemodynamic collapse indicated by the measurement of the cardiovascular parameters, as well as end-organ damage. It is argued that the shift may have caused compression of brainstem autonomic centers, akin to those occurring in brain herniation-related coma in humans, but no evidence is provided that the small displacement reported is capable of such profound mechanical effects and damage on the brainstem.

Therefore, while the new data show a plausible cause underlying the mortality, i.e., hemodynamic collapse, the mechanisms of such profound and presumably lethal cardiovascular changes remain unclear. Statements that the cardiovascular decompensation and death is due to “exacerbated post-stroke brain edema” do not seem justified by the data. Overall, it remains puzzling why these devastating effects are observed only in the late stages of injury.

(Remarks on code availability)

Referee #2

(Remarks to the Author)

The authors have addressed all my comments.

(Remarks on code availability)

Response Summary: We are grateful to the three reviewers for their thoughtful comments on our manuscript titled “Dynamic fibroblast-immune interactions shape wound healing after brain injury”. In this revised manuscript, we provide substantial new data that strengthens our previous conclusions and expands on the work in both scope and impact. Below, we broadly summarize our response to four key conceptual questions raised by reviewers, followed by a detailed point-by-point response to each individual reviewer and associated comments.

(1) What novel insights are provided into how specific brain injury-associated fibroblast states functionally contribute to brain recovery or long-term brain dysfunction? In entirely new data, we demonstrate that late injury-associated fibroblasts upregulate the chemokine CXCL12 to promote lesional T cell retention and regulation (1a). We also expand upon our previous findings to show that the brain myofibroblast state is functionally key in acute to sub-acute time periods after brain injury but is largely dispensable at chronic time points post-injury (1b). Together, we believe our revised manuscript contains significant conceptual advances over previous characterizations of brain fibroblast functions.

(1a) A lymphocyte-interactive fibroblast state emerges late after injury and promotes the accumulation and suppression of chronic brain lymphocytes. To investigate discrete functional roles for late fibroblasts, we identified upregulation of *Cxcl12*, a chemokine known to recruit multiple types of immune cells, including T cells. To directly test if fibroblast CXCL12 promotes the accumulation of late lesional lymphocytes, we conditionally delete *Cxcl12* in the fibroblasts of adult mice. We show that loss of this chemokine impaired T cell accumulation in fibroblast-dense lesions, dysregulating brain lymphocyte cytokine expression and chronic neuroinflammation.

(1b) A myofibroblast state is functionally critical in early but not late brain injury. To expand on our findings that the myofibroblast state (TGF β -driven pro-fibrotic fibroblast) contributes to wound healing, we now define a specific temporal window when myofibroblast function is required. We use a new *Cthrc1* myofibroblast deleter strain to show increased lesion size in the specific absence of early myofibroblasts. We also show that loss of late fibroblast TGF β signaling is dispensable (tamoxifen induction of *Col1a2*^{creER}; *Tgfb2*^{flox} mice at 14dpi).

We define two temporally discrete and functionally beneficial roles of brain fibroblasts after injury: an early wound-healing role for brain myofibroblasts, coordinated via lesional TGF β , and a late lesional fibroblast role that organizes lymphocyte positioning and dampens their function to fine-tune chronic brain immunity. These early and late fibroblast roles represent distinct therapeutic targets that are present at distinct temporal stages of brain injury and recovery.

(2) How does loss of the myofibroblast state increase mortality in the tMCAO stroke model? We present new data showing that the myofibroblast state locally restricts tMCAO-driven subacute brain injury. To further mechanistically characterize how early myofibroblasts were limiting mouse mortality after tMCAO, and to address valid concerns concerning peripheral fibroblast impacts on brain-injury-associated phenotypes, we now provide a more comprehensive characterization of both brain and peripheral tissue in control and “myofibroblast-impaired” *Tgfb2* cKO mice. While we find no differences in peripheral organs – or mortality – between genotypes of uninjured mice, tMCAO in “myofibroblast-impaired” cKO mice induces excessive brain swelling and circulating antibody extravasation, consistent with worsened edema or hemorrhage; loss of mature oligodendrocytes; and an increase in neuronal degeneration. Collectively, these data suggest exacerbated brain damage just prior to the appearance of increased mortality in cKO mice, strongly linking local brain myofibroblasts to the observed mortality.

(3) What are the cellular source(s) of brain injury-associated fibroblasts? We now perform comprehensive fate mapping of fibroblasts and mural cells to demonstrate that injury-associated lesional fibroblasts derive primarily from pre-existing brain fibroblasts and not mural cells such as pericytes or smooth muscle cells. We also provide new data with our collaborator, Dr. Rich Daneman, highlighting a similar fibroblast-derived fibroblast response to a second murine model of stroke (dMCAO). These data demonstrate a “fibroblast-derived fibroblast” ontogeny that is conserved across at least the two mouse models of brain injury tested here.

(4) Can the lesional fibroblast states – observed via single nuclear and spatial transcriptomics – be validated by microscopy? We use fibroblast state markers identified via our snRNAseq data, in conjunction with quantitative lineage tracing, to refine the spatiotemporal map of brain injury-associated fibroblast evolution, confirming the presence and topography of all four previously defined late lesional fibroblast states – as well as the myofibroblast origin for each of these states – with predicted fibroblast state reductions in myofibroblast-impaired mice (cKO). Using fibroblast reporter mice, we also quantify reductions in fibroblast density after brain injury across our three approaches to directly or indirectly perturb brain fibroblasts (i.e., *Tgfbr2* cKO, $\alpha_v\beta_8$ -blockade, clodronate treatment). These data validate our prior single nuclear transcriptomics and demonstrate a fibroblast evolutionary trajectory through the myofibroblast state and into multiple, spatially discrete late fibroblast states.

Referee #1 (Remarks to the Author):

Summary: This paper investigates the interaction of brain resident fibroblast with other non-neuronal cells in the setting of focal brain injury produced by photothrombosis, trauma, or transient occlusion of the middle cerebral artery (tMCAO) in mice, and endothelin-induced focal cortical strokes in non-human primates. It was found that fibroblasts infiltrate the lesions in all models. Transcriptomic analysis identified early (transient) and late (more persistent) fibroblast clusters within the lesion. The early cluster (myofibroblasts) was associated with evidence of TGFbeta signaling and profibrotic markers and was inferred to be mechanistically linked to the innate immune response induced by the lesion (microglia and macrophages). The “late” fibroblasts were transcriptomically diverse and included fibroblasts thought to interact with lesional lymphocytes, suggesting a transition from innate to adaptive immune responses. Additional clusters included “late myofibroblasts”, possibly involved in vascular remodeling, and arachnoid and dural fibroblasts. Interaction of fibroblasts with macrophages was required for extracellular matrix coverage of the lesion, a process in part related to TGFbeta signaling. Suppression of TGFbeta signaling led to an increase in inflammatory cells in the lesion, and pharmacological experiments suggested involvement of the integrin $\alpha_V\beta_8$. In mice with suppression of fibroblast TGFbeta signaling the mortality after tMCAO at 14 days was increased. It is concluded that the brain fibroblasts coordinate reparative responses after brain injury.

This study addresses the role of brain fibroblast, a cell about which little is known, in the transcriptomic and cellular processes underlying brain injury and repair. The study makes extensive use of transcriptomic approaches, including spatial transcriptomics, reporter mice to track the fibroblast, as well Cre-lox based methods to investigate the role of select signaling pathways. While the experiments provide an extensive description of the fate and diversity of brain fibroblasts and associated cells, the impact of the study is reduced by methodological shortcomings and lack of functional evidence for a role of these cells in the outcome of brain injury.

We thank the reviewer for their constructive comments and appreciate the opportunity to provide an improved manuscript that addresses concerns. To better parse the functional contributions of brain fibroblast states in the outcome of brain injury, we have now added new data from existing and novel mouse genetic models, including $Col1a2^{creER}$; $Tgfr2^{flox}$, $Cthrc1^{creER}$; $Rosa26^{DTA}$, and $Col1a2^{creER}$; $Cxcl12^{flox}$ and described in detail below, to define the functional importance of a brain fibroblast evolution through discrete temporal states and positionings. This substantial new data, including all of **Figure 5**, emphasizes a specific functional role for the early TGF β -driven myofibroblast state in limiting the extent of sub-acute brain damage, and a distinct role for late lymphocyte interactive, CXCL12-producing fibroblasts in recruiting brain lymphocytes near lesional fibroblasts and restricting brain lymphocytes and their IFN γ production. Together, these results support distinct roles for early brain myofibroblasts and later brain lymphocyte-organizing fibroblasts, both of which are functionally adaptive in the response to brain injury.

To shore up potential methodologic shortcomings, we now provide extensive new data throughout the manuscript to strengthen and refine our conclusions. We directly investigate the potential peripheral effects of fibroblast *Tgfr2* deletion and perform a more extensive analysis of tMCAO model at earlier time points post-injury. Our new data detailed below suggest that the local brain myofibroblast state is required – across at least two models of brain injury – to limit the extent of brain damage and perilesional tissue loss.

The impact of the fibroblast on the functional and neuropathological outcome of the lesion remains unclear. Was the impact of the lesion on brain structure and function affected by the manipulations of brain fibroblasts? Was there an effect on neurons? How were the sensory-motor deficits induced by the lesion (J Cereb Blood Flow Metab. 2021;41: 2439) modified by changes in the fibroblasts? Was there less recovery of neuronal function? Was the neuronal connectivity that contributes to functional repair processes affected by the genetic manipulations of the fibroblasts?

We thank the reviewer for these important questions on the functional and neuropathological outcomes of fibroblast responses to brain injury. We previously included data demonstrating that fibroblast *Tgfr2* conditional knockout (cKO) mice have larger brain lesion sizes (**Fig. 4h**) and enhanced perilesional inflammation (**Fig. 4i, Extended Data Fig. 8i**), consistent with increased perilesional neuronal loss at sub-acute time points post-injury. However, to more specifically parse the effects of fibroblast manipulation, we have now conducted an analysis of neuronal and glial dynamics after photo-thrombotic (PT) brain injury, which is detailed below. In summary, our neuropathologic assessments only demonstrated small differences between cKO and WT mice, although our PT injury model is less severe than the one utilized in the reference work above. Therefore, to orthogonally address these important questions, we have also included an expanded analysis in the more severe tMCAO stroke model that addresses many of these same reviewer questions in a more functionally severe and clinically relevant stroke model.

Effect of PT injury on cortical neurons: we observed a small decrease in cortical neuronal density in the injured hemispheres of both control and cKO mice (relative to uninjured hemispheres); however, we did not observe a significant difference between genotypes, with similar results in the striatum, a region functionally downstream of the injured cortex (**Fig. R1.1, below, top**). Additionally, we detected minimal differences in myelin-producing mature oligodendrocytes (SOX10⁺ ASPA⁺) or their precursors (OPCs, SOX10⁺ ASPA⁻) between injured vs. uninjured tissue or in conditional knockout vs. control mice, though cKO mice showed a trending deficit in oligodendrocyte precursors, which expand after injury in controls (**Fig. R1.1, below, middle/bottom**). We identified an accumulation of cleaved caspase 3 (cCasp3), a marker of apoptosis, especially within the corpus callosum of injured cKO mice (**Extended Data Fig. 8q,r, reproduced below**). Largely extracellular, these cCasp3⁺ puncta have been previously described

as markers of generalized white matter damage after brain injury¹, consistent with exacerbated injury in the absence of functional CNS (myo)fibroblasts. Collectively, these data indicate that loss of the myofibroblast state (cKO) exacerbates PT injury-induced apoptosis within white matter tracts, though specific CNS cell populations outside of directly affected lesional areas are not detectably impacted.

Fig. R1.1: Minimal changes in neuronal and oligodendrocyte populations between control and Tgfr2 cKO mice

Extended Data Fig. 8q,r: Elevated cCasp3 staining in corpus callosum of cKO mice

Similarly, we did not detect major sensory or motor deficits as assessed via the Rotarod test (sensorimotor integration and balance) or the tail-flick test (sensory perception) (Fig. R1.2, below), although the impact of our PT stroke on these phenotypes was also modest.

Fig. R1.2: Minimal behavioral differences between control and cKO mice via tail flick or rotarod

Our photothrombotic stroke protocol – optimized for reproducibility in lesion size, which allowed our careful microscopy and transcriptomic spatial mapping – involves a much briefer exposure to light than in the cited reference (2 minutes vs. 15 minutes), potentially explaining the milder neuropathological sequelae we observed relative to the cited reference.

For this reason, we chose to focus our further work in the PT model on the analysis of brain “organ-level effects” of fibroblast perturbation, including lesion size and neuroinflammation. Using new “myofibroblast deleter” *Cthrc1*^{CreER}; *Rosa26*^{DTA} mice (**Fig. 4k,l**) and a late tamoxifen regimen in fibroblast *Tgfr2* cKO (**Fig. 4m,n**, **Extended Data Fig. 8u**, and **response 2, below**), our new data demonstrate the specific functional role for the TGF β -driven myofibroblast state at subacute but not chronic time points post-injury. We have also included completely novel data regarding a distinct functional role for late immune-interactive fibroblasts after PT injury (**Fig. 5**, **Extended Data Fig. 10**).

Col1A2 is expressed widely in the body and the paper makes extensive use of Col1a2-Cre-ERT mice. While this approach is appropriate for cell-tracking experiments, the functional impact of brain fibroblast cannot be assessed by using Col1a2-Cre mice since the gene of interest will also be deleted in peripheral organs. This is a problem for the survival studies in which Col1a2creER; *Tgfr2* flox mice were used. In these and other similar studies, effects on peripheral organs need to be ruled out.

We thank the reviewer for raising this important methodologic concern. We had previously shown data that the neuroinflammatory effects of fibroblast perturbation in the PT injury model are limited to the injured region of the cortex, with no effects in the contralateral brain (**Fig. 4i**, **Extended Data Fig. 8i**), adjacent meninges (**Extended Data Fig. 8j,k**), peripheral blood (**Extended Data Fig. 8l,m**), or spinal cord (**Extended Data Fig. 8n-p**). Nonetheless, we agree that the potential impacts of peripheral organ fibroblasts on the response to brain injury need to be ruled out. Therefore, we have added three new lines of data to address this concern.

1. To test if impacts on peripheral fibroblasts were sufficient to drive the tMCAO stroke-associated mortality, we have added survival data from tamoxifen-treated and sham-operated, or resting *Tgfr2* cKO (*Col1a2creER*; *Tgfr2* flox) and control mice (**Fig. 5b**), demonstrating that cKO “myofibroblast-impaired” mortality only occurs in the context of tMCAO brain injury, suggesting any potential loss of peripheral organ myofibroblasts in cKO mice was not sufficient to drive mortality.
2. To test if impacts on peripheral fibroblasts were sufficient to drive the observed PT stroke-associated enlarged lesions and delayed neuroinflammation, we used *Col1a2*^{CreER}; *Tgfr2*^{flox} mice with tamoxifen recombination delayed until after the early brain myofibroblast temporal window. This regimen still recombined both peripheral fibroblasts and brain lesional

fibroblasts, but it enabled the full brain-myofibroblast response to occur prior to this “late” recombination. In these experiments, we observed no increase in brain lesion size (**Fig. 4m**, reproduced **below**), in contrast to the observed increased lesion size with early tamoxifen recombination in cKO mice. We additionally found no increase in (neutrophilic) neuroinflammation in the cKO (**Fig. 4n**, reproduced **below**), even after an equivalent tamoxifen treatment period (**Fig. R1.3**, **below**).

Fig. 4m,n: Late tamoxifen induction does not recapitulate increased lesion size or neuroinflammation in cKO mice

Fig. R1.3: Timecourse of myeloid neuroinflammatory infiltrates after late tamoxifen recombination

- To further explore possible peripheral organ effects of fibroblast *Tgfr2* loss, we also analyzed serum levels of ALT (a liver enzyme elevated during acute hepatotoxicity) and creatinine (a proxy for kidney function) in control and cKO mice. We found no signs of organ dysfunction in resting mice (**Extended Data Fig. 10a,b**, reproduced **below**). Hematoxylin and eosin (H&E) staining of peripheral organs, including the liver, kidney, and lung, similarly revealed no overt signs of organ dysfunction, inflammation, or structural alterations (**Extended Data Fig. 10c**, reproduced **below**).

Extended Data Fig. 10a-c: no evidence of peripheral organ dysfunction in *Tgfr2* cKO mice

Together, these data suggest that – while peripheral fibroblasts are recombined in *Tgfr2* cKO mice – peripheral loss of fibroblast TGF β signaling, and the associated myofibroblast state, does not detectably influence peripheral organ function, nor is peripheral loss of TGF β signaling temporally linked to brain-injury-related phenotypes including tMCAO mortality, PT injury lesion size, or delayed neuroinflammation. These findings are also consistent with the relative sparsity of TGF β -activated pro-fibrotic myofibroblasts in peripheral organs at rest^{2,3}; instead, myofibroblasts are well appreciated to emerge in settings of fibrosis, cancer, and chronic inflammation. Unfortunately, we are not aware of any validated fibroblast Cre lines with sensitive and specific recombination within, and limited to, brain fibroblasts – or fibroblasts in other organs – although we are actively working to generate and validate such tools. As such, we do not rule out all impacts of peripheral fibroblasts during brain injury. While these limitations also hold true for a vast majority of studies on fibroblasts and their states in other organs and systems, these points are now explicitly raised in the discussion.

These survival experiments also raise the question: why did the mice die from deletion of TGFbeta receptor 2 in fibroblasts? How did changes in the extracellular matrix, cellular composition, etc. in the brain lesion lead to their demise? Were there effects on other vital organs?

We thank the reviewer for raising this important question asking why tMCAO injured cKO mice die. As discussed above, we were unable to detect any differences in cKO mice related to vital peripheral organ structure or function (**Extended Data Fig. 10a-c**), and the lack of mortality in sham-operated and resting cKO mice (**Fig. 5b**) suggested that mortality is related to brain injury.

To directly address potential mechanisms underlying the brain injury-induced mouse death, we have now added data from a comprehensive analysis of tMCAO brains from control and cKO mice at 3dpi, before significant mortality was detected in either genotype. We find that cKO brains had exacerbated ischemic or secondary/inflammatory damage, including relatively increased brain swelling (**Fig. 5c**, reproduced **below**) and blood-brain-barrier permeability, as assessed via antibody extravasation (**Extended Data Fig. 10d,e**), consistent with increased brain bleeding and/or edema; a loss of mature oligodendrocyte density⁴ (**Fig. 5d**, reproduced **below**); and an increase in degenerating neurons (FluoroJade C⁺, **Fig. 5e**, reproduced **below**). Analysis of later time points post tMCAO was hindered by mortality and potential selection bias of surviving mice. Nevertheless, the impacts of myofibroblast attenuation – already visible by 3dpi – suggest a failure to contain ischemic and/or inflammatory damage and a loss of vulnerable cells in the surrounding brain, resulting in loss of critical neuronal tissue and organismal death.

Fig. 5c-e: *Tgfr2* cKO brains show elevated swelling, decreased oligodendrocytes, and increased neuronal degeneration

Emx1-Cre is expressed predominantly in excitatory neurons (J Neurosci 22:6309, 2002). The results *Emx1*^{cre}; *Itgb8*^{fllox} mice do not necessarily implicate glial cells. Furthermore, peripheral effects on autonomic ganglia with a potential impact on the immune system need to be excluded (Y. Ning et al eNeuro, 2022).

We agree with the reviewer and have taken several steps to support our prior conclusions using *Emx1*^{cre}, which are detailed below. Further, we have now phenocopied our *in vivo* results using *hGfap*^{cre}, which is restricted to the CNS⁵, with a similar reduction in brain fibrosis after PT injury (**Fig. 4r,s**, reproduced below). Collectively, our data are consistent with a CNS glial contribution to TGF β activation and wound healing after brain injury. However, as we have not formally ruled out a role for CNS neurons, we have also updated our language throughout the manuscript to reflect the glial *and* neuronal recombination pattern of both *Emx1*^{cre} and *hGfap*^{cre}. Given the integrin expression patterns discussed below, we continue to favor a glial source, as now detailed in the discussion.

Fig. 4r-s: *hGfap*-restricted *Itgb8* deletion recapitulates reduction in fibrosis

We agree that *Emx1* is not restricted to glial cells, with expression in other brain subsets such as excitatory neurons. However, our snRNAseq analysis shows that glial cells uniquely co-express *Emx1* with both *Itgav* and *Itgb8* (**Extended Data Fig. 8v**). While some fibroblast subsets express *Itgav/Itga8*, fibroblasts are not recombined by *Emx1*^{cre}; similarly, while neurons are recombined by *Emx1*^{cre}, they do not co-express detectable *Itgav/Itgb8* (**Extended Data Fig. 8v**, with minimal *Itgb8* expression, **Fig. R1.4**, below). This data suggests that neurons do not co-express the $\alpha_v\beta_8$

components that can drive TGF β activation and that are implicated by our genetic studies in the brain injury models.

Fig. R1.4: Expression of *Itgb8* across snRNAseq dataset

To orthogonally support our snRNAseq data, and in addition to previously included *Itgb8*^{TdT} reporter imaging (**Extended Data Fig. 8ad**), we have performed immunostaining using ADWA11 (anti- $\alpha_v\beta_8$), demonstrating glial expression of this integrin at rest (**Fig. R1.5, below, top**). Similar results were observed in the jugulodose ganglion (a peripheral autonomic ganglion, as mentioned by the reviewer above), with significant co-localization with GFAP+ ganglionic cells, though the overall expression of $\alpha_v\beta_8$ appeared lower at this site (**Fig. R1.5, below, bottom**). Thus, while *Emx1*^{cre} also recombines CNS neurons and autonomic ganglia neurons, $\alpha_v\beta_8$ expression is predominantly limited to glial cells at both sites.

Fig. R1.5: Glial expression of integrin $\alpha_v\beta_8$ in cortex and jugulodose ganglion

To directly investigate possible *Emx1*^{cre}-mediated impacts on peripheral autonomic ganglia and subsequent modulation of peripheral immune cells that could impact our injury-associated phenotypes, we performed flow cytometric analysis of peripheral blood from *Emx1*^{cre}; *Itgb8*^{flox} mice and controls (**Fig. R1.6, below**). We saw no significant changes among global immune cells in

the blood, including both lymphocyte and myeloid cell subsets. We would be happy to include this data in our manuscript if desired, but it is currently included as reviewer-only (R) data.

Fig. R1.6: Immune populations in controls and *Emx1^{cre}; Itgb8^{flox}* mice (“cKO”)

Late fibroblasts are said to support lymphocyte survival *ex vivo*. In these studies, the whole brain lesion was added to the lymphocytes. It is unclear if the critical supporting cells are fibroblasts or other cells in the lesion.

We thank the reviewer for raising this important concern and have updated the manuscript to reflect this limitation of our *ex-vivo* lesional coculture system. While a purified fibroblast coculture would more precisely identify the source of T cell support, the small size and dense ECM of fibrotic brain lesions have proven incompatible with live lesional fibroblast isolation and coculture. Instead, we now provide orthogonal data that fibroblast expression of *Cxcl12* is required for full accumulation of late lesional T cells, with accompanying regulation of their IFN γ production. This new data suggests that fibroblast recruitment and/or support of T cells via CXCL12 promotes local lesional T cell accumulation and/or persistence while limiting brain T cell dispersal and excessive neuroinflammation. Those data and relevant conclusions are more completely described further below in our response letter. Nonetheless, we have also responded to this valid point with two independent lines of evidence, detailed below, where we demonstrate that (1) dural meningeal fibroblasts are sufficient to directly support T cells *in vitro*, whereas (2) myeloid cell depletion from *ex vivo* PT lesions did *not* impact their T cell support.

- (1) To determine whether direct fibroblast-lymphocyte interactions are sufficient to support T cell survival, we performed a coculture experiment using fibroblasts sorted from brain dural meninges and pooled from 18 mice. Of note, dural fibroblasts are potentially distinct from lesion-resident “lymphocyte-interactive fibroblasts” identified *in vivo*, though dural-like fibroblasts and lymphocyte-interactive fibroblasts are transcriptomically similar (**Extended Data Fig. 9g-i**), suggesting a degree of functional similarity. Similarly to *ex vivo* PT lesions, purified meningeal fibroblasts supported T cell survival at levels comparable to lung fibroblasts (**Fig. 3t**, reproduced **below**; **Extended Data Fig. 6x**) without impacting T cell proliferation (data not shown). We and others have previously described similar abilities of peripheral fibroblasts from the lung (and elsewhere) to directly support lymphocyte subsets, acting via both contact-dependent and soluble signals⁶.

Fig. 3t: isolated meningeal fibroblasts support T cell survival ex vivo

- (2) To test for direct contributions of myeloid cells including macrophages and reactive microglia, we also performed *ex vivo* myeloid cell depletions using clodronate liposomes or Cx3cr1^{creER}; Rosa26^{DTR} mice (with liposomes or DTx added to in vitro culture media), and we did not observe significant changes in T cell survival (**Fig. R1.7, below**).

Fig. R1.7: No significant survival defect after myeloid cell depletion from ex vivo cocultures

We have updated our language throughout the manuscript to reflect additional *in vivo* possibilities, including roles for myeloid cell subsets that are also fibroblast-associated.

TGFbeta signaling within the lesion is extensively examined using transcriptomics and KO approaches. Is there biochemical evidence of TGFbeta in the lesion?

To provide biochemical evidence for TGFβ in the sub-acute PT brain lesion, we have now added nuclear flow cytometry data involving lesions, contralateral cortex, ipsilateral cortex, and dural meninges of Pdgfrα^{GFP} mice at 7dpi, staining for phospho-Smad3, a canonical downstream effector of active TGFβ signaling. We found that lesional fibroblast nuclei (GFP^{int}) expressed higher levels of pSmad3 than fibroblast nuclei from other brain regions or the dural meninges (**Extended Data Fig. 4a** and **Fig. R1.8, below, left**). Lesional fibroblast nuclei also expressed higher pSmad3 than non-fibroblast nuclei (**Extended Data Fig. 4b** and **Fig. R1.8, below, right**). Collectively, these data support spatially restricted TGFβ signaling specifically in lesional fibroblasts.

Fig. R1.8: *pSmad3* signal is enriched within lesional fibroblast nuclei

Referee #2 (Remarks to the Author):

The study by Ewing-Crystal et al. investigates the formation of fibrotic scar tissue after ischemic lesions to the brain. The authors address the origin and dynamic interaction of lesional fibroblasts with macrophages, lymphocytes, and perilesional glial cells and identify TGF β signaling as a key pathway coordinating CNS fibrosis. While the topic is highly relevant, the study rather confirms the known with additional expression analysis but does not significantly advance the conceptual understanding of CNS fibrosis. Fibrotic scarring in response to ischemic stroke lesions has been reported for mice and humans. It was also previously established that scar-forming fibroblasts originate from perivascular cells (fibroblasts/pericytes) that transition through a proliferative myofibroblast state in response to different kinds of CNS lesions, including cortical stroke (PMID: 33526922; 34535655; 23966707). The presented data on the cellular origin of scar-forming fibroblasts are rather confusing than addressing current limitations. The different experiments are not well aligned, and some data presentations and interpretations have flaws. Several critical points are listed below:

We thank the reviewer for emphasizing the relevance of our study. We are also grateful for the opportunity to respond to the comments above and better situate our findings in the context of a rapidly evolving field. As referenced above, previous studies have shown that stromal cells expand in CNS pathology including stroke⁷⁻¹⁰; that these stromal cells exhibit a proliferative or myofibroblast state in EAE¹¹ and spinal cord injury (SCI)^{12,13}; and that the ontogeny of these cells may depend on microanatomical context and specific pathology^{10,12-14}. However, we believe that our data – including significant new data in this resubmission – provides unique insights into distinct temporal and functional roles of brain injury-associated fibroblasts that will advance the field's understanding of CNS fibroblast injury responses. First, we address the broader points raised by the reviewer, including questions about fibroblast ontogeny and manuscript significance, followed by a detailed point-by-point response to all reviewer comments.

- (1) The origin of scar-forming brain fibroblasts: We extend our data on the cellular origin of scar-forming fibroblasts, using quantitative lineage tracing to show a predominant fibroblast origin for injury-responsive brain fibroblasts in two models of murine stroke, as has also been shown in EAE¹¹, SCI^{12,13}, and intracerebral hemorrhage¹⁴. We present new resting and post-injury fibroblast lineage tracing data, detailed below, to parse and quantify a fibroblast origin after PT injury; we also present new pericyte lineage tracing studies and quantify our smooth muscle lineage tracing to directly address the possibility of alternate fibroblast sources. Using a novel technique of micro-anatomically-restricted CreERT activation, we implicate dural meningeal fibroblasts in the stroke fibroblast response, representing a significant improvement in quantitative assessment of the microanatomical origin of stroke-associated

fibroblasts¹⁵. In collaboration with Dr. Richard Daneman (UC, San Diego), a previous collaborator¹¹ and expert in mural cells associated with the blood-brain barrier, we find a similar pre-existing fibroblast – rather than pericyte – origin for lesional fibroblasts in a second stroke model (dMCAO). *These data are more completely detailed in response to R2 comments 1 and 2 below.*

- (2) Advancing the conceptual understanding of CNS fibrosis. To investigate discrete functional roles for *late lesional* fibroblasts, including lymphocyte interactive fibroblasts, we identified the chemokine gene *Cxcl12* as upregulated in late brain injury-associated fibroblasts as compared with early myofibroblasts. We conditionally and inducibly deleted *Cxcl12* in fibroblasts in adult mice, using *Col1a2^{creER}; Cxcl12^{fllox}* mice (**Figure 5**, with associated **Extended Data Fig. 10**). At chronic timepoints after injury (21dpi), we observed that *Col1a2^{creER}; Cxcl12^{fllox}* mice had decreased perilesional T cells, as assessed via confocal imaging (**Fig. 5f-i**, reproduced **below**; **Extended Data Fig. 10i-j**), along with a decrease in macrophage MHC-II levels and perilesional MHCII⁺ cells (**Extended Data Fig. 10m,n**), consistent with loss of a type 1 (i.e., IFN γ -competent) lesional T cell population¹⁶. However, total cortical T cells were *not* decreased, as assessed by flow cytometry (**Extended Data Fig. 10o,p**), indicating a specific loss of perilesional T cell *localization* in the absence of fibroblast-derived CXCL12.

Fig. 5f-i: *Col1a2^{creER} Cxcl12^{fllox}* mice show reduced perilesional localization of T-cells

To assess the consequence of interrupting this fibroblast-T cell chemokine network, we assessed lymphocytic cytokine expression via *ex vivo* restimulation. T cells from *Cxcl12* cKO mice expressed higher levels of effector cytokines including IFN γ (CD4⁺ and CD8⁺) and IL-17A (CD4⁺) (Fig. 5k-m, Extended Data Fig. 10q-s, reproduced below), implying the loss of a regulatory mechanism that – in wildtype mice – constrains T cell cytokine expression and dispersal after injury. Additionally, we observed a lesioned-hemisphere-specific increase in neutrophils, consistent with an altered cortical immune tone (Fig. 5o, reproduced below). We validated that loss of CNS fibroblast *Cxcl12* does not impact lesion size (Fig. 5j), overall lesional fibrosis (Extended Data Fig. 10k), lymphocyte-interactive fibroblast expansion (Extended Data Fig. 10l), or lymphocytic expression of suppressive/inhibitory cytokines (IL-10, Fig. 5n). Loss of *Cxcl12* within peripheral fibroblasts does not cause detectable immune changes in blood or meninges (Extended Data Fig. 10t-y). Together, this new and extensive data indicates that late lesional fibroblasts, predominantly including lymphocyte interactive fibroblasts, express chemokines required for perilesional T cell localization and/or persistence within *de novo* lymphocytic niches. In the absence of the chemokine signal CXCL12, T cells are mislocalized, and their effector cytokine expression is unleashed, contributing to an associated dysregulation of chronic cortical innate immune tone (Fig. 5p, reproduced below).

Fig. 5k-p: *T* cell mislocalization in *Col1a2^{creER} Cxcl12^{flox}* mice correlates with increased cytokine expression and chronically dysregulated immune tone

Extended Data Fig. 10q-s: *Col1a2^{creER} Cxcl12^{flox}* show an increase in cytokine-expressing *T* cells

Collectively, these new data suggest that – in contrast to the early wound-healing role played by myofibroblasts – late lesional fibroblast subset(s) spatially coordinate lymphocytic function and fine-tune chronic brain immune tone. This novel conceptual role, in addition to expanding our understanding of CNS fibroblasts as agents of wound healing, represents a distinct potential therapeutic opportunity to address chronic neuropsychiatric sequelae of brain injury.

- (3) Advancing the conceptual understanding of CNS fibrosis. We demonstrate a specific and beneficial functional role for the transient myofibroblast state in promoting wound healing and limiting neuroinflammation. This role is distinct from the role of proliferative fibroblasts in EAE, where fibroblasts (and fibrosis) actually impair oligodendrocyte migration and exacerbate pathology¹¹. Moreover, while global stromal cell ablation prevents tissue regeneration after spinal cord injury, a mild reduction enhances regeneration^{17,18}, suggesting context-dependent or dose-dependent roles of CNS stromal cells. In contrast, we present new data from novel *Cthrc1^{creER}; Rosa26^{DTA}* “myofibroblast deleter” mice. In these mice, even the modest myofibroblast ablation achieved impairs wound healing after stroke, quantified as an increase in lesion size (Fig. 4k,l, reproduced below).

Fig. 4k,l: *Cthrc1^{creER} Rosa26^{DTA}* myofibroblast delete mice show increased lesion size

Additionally, to determine the temporal specificity of the brain myofibroblast response, we now include new data involving *Tgfb2* cKOs with tamoxifen recombination delayed until 14dpi – after most myofibroblasts have transitioned into late lesional fibroblasts, as suggested by sequencing and validated by *Cthrc1* lineage tracing. In this tamoxifen regimen, early lesional fibroblasts remain TGF β responsive, but late lesional fibroblasts do not; therefore, we were able to isolate the role of the TGF β -response to early myofibroblasts, while controlling for the effects of TGF β on other fibroblast subsets both in the periphery and within the CNS. In these mice, we observed no change in lesion size 2 weeks after tamoxifen administration (**Fig. 4m**, reproduced **below**). Additionally, ablation of fibroblast *Tgfb2* after closure of this apparent “myofibroblast window” did not drive a secondary neutrophilic or monocytic neuroinflammation (**Fig. 4n**, reproduced **below**). Collectively, these data are consistent with a specific pro-reparative role for transient, early myofibroblasts after stroke – a significant advance over our previous understanding of *global* fibroblast functions in separate CNS insults, and adding significant nuance to emerging literature which has so far suggested – using pan-fibroblast tools – that fibroblast *attenuation* may be an appropriate therapeutic strategy^{18,19}.

Fig. 4m,n: Late tamoxifen induction does not recapitulate increased lesion size or neuroinflammation

- (4) Advancing the conceptual understanding of CNS fibrosis: We now define not only an early proliferative and migratory myofibroblast state, as has been identified in other CNS pathology^{11–13}, but also divergent chronic states that all derive from transient myofibroblasts. We now fully confirm these transcriptomic states via immunofluorescent imaging, validating

the presence and topography of late inner-lesion fibroblasts, lymphocyte interactive fibroblasts, and dural- and leptomeningeal-like fibroblasts across two models of brain injury. We integrate these markers with our myofibroblast lineage tracing studies – using $Cthrc1^{creER}$ – to confirm a myofibroblast origin for most fibroblasts from each late fibroblast state.

First, we visualized that α SMA⁺ myofibroblasts expand by 7dpi and localize to the early lesion border and lesion core but disappear by 21dpi (**Extended Data Fig. 4c-e**, reproduced **below**), consistent with our transcriptomic data (**Fig. 2b,e**, **Fig. 3a**) and with timed lineage tracing data using the $Cthrc1^{creER}$ (**Fig. 2h-n**) or $Acta2^{creER}$ (**Extended Data Fig. 4g-l**) myofibroblast-lineage tracking mice. We also validated the late lesional fibroblast subsets (identified in **Fig. 2b** and **Fig. 3a**) and their predicted spatial topography (based on integration of single nuclear and spatial transcriptomic data; **Fig. 3c**, **Extended Data Fig. 6a**). After identifying highly and differentially expressed candidate genes (**Extended Data Fig. 6b**), we positively identified late inner fibroblasts using FGF13; lymphocyte-interactive fibroblasts using CD80; altered dural fibroblasts using *Alpl*; and leptomeningeal fibroblasts, including arachnoid-like and pial-like fibroblasts, with the combination of *Aldh1a2* and Laminin¹³ (with *Aldh1a2* expressed in both subsets and *Lama1* preferentially expressed in pial-like fibroblasts) (**Fig. 3d**, reproduced **below**). Each subset, as defined by protein-level expression, localized as bioinformatically predicted and summarized in **Fig. 3e** (reproduced **below**). As expected, late fibroblast markers were sparse at early timepoints (**Extended Data Fig. 6c,d**), though *Aldh1a2* and Laminin-1 were persistent across time points, likely reflecting an early population of pial-like fibroblasts observed in our transcriptomic data (**Fig. 3a**).

Extended Data Fig. 4c-e: A transient myofibroblast response predominates subacute time points

Fig. 3d,e: Topography of late fibroblast subsets by immunofluorescent imaging

To further explore this lesional fibroblast trajectory, we performed fluorescent imaging for the above subset markers in *Cthrc1^{creER}; Rosa26^{TdT}* lineage tracer mice; in these mice, all fibroblasts that passed through a myofibroblast state express *TdT*. We found that most fibroblasts from each marker-defined subset were *TdT*⁺ (**Extended Data Fig. 6k**, reproduced **below**), confirming that myofibroblasts give rise to (at least) a majority of cells in all late fibroblast states. Overall, we believe this substantial new data supports insights derived from our bioinformatic analyses and lineage tracing experiments, providing a stronger foundation for our existing and new data (detailed below) concerning specific and divergent roles for temporally distinct fibroblast states.

Extended Data Fig. 6k: myofibroblasts give rise to late fibroblasts

(5) Advancing the conceptual understanding of CNS fibrosis. While previous studies have identified transcriptional hallmarks of TGF β signaling in CNS injury-responsive fibroblasts¹³, a full understanding of this coordinated multicellular process – incorporating integrated pharmacologic, genetic, and transcriptomic approaches – is essential. Here, we elucidated the cellular and molecular mechanism by which post-injury brain fibrotic lesion deposition occurs, implicating CNS resident integrin $\alpha_v\beta_8$ -expressing cells, along with resident and recruited myeloid cells as TGF β 1 sources, in TGF β -mediated CNS fibrosis. We believe these cellular and temporal integrations are both critical and novel elements in an improved understanding of CNS fibroblast states and functions.

We believe our revised manuscript, including substantial new data, significantly advances the conceptual understanding of brain fibroblasts and their functional and spatial states, moving beyond the rapidly evolving literature and existing concepts of CNS stromal cell biology. We have also clarified and aligned existing data and descriptions, and we have updated our manuscript throughout to more precisely and quantitatively address the origin and function of brain fibroblasts.

1. Col1a2+ cells contribute to about 50% of the lesional (myo)fibroblasts. An undefined part of this contribution comes from Col1a2+ cells in the dura/meninges. What is the cellular origin of the remaining fibroblasts?

2. Using lineage tracing, a previous study (PMID: 34535655) reported that Slc1a3/Glast+ perivascular cells contributed ~80% to all scar-forming PDGFR β + stromal cells after cortical ischemic lesions. Considering the reported 50+% derived from the Col1a2-CreERT2 line, there needs to be at least a substantial overlap regarding the cellular origin. This should be addressed or at least discussed.

We thank the reviewer for raising these important questions and have addressed them via several experimental approaches detailed below, adding substantial new data to strengthen our conclusions that pre-existing fibroblasts are the major source of brain lesional fibroblasts.

1. We have now quantified the baseline efficiency of $Col1a2^{creER}$ in resting, uninjured brain tissue, using $Col1a1^{GFP+}$ cells as a denominator to represent the total and current “fibroblast pool”²⁰ (**Extended Data Fig. 1i**, reproduced below). We found that $Col1a2^{creER}$ is highly specific for fibroblasts in the brain – recombining only in $Col1a1^{GFP+}$ cells (**Extended Data Fig. 1j**, reproduced below) and validating our use of this tool as a fibroblast-specific *Cre*. However, its sensitivity (i.e., efficiency; TdT+ as a proportion of $Col1a1^{GFP+}$) is only 50% in *resting brain* fibroblasts. This resting efficiency is consistent with the proportion of lineage-traced lesional fibroblasts observed after injury (**Extended Data Fig. 1k,l**, reproduced below), suggesting that the observed 50% recombination rate after injury is *not* due to a separate source of lesion-infiltrating fibroblasts (e.g., pericytes that upregulate *Col1a2*). Any emergence of new non-lineage-traced $Col1a1^{GFP+}$ fibroblasts (TdT-) would reduce the post-injury lineage-traced proportion below the baseline efficiency of 50%. Rather, our data suggests a technical impact of an inefficient *Col1a2*-driven CreERT activity, perhaps due to a relatively weaker fibroblast *Col1a2* expression prior to injury, but remains consistent with a predominantly fibroblast-derived response to injury. The higher recombination efficiency with tamoxifen treatment *after* PT injury (**Extended Data Fig. 1l**, reproduced below) is consistent with the known upregulation of *Col1* expression within fibroblasts after injury¹⁴.

Extended Fig. 1i-l: Preserved fibroblast recombination efficiency at rest and after injury

2. To formally address the possibility of a pericyte contribution to the fibrotic scar fibroblasts, as referenced above, we have now included new data from lineage tracing in pericyte reporter mice ($Ng2^{creER}$; $Rosa26^{TdT}$). Unlike *Cre* alleles used in prior studies^{12,17,18}, this line does not significantly recombine fibroblasts. While it does efficiently recombine oligodendrocyte lineage cells, $Ng2^{creER}$ has been used to quantify pericyte contributions to fibrotic scarring in other contexts^{11,13,15}. Using overlap with CD31 as a proxy for perivascular pericyte location, we found that ~84% of TdT⁺ cells (i.e., $Ng2$ -lineage⁺ cells) are perivascular (vs. just ~23% of oligodendrocyte-lineage cells, SOX10⁺), consistent with significant pericyte recombination by this $Ng2^{creER}$ line, as expected (**Fig. R2.1**, below). Additionally, we quantified the co-positivity of TdT with markers of oligodendrocyte-lineage cells (SOX10) or pericytes (Desmin; **Extended Data Fig. 1p**, reproduced below). While recombination was more efficient within the oligodendrocyte lineage, pericyte recombination was sufficient to proceed (**Extended Data Fig. 1q**, reproduced below). We observed minimal contribution of $Ng2$ -lineage-traced cells to the pool of lesional fibroblasts (**Extended Data Fig. 1o,s**, reproduced below). While

$Ng2$ -lineage⁺ cells did expand in perilesional locations, they were confined to GFAP⁺ regions of astroglial scarring, unlike the $Col1a2$ -lineage⁺ fibroblasts that resided within the lesion. Finally, in pericyte/fibroblast double reporter mice ($Ng2^{creER}$; $Rosa26^{Tdt}$; $Col1a1^{GFP}$), we observed minimal overlap between $Ng2$ -lineage⁺ cells (including pericytes) and $Col1a1^{GFP+}$ fibroblasts by 21dpi (and as compared with $Col1a2^{creER}$ lineage tracing; **Fig. R2.2, below**).

Fig. R2.1: $Ng2^{creER}$; $Rosa26^{Tdt+}$ cells exhibit a perivascular topography

Extended Data Fig. 1o-q,s: $Ng2$ -lineage⁺ cells (including pericytes) do not detectably infiltrate lesions

Fig. R2.2: Minimal overlap of $Ng2$ -lineage⁺ cells and $Col1a1^{GFP+}$ fibroblasts at 21dpi

These results could be interpreted to contrast with the reported contribution of a $Glast$ ⁺ pericyte population after spinal cord injury (referenced above). However, our own sequencing data (**Fig. R2.3, below**) reveals substantial expression of $Glast$ (i.e., $Slc1a3$) within PT lesional fibroblasts at all time points, confirming previous findings^{19,21}. Subsequent analysis from the cited group found that 98.3% of fibroblasts (lineage-traced with a $Col1a1^{creER}$ allele) expressed $Slc1a3$, and that these fibroblasts significantly contributed to the stromal scar after spinal cord

injury¹², potentially consistent with a *Slc1a3*-lineage⁺ fibroblast origin for stromal cells after PT injury.

Fig. R2.3: Expression of *Slc1a3* (*Glast*) across fibroblast subsets and timepoints

- To directly investigate if fibroblasts, and not pericytes, predominantly contribute to CNS fibrosis in a discrete model of brain injury, we have now included new data, generated in collaboration with Dr. Richard Daneman at UCSD, involving distal middle cerebral artery occlusion (dMCAO), a distinct model of stroke involving permanent electrocoagulation of the distal MCA. By 14dpi, lineage-traced fibroblasts (*Col1a2*^{creER}; *Rosa26*^{TdT}) infiltrated and expanded within the fibrotic lesion (marked by *Col1* and surrounded by a *GFAP*⁺ glial scar, as in the PT injury model; **Extended Data Fig. 1t**, reproduced **below**). In contrast, lineage-traced pericytes (*Atp13a5*^{creER}; *Rosa26*^{TdT}) maintained a perivascular distribution and did not visibly expand within the lesion (**Extended Data Fig. 1u**, reproduced **below**), suggesting that the biology of fibroblast-derived fibroblasts observed in PT injury may extend across ischemic brain injuries, and potentially other categories of CNS injury.

Extended Data Fig. 1t,u: Fibroblasts – but not pericytes – expand within lesions after dMCAO

Collectively, these data indicate that a large majority of lesion-resident fibroblasts derive from pre-existing fibroblasts, though we cannot rule out that a small population of pericytes not recombined by our pericyte reporters may contribute to the pool of lesional fibroblasts. The differences in reported pericyte contributions to brain fibroblast ontogeny across studies may represent an important distinction in the biology of traumatic spinal cord injury and stroke, or between spinal cord and brain. Indeed, fibroblasts were reported to contribute more to white-matter-associated spinal cord injuries – with closer proximity to the spinal meninges, analogous to our cortical injury model – whereas pericytes contributed more to deeper grey-matter-associated injuries¹². These possibilities are now raised in the revised manuscript discussion.

3. Along the same line, the authors show results from lineage tracing of Gli1+ cells using Gli1-CreERT2xR26-TdTomato mice. Judging from the pictures shown in Extended Data Fig. 1i, it seems Gli1+ cells contribute substantially to scar-forming fibroblasts. However, quantification is missing. Do Gli1+ cells contribute more fibroblasts to the scar? In this case, why did the authors not use this line instead? Is the contribution similar to the Col1a2 line? This would further substantiate the evidence for another source besides fibroblasts.

We thank the reviewer for raising this issue. We have now clarified this point by emphasizing in the text that we use *Gli1* as a marker of fibroblast subset(s), as appreciated in peripheral organs. *Gli1* also labels border-associated fibroblasts in the CNS (**Fig. R2.4, below**); qualitative lineage tracing data was originally included to orthogonally highlight that pre-existing fibroblasts contribute

to the injury-responsive pool of fibroblasts. However, *Gli1* marks both fibroblasts and a subset of astrocytes in the CNS (**below**), and therefore the *Gli1^{creER}* line is not well suited for manipulation of fibroblasts or definitive lineage tracing, leading to our preferred use of the *Col1a2^{creER}* line throughout the manuscript. We have now added quantification of lesional *Gli1*-lineage⁺ cells (**Extended Data Fig. 1s**); *Gli1^{creER}* recombines fewer cells than *Col1a2^{creER}*, potentially reflecting a subset of *Gli1*⁺ fibroblasts contributing to the lesional pool, or alternatively reflecting varying CreERT efficiencies.

Fig. R2.4: *Gli1^{creER}*; *Rosa26^{TdT}* labels border-associated fibroblasts and astrocyte subsets

4. *Col1a2*-CreERT2;tdTom mice label perivascular fibroblasts and all meningeal fibroblasts (dura, arachnoid, and pia). It is unclear from their lineage tracing experiments using the *Col1a2*-CreERT2 line (Ext Data Fig. 1h, l-q) what is the source of lesional (myo)fibroblasts. Do all these different fibroblast populations contribute to lesional myo(fibroblasts)?

5. The authors perform sparse labeling of dural fibroblasts and find sparse labeling of lesional fibroblasts. Once more, the interpretation of these results is unclear. What happens if dural meningeal fibroblasts are efficiently recombined? Do 50% of all lesion fibroblasts derive from the dural fibroblasts? These questions need to be clarified.

We thank the reviewer for calling attention to this lack of clarity, and we have revised our language to more clearly and precisely reflect the data. In short, and in part as described above, lineage tracing in fibroblast and mural cell reporters suggests that lesional fibroblasts predominantly derive from pre-existing fibroblasts, with the referenced “50%” labeling efficiency arising from *Cre* inefficiency rather than a separate fibroblast source. Given this apparent fibroblast-derived response, we also used sparse labeling of dural fibroblasts to separately ask the question of the *microanatomical origin* of these fibroblasts. Transcranial 4-OHT – the bioactive, short-lived component of tamoxifen – sparsely recombined dural fibroblasts (8-10%; **Extended Data Fig. 1w,z**) but did not detectably recombine perivascular fibroblasts or leptomeningeal (arachnoid, pial) fibroblasts, which in our hands remain largely attached to the brain during removal from the cranium (**Extended Data Fig. 1x-z**, and see **response 6**, below).

Transcranial 4-OHT resulted in a similar proportion of fibroblasts expressing TdT in the *lesion* as is observed in the *dura* (**Extended Data Fig. 1aa,ab**). This data suggests a large proportion of PT lesional myofibroblasts derive from dural fibroblasts, as a significant contribution from *non-lineage-traced* fibroblasts (e.g., perivascular or leptomeningeal fibroblasts) would *lower* the lesional transcranial lineage tracing proportion relative to the dural meninges (i.e., less than 8-

10%). Unfortunately, as transcranial application of 4-OHT requires invasive surgery, 4-OHT can only be applied once, limiting the efficiency of the technique. Nevertheless, these results are collectively consistent with a fibroblast injury response that (1) derives from pre-existing *Col1a2*-lineage⁺ fibroblasts, which (2) largely arise from the dural meninges. However, absent a *Cre* allele sensitively and specifically recombining in arachnoid/pial fibroblasts, we do not rule out a contribution from leptomeningeal or perivascular fibroblasts, and we have clarified this issue in the text.

6 Ext. Data Fig. 1: Scheme in (l) should contain arachnoid represented. It is unclear whether the arachnoid meningeal layer is labeled or not by transcranial application of 4-OH-tamoxifen, as this meningeal layer is omitted from images (n) and analyses (o). Images in (n) should be labeled with DAPI and provide evidence that the arachnoid meningeal layer is not recombined by transcranial application of 4-OH-tamoxifen (higher resolution required). In addition to perivascular and pial fibroblasts, quantifications in (o) should include arachnoid and dura meningeal fibroblast numbers, as intraperitoneal tamoxifen injection results in recombination of all *Col1a2*⁺ fibroblasts.

We appreciate the reviewer's inclusion of arachnoid fibroblasts in this discussion. Based on the separation of the leptomeninges (pial and much of arachnoid layers) from the dural meninges during our microdissection, we are treating brain-associated meninges – visible in coronal sections – as leptomeninges, including both pial *and* many arachnoid cells. We have now included higher resolution images (**Extended Data Fig. 1x**) and updated schematics (**Extended Data Fig. 1v**), labels (**Extended Data Fig. 1z**), and the manuscript text to emphasize that neither the pial nor arachnoid leptomeninges are detectably recombined by transcranial 4-OHT. To further test the lack of arachnoid meningeal recombination by transcranial 4-OHT, we have also included high-resolution images of *Aldh1a2* and E-Cadherin, markers of arachnoid meningeal fibroblasts^{22,23} (**Extended Data Fig. 1y**, reproduced **below**). Moreover, we have included data from dural meninges in our analysis of perivascular and leptomeningeal recombination (**Extended Data Fig. 1z**, also shown as normalized numbers in **Extended Data Fig. 1ab**).

Extended Data Fig. 1y: Arachnoid fibroblasts are not detectably recombined by transcranial 4-OHT

6.....From images in (p), it does not seem Col1a2-tdTom+ cells are positive for ER-TR7. Have these cells lost ER-TR7 expression? Or are these cells not expressing ER-TR7 (reticular fibroblast ECM marker)? Or are these Col1a2+ cells from the skull that are not fibroblasts (e.g., Osteoblasts / Bone cells)?

Based on confocal imaging, the fibroblasts in **Extended Data Fig. 1aa** do express ER-TR7 (**Fig. R2.5, below**), albeit more dimly. We have clarified the text to emphasize that ER-TR7 is an ECM molecule (likely Collagen VI²⁴) that is enriched in extracellular spaces and often used as a proxy for fibroblast-inhabited regions^{11,13}. While we cannot rule out a contribution from bone-associated Col1a2⁺ cells from microscopy alone, our snRNAseq lesional data did not reveal a population of PT lesional osteoblasts or osteocytes, although these subsets were identified in the dural meninges specimens.

Fig. R2.5: Optical slice view (via confocal imaging) of **Extended Data Fig. 1aa**

7. Are “meninges” quantifications in Ext Data Fig. 1e referring to dura meningeal fibroblasts only or fibroblasts from the dura and leptomeninges? The description of this figure in the main text implies that quantifications refer to the dura mater only. Why are not all meningeal fibroblasts quantified? Also, bear in mind that not all fibroblasts in the meninges express *Pdgfra* (arachnoid layer fibroblasts do not express *Pdgfra*). See <https://doi.org/10.1016/j.neuron.2023.09.002>). Therefore, *Pdgfra*-GFP mice might not be the best option for this quantification. Why are Col1a2-CreERT2;Rosa26cGFP mice not used instead?

Meninges quantifications in our nuclear flow cytometry data (**Extended Data Fig. 1f**) refer to the dural meninges. We have found that dural meninges can be readily removed with the cranium (skullcap), whereas the leptomeninges – especially in the region of the fibrotic lesion – are difficult to reproducibly and accurately remove from the brain. As quantification of dural fibroblasts serves only as a positive control to provide context for the fibroblast-rich PT lesion, we did not include additional meningeal layers for this particular experimental approach. Of note, we and others have previously documented that this specific mouse *Pdgfra* reporter expresses a particularly stable and long-lived nuclear-localized GFP, and therefore GFP⁺ cells in our system likely include stromal cells/fibroblasts that have previously expressed *Pdgfra* but with low levels of current *Pdgfra*. We have clarified this issue in the text.

To further compare the sensitivity of these two mouse reporter strains, we have now directly compared *Pdgfra*^{GFP} mice with Col1a2^{creER}; Sun1^{GFP} mice induced with tamoxifen beginning on day 0 (**Fig. R2.6, below**). Interestingly, *Pdgfra*^{GFP} nuclei represented a slightly *higher* proportion of total PT lesional nuclei, as compared with a “gold standard” for fibroblasts, the Col1a2^{creER}; Sun1^{GFP} mice, and suggesting *Pdgfra*^{GFP} is efficiently marking at least a majority of injury-

associated PT fibroblasts. We opted to use $Pdgfra^{GFP}$ for the above reasons, and we can include this data if desired.

Fig. R2.6: Lesional fibroblast nuclei (GFP^+) in $Pdgfra^{GFP}$ or $Col1a2^{creER}; Rosa26^{Sun1GFP}$ mice

8. The recombination induction protocols are complicated to interpret for experiments involving $Col1a2$ -CreERT2;tdTom mice. For some experiments (tMCAO, Fig. 1d; PT stroke Ext Data Fig. 1h), recombination is induced before injury, and results show lineage tracing of $Col1a2^+$ cells. For some other experiments, recombination is induced before and after injury (PT stroke, Ext Data Fig. 1b), or recombination is induced after injury (5-7 days after injury, PT stroke, Fig. 1b). It is unclear why all these experiments use different approaches for recombination, as results are difficult to interpret...

We appreciate these concerns and have revised our text and figure legends throughout for clarity and to appropriately emphasize one of the following for each experiment described.

- Tamoxifen induction exclusively before the injury was used for the purpose of lineage-tracking pre-existing fibroblast (or other stromal cell) populations (e.g., **Fig. 1b,d, Extended Data Fig. 1c,i,k,m,o,r,t,u**).
- Tamoxifen induction before and after injury was used (e.g., **Extended Data Fig. 1d**) in order to overcome the inefficiency of the $Col1a2$ allele, which is only able to recombine ~50% of fibroblasts at rest but can recombine virtually all $Col1a1^{GFP+}$ cells after injury (**Extended Data Fig. 1l**). Recombination before PT injury was also used for deletion of genes (e.g. $Tgfbr2$) so that the associated proteins would already be absent at the time of initial injury. In this context, post-injury tamoxifen induction enabled us to maintain the deletion of a gene in the face of potential selective pressure, which favors the rebound/emergence of $Tgfbr2$ -sufficient fibroblasts that had not been initially recombined.
- Tamoxifen induction exclusively after an injury was, in general, not performed, except when aimed at explicitly characterizing Cre activity in the late post-injury phase (to characterize Cre efficiency, e.g., in **Extended Data Fig. 1l**, or to characterize gene expression trajectory, e.g., **Fig. 2l,m, Extended Data Fig. 4j-l**). For example, these approaches allowed us to show that $Cthrc1$ transcript (Cre -ERT activity) was no longer expressed by lesional fibroblasts at late time points post-PT injury.

We appreciate the chance to clarify our manuscript and rationale for using these distinct tamoxifen regimens to answer distinct questions. We believe that each adds to our full understanding of fibroblast origins, dynamics, and functions, and we hope they are now described with more clarity in the revised manuscript.

8...If the aim is to visualize fibroblasts at different stages after injury and across different types of lesions, then the authors should use Col1a1-GFP mice consistently across all models and time points as in Fig. 1f. In addition, for CCI experiments, the authors show the presence of Col1 and ER-TR7 (recognizing collagen type VI) ECM deposition in the lesion. This does not provide direct evidence that fibroblasts populate these lesions. Once more, Col1a1-GFP mice should be employed for CCI experiments.

We agree that Col1a1^{GFP} provides a more straightforward way of presenting fibroblast kinetics, while also providing more convincing evidence for the presence and expansion of fibroblasts than fibroblast-associated ECM staining. We have therefore performed confocal microscopy on Col1a1^{GFP} brains after traumatic brain injury (CCI model), revealing the infiltration of GFP⁺ fibroblasts, with paired lineage tracing data (**Extended Data Fig. 1b,c**, reproduced below) to highlight their fibroblast ontogeny. Of note, microscopy from Cthrc1^{creER}; Rosa26^{TdT} mice suggests that CCI-associated fibroblasts also pass through a myofibroblast state (**Extended Data Fig. 4p**, reproduced below). We have added quantification of Col1a1^{GFP} fibroblast expansion across all the orthogonal models of fibroblast perturbation utilized (clodronate, ADWA11, and *Tgfr2* cKO; please also see **responses 18 and 19**, below).

Extended Data Fig. 1b,c: Fibroblasts infiltrate after TBI model (CCI)

Extended Data Fig. 4p: TBI-induced fibroblasts pass through a Cthrc1-lineage⁺ state

9. Lines 100-103 “Fibroblasts were first detected near the leptomeninges by 4dpi (Extended Data Fig. 1b) and had surrounded and infiltrated the lesion by 14-21dpi (Fig. 1f, Supplementary Video 1), where they persisted for at least one year post-injury (Fig. 1g, Extended Data Fig. 1c).” – This description needs to be refined as fibroblasts are clearly visible in the lesion already at 7 days after PT injury (see Fig. 1f).

We have updated the text to reflect that fibroblast infiltrate the lesion by 7dpi but do not fully surround it until 14dpi.

10. Legends need refinement to describe what is shown in the images. For example, Ext Data Fig 1f-g, “Vascular remodeling after PT injury, with PdgfraGFP+ nuclei (f) or fibroblasts (g) visualized in perivascular spaces”. These images show the lesion and not perivascular spaces. Ext. Data Fig. 1f shows no vascular remodeling.

We have updated our legends accordingly.

11. It is unclear how many fibroblasts and other cell types are included in the snRNAseq analyses. Only the number of total nuclei is stated. Please state the number of analyzed nuclei instead.

We have updated the text and methods to reflect the analyzed cell types for each of our two RNA-Seq experiments. For experiment 1, analyzed cells comprise 8,096 fibroblasts; 189 mural cells; 4,568 myeloid cells/microglia; 548 T cells; 11,216 neurons; 2,026 astrocytes; 537 oligodendrocytes; 94 oligodendrocyte precursor cells; 470 endothelial cells; 259 neural progenitor cells; and 184 unassigned nuclei. For experiment 2, analyzed cells comprise 18,455 fibroblasts; 496 mural cells; 24,302 myeloid cells/microglia; 2,271 T cells; 5,994 neurons; 2,781 astrocytes; 1,669 oligodendrocytes; 2,399 oligodendrocyte precursor cells; and 1,703 endothelial cells. For space reasons, the number of analyzed fibroblasts and mural cells have been added to the main text, while additional cell types have been added to the Methods.

12. As specified in the materials and methods section, the authors excluded pericytes from their murine snRNAseq analysis (for example in Fig. 2e and Fig. 3b). Given that these cells are a potential source of lesional (myo)fibroblasts, the analyses would be substantially enhanced if these cells were not removed. The authors should include these cells in the analyses.

We agree that pericytes are an important cell type in the context of CNS injury, and we have updated the figures and methods as suggested by the reviewer.

13. Refrain from using the term “late myofibroblasts” for 21dpi. Fibroblasts at this stage are not myofibroblasts. As the authors show in fig. 2j-m, the myofibroblast stage is transient and very few cells exhibit myofibroblast features from 7-9 days after injury, with no lesional fibroblasts being labeled with Cthrc1;tdTomato at 21dpi (only a few cells in the meninges are tdTom+).

We agree that this was an imprecise term and that this fibroblast state does not represent a *bona fide* myofibroblast identity. Therefore, we have updated our terminology throughout the manuscript. These fibroblasts are now referred to as “late inner” lesional fibroblasts, reflecting their unique spatiotemporal topography, as determined by new experiments via immunofluorescent microscopy (**Fig. 3d, Extended Data Fig. 6a-k**).

14. Ext Data Fig. 5a shows that Cthrc1;tdTom+ dural fibroblasts react to PT injury in the meninges but seemingly do not contribute to lesional (myo)fibroblasts (contrary to Col1a2+ lineage traced dural fibroblasts – Ext Data Fig.1 l-q). Specify in the figure legend and/or text when recombination was induced for this experiment and provide an interpretation of why you think Cthrc1;tdTom+ dural fibroblasts and Col1a2-tdTom+ dural fibroblasts respond differently to PT injury.

We thank the reviewer for raising this issue and have clarified the legend to reflect that recombination in this experiment was induced beginning on day -1 to enable full recombination by 2dpi and continued throughout the injury time course. Therefore, these images (now **Extended Data Fig. 4f**) reveal an emergent dural myofibroblast response to injury. As *Cthrc1*-lineage⁺ cells are relatively rare in the resting meninges, this represents a post-injury upregulation of *Cthrc1* in the dural meninges, similar to that seen in the lesional cortex (**Fig. 2h-j,l,m**), and is also consistent

with our model of dural fibroblast-derived lesional fibroblasts. This approach is distinct from lineage-tracing ontogeny experiments, where fibroblasts are induced solely *prior* to injury; in this lineage-tracing regimen, *Col1a2*-lineage⁺ fibroblasts contributed to the lesion-associated pool, while *Cthrc1*-lineage⁺ fibroblasts – absent in resting cortex and sparse in resting meninges – did not (**Fig. 2k**). This divergence likely reflects the sparsity of *Cthrc1*-lineage⁺ myofibroblasts in most resting tissues of mice, including in the naïve brain and associated meninges.

15. Legend is missing for Ext Data Fig 5h.

We apologize for the error and have corrected the legend.

16. In Ext Data Fig. 5c tamoxifen induction should be from -16 until -14 days (and not from 16 to 14 days). The same in Fig. 2i.

We have corrected the figures to make the negative signs more visible (now **Fig. 2k, Extended Data Fig. 4i**).

17. Fig 3: it would be interesting to visualize lymphocytes on the spatial data and cell-to-cell communication with the late-fibroblast population to understand how these populations interact.

Unfortunately, lymphocyte genes are sparse in many spatial transcriptomics datasets, including our own. This is likely related to the relatively rarer lymphocytes inhabiting similar spaces to fibroblasts, macrophages/microglia, and other dominant brain-resident cells, making meaningful detection difficult relative to more sensitive approaches such as single nuclear, flow cytometric, or microscopy-based approaches. Nonetheless, lymphocyte transcripts are detected (including *Cd3e*, *Cd4*, and *Cd8a*) and are consistent with our imaging data: transcripts are minimal at early time points – with variable peri-ventricular transcripts – and increased at chronic time points near the lesion (**Fig. R2.7, below**).

Fig. R2.7: Lymphocyte transcripts in Visium data

Additionally, we have now performed cell-cell interaction analysis (**Fig. R2.8, below**, via CellPhoneDB and using our single nuclear RNA sequencing data) between T cells – including 2 main clusters of conventional and $\gamma\delta$ T cells – and late lymphocyte-interactive fibroblasts

(“Fibroblasts”). Our analysis highlights potential interactions involving costimulatory molecules (e.g., *Cd80*), chemokines (e.g., *Cxcl12*), and integrin ligands (e.g., *Vcam1*, *ICAM1*). We would be happy to add this data to the manuscript but have not included it for now in lieu of a more direct analysis of fibroblast-lymphocyte interactions (**Figure 5**).

Fig. R2.8: Cell-cell interaction analysis between lymphocyte-interactive fibroblasts and T cells (CellphoneDB)

18. In Fig. 4a, ER-TR7 EMC deposition and lesion size are altered between control and clodronate groups. Does the total number of lesional fibroblasts (Col1a2-CreERT2;tdTom+ cells) change as well? Additionally, are only lesional fibroblasts in the inner lesion core (macrophage-contacting fibroblasts) affected, and are microglia-contacting lesional fibroblasts unaffected/less affected? Provide quantifications as this would enhance the analyses and provide relevant information.

We thank the reviewer for this comment and have added quantification of Col1a1^{GFP+} fibroblasts after clodronate treatment. Clodronate-treated mice show a reduction in overall GFP⁺ fibroblast density (**Extended Data Fig. 7f**, reproduced **below**); we observed a relative redistribution of fibroblasts from the dense border to the sparser inner core (**Fig. R2.9**, **below**; **Extended Data Fig. 7g,h**, reproduced **below**). Notably, both macrophages and microglia are enriched in the outer lesion border (**Extended Data Fig. 5c-i**).

Extended Data Fig. 7f-h: Reduction in fibroblasts after clodronate treatment

Fig. R2.9: Redistribution of fibroblasts to sparser inner core after clodronate treatment

19. Similar quantifications should be provided in Fig. 4f (control versus cKO) and Fig. 4j (IgG versus ADWA11), does the total number of Col1a2-CreERT2;tdTom+ cells change between conditions? In addition, are only lesional fibroblasts in the inner lesion core (macrophage-contacting fibroblasts) affected, and are microglia-contacting lesional fibroblasts unaffected/less affected? Provide quantifications.

We have now added quantification of fibroblasts in both *Tgfb2* conditional knockout (cKO) mice (using Rosa26^{TdT} driven by Col1a2^{creER}) and in ADWA11-treated Col1a1^{GFP} mice. *Tgfb2* cKO mice show decreased fibroblast density (**Extended Data Fig. 8c**, reproduced below). Specifically, fibroblasts in the “inner” lesion core were much sparser (**Extended Data Fig. 8d**); meanwhile, fibroblasts in the lesion border maintained their relative density (e.g. the border was preserved, **Extended Data Fig. 8e**), though the border itself becomes thinner (**Extended Data Fig. 8f**), summarized in **Extended Data Fig. 8g**. We did not observe a relative redistribution of fibroblasts between lesional regions (**Fig. R2.10**, below).

Extended Data Fig. 8c-g: Reduction in inner core fibroblasts and thinner fibroblast border in *Tgfb2* cKO mice

Fig. R2.10: No redistribution of fibroblasts in *Tgfb2* cKO mice

ADWA11-treated mice also showed an overall decrease in fibroblast density (**Extended Data Fig. 8w**, reproduced **below**), also predominantly due to decreased density within the inner lesion (**Extended Data Fig. 8x,y**, reproduced **below**). These decreases were milder as compared with cKO mice, consistent with our sequencing data showing that most fibroblast subsets were not numerically impacted by ADWA11, with the exception of “late inner fibroblasts” (**Fig. 4t**, **Extended Data Fig. 9j**), though TGF β -associated gene expression was significantly impaired (**Fig. 4u**, **Extended Data Fig. 9q,r**). Similarly to the cKO mice, we did not observe a relative redistribution of fibroblasts between lesional regions (**Fig. R2.11**, **below**).

Extended Data Fig. 8: Reduction in inner core fibroblasts in integrin $\alpha_v\beta_8$ -blockaded mice

Fig. R2.11: No redistribution of fibroblasts in integrin $\alpha_v\beta_8$ -blockaded mice

20. Across Figs. 2n, 3j, 3k, 4b, 4c, 4g, 4h, 4k, 4l, 4n, 4s, and 4t, what do individual data points in the plots represent? It is unclear if each dot represents an individual animal or the number of slices analyzed. For transparency and statistical purposes, each dot in the plots should represent one animal, and this should be used for statistical tests. The same applies to all Ext Data Fig. plots.

We have updated all figures and legends such that each point in all plots represents one animal, and statistical tests have been updated accordingly. In a small number of situations where intra-animal variability may be relevant, we have *also* included per-slice values – on the same graphs but in distinct colors and sizes – and clarified this in the figure legends.

21. For experiments presented in Fig. 4d-h using Col1a2-CreERT2;tdTom;Tgfb2flox cKO mice, recombination is induced before injury and after injury. This essentially means that any cell that upregulates Col1a2 after injury (e.g., pericytes, smooth muscle cells, etc.) is also blinded to TGFbeta signaling. For this experiment to be restricted to fibroblasts, recombination needs to be induced before injury followed by a washout period, and then the injury should be performed.

While our lineage tracing data in fibroblast reporters (**Extended Data Fig. 1k,l**), pericyte reporters (**Extended Data Fig. 1o**), and smooth muscle reporters (**Extended Data Fig. 1r**) suggest that pericytes and smooth muscle cells do not make detectable contributions to the injury-associated fibroblast pool (**Extended Data Fig. 1s**), we agree that our tamoxifen regimen captures all cells that upregulate *Col1a2* after injury. Given the inefficiency of the *Col1a2*^{creER} allele (50% at baseline, >90% after injury, **Extended Data Fig. 1i,l**), post-injury tamoxifen was necessary to fully recombine even pre-existing fibroblasts. Additionally, fibroblasts recombined prior to injury – being unable to proliferate or otherwise expand – may fail to fill the lesional niche, which could lead to compensatory expansion of un-recombined pre-existing fibroblasts, particularly given the selective pressure of the lesional environment (including high levels of TGF β , integrins, inflammatory signals, etc.). We directly tested this hypothesis by limiting tamoxifen to a pre-injury period, without follow-up tamoxifen induction after injury, and observed no significant decrease in ECM deposition or increased lesion size (**Fig. R2.12, below**). Finally, even if injury-associated fibroblasts do arise from non-fibroblast cellular ontogenies in some settings, their fibroblast identity by 7dpi (as defined by *Col1a2* expression and snRNAseq) makes these cells pertinent to our work, which also seeks to define the roles of injury associated fibroblasts independently of their ontogeny. These points are now clarified within the manuscript and discussion.

Fig. R2.12: Tamoxifen pre-treatment is insufficient to prevent fibroblast expansion

22. Fig 4p: the heatmap should be replaced by violin plots to determine if the number of cells expressing the mentioned genes is affected.

We have added violin plots to highlight decreased expression of myofibroblast-related genes among myofibroblasts (**Extended Data Fig. 9q**) and to complement new (**Extended Data Fig. 9k**) and existing data (**Fig. 4t**) showing a decrease in myofibroblasts. We have left **Fig. 4u** in the manuscript to emphasize relative differences between groups.

23. Why use *Emx1-Cre* to delete *Itgb8* in injury-induced reactive glia? In *Emx-Cre;Itgb8flox* mice, *Itgb8* is deleted in glia and neurons across embryonic and postnatal development, which can greatly affect the interpretation of the results. To provide solid results without confounders, the authors should use an inducible CreERT2 mouse line with a glia-specific promoter instead.

We thank the reviewer for raising this issue, which was also raised by Reviewer 1, and agree that *Emx1*^{cre} will recombine glial cells and neurons across development; indeed, the developmental effects of *Emx1*^{cre}-induced *Itgb8* deletion, and subsequent impaired TGF β signaling, have been well characterized^{25,26}. Therefore, we have added new data to our manuscript demonstrating that conditional deletion with the constitutive glial *Cre* allele *hGfap*^{cre} also caused a significant reduction in fibrosis (**Fig. 4r,s**, reproduced below). While *hGfap*^{cre} (like *Emx1*^{cre}) is now known to

recombine subsets of neurons, it is expressed later during development than $Emx1^{cre}$. As a result, conditional deletion of $Itgb8$ by $hGfap^{cre}$ does not cause developmental abnormalities seen with $Emx1^{cre}$, and conditional knockout mice show no neuropathological changes²⁶.

Fig. 4r,s: *hGfap-restricted Itgb8 deletion recapitulates reduction in fibrosis*

Interestingly, conditional deletion of $Itgb8$ with $Aldh11^{creER}$, an inducible astrocytic $CreERT$ line, did not recapitulate a significant reduction in fibrosis (**Fig. R2.13, below**), although inefficient $Itgb8$ deletion efficiency in this line precludes definitive interpretation of these results. Collectively, our new evidence, along with our complementary pharmacological data from ADWA11 $\alpha_v\beta_8$ blocking experiments, is most consistent with the redundant expression of $Itgb8$ across glial subsets – as suggested by snRNAseq (**Extended Data Fig. 8v**) – with resulting partial abrogation of fibroblast expansion after astrocyte-restricted deletion. Nevertheless, we do not rule out potential confounding effects caused by deletion of $Itgb8$ in neurons and/or during development and have updated our manuscript text accordingly. We can include the below reviewer-only data in our manuscript if desired.

Fig. R2.13: *Fibrosis after PT injury (14dpi) in controls or $Aldh11^{creER}; Itgb8^{floX}$ mice*

24. In the discussion, put your findings into context with previous literature classifying microglia/macrophages into previously described polarization states.

We appreciate this suggestion and now more thoroughly position our data within previously described microglial/macrophage states in the revised discussion.

Referee #3 (Remarks to the Author):

This manuscript by Ewing-Crystal et al. uses multiple approaches (for example, single-nuclei RNAseq, spatial transcriptomics and transgenic mouse models) to investigate the role of fibroblast subpopulations as regulators of CNS wound healing and neuroinflammation following brain injury. This is a well-written manuscript, with clear figures presented throughout, however I think a significant amount of further work is required to fully leverage the datasets presented. For example, to add significant conceptual advance, can the authors manipulate a specific fibroblast subpopulation (identified and fully validated from their single nuclei RNAseq datasets) to demonstrate that these new datasets facilitate precision medicine approaches (via specific targeting of the critical fibroblast subpopulation(s)) in the context of brain injury and wound healing? My specific comments are listed below:

We thank the reviewer for their positive comments on the manuscript, figures, and multimodal approach to defining CNS fibroblasts. We agree that our presented datasets and prior functional analysis invited further validation and manipulation of brain fibroblast states. To address this and further the conceptual advance of our work, we have now added significant new data aimed at (1) fully and orthogonally validating the emergence of a transient myofibroblast state that resolves into distinct late fibroblast states; (2) defining a specific and temporal role for the early myofibroblast state in limiting extent of CNS perilesional damage and restraining secondary neuroinflammation; and (3) defining a novel role for late lesional fibroblasts and their CXCL12 production in providing long-term brain T cell niches and dampening T cell cytokine expression. We have also added data throughout the manuscript with expanded bioinformatic comparisons between fibroblast and macrophage biology in our murine PT stroke and single-cell/nuclear transcriptomic datasets from non-human primates and human CNS injury. Overall, we believe that our new data, detailed below, represents a significant conceptual advance over current understandings of CNS fibroblast heterogeneity and discrete temporal functions after injury, with multiple areas of clinical relevance to brain and spinal cord diseases. We are grateful for the opportunity to elaborate on these points of response.

1. Figure 1, line 111: the authors use a Col1a1GFP; Col1a2creER;R26TdT transgenic mouse approach. How efficient was the initial labelling (ie baseline recombination rate / TdTomato report in Col1a2+ cells in Col1a2CreER mice) in uninjured tissue? Furthermore, has the Col1a2creER mouse previously been shown to be specific in the CNS for fibroblasts? Does it only label fibroblasts? What proportion of total CNS fibroblasts does it label? Data should be presented regarding this.

- the authors state that 'By 21dpi, over half of lesional fibroblasts (GFP+) were derived from pre-existing Col1-expressing fibroblasts (TdT+; Extended Data Fig. 1h)'. Where did the remaining lesional fibroblasts derive from? The authors then describe lineage tracing of meningeal fibroblasts with 'an equal frequency of TdT+ fibroblasts in the dura and lesion after PT injury (~10%)', but it remains unclear to the reader as to where the remaining 50% of lesional fibroblasts arise from.

We thank the reviewer for raising these important questions, which were also raised by Reviewer 2, and have addressed them via several experimental approaches detailed below, adding substantial new data to strengthen our conclusions that pre-existing fibroblasts are the major source of brain lesional fibroblasts.

1. *Col1a2* was validated as a specific marker of fibroblasts in the CNS by comparison of *Col1a2*-lineage⁺ cells with cells actively expressing *Col1a1*^{GFP}, a well-established specific fibroblast

reporter^{8,11,20}, with the 2 alleles displaying almost complete overlap. However, the exact $Col1a2^{creER}$ allele used throughout our manuscript has not been previously used or validated in the CNS. Therefore, to directly test the specificity of this allele, we quantified the proportion of $Col1a2$ -lineage⁺ cells that express $Col1a1^{GFP}$, finding that virtually all TdT⁺ cells are also GFP⁺ (i.e., are fibroblasts, **Extended Data Fig. 1j**, reproduced **below**). This supports the observation that lineage-traced cells are found only in border regions such as the meninges, choroid plexus, and larger perivascular Virchow-Robbins spaces (**Fig. 1a**), rather than throughout the brain parenchyma, as with other pericyte or pan-stromal cell reporters; **Extended Data Fig. 1n**.

Extended Fig. 1j: $Col1a2^{creER}$ specifically recombines in fibroblasts ($Col1a1^{GFP+}$)

- We have now quantified the baseline efficiency of $Col1a2^{creER}$ in resting uninjured brain tissue, using $Col1a1^{GFP+}$ cells as a denominator to represent the total and current “fibroblast pool”²⁰ (**Extended Data Fig. 1i**, reproduced **below**). We found that the $Col1a2^{creER}$ allele sensitivity (i.e., efficiency; TdT⁺ as a proportion of $Col1a1^{GFP+}$) was only 50% in *resting brain* fibroblasts. This resting efficiency was consistent with the proportion of lineage-traced lesional fibroblasts observed *after* injury (**Extended Data Fig. 1k,l**, reproduced **below**), suggesting that the observed 50% recombination rate after injury was *not* due to a separate source of lesion-infiltrating fibroblasts (e.g., pericytes that upregulate $Col1a2$). Any emergence of a new non-lineage-traced (TdT⁻) $Col1a1^{GFP+}$ fibroblast population would reduce the post-injury lineage-traced proportion below the baseline efficiency of 50%. Rather, our data suggests a technical effect of inefficient $Col1a2$ -driven CreERT activity, perhaps due to a relatively weaker $Col1a2$ expression at rest, but consistent with a predominantly fibroblast-derived response to injury and an equal contribution of recombined and non-recombined (e.g., TdT⁺ and TdT⁻) fibroblasts to the injured pool. The higher recombination efficiency with tamoxifen treatment observed *after* PT injury (**Extended Data Fig. 1l**) is consistent with the known upregulation of $Col1$ expression within fibroblasts after injury¹⁴.

Extended Fig. 1i,k,l: Preserved fibroblast recombination efficiency at rest and after injury

3. To orthogonally validate the specificity of $Col1a2^{creER}$, and to formally address the possibility of an alternate pericyte contribution to the fibrotic scar fibroblasts, we have now included new data from lineage tracing in pericyte reporter mice ($Ng2^{creER}; Rosa26^{TdT}$). Unlike *Cre* alleles used in prior studies^{12,17,18}, this line does not significantly recombine *bona fide* fibroblasts. $Ng2^{creER}$ has been used to quantify pericyte contributions to fibrotic scarring in other contexts^{11,13,15}, although it also recombines within oligodendrocyte lineage cells in the CNS. Using overlap with CD31 as a proxy for perivascular pericyte location, we found that ~84% of $Ng2$ -lineage⁺ cells (TdT^+ cells) were perivascular, as compared with ~23% of oligodendrocyte-lineage cells marked by $SOX10^+$, consistent with significant pericyte recombination by this $Ng2^{creER}$ line, as expected (**Fig. R3.1, below**). Additionally, we quantified the co-positivity of TdT with markers of oligodendrocyte-lineage cells ($SOX10$) or pericytes ($Desmin$; **Extended Data Fig. 1p**, reproduced below). While recombination was more efficient within the oligodendrocyte lineage, pericyte recombination was sufficient for microscopy analysis (**Extended Data Fig. 1q**, reproduced below). We observed minimal contribution of $Ng2$ -lineage-traced cells to the pool of lesional fibroblasts (**Extended Data Fig. 1o,s**, reproduced below). While $Ng2$ -lineage⁺ cells did expand in perilesional locations, they were confined to $GFAP^+$ regions of astroglial scarring, unlike $Col1a2$ -lineage⁺ fibroblasts that resided within the lesion. Moreover, in double pericyte/fibroblast reporter mice ($Ng2^{creER}; Rosa26^{TdT}; Col1a1^{GFP}$), we observed minimal overlap between $Ng2$ -lineage⁺ cells (including pericytes) and $Col1a1^{GFP+}$ fibroblasts by 21dpi (and as compared with $Col1a2^{creER}$ lineage tracing; **Fig. R3.2, below**). Similarly, we have included quantification of lineage tracing from smooth muscle reporter mice, finding that smooth muscle cells do not detectably contribute to lesional fibroblasts (**Extended Data Fig. 1r,s**, reproduced below).

Fig. R3.1: $Ng2^{creER}; Rosa26^{TdT+}$ cells exhibit a perivascular topography

Extended Data Fig. 1o-s: $Ng2$ -lineage⁺ cells (including pericytes) and $Acta2$ -lineage⁺ cells (smooth muscle) do not detectably infiltrate lesions

Fig. R3.2: Minimal overlap of Ng2-lineage⁺ cells and Col1a1^{GFP+} fibroblasts at 21dpi

- To directly investigate if fibroblasts, and not pericytes, predominantly contribute to CNS fibrosis in other models of brain injury, we have also included new data, generated with our collaborator Dr. Richard Daneman at UCSD, involving distal middle cerebral artery occlusion (dMCAO), a distinct model of stroke involving permanent electrocoagulation of the distal MCA. By 2 weeks after injury, lineage-traced fibroblasts (Col1a2^{creER}; Rosa26^{Tdt}) infiltrate and expand within the fibrotic lesion (marked by Col1 and surrounded by a GFAP⁺ glial scar, as in PT injury; **Extended Data Fig. 1t**, reproduced **below**). In contrast, lineage-traced pericytes (Atp13a5^{creER}; Rosa26^{Tdt}) maintain a perivascular distribution and do not visibly expand within the lesion (**Extended Data Fig. 1u**, reproduced **below**), suggesting that the biology of fibroblast-derived fibroblasts observed in PT injury extends across ischemic brain injuries and potentially other types of CNS injury and disease.

Extended Data Fig. 1t,u: Fibroblasts – but not pericytes – expand within lesions after distal MCAO

Collectively, these data indicate that a large majority of brain lesion-resident fibroblasts derive from pre-existing fibroblasts, though we cannot rule out that a small population of pericytes – not recombined by our pericyte reporter lines tested – may contribute to the pool of lesional fibroblasts.

The transcranial 4-OHT lineage tracing technique referenced above by the reviewer is aimed at determining the micro-anatomical origin of *Col1a2*-lineage⁺ fibroblasts, with equal recombination rates in dural meninges and lesion suggesting a largely dural origin (see **reviewer 2, response 5** for further details). We apologize for the lack of clarity on these issues and have updated our text accordingly.

And given that the knock-in reporter was *Col1a1*-GFP, why did the authors use a *Col1a2*creER rather than a *Col1a1*CreER?

The *Col1a2*^{creER} allele was already present in collaborating labs and had been intercrossed to relevant *Tgfb2*^{flox} alleles. Given the observed specificity of *Col1a2*^{creER} for *Col1a1*-expressing cells and strong co-expression of *Col1a1* and *Col1a2* in our snRNAseq dataset, we opted to continue with the *Col1a2*^{creER} allele for the sake of consistency.

Lines 91-93, Fig 1b: Col1a1 GFP knock-in and Col1a2CreER TdTom report appears in some cases to label different populations and no quantitation of overlap is presented here. Please show data regarding this and discuss why the two lines were chosen.

The mouse shown in **Fig. 1b** was treated with tamoxifen on days -16 through -14 prior to the injury, i.e., TdT⁺ fibroblasts are lineage-traced cells and derive from pre-existing fibroblasts that were expressing high enough levels of *Col1a2* to be recombined at the time of injection. In contrast, Col1a1^{GFP} marks cells actively expressing *Col1a1* and is, therefore, sensitive to post-injury upregulation. As discussed **above**, the lineage-tracing efficiency of the Col1a2^{creER} allele is ~50%, explaining why some cells in **Fig. 1b** appear to express GFP but not TdT (**Extended Data Fig. 1i**), whereas all cells expressing TdT are GFP⁺ (**Extended Data Fig. 1j**). We have clarified the above points in the **Fig. 1** legend. The Col1a1^{GFP} line was chosen for its high sensitivity and ability to dynamically highlight fibroblast expansion, while the Col1a2^{creER} line was chosen for its specificity and compatibility with temporally restricted induction.

-Line 105 Ext Fig 1e: the authors use a nuclei PDGFRalpha-GFP knock-in line to help quantitate fibroblast numbers – how did the reporting / marking of CNS fibroblast populations in this PDGFRa-GFP line compare with col1a1-GFP and the col1a2CreER lines also used – this is not clear. Again data regarding degree of overlap, and also whether PDGFRa-GFP marks a subpopulation of CNS fibroblasts should be presented and discussed.

The Pdgfra^{GFP} line was used for nuclear flow cytometry, given the difficulty of generating single-cell suspensions of intact fibroblasts and the lack of detectable nuclear GFP (Col1a1^{GFP}) or TdT (Col1a2^{creER}; Rosa26^{TdT}) signal within fibroblast nuclei of those lines. The Pdgfra^{GFP} line has been extensively used in peripheral organs as a pan-fibroblast marker^{6,27,28}. While it has not been extensively used in the CNS, perhaps secondary to oligodendrocyte-lineage expression of *Pdgfra*, this allele is a hyper-stable nuclear-localized reporter of a well-validated marker of CNS fibroblasts²⁰ and is expected to mark all fibroblasts in the CNS. Indeed, confocal imaging demonstrated labeling of both meningeal and perivascular fibroblasts at rest (**Fig. R3.3, below, top left**) and labeling of fibroblasts throughout the lesion after PT injury (**Fig. R3.3, below, top right**; oligodendrocyte-lineage cells labelled in brown), while our single-nuclear RNAseq data revealed *Pdgfra* expression across all fibroblast subsets (**Fig. R3.3, below, bottom left**). While we have not directly compared Pdgfra^{GFP} expression with either of the referenced fibroblast reporter lines in the same mouse, Pdgfra^{GFP+} cells make up an even *higher* proportion of lesional nuclei than Col1a2^{creER}; Sun1^{GFP+} cells (**Fig. R3.3, below, bottom right**). This difference likely represents the relative inefficiency of the inducible Col1a2^{creER} and/or Cre-inducible nuclear GFP (see **reviewer 2, response 7** for more details). Collectively, these data suggest that Pdgfra^{GFP} is a highly sensitive, if not specific, pan-CNS-fibroblast reporter.

Fig. R3.3: $\text{Pdgfra}^{\text{GFP}+}$ marks border-associated fibroblasts that expand after injury, comparable to $\text{Col1a2}^{\text{creER}}$

2. Figure 2, Ext Data Fig 3, line 158: ‘Early myofibroblasts spatiotemporally correlate with lesional profibrotic macrophages and damage-associated microglia’.

The data described here (first half of fig 2 and Ext fig 3) are in silico based, descriptive data. The authors should validate and annotate the proposed key fibroblast subpopulations in this section at the protein level with immunofluorescence staining using markers derived from the single-nuclei seq datasets, as this is a key dataset (with proposed fibroblast subpopulations) that underpins the rest of the manuscript. It would be important to show these protein-level data longitudinally in the PT injury model to delineate and confirm when and where these fibroblast subpopulations are manifest post-injury.

We thank the reviewer for this excellent suggestion, which is highly relevant given the importance of fibroblast state evolution to our story. We have now added extensive new data aimed at validating fibroblast subset heterogeneity, kinetics, and positioning using immunofluorescent microscopy. First, we visualized that αSMA^+ myofibroblasts expand by 7dpi and localize to the early lesion border and lesion core but disappear by 21dpi (**Extended Data Fig. 4c-e**, reproduced below), consistent with our transcriptomic data (**Fig. 2b,e**, **Fig. 3a**) and with timed lineage tracing data using the $\text{Cthrc1}^{\text{creER}}$ (**Fig. 2h-n**) or $\text{Acta2}^{\text{creER}}$ (**Extended Data Fig. 4g-l**) myofibroblast-tracking mice. We also validated the late lesional non-myofibroblast (αSMA^-) fibroblast subsets (identified in **Fig. 2b** and **Fig. 3a**) and their predicted spatial topography (based on integration of single nuclear and spatial transcriptomic data; **Fig. 3c**, **Extended Data Fig. 6a**). After identifying highly and differentially expressed candidate genes (**Extended Data Fig. 6b**), we positively identified late inner fibroblasts using FGF13; lymphocyte-interactive fibroblasts using CD80; altered dural fibroblasts using *Alpl*; and leptomeningeal fibroblasts, including arachnoid-like and pial-like fibroblasts, with the combination of *Aldh1a2* and *Laminin*¹³ (with *Aldh1a2* expressed in both subsets and *Lama1* preferentially expressed in pial-like fibroblasts) (**Fig. 3d**, reproduced below). Each subset, as defined by protein-level expression, localized as bioinformatically predicted and summarized in **Fig. 3e** (reproduced below). As expected, late fibroblast markers were sparse at early timepoints (**Extended Data Fig. 6c,d**), though *Aldh1a2* and *Laminin-1* were

persistent across time points, potentially reflecting an early population of pial-like fibroblasts observed in our transcriptomic data (Fig. 3a).

Extended Data Fig. 4c-e: A transient myofibroblast response predominates subacute time points

Fig. 3d,e: Topography of late fibroblast subsets by immunofluorescent imaging

To further explore this lesional fibroblast trajectory, we performed fluorescent imaging for the above subset markers in Cthrc1^{creER}; Rosa26^{Tdt} lineage tracer mice; in these mice, all fibroblasts that passed through a myofibroblast state express *Tdt*. We found that most fibroblasts from each marker-defined subset were Tdt⁺ (**Extended Data Fig. 6k, reproduced below**), confirming that myofibroblasts give rise to (at least) a majority of cells in all late fibroblast states. Overall, we believe this substantial new data supports insights derived from our bioinformatic analyses and lineage tracing experiments, providing a stronger foundation for our existing and new data (detailed below) concerning specific and divergent roles for temporally distinct fibroblast states.

Extended Data Fig. 6k: myofibroblasts give rise to late fibroblasts

Also when the authors draw cross-species comparisons (lines 177-186), it would be more compelling for the reader if they could show (at least in some of the other species) immunofluorescence staining confirming at the protein level corollary subpopulations in the other species / injury states.

Unfortunately, validated antibodies for relevant markers and additional tissue from marmoset were not readily available, and we plan to perform further cross-species analysis in the future. Nevertheless, focusing on other injury states (as referenced above), we have validated the presence of the myofibroblast state in tMCAO and TBI (as suggested by bioinformatic analysis, **Extended Data Fig. 3i-q**) via lineage tracing experiments in *Cthrc1*^{creER}; *Rosa26*^{TdT} mice, which highlight the presence of *Cthrc1*-lineage⁺ fibroblasts across models (**Extended Data Fig. 4o,p**, reproduced **below**). Additionally, we validated the presence of late fibroblast states observed after PT injury – including late inner fibroblasts, lymphocyte interactive fibroblasts, and dural/leptomeningeal-like fibroblasts – after tMCAO injury (**Extended Data Fig. 6l**, reproduced **below**). Notably, in this setting, fibroblast subset localization mirrors PT injury, with leptomeningeal fibroblasts immediately adjacent to the brain gliotic parenchyma, lymphocyte-interactive fibroblasts present in clusters beneath the leptomeninges, dural meningeal fibroblasts further from the brain parenchyma, and “late inner” fibroblasts in the lesion core.

Extended Data Fig. 4o,p: Fibroblasts pass through a *Cthrc1*-lineage⁺ state after *tMCAO* and *TBI* (CCI)

Extended Data Fig. 6l: Late fibroblast states are observed after *tMCAO*

3. The authors use ER-TR7 throughout to label fibroblasts, however in the literature this appears to be a marker of an ECM protein (collagen VI). Can the authors also use specific fibroblast-

marking antibodies throughout the manuscript to mark (and quantify) the fibroblast subpopulations of interest.

As referenced above, ER-TR7 is a marker of collagen VI that we and others use as a proxy for fibroblasts^{11,13}, given its specific colocalization with lineage-traced fibroblasts (**Fig. 1b**). ER-TR7 is also useful as a measure of fibroblast ECM deposition in addition to fibroblast expansion. Nevertheless, we appreciate the reviewer's suggestion; to further validate cellular effects across models, we have now performed follow-up experiments for key relevant fibroblast manipulations (e.g., $\alpha_v\beta_8$ -blockade, clodronate treatment, *Tgfb2* cKO) using fibroblast reporter mice (*Col1a1*^{GFP} or *Col1a2*^{creER}; *Rosa26*^{TdT}), revealing decreases in fibroblast expansion across lesional regions after each manipulation (**Extended Data Fig. 7f-h**, **Extended Data Fig. 8c-e,w-y**; see also **reviewer 2, response 18,19**, above for a more detailed discussion). As discussed above, we have also optimized specific fibroblast state markers and quantified their expression across time points via immunofluorescent imaging (see **response 2**, above; **Fig. 3d,e**, **Extended Data Fig. 6a-d,k,l**). To complement fibroblast reporter data, we stained for these markers in *Tgfb2* cKO mice at 7 and 21dpi, confirming decreases across late fibroblast states predicted in our sequencing data and reflected in our lineage tracing data (**Extended Data Fig. 9k-p**, and **response 9**, below).

4. Line 230: 'To investigate potential mechanisms underlying this profibrotic state, we performed ligand-receptor analysis, which predicted that SAM/DAM myeloid cells could signal to myofibroblasts via profibrotic signals...' The proposed ligand-receptor analyses should be functionally interrogated using co-cultures of SAM/DAM myeloid cells and myofibroblasts with modulation of the proposed key ligand-receptor interactions.

We agree with the reviewer that our *ex vivo* coculture system could be well suited for the interrogation of interactions between fibroblasts and SAM/DAM. To begin to investigate the ability of myofibroblast-containing lesions to modulate myeloid cells, we cocultured 7 or 21dpi lesions with freshly sorted brain microglia. Coculture of microglia with 7dpi lesions increased expression of several DAM markers including CD11c, CD9, CD63, and TREM2, leading to a higher proportion of DAMs than microglia in monoculture (though monoculture also caused upregulation of DAM markers relative to the sorted population) (**Extended Data Fig. 5y**, reproduced **below**). Coculture with 21dpi lesions, containing fewer myofibroblasts, caused less of an increase in DAMs. Collectively, these data – in keeping with our findings *in vivo* (**Fig. 4v,w**) – suggest that myofibroblast-rich lesions can modulate myeloid identities and promote a “damage/disease-associated/pro-fibrotic” myeloid cell phenotype.

Extended Data Fig. 5y: 7dpi lesions upregulate DAM markers in microglia ex vivo

Given the difficulty of isolating brain fibroblasts (i.e., separating fibroblasts from their endogenous profibrotic-myeloid-cell-rich environment), it was more difficult to interrogate the effects of macrophages on lesional fibroblasts *ex vivo*. Using bulk lesional RNA expression, we were unable to detect differences in profibrotic genes within lesions cultured with or without profibrotic macrophages, precluding the interrogation of specific pathways (data not shown). A more thorough investigation of fibroblast-macrophage interactions, both *in vitro* and *in vivo*, is of significant interest but we feel outside the scope of this paper. Further, we emphasize that our *in vivo* data suggests that macrophage-expressed *Tgfb1* is at least one such signal that drives the brain fibrotic injury response (**Extended Data Fig. 7n,o**).

Immunofluorescence staining-based assessment of SAM/DAM topography relative to the key myofibroblast subpopulation(s) should also be performed.

We have added imaging of SAM and DAM topography, using a shared SAM/DAM marker (FABP5), which reveals SAM/DAM near *Cthrc1*-lineage⁺ myofibroblasts at 7dpi (**Extended Data Fig. 5s**, reproduced **below**). We observed cells of both monocytic and microglial lineage expressing this marker in myofibroblast-enriched locations (**Extended Data Fig. 5t,u**, reproduced **below**), supporting our hypothesis of myeloid profibrotic state convergence across myeloid cell monocyte-derived macrophage and microglial ontogenies.

Extended Data Fig. 5s-u: Immunofluorescent imaging of SAM/DAM populations

5. The authors use Visium-based spatial transcriptomics, however they should also mention the limited resolution of this method and how each spot (which is 55um in size) will comprise multiple cell types. For example, in Ext Fig 2i the authors show the enriched fibroblast spots are also enriched for proliferative genes, however in Ext Fig 3h one can see many other cell types have an increased proliferative signature.

We thank the reviewer for this suggestion and have modified and clarified our language throughout the manuscript and figure legends to reflect the limited resolution of our spatial transcriptomic data.

Furthermore, do the authors have supplementary tables listing all the signatures used to identify lineages and subpopulations and cell identity scores?

We used well-validated markers to identify common lineages, which have now been added to our Methods section (e.g., *Col1a1*, *Col1a2*, *Pdgfra* [fibroblasts]; *Cspg4* [mural]; *Itgam* [myeloid]; *P2ry12*, *Sall1* [microglia]; *Cd3e*, *Cd4*, *Cd8* [T cells]; *Rbfox3* [neurons]; *Dcx*, *Prom1* [neural progenitors]; *Gfap*, *Aldh1l1* [astrocytes]; *Mbp*, *Olig2*, *Sox10* [oligodendrocytes]; *Olig2*, *Sox10*, *Pdgfra* [oligodendrocyte precursor cells]; and *Pecam1* [endothelial cells]. All identities were checked against results from common identity assignment algorithms (e.g., SCType²⁹).

6. Lines 230-238, ligand-receptor analysis. The authors show in Ext Fig 6m and Fig 2q the spatial location of the cells of interest using Visium data, however as discussed above this is not at single cell level (spots are 55um in size and 100um apart). Can the authors validate their findings using immunofluorescence staining to confirm whether the subpopulations of interest co-locate in the regions mentioned? This would add further strength to Fig 2r.

We thank the reviewer for raising this concern and have updated the text accordingly. As referenced above (see **response 4**), we have added additional immunofluorescent data validating the presence of SAM and DAM populations, using the marker FABP5 in combination with respective lineage tracer alleles (**Extended Data Fig. 5s-u**). Both populations colocalize to regions of emerging fibrosis, in accordance with the global distribution of monocyte- and microglia-derived cells (**Fig. 2o**).

7. Line 282: 'late myofibroblasts expressed markers associated with smooth muscle differentiation...' – the late myofibroblasts should be further validated in tissue with multiplex immunofluorescence staining using markers derived from the *in silico* data.

Similarly, regarding the statement 'The final three late fibroblast (21dpi) clusters were identified as pial, arachnoid, and altered dural fibroblast states, based on resting leptomeningeal fibroblast distribution and meningeal layer signatures' – these 3 fibroblast subpopulations should also be validated in tissue with multiplex immunofluorescence staining (in addition to the integration with spatial sequencing already presented).

As discussed above (see **response 2**), we have added new immunofluorescent imaging-based validation of all late fibroblast states, confirming our *in silico* predictions. Specifically, imaging of altered dural and leptomeningeal fibroblasts (Alpl and Laminin/Aldh1a2; **Fig. 3d,e, Extended Data Fig. 6a-d,k**) highlights the presence of layered dural-like and leptomeningeal fibroblast populations along the lesion border. Meanwhile, late inner fibroblasts (previously named "late myofibroblasts", FGF13⁺; **Fig. 3d, Extended Data Fig. 6c,d,k**) are enriched within the late lesional core. We have also added data comprising immunofluorescent staining for the markers referenced above (and pertaining to smooth muscle differentiation, **Extended Data Fig. 6e**). We observe expression of CDH18 among fibroblasts enriched at 21dpi and localized to the inner lesional core (**Extended Data Fig. 6f,g, reproduced below**), consistent with higher *Cdh18* expression by late inner fibroblasts. Additionally, we observed high immunopositivity for SEMA3C, a secreted molecule highly and specifically expressed by late inner fibroblasts, within the core of the lesion but *not* the fibroblast-rich border or brain parenchyma (**Extended Data Fig. 6h,i, reproduced below**). Collectively, these data support our bioinformatic characterization of late fibroblast states.

Extended Data Fig. 6f-i: Immunofluorescent analysis of marker expression within “Late Inner” fibroblasts

8. Figure 4f – in addition to showing the TdTom micrographs, the number of brain lesional fibroblasts should be quantitated using specific immunofluorescence markers and counting.

9. Line 371: ‘cKO mice had a profound loss of early myofibroblast clusters at 7dpi and a substantial reduction in late states at 21dpi, including late myofibroblasts, meningeal-like, and lymphocyte-interactive fibroblasts, supporting our model of sequential state transitions (Fig. 4o, Extended Data Fig. 10k)’. The authors should validate (and quantitate) these findings at the protein level with multiplex immunofluorescence staining and counting of the various myofibroblast subpopulations.

We thank the reviewer for making these important requests to validate a loss of fibroblasts (and their specific states) in *Tgfb2* cKO mice, and we have addressed them as follows:

1. We have quantified total fibroblasts (Col1a2^{creER}; Rosa26^{TdT}), as shown in **Fig. 4f** in control and cKO mice, revealing a decrease in fibroblast density, especially within the inner lesional core, in cKO mice (**Extended Data Fig. 8c-g**, reproduced below). Similar analysis was performed after clodronate treatment and $\alpha_v\beta_8$ -blockade (**Extended Data Fig. 7f-h**, **Extended Data Fig. 8w-y**; see also **Reviewer 2, response 18,19**, above).

Extended Data Fig. 8c-g: Reduction in inner core fibroblasts and thinner fibroblast border in *Tgfr2* cKO mice

2. Combining this fibroblast reporter analysis with our fibroblast state-specific antibodies (see **response 2**, above), we have quantified fibroblast subpopulations after injury across control and *Tgfr2* cKO (**Extended Data Fig. 9k-p**, reproduced below). We observed a loss of most fibroblast states in cKO mice, in accordance with our transcriptional data (**Fig. 4t**). Specifically, we found a decrease in myofibroblasts at early timepoints, along with a decrease in lymphocyte interactive and dural-like fibroblasts – and a trending decrease in late inner fibroblasts – at late timepoints. As discussed in our manuscript, the relative preservation of leptomeningeal (*Aldh1a2*⁺) fibroblasts, also observed *in silico* (**Fig. 4t**), may suggest a strong selective pressure for restoration of the lesional border after injury.

Extended Data Fig. 9k-p: Reductions across fibroblast states in *Tgfr2* cKO mice

10. Line 132: the authors state the 4 fibroblast clusters had distinct temporal, molecular and microanatomical patterns – however the authors only show *Col1a1* gene expression in the Visium images (which marks all 4 clusters) – please add data demonstrating the distinct microanatomical location of these fibroblast populations.

We appreciate the chance to clarify this issue. While the panels referenced after the cited sentence demonstrate the spatial, temporal, and molecular characteristics of all 4 clusters in aggregate (including *Col1a1* expression), subsequent sentences individually highlight and reference **Fig. 11** and **Extended Data Fig. 2e-f**, which collectively describe the temporal and microanatomical heterogeneity between each of these individual clusters.

11. Substantial amounts of data based on transgenic mouse approaches are presented, however to further enhance the manuscript can the authors show evidence of corollary myofibroblast subpopulation / immune cell subset interactions in diseases where wound healing occurs in human brain injury.

To investigate the potential for parallel myofibroblast-SAM/DAM interactions across species, we mapped our profibrotic SAM score onto snRNAseq data from marmoset stroke and human TBI (**Extended Data Fig. 5v,w**, reproduced **below**). In each case, we identified the expression of SAM markers by subsets of myeloid cells. Additionally, TGFB1 was highly and uniquely expressed by myeloid cells in both models (**Extended Data Fig. 7k,l**, reproduced **below**). These new bioinformatic analyses – in conjunction with existing evidence highlighting the presence of myofibroblasts in these models (**Extended Data Fig. 3i-q**) – suggest conserved myofibroblast-macrophage/microglial interactions across mammalian species and CNS pathologies.

Extended Data Fig. 5v,w: SAM scores within marmosets and human patients after brain injury

Extended Data Fig. 7k,l: TGFB1 expression within marmosets and human patients after brain injury

12. Line 250 Fig 3d: In addition to the flow cytometry data presented can the authors also show immunofluorescence staining to delineate the spatial location of the T cell population relative to the lymphocyte-interactive fibroblasts (not simply show ER-TR7 staining).

We appreciate this suggestion, and – based on fibroblast state marker identification and validation referenced above (see **response 2**; **Fig. 3d,e**, **Extended Data Fig. 6b-d**) – we have now added data using the lymphocyte interactive fibroblast marker CD80. Specifically, we have included immunofluorescence imaging and quantification revealing a strong spatial association between lesional T cells and border-associated CD80⁺ lymphocyte-interactive fibroblasts that increases across late time points after PT injury (**Fig. 3p,q**, reproduced **below**; **Extended Data Fig. 6s**).

Fig. 3p,q: T-cells persist in close proximity to lymphocyte interactive (CD80⁺) fibroblasts

13. Line 150: please provide details of the genes used for the TGFb score and the data and experimental information used to generate this score. What were the primary cells used here, where/how were they obtained?

As described in Sbierski-Kind et al.³⁰, primary lung fibroblasts were sorted from healthy mice, were cultured for 6 days, and were subsequently treated with TGFβ (1ng/mL) or PBS for 48 hours prior to RNA isolation and bulk RNA sequencing (with further details on sorting strategies and culture conditions in the Methods section of the cited reference). Subsequently, genes upregulated by TGFβ (relative to PBS; log2FC > 0, q < 0.05) were used to construct a fibroblast “TGFβ module” score. This information is also elaborated in our Methods section.

14. Figure 4u,v: functional interrogation experiment using conditional KO, Col1a2creER; Tgfr2flox mice in the context of the tMCAO model.

It is perhaps somewhat unsurprising that KO of Tgfr2 on Collagen1a2-expressing myofibroblasts results in impaired fibrotic scar formation in the CNS. To fully leverage the authors’ interesting findings (using single-nuclei RNA seq to generate precise annotation of fibroblast subpopulations during CNS injury and wound healing), and to also add conceptual advance, can the authors use a Cre-driver approach to either delete a gene within, or deplete (eg using a Cre-DTR system) a specific fibroblast subpopulation which alters wound healing to demonstrate that their new datasets facilitate precision medicine approaches (via specific targeting of the critical fibroblast subpopulation(s)) in the context of brain injury and wound healing?

We thank the reviewer for this critical feedback that further work was required to fully leverage our transcriptional annotation and to demonstrate its functional relevance. We have addressed these points by adding significant additional data aimed at showing (1) a specific, orthogonally validated functional role for the early myofibroblast state in regulating wound healing, and (2) a distinct and novel role for late fibroblast subsets – and their *Cxcl12* expression – that regulates brain T cell localization and cytokine expression after PT injury, with long-lasting effects on cortical immune tone. Together, these data, described in detail below, highlight fibroblast dynamic functional specialization that mirrors phenotypic state evolution, suggesting that future therapeutic interventions can target specific temporal windows – and topographic states – of CNS injury-associated fibrosis to achieve distinct goals.

With the goal of testing fibroblast state-specific roles in wound healing, we first focused on the myofibroblast state. While our previous data utilized the pan-fibroblast Col1a2^{creER} line, we have

now added data involving ablation experiments in $Cthrc1^{creER}; Rosa26^{DTA}$ mice. The $Cthrc1^{creER}$ strain was recently published by our collaborator, Dr. Dean Sheppard, as a sensitive and specific tool to mark myofibroblasts that drive lung fibrosis³. In $Cthrc1^{creER}; Rosa26^{DTA}$ mice, administration of tamoxifen causes specific – albeit partial – DTA-induced deletion of $Cthrc1^+$ brain myofibroblasts, present only during the first 1-2 weeks after brain injury (as demonstrated by lineage tracing, **Fig. 2h-n**). Using these mice, we found a reduction in brain lesional ECM (ER-TR7 coverage) by 21dpi, and even this modest deletion was sufficient to drive an increase in PT lesion size (**Fig. 4k,l**, reproduced **below**). Together, this novel data supports the temporal role of the myofibroblast state in limiting brain injury after stroke. Further, it is consistent with our previously described fibroblast cKO mice, in which all fibroblasts can be induced to lose sensitivity to TGF β signaling, effectively blocking myofibroblast state formation.

Fig. 4k,l: $Cthrc1^{creER}; Rosa26^{DTA}$ myofibroblast delete mice show increased lesion size

To orthogonally validate the timing of myofibroblast contributions to brain injury and healing, we also employed temporally-restricted recombination in the *Tgfb2* cKO mice, delaying tamoxifen induction until 14dpi – after most myofibroblasts have transitioned into late lesional fibroblasts, as suggested by sequencing and validated by our *Cthrc1* lineage tracing. In this regimen, early lesional fibroblasts remain TGF β -responsive, but late lesional fibroblasts do not. Therefore, we were able to isolate the role of the TGF β -response to early myofibroblasts while controlling for the effects of TGF β on other fibroblast subsets both in the periphery and within the CNS. In these mice, we observed no change in lesion size 2 weeks after late tamoxifen administration (**Fig. 4m**, reproduced **below**). Additionally, ablation of fibroblast *Tgfb2* after closure of the “myofibroblast window” did not lead to a secondary neutrophilic or monocytic neuroinflammation (**Fig. 4n** and **Fig. R3.4**, **below**). We continued to observe a mild decrease in ECM coverage, suggesting that TGF β could play modest roles in ECM secretion across late injury subsets and time points (**Extended Data Fig. 8u**). Together, these experiments indicate that $Col1a2^{creER}$ – while not subset-specific – only generates relevant phenotypes within a transient temporal window wherein brain myofibroblasts predominate. In combination with the *Cthrc1* myofibroblast ablation experiments described above, these data are consistent with a myofibroblast-specific role in limiting excessive injury and secondary neuroinflammation, adding significant nuance to emerging literature which has so far suggested, often using pan-fibroblast tools, that fibroblast *attenuation* may be an appropriate therapeutic strategy^{18,19}.

Fig. 4m,n: Late tamoxifen induction does not recapitulate increased lesion size or neuroinflammation

Fig. R3.4: Time course of myeloid neuroinflammatory infiltrates after late tamoxifen recombination in cKO mice

To further add to the conceptual advance of our work, we also provide data to define at least one functional role of late fibroblast subsets. As lymphocytes become increasingly dominant in the late fibroblastic lesions of the brain, we focused our efforts on this axis, including the chemokine *Cxcl12* (Fig. 3g). CXCL12 is a well-described chemokine that acts via CXCR4 to regulate an array of immature and mature hematopoietic cells, including T cells³¹, with previously described roles in coordinating dural meningeal T cell positioning³¹. We found that *Cxcl12* was highly expressed by late lesional lymphocyte interactive fibroblasts; while not unique to this fibroblast subset, it was minimally expressed in early lesional myofibroblasts (Extended Data Fig. 10g,h). As such, we tested the role of fibroblast-derived CXCL12 in our PT injury model, generating *Col1a2^{creER}; Cxcl12^{fllox}* mice where post-injury tamoxifen induction conditionally deletes *Cxcl12* in fibroblasts in adult mice.

At chronic time points after injury (21dpi), we observed that mice in which all fibroblasts were induced to lack CXCL12 (*Col1a2^{creER}; Cxcl12^{fllox}*) had reduced perilesional T cells, as assessed via confocal microscopy (Fig. 5f-i, reproduced below; Extended Data Fig. 10i,j), along with a decrease in macrophage MHC-II levels and perilesional MHC-II⁺ cells (Extended Data Fig. 10m,n), and consistent with a reduction in the type 1, IFN γ -competent lesional T cell population¹⁶. Of interest, total cortical T cells were *not* decreased, as assessed by flow cytometry (Extended Data Fig. 10o,p), indicating a specific loss of perilesional T cell localization in the absence of fibroblast-derived CXCL12.

Fig. 5f-i: *Col1a2^{creER} Cxcl12^{fllox}* mice show reduced perilesional localization of T-cells

To assess the consequence of interrupting this fibroblast-T cell chemokine network, we assessed lymphocytic cytokine expression via *ex vivo* restimulation. T cells from *Cxcl12* cKO mice expressed higher levels of effector cytokines including IFN γ (CD4⁺ and CD8⁺) and IL-17A (CD4⁺) (Fig. 5k-m, Extended Data Fig. 10q-s, reproduced below), implying the loss of a regulatory mechanism that – in wildtype mice – constrains brain T cell cytokine expression and dispersal after injury. We also observed a lesioned-hemisphere-specific increase in neutrophils, consistent with an altered cortical immune tone (Fig. 5o, reproduced below). We validated that loss of CNS fibroblast *Cxcl12* does not impact brain lesion size (Fig. 5j), overall lesional fibrosis (Extended Data Fig. 10k), lymphocyte-interactive fibroblast expansion (Extended Data Fig. 10l), or lymphocyte expression of suppressive/inhibitory cytokines (IL-10, Fig. 5n). Additionally, inducible loss of *Cxcl12* within fibroblasts of adult mice does not cause detectable immune changes in blood or meninges (Extended Data Fig. 10t-y). Together, this new and extensive data indicates that late lesional fibroblasts, predominantly including lymphocyte interactive fibroblasts, express CXCL12 – and other chemokines and cytokines – required for perilesional T cell localization and/or persistence within *de novo* lymphocytic niches. In the absence of the CXCL12 chemokine signal, T cells were mislocalized and their effector cytokine expression was further unleashed (Fig. 5p, reproduced below).

Fig. 5k-p: T cell mislocalization in *Col1a2^{creER} Cxcl12^{fllox}* mice correlates with increased cytokine expression and chronically dysregulated immune tone

Extended Data Fig. 10q-s: *Col1a2^{creER} Cxcl12^{fllox}* mice have increased brain cytokine-expressing T cells after PT injury

Collectively, the above experiments draw on insights from our snRNAseq data to highlight a transition from brain lesional myofibroblasts, directly involved in physical wound-healing and constraint of neuroinflammation, to late lesional fibroblasts, including lymphocyte interactive fibroblasts that express *Cxcl12*, with distinct roles in regulating cortical lymphocytes and associated brain immune tone. Given the long-term sequelae of stroke – including risk of psychiatric disease, epilepsy, sleep disturbances, and neurodegeneration – as well as the emerging promise of immune therapies after CNS injury^{32,33}, an understanding of dynamic fibroblast-immune interactions and their spatiotemporal dynamics offers direct and meaningful applications to precision medicine approaches for brain and spinal cord injuries, and likely for a broader swath of disease states.

15. Line114: the authors use dual labelling with *Col1a1*-GFP and *Col1a2*-TdT mice, however then call them 'Col1-expressing (TdT+)' – please be specific and consistent in the text – eg call these 'Col1a2-expressing (TdT+)'

We appreciate this feedback and have updated the text accordingly.

16. Ext Fig 1h has confusing axis labeling, which does not match the schematic. In the figure legend it states 7-9dpi but the schematic and IHC shows 21dpi – please clarify.

We apologize for the confusion and have updated the legend (see **Extended Data Fig. 11**, now with additional uninjured data) to clarify the following: animals labeled “lineage traced” (middle column) were treated with tamoxifen at days -16 through -14 (~two weeks prior to injury). Animals labeled “post-injury induction” (right column) were treated with tamoxifen on days 7-9. In both cases, animals were harvested and imaged at 21 dpi.

17. Line 115: the authors use Gli1-TdT along with some I.F. stains to show that lesional fibroblasts express fibroblast markers. Please justify the use of the Gli1 strain in the text. Please also state 'n' of mice used at each timepoint. There are no scale bars presented. Please also show an overlay image if these are all the same mice for each IF micrograph.

We thank the reviewer for raising this issue. The text has been updated to reflect that the *Gli1^{creER}* allele is used as an orthogonal and well-validated fibroblast reporter in peripheral organs⁶, with a particular tendency to label adventitial “universal” fibroblasts with elevated mesenchymal progenitor capacity^{6,34}. Here, we use this tool to orthogonally demonstrate that lesion-infiltrating fibroblasts arise from pre-existing *Gli1⁺* fibroblasts, which are broadly present in the resting dural meninges. Quantification of *Gli1*-lineage⁺ cells (**Extended Data Fig. 1s**), as well as scale bars

and relevant “n” values (**Extended Data Fig. 1n,s**), have also been added. As the different images represent distinct adjacent slices, there is no overlay image to show, and this has been clarified in the legend.

18. Ext. Fig 1k, please show quantification of this and higher mag images.

We have added high-magnification images and quantification of both *Acta2*-lineage⁺ and *Ng2*-lineage⁺ cells (representing smooth muscle cell-derived and pericyte-derived cells, respectively; **Extended Data Fig. 1o-s**).

19. Fig 1h: it is unclear in the methods and figure legend if the upper I.F. images were derived from the same sections as the lower spatial, or were these consecutive sections? Please clarify this.

We have updated the legend and methods to clarify that, as per the 10X Visium workflow, IF images were derived from the same sections used for spatial sequencing data, with imaging performed prior to the 10X workflow of de-coverslipping, tissue permeabilization, RNA capture, and library construction.

20. Line 154: can the authors please validate this increased pro-fibrotic fibroblast response using I.F. eg using Ki67 and a marker of the pro-fibrotic fibroblasts.

We have updated the text to emphasize our existing data (**Extended Data Fig. 4m,n**) showing that EdU incorporation within fibroblasts, defined by a cytoplasmic TdTomato and a nuclear GFP, both driven by *Col1a2*^{creER}, was elevated within fibroblasts at 7dpi – when myofibroblasts predominate – and diminished by later time points, when myofibroblasts were mostly absent. We have corrected an erroneously omitted figure caption for **Extended Data Fig. 4m**. Unfortunately, lesional autofluorescence renders Ki67 staining difficult to interpret (data not shown).

21. Line 177: adult marmoset data. The authors reference a previous paper in the Ext Fig 4a legend, however also cite methods in the manuscript regarding generation of marmoset snRNAseq data but this doesn't appear to be referenced in the manuscript text. Please clarify this in the methods and main text.

The methods and text have been clarified to reflect that previously published data from marmoset stroke (7dpi, **Extended Data Fig. 3i**) was integrated with new resting marmoset data (referenced above, **Extended Data Fig. 3j**) to generate a comprehensive picture of marmoset CNS fibroblasts at rest and after injury. Methods regarding the generation of snRNAseq refer to the new resting data, though they are identical to previously published methods used to generate injured data.

22. Line 180: there is no reference provided for human paper cited. Also, the fibroblast cluster is very small – please state how many cells were in this. The number of spots in Ext Fig 4f doesn't seem to align with the violin plot presented in Ext Fig 4h – ie the violin plot looks like it contains far more cells – please clarify this by also presenting the number of cells (in addition to the gene expression) in the violin plots. Also please clarify what the control tissue was in these datasets.

The reference, previously located in the figure legend, has been to the main text. Both the UMAP (now **Extended Data Fig. 3o**) and the violin plots (now **Extended Data Fig. 3q**) contain 77 fibroblast nuclei, and this value has been added to the figure legend. It is unclear whether these low cell numbers reflect TBI biology, tissue sampling/processing techniques unsuited to fibroblast capture, or the acute time points of collected samples (4 hours to 8 days post-injury). The legend

has been clarified to reflect that control tissue comprised region-matched post-mortem samples from patients who died of non-neurological causes³⁵.

23. Line 182: As GBM is a cancer, the authors should state that the fibroblasts in these datasets are cancer-associated fibroblasts (unless I have misunderstood their reasoning here). The links presented here are somewhat tenuous - only some genes are shared between the datasets - it would potentially be more powerful to show a signature of myofibroblasts.

We appreciate this suggestion and have updated the legend text accordingly to reflect the expression of myofibroblast-associated genes among cancer-associated fibroblasts. As a “myofibroblast score” or signature, we used the “TGF β score” presented in **Fig. 1n** and explained above (see **response 13**), comprising all genes upregulated by lung adventitial fibroblasts (sorted as PDGFR α^+ Sca1 $^+$) after treatment *in vitro* with TGF β . Indeed, all 3 models show an increase in this score in relevant compartments, including 7dpi fibroblasts in marmoset stroke (relative to resting fibroblasts, **Extended Data Fig. 3n**); TBI-associated fibroblasts in human patients (relative to resting fibroblasts, **Extended Data Fig. 3q**); and cluster 0, along with smaller clusters 1 and 3, in GBM (**Extended Data Fig. 3w**).

24. Line 206: the authors discuss subsets of microglia and macrophages and utilise multiple reporter strains – can the authors please clarify further which strains are being used to label which subpopulations of macrophages and microglia.

We have further clarified these points in the manuscript text. Microglia-derived cells are labeled using P2ry12^{creER}; Rosa26^{TdT} (**Extended Data Fig. 5d**). Monocyte-derived cells are labeled using Ccr2^{creER}; Rosa26^{TdT} (**Extended Data Fig. 5g**). Border-associated macrophages are labeled using PF4^{cre}; Rosa26^{TdT} (**Extended Data Fig. 5l**). Iba1 is used to highlight all of these myeloid cell populations (**Fig. 2o**). For clarity, we have removed one image of Ccr2^{RFP}, a related monocyte and early monocyte-derived macrophage marker.

25. Ext Fig 6c-f: The authors describe the spatial changes in microglia, infiltrating monocytes and BAM/PVM, summarising this in panel ‘f’ – it is a nice spatial interpretation of these changes, however there is no mention of ‘n’ used to generate these results, nor any quantification to back this up. Please add analyses of these images.

We thank the reviewer for raising this concern and agree that quantification of myeloid cell topography would strengthen our conclusions. We have updated the manuscript to reflect the number of mice analyzed per reporter line, and we have performed quantitative spatial analysis to confirm our interpretation of the images referenced above. Microglia preferentially localize to the “outer” and “border” regions of the PT lesion (with the highest density within the border but the highest proportion outside the lesion; **Extended Data Fig. 5d-f**, reproduced **below**). In contrast, monocyte-derived cells preferentially localize to the “inner” and “border” regions (**Extended Data Fig. 5g-i**, reproduced **below**). BAMs show less of a microanatomical preference, with higher density in the inner and border regions but a higher proportion found in the border/outer regions (**Extended Data Fig. 5l-n**).

Extended Data Fig. 5e,f,h,i: Analysis of myeloid cell topography by lineage

26. Fig 3m,Ext Fig 7m+q: how are the authors controlling for the lesion area at 21dpi already having more (lesional) T cells at time of isolation and therefore culture? Similarly for 7dpi, which would have fewer (lesional) T cells also. Please clarify this.

T cells added to coculture experiments are pre-labeled with CTV fluorescence prior to plating, enabling the separation of CTV-negative cells already within the lesion (which may vary with lesional timepoint) from CTV-positive T cells added (which are standardized at 100,000 per well). All reported T cell metrics are generated from CTV⁺ T cell gates. This approach also allowed for the determination of (lack of) T cell proliferation (CTV dilution) in this experimental system.

Citations:

1. Glushakova, O. Y. *et al.* Role of Caspase-3-Mediated Apoptosis in Chronic Caspase-3-Cleaved Tau Accumulation and Blood-Brain Barrier Damage in the Corpus Callosum after Traumatic Brain Injury in Rats. *J Neurotrauma* **35**, 157–173 (2018).
2. Buechler, M. B. *et al.* Cross-tissue organization of the fibroblast lineage. *Nature* (2021) doi:10.1038/s41586-021-03549-5.
3. Tsukui, T., Wolters, P. J. & Sheppard, D. Alveolar fibroblast lineage orchestrates lung inflammation and fibrosis. *Nature* 1–8 (2024) doi:10.1038/s41586-024-07660-1.
4. Dewar, D., Underhill, S. M. & Goldberg, M. P. Oligodendrocytes and Ischemic Brain Injury. *J Cereb Blood Flow Metab* **23**, 263–274 (2003).
5. Zhuo, L. *et al.* hGFAP-cre transgenic mice for manipulation of glial and neuronal function in vivo. *Genesis* **31**, 85–94 (2001).
6. Dahlgren, M. W. *et al.* Adventitial Stromal Cells Define Group 2 Innate Lymphoid Cell Tissue Niches. *Immunity* **50**, 707–722.e6 (2019).
7. Makihara, N. *et al.* Involvement of platelet-derived growth factor receptor β in fibrosis through extracellular matrix protein production after ischemic stroke. *Exp Neurol* **264**, 127–34 (2015).
8. Kelly, K. K. *et al.* Col1a1+ perivascular cells in the brain are a source of retinoic acid following stroke. *BMC Neuroscience* **17**, 49 (2016).
9. Fernández-Klett, F. *et al.* Early loss of pericytes and perivascular stromal cell-induced scar formation after stroke. *J Cereb Blood Flow Metab* **33**, 428–439 (2013).
10. Dias, D. O. *et al.* Pericyte-derived fibrotic scarring is conserved across diverse central nervous system lesions. *Nature Communications* **12**, 5501 (2021).
11. Dorrier, C. E. *et al.* CNS fibroblasts form a fibrotic scar in response to immune cell infiltration. *Nature Neuroscience* **24**, 234–244 (2021).
12. Holl, D. *et al.* Distinct origin and region-dependent contribution of stromal fibroblasts to fibrosis following traumatic injury in mice. *Nat Neurosci* 1–14 (2024) doi:10.1038/s41593-024-01678-4.
13. Xue, X. *et al.* Heterogeneous fibroblasts contribute to fibrotic scar formation after spinal cord injury in mice and monkeys. *Nat Commun* **15**, 6321 (2024).
14. Xu, L. *et al.* Fibroblasts repair blood-brain barrier damage and hemorrhagic brain injury via TIMP2. *Cell Reports* **41**, (2022).
15. Soderblom, C. *et al.* Perivascular fibroblasts form the fibrotic scar after contusive spinal cord injury. *J Neurosci* **33**, 13882–13887 (2013).
16. Garber, C. *et al.* T cells promote microglia-mediated synaptic elimination and cognitive dysfunction during recovery from neuropathogenic flaviviruses. *Nat Neurosci* **22**, 1276–1288 (2019).
17. Göritz, C. *et al.* A Pericyte Origin of Spinal Cord Scar Tissue. *Science* **333**, 238 (2011).
18. Dias, D. O. *et al.* Reducing Pericyte-Derived Scarring Promotes Recovery after Spinal Cord Injury. *Cell* **173**, 153–165.e22 (2018).
19. Dorrier, C. E., Jones, H. E., Pintarić, L., Siegenthaler, J. A. & Daneman, R. Emerging roles for CNS fibroblasts in health, injury and disease. *Nature Reviews Neuroscience* (2021) doi:10.1038/s41583-021-00525-w.
20. Bonney, S. K., Sullivan, L. T., Cherry, T. J., Daneman, R. & Shih, A. Y. Distinct features of brain perivascular fibroblasts and mural cells revealed by in vivo two-photon imaging. *Journal of Cerebral Blood Flow & Metabolism* **42**, 966 (2021).
21. Vanlandewijck, M. *et al.* A molecular atlas of cell types and zonation in the brain vasculature. *Nature* **554**, 475–480 (2018).
22. DeSisto, J. *et al.* Single-Cell Transcriptomic Analyses of the Developing Meninges Reveal Meningeal Fibroblast Diversity and Function. *Developmental Cell* **54**, 43–59.e4 (2020).
23. Betsholtz, C. *et al.* Advances and controversies in meningeal biology. *Nat Neurosci* 1–17 (2024) doi:10.1038/s41593-024-01701-8.

24. Schiavinato, A., Przyklenk, M., Kobbe, B., Paulsson, M. & Wagener, R. Collagen type VI is the antigen recognized by the ER-TR7 antibody. *Eur J Immunol* **51**, 2345–2347 (2021).
25. Arnold, T. D. *et al.* Impaired α V β 8 and TGF β signaling lead to microglial dysmaturation and neuromotor dysfunction. *Journal of Experimental Medicine* **216**, 900–915 (2019).
26. McKinsey, G. L. *et al.* Radial glia control microglial differentiation via integrin avb8-dependent trans-activation of TGFB1. 2023.07.13.548459 Preprint at <https://doi.org/10.1101/2023.07.13.548459> (2023).
27. Cautivo, K. M. *et al.* Interferon gamma constrains type 2 lymphocyte niche boundaries during mixed inflammation. *Immunity* **55**, 254-271.e7 (2022).
28. Hu, K. H. *et al.* Transcriptional space-time mapping identifies concerted immune and stromal cell patterns and gene programs in wound healing and cancer. *Cell Stem Cell* **30**, 885-903.e10 (2023).
29. Ianevski, A., Giri, A. K. & Aittokallio, T. Fully-automated and ultra-fast cell-type identification using specific marker combinations from single-cell transcriptomic data. *Nat Commun* **13**, 1246 (2022).
30. Sbierski-Kind, J. *et al.* Group 2 innate lymphoid cells constrain type 3/17 lymphocytes in shared stromal niches to restrict liver fibrosis. 2023.04.26.537913 Preprint at <https://doi.org/10.1101/2023.04.26.537913> (2023).
31. Rustenhoven, J. *et al.* Functional characterization of the dural sinuses as a neuroimmune interface. *Cell* (2021) doi:<https://doi.org/10.1016/j.cell.2020.12.040>.
32. Gao, W. *et al.* Engineered T cell therapy for central nervous system injury. *Nature* 1–9 (2024) doi:10.1038/s41586-024-07906-y.
33. Heindl, S. *et al.* Chronic T cell proliferation in brains after stroke could interfere with the efficacy of immunotherapies. *Journal of Experimental Medicine* **218**, (2021).
34. Kramann, R. *et al.* Perivascular Gli1+ progenitors are key contributors to injury-induced organ fibrosis. *Cell Stem Cell* **16**, 51–66 (2015).
35. Garza, R. *et al.* Single-cell transcriptomics of resected human traumatic brain injury tissues reveals acute activation of endogenous retroviruses in oligodendroglia. 2022.09.07.506982 Preprint at <https://doi.org/10.1101/2022.09.07.506982> (2023).

Reviewer Response Summary: We are grateful to the three reviewers for their thoughtful feedback on our revised manuscript, entitled “Dynamic fibroblast-immune interactions shape wound healing after brain injury”. In this revision response, we provide substantial new data that strengthens existing conclusions and enhances the conceptual scope and functional relevance of our work. Below, we summarize the major new additions to our manuscript, followed by a detailed point-by-point response to individual reviewer comments. Briefly:

1) We delineate a mechanism underpinning the death of adult “*Tgfb2* cKO” mice (*Col1a2*^{creER}; *Tgfb2*^{fllox}). New data using the tMCAO stroke model demonstrate that *Tgfb2* cKO mice have elevated vasogenic brain edema, worsened brain midline shift (a clinically validated predictor of poor stroke outcome in humans, typically driven by elevated intracranial pressure), and peripheral cardiovascular decompensation with associated end-organ damage and death.

2) We demonstrate that the myofibroblast response functionally impacts the recovery from local brain injury. Returning to the photothrombotic (PT) cortical injury model, which under our experimental setup causes local cortical damage and no increase in animal mortality, we provide new data that identify movement deficits and behavioral alterations after PT stroke in *Tgfb2* cKO mice versus controls using unsupervised behavioral analysis by Motion Sequencing (MoSeq). As these data were not directly requested by the reviewer, we have currently included them as “reviewer only” but can add them to the manuscript if requested, as we feel this is an orthogonal approach that demonstrates a functionally important and local contribution of myofibroblasts in a distinct model of focal brain injury (PT injury).

3) We further address the pathophysiologic relevance of fibroblast-lymphocyte interactions at late time points after brain injury. Using the PT injury model, we provide new data that includes single-nucleus RNA sequencing of injured brains from mice without fibroblast-derived CXCL12 (*Col1a2*^{creER}; *Cxcl12*^{fllox}, tamoxifen induced; “*Cxcl12* cKO”) and controls at 21 days post-injury in the photothrombotic (PT) brain injury model. We previously found that *Cxcl12* cKO mice had decreased brain-lesional lymphocytes with preserved total brain lymphocytes and elevated IFN γ expression per lymphocyte. Our new snRNAseq data reveal evidence of elevated brain IFN γ signaling programs within parenchymal neurons as well as dysregulated lesional immune states. Together, these findings support a distinct role for late “lymphocyte-interactive” CXCL12-producing lesional fibroblasts in recruiting and retaining brain lymphocytes, and thus positionally regulating lymphocytic impact on distal neuronal function.

A point-by-point response to each reviewer follows:

Referee 1: p. 2-10

Referee 2: p. 11-17

Referee 3: p. 18

Please note that changes to the manuscript text since the last submission are highlighted in red.

Figures in this response letter labeled as “R” are currently included as reviewer-only figures and are not currently within the updated manuscript.

Referee #1:

(Reviewer remarks to the Authors in blue – Authors' responses in black)

Summary: The revised paper provides further support to the conclusions of the paper related to the fate and contribution of brain fibroblasts to the ECM and fibrotic changes occurring after brain injury, and interaction with immune cells. The work is very detailed and comprehensive and provides new cellular and signaling insights into the formation of the tissue scar.

However, the role of the fibroblast in the origin and evolution of the damage remains less clear. The massive early mortality after MCA occlusion remains largely unexplained. The new data presented do not provide sufficient evidence to conclude that the mice die as a consequence of enhanced brain injury.

Similarly, the pathophysiological implications of the late interactions of fibroblasts with lymphocytes are also unclear.

We thank the reviewer for their positive evaluation of our revised manuscript. We have added substantial new data to address these important concerns. We first address reviewer concerns, detailed together below, related to why mice with “TGFβ-blind fibroblasts” ($Col1a2^{creER}; Tgfb2^{flox}$ – hereafter called “*Tgfb2* cKO”) and injured via tMCAO have increased mortality.

However, the role of the fibroblast in the origin and evolution of the damage remains less clear. The massive early mortality after MCA occlusion remains largely unexplained. The new data presented do not provide sufficient evidence to conclude that the mice die as a consequence of enhanced brain injury.

Figure 5c,d,e: How can these small differences in hemispheric volume, oligodendrocytes and fluoro jade-stained neuron explain 90% mortality?

Was there early systemic hypoxia, hypotension, hypothermia which could explain a life ending catastrophe? Did the mice had epileptiform activity which can also be a cause of cerebral death?

(1) Mechanism of *Tgfb2* cKO mouse death after tMCAO brain injury. We provide substantial new data to define this mechanism. Briefly, as additional support for our finding of exacerbated brain damage, we now show an increase in brain midline shift in *Tgfb2* cKO mice, a well-characterized clinical correlate of stroke severity and an independent predictor of mortality in human stroke patients, along with additional evidence of vascular leakage and vasogenic edema (detailed in response to specific requests, below). We also provide evidence of cardiovascular collapse, including bradycardia, hypotension, and end-organ damage, ultimately leading to death.

New data demonstrate a ~150% increase in midline shift in *Tgfb2* cKO mice versus controls (**Fig. 5d**, reproduced **below**). Midline shift, or horizontal displacement of brain parenchyma, is a sign of elevated intracranial pressure (ICP), which typically peaks at ~4 days after stroke¹. While the cause of death in most severe acute strokes remains unknown, midline shift is one of the only clinically validated

correlates of death after stroke^{1,2} and is a clinical predictor of mortality³ due to its relationship to brainstem herniation⁴.

While we agree with the reviewer that the observed differences in brain hemispheric volume are modest, the magnitude of the effect in control mice (18% volume increase vs. uninjured hemisphere) is consistent with the magnitude of the effect in human post-stroke edema⁵. *Tgfb2* cKO mice had a 31% volume increase (injured/uninjured). In patients, increasing edema predicts neurological deterioration⁵ and mortality⁶, with increased death in stroke patients with edema⁷. Collectively, our findings of exacerbated brain edema, in conjunction with the previously observed death/degeneration of neurons and oligodendrocytes, suggest that impairing myofibroblasts causes excess injury-induced brain swelling and tissue loss/dysfunction at a level thought to cause mortality in human patients.

To further define the precise mechanism of animal death, we performed comprehensive monitoring of vital signs at 3dpi (or earlier, for decompensating animals). Injury induced a decrease in heart rate that was more than doubled in *Tgfb2* cKO mice (**Fig. 5g**, reproduced below). Notably, this effect was specific to injury, and uninjured mice did not display differences between genotypes. Similarly, tMCAO induced severe decreases in systolic, diastolic, and mean arterial blood pressure in injured *Tgfb2* cKO mice relative to uninjured and control injured mice

(**Fig. 5h**, **Extended Data Fig. 10e,f**, reproduced below).

Other reports indicate cardiovascular dysfunction and bradycardia after tMCAO, attributed largely to brain injury-induced cardiac denervation^{8,9}. Additionally, hypotension is a common sequela of human stroke¹⁰ and is associated with poor outcomes and mortality^{11,12}. Accordingly, our data suggest an exacerbation of brain injury-induced peripheral dysfunction, consistent with elevated brain damage in *Tgfb2* cKO mice. This dysfunction may stem from swelling, compression, and/or destruction of brain tissue containing heart-innervating autonomic neurons or may involve dysfunction of circuitry upstream of medullary control centers. We propose that severe cardiovascular decompensation, downstream of exacerbated brain damage, contributes to global hypoperfusion and death.

We also observed signs of end-organ damage, including elevation in serum levels of the liver enzyme ALT in injured *Tgfb2* cKO mice (relative to both injury-control and genotype-control groups; **Extended Data Fig. 10g**, reproduced below), and increased levels of cCasp3⁺ apoptotic cells in injured but not uninjured *Tgfb2* cKO livers (**Extended Data Fig. 10h,i**, reproduced below). These data support exacerbated liver hepatocyte damage in *Tgfb2* cKO mice after tMCAO injury, likely downstream of cardiovascular hypoperfusion.

We observed no deficit in kidney function (i.e., serum creatinine) or lung function (arterial oxygen saturation) between either resting vs. injured or control vs. *Tgfb2* cKO mice (**Extended Data Fig. 10j,k**, reproduced below). No visible seizures were observed while taking vitals measurements or on harvest days, though we did not rule out subclinical epileptiform activity. Finally, while stroke induced weight loss and decreased body temperature, we did not observe differences by genotype (**Fig. R2.1.1**, below).

Together, these comprehensive new data provide a compelling mechanism for elevated mortality in the *Tgfbr2* cKO mice after tMCAO injury, involving exacerbated post-stroke brain edema and downstream cardiovascular decompensation and death.

(2) While the data delineated above support a critical role for TGF β -driven myofibroblasts after severe brain injury (tMCAO stroke model), we also present data that demonstrate local functional consequences of a loss of the myofibroblast state in photothrombotic (PT) injury, a milder brain cortical damage model that does not lead to animal mortality. These new data support our previous findings that PT-injured *Tgfbr2* cKO mice have worsened cortical lesion size, chronic-stage cortical inflammation, and white matter tract injury.

To comprehensively assess whether *Tgfbr2* cKO mice had functional evidence of increased brain damage after PT injury, we applied the Motion Sequencing (MoSeq) pipeline¹³. This paradigm uses unsupervised machine learning to analyze 3D video recordings of individual mice freely moving in an open field, capturing a broad range of mouse behaviors in an unbiased manner. We recorded control versus *Tgfbr2* cKO mice three weeks post-PT stroke in a circular open-field arena for 20 minutes (n=16 control, n=13 *Tgfbr2* cKO)¹⁴. Scalar values, including mouse height (depth from camera), speed, angle (heading direction), center of body, and position within the arena, were extracted from each video frame. After extraction, we applied the MoSeq analysis pipeline uses an autoregressive hidden Markov model (AR-HMM) to identify behavioral “syllables”, or modular components of behavior (**Fig. R2.1.2a, below**).

We observed significant differences in behavioral patterns between control injured versus *Tgfbr2* cKO injured mice, based on syllables that were differentially observed between the two groups (**Fig. R2.1.2b, below**). Notably, we observed significant movement asymmetry in *Tgfbr2* cKO mice, all of which had received right hemispheric injury, with enrichment in behaviors that moved forward but listed leftwards, or counterclockwise (**Fig. R2.1.2c, below**). These data are consistent with focal weakness contralateral to the injured hemisphere. We also observed a decrease in rearing behavior in injured *Tgfbr2* cKO mice relative to injured controls (**Fig. R2.1.2.d, below**), suggesting more functional impairment after PT injury. These two features seem to account for many of the differentially observed syllables between the genotypes.

Fig. R2.1.2: PT-injured *Tgfr2* cKO mice show altered behavioral patterns characterized by increased counterclockwise (leftward) movements and decreased rearing. **(a)** Schematic of MoSeq data acquisition and analysis. Paradigm was performed on 16 control and 13 *Tgfr2* cKO mice, all of whom underwent PT stroke three weeks prior to testing. **(b)** Syllable usage per experimental group, sorted by differential usage. Purple diamonds indicate statistically significant differences in the usage of the indicated syllable between the control and cKO groups (Mann-Whitney test with Holm-Sidak method). **(c)** Representative images of enriched syllables demonstrating movement asymmetry in cKO mice (note that mice were alone in the arena during testing; extracted data were combined into “crowd movies” for representative images). Curvature of arrows indicates directionality of mouse trajectory in subsequent frames. Red dots indicate detection of the associated syllable by MoSeq. Quantification shows the average angle of each mouse over the course of the entire video recording. A positive radian indicates a counterclockwise angle. **(d)** Representative images of enriched syllables showing decreased rearing in cKO mice. Height (rearing) appears as a more yellow signal. Quantification shows the extent of rearing of the mouse, with the average height of each mouse over the course of the entire video recording.

Similarly, the pathophysiological implications of the late interactions of fibroblasts with lymphocytes are also unclear.

Response: To further explore the pathophysiological impacts of fibroblast-lymphocyte interactions at late time points after brain injury, we profiled injured brains from mice without fibroblast-derived CXCL12 by single nucleus RNA sequencing (*Col1a2^{creER}; Cxcl12^{fllox}*, tamoxifen induced; “*Cxcl12* cKO”) at 21 days post-injury in the photothrombotic (PT) brain injury model (**Fig 5r,s**, reproduced **below**). We micro-dissected PT lesions/peri-lesional tissue and profiled them separately from the brain parenchyma to preserve this metadata. Our previous data showed that *Cxcl12* cKO mice (in which fibroblasts are genetically induced to stop production of the chemokine CXCL12 that can attract diverse CXCR4⁺ immune cells) had decreased lesion-associated lymphocytes with preserved total brain lymphocytes and elevated IFN γ expression per lymphocyte. Broadly, we now show transcriptomic evidence that supports and elaborates upon these findings, including

evidence of elevated IFN γ signaling pathways in brain parenchymal neurons. These data suggest that regulatory fibroblast-lymphocyte interactions constrain cellular immunity and indirectly modulate brain parenchymal function at late timepoints, with implications for chronic sequelae of CNS injuries.

We did not observe robust evidence of fibroblast cell-intrinsic effects of loss of *Cxcl12*, as all expected transcriptomic “states” previously defined in our work were identified (**Extended Data Fig. 10ag**, reproduced **below**), with minimal changes in lymphocyte-interactive fibroblast state and abundance between genotypes (**Extended Data Fig. 10ah**, reproduced **below**), consistent with unaltered CD80 staining by microscopy (**Extended Data Fig. 10q**).

Low numbers of T cell nuclei were recovered, most of which were lesional. Nonetheless, we identified T cells that included both type 1 and type 17/ $\gamma\delta$ T cell subsets (**Extended Data Fig. 10ai-ak**, reproduced **below**). Notably, *Cxcl12* cKO mice showed a decrease in type 1 T cells, consistent with the loss of lesional T cell localization seen in our imaging data (**Fig. 5t**).

We also identified expected myeloid

cell states (**Extended Data Fig. 10al,am**, reproduced **below**) and found that lesional myeloid cells from *Cxcl12* cKO mice had a decreased IFN γ response score (generated based on a gene set induced in microglia after *in vivo* treatment with recombinant IFN γ)¹⁵ (**Fig. 5u**, reproduced **below**). This is consistent with our microscopy data that showed loss of lesional localization among type 1 T cells and a decrease in myeloid cells expressing MHC-II, a well-known target of IFN γ signaling (**Extended Data Fig. 10r**). We also observed an increase in DAM/SAM microglia/macrophages, including a SAM subset enriched for expression of pro-inflammatory/M1-like genes (**Extended Data Fig. 10an,ao**, reproduced **below**). Collectively, these data provide orthogonal validation that the inducible loss of fibroblast *Cxcl12* results in impaired lesional-

Fig. 5r,s: snRNA sequencing at 21dpi in control and *Col1a2*^{creER}; *Cxcl12*^{fllox} (*Cxcl12* cKO) mice, analyzing tissue from both cortical lesions and adjacent “uninjured” cortex

Extended Data Fig. 10ag,ah: Fibroblast states are not markedly altered in *Cxcl12* cKO mice

Extended Data Fig. 10ai-ak, Fig. 5t: *Cxcl12* cKO mice show a loss of lesional type 1 T cells

localization of a type 1 IFN γ -competent T cell population and alterations of the lesional myeloid cell immune milieu.

We next analyzed neurons, given known roles for lymphocyte-derived cytokines in modulating neuronal function^{16–18}, including via promotion of inhibitory tone^{15–17,19,20}. We constructed a neuronal IFN γ response signature comprising genes upregulated within neurons after CNS intraparenchymal injection with IFN γ ²¹. The IFN γ response signature was elevated in both excitatory and inhibitory neurons from Cxcl12 cKO mice (**Fig. 5v,w**, reproduced below). Gene set enrichment analysis (GSEA) identified processes involved in synaptic organization and excitatory signaling that were enriched in control neurons (i.e., de-enriched in Cxcl12 cKO, associated with elevated IFN γ signaling; **Extended Data Fig. 10ap**, reproduced below). Collectively, these data are consistent with a role for lymphocyte-derived IFN γ in altering neuronal synaptic function, and with a role for lymphocyte-interactive fibroblasts in constraining this function.

Extended Data Fig. 10ap, reproduced below). Collectively, these data are consistent with a role for lymphocyte-derived IFN γ in altering neuronal synaptic function, and with a role for lymphocyte-interactive fibroblasts in constraining this function.

Together, these data support a pathophysiologic role for late lesional fibroblasts and their lymphocyte interactions in recruiting/supporting type 1 lymphocytes while constraining their IFN γ cytokine expression to the lesional area and thus also limiting IFN γ signaling to neurons in the adjacent parenchyma. We also observed transcriptomic evidence of alterations in neuronal function. Together, these data also suggest a broader role for brain border fibroblasts, and particularly “lymphocyte-interactive” states, in regulating immune cell positioning and functional impacts after injury, and we have ongoing efforts to further define consequences of their disruption.

Line 66: As stated in the cited review, the role of fibroblasts in CSF dynamics has been suggested but not demonstrated.

We thank the reviewer for raising this point and have updated the text accordingly.

Line 213: How was the Pf4^{cre} mouse characterized with respect to BAM specificity? Was this mouse used in the Drieu et al paper cited?

We thank the reviewer for this important question. We previously characterized Pf4^{cre}²⁴ (originally used as a megakaryocyte lineage tracing tool^{25,26}) as a sensitive and specific allele to mark border-associated macrophages (BAMs). Specifically, we showed that most BAMs (CD206⁺ or LYVE1⁺) are Pf4-lineage⁺, including virtually all dural, leptomeningeal, and perivascular macrophages, as well as 80% of choroid plexus macrophages. We observed minimal recombination in microglia (~5% by flow cytometry). Additionally, we and others have shown that Pf4 is a sensitive and specific marker of BAMs²⁷. These data are consistent with our observations in this study, where all CD206⁺ BAMs but only a small minority of microglia (Cx3cr1^{GFP+}, CD206⁻) are recombined (**Fig. R2.1.3, below**). Of note, we cited *Drieu et al.* as a relevant reference regarding BAM physiology and functional roles, but the authors did not use the Pf4^{cre} allele²⁸. We have now updated our manuscript with the relevant references.

Line 545: How does elevated IGG staining support increased bleeding and edema? Did you document brain bleeding?

We thank the reviewer for this question and for the chance to clarify our findings. The reviewer is correct that we did not document gross brain bleeds, and we have clarified this in the text. However, we now provide evidence demonstrating extravasation of blood products and red blood cell-associated markers, consistent with increased vasogenic edema in *Tgfr2* cKO mice.

First, while we previously showed increased IgG in the ipsilateral hemisphere of injured *Tgfb2* cKO mice (tMCAO injury model, **Extended Data Fig. 10b**), we now demonstrate a similar increase in parenchymal IgM, reflecting more severe blood-brain barrier (BBB) dysfunction due to its larger size relative to IgG³⁸ (**Extended Data Fig. 10c**, reproduced **below**). We did not observe increased IgM in contralateral hemispheres, suggesting that baseline BBB function was intact. Brain damage leads to BBB disruption and extravasation of proteins that are typically excluded from brain parenchyma³¹ into perivascular and parenchymal spaces^{32–36}. Therefore, blood-borne proteins, including IgG, are proxies for BBB damage^{35–37}.

Second, we now show evidence of blood components in the parenchyma based on increased staining for the red blood cell (RBC) marker Ter119 in the injured hemisphere, which was further increased in *Tgfb2* cKO mice (**Extended Data Fig. 10d**, reproduced **below**). RBC extravasation, in addition to signaling gross or microscopic bleeding, can be a sign of severe BBB damage/leakage^{29,30}.

Collectively, these findings suggest exacerbated BBB damage/dysfunction after tMCAO in *Tgfb2* cKO mice, consistent with recent studies showing a role for CNS fibroblasts in repairing the injured BBB³⁷. Together with our data showing an increase in lesioned hemisphere size, these data suggest elevated vasogenic edema, a common and dangerous complication of stroke wherein BBB disruption enables extracellular buildup of fluid (as well as blood-borne products), leading to elevated intracranial pressure (ICP), decreased cerebral perfusion, hypoxia, and death^{39–41}. Midline shift is one clinically relevant readout of post-stroke edema that can independently predict mortality and other poor outcomes^{1,2}, and which we also observed in *Tgfb2* cKO mice (**Fig. 5d**, shown **above**). As noted by the reviewer, we did not observe gross hemorrhage on sectioning or by T1-flash MRI (**Fig. R2.1.4**, **below**).

Referee #2:

Summary: I was pleased to see that the authors took the criticism of their manuscript seriously. I acknowledge the effort they have made, and the revised version shows substantial improvement. However, the conceptual advancement and clinical relevance remain somewhat limited. While the authors attempted to explain the increased mortality rate following *Tgfr2* deletion in fibroblasts, the reported brain swelling/edema formation—likely causing neuronal and oligodendrocyte loss—is not clearly linked to the reported roles of myofibroblasts. It is more likely that the edema formation is related to off-target effects on astrocytes and their water channels.

We thank the reviewer for their positive comments regarding our revised manuscript, and we reiterate our appreciation of their insight throughout this process.

Conceptual advancement and clinical relevance: This revision includes new data that extends the clinical relevance of our findings by providing a more detailed mechanistic explanation for *Tgfr2* cKO mouse death after tMCAO stroke, as detailed below. We also present new data showing functional deficits in *Tgfr2* cKO after the more focal (local) PT injury model, currently presented as “reviewer only” data. To strengthen our existing findings and expand the conceptual advancement offered by our manuscript, we also include new transcriptomic data from mice with *Cxcl12*-deficient fibroblasts after PT brain injury. These data are detailed in the response to Reviewer #1 (**above**). We have additionally addressed individual reviewer concerns regarding our methods, dMCAO injury data, and characterization of LAMA1⁺ pial-like fibroblasts, as detailed further below. Collectively, our updated findings emphasize temporally and mechanistically distinct roles for both early and late brain fibroblast subsets (states). We believe they represent a substantial conceptual advance over current understandings of fibroblast roles after CNS injury.

Role of edema and relationship to mortality and fibroblasts in the *Tgfr2* cKO mice after tMCAO injury: We now show an increase in brain midline shift in *Tgfr2* cKO mice after tMCAO, a well-characterized clinical correlate of stroke severity and an independent predictor of mortality in human stroke patients, along with additional evidence of vascular leakage and vasogenic edema. We also provide a clinically relevant mechanism by which this edema may drive animal mortality, involving bradycardia, hypotension, and end-organ damage.

New data demonstrate a ~150% increase in brain midline shift in *Tgfr2* cKO mice versus controls (**Fig. 5d**, reproduced **below**). Midline shift, or horizontal displacement of brain parenchyma, is a sign of elevated intracranial pressure (ICP), which typically peaks at ~4 days after stroke¹. While the cause of death in most severe acute strokes remains unknown, midline shift is one of the only clinically validated correlates of death after stroke^{1,2} and is a clinical predictor of mortality³ due to its relationship to brainstem herniation⁴.

We also provide further evidence of vascular leakage and vasogenic edema likely contributing to this midline shift. First, while we previously showed increased IgG in the ipsilateral hemisphere of injured *Tgfr2* cKO mice (tMCAO injury model, **Extended Data Fig. 10b**), we now demonstrate a similar increase in parenchymal IgM, reflecting more severe blood-brain barrier (BBB) dysfunction due to its larger size relative to IgG³⁸ (**Extended Data Fig. 10c**, reproduced **below**). We did not observe increased IgM in contralateral hemispheres, suggesting that baseline BBB function was intact. Second, we now show evidence of blood components in the parenchyma based on increased staining for the red blood cell (RBC) marker Ter119 in the injured hemisphere, which was further increased in *Tgfr2* cKO mice (**Extended Data Fig. 10d**, reproduced **below**). RBC extravasation, in addition to signaling gross or microscopic bleeding, can be a sign of severe BBB damage/leakage^{29,30}. Collectively, these findings suggest exacerbated BBB damage/dysfunction after tMCAO in *Tgfr2* cKO mice, consistent with recent studies showing a role for CNS fibroblasts in repairing the injured BBB³⁷. Together with our data showing an increase in lesioned hemisphere size, these data suggest elevated vasogenic edema, a common and dangerous complication of stroke wherein BBB disruption enables extracellular buildup of fluid (as well as blood-borne products), leading to elevated intracranial pressure (ICP), decreased cerebral perfusion, hypoxia, and death³⁹⁻⁴¹.

Of note, the magnitude of the observed edema in control mice (18% volume increase vs. uninjured hemisphere) is consistent with the magnitude of the effect in human post-stroke edema⁵, whereas *Tgfr2* cKO mice had a 31% volume increase (injured/uninjured). In patients, increasing edema predicts neurological deterioration⁵ and mortality⁶, with increased death in stroke patients with edema⁷. Collectively, our findings of exacerbated brain edema, in conjunction with the previously observed death/degeneration of neurons and oligodendrocytes, suggest that impairing myofibroblasts causes excess injury-induced brain swelling and tissue loss/dysfunction at a level thought to cause mortality in human patients.

To further define the precise mechanism of animal death, we performed comprehensive monitoring of vital signs at 3dpi (or earlier, for decompensating animals). Injury induced a decrease in heart rate that was more than doubled in *Tgfr2* cKO mice (**Fig. 5g**, reproduced **below**). Notably, this effect was specific to injury, and uninjured mice did not display differences between genotypes. Similarly, tMCAO induced severe decreases in systolic, diastolic,

and mean arterial blood pressure in injured *Tgfr2* cKO mice relative to uninjured and control injured mice (**Fig. 5h, Extended Data Fig. 10e,f**, reproduced **below**). Other reports indicate cardiovascular dysfunction and bradycardia after tMCAO, attributed largely to brain injury-induced cardiac denervation^{8,9}. Additionally, hypotension is a common sequela of human stroke¹⁰ and is associated with poor outcomes and mortality^{11,12}.

Accordingly, our data suggest an exacerbation of brain injury-induced peripheral dysfunction, consistent with elevated brain damage in *Tgfr2* cKO mice. This dysfunction may stem from swelling, compression, and/or destruction of brain tissue containing heart-innervating autonomic neurons or may involve dysfunction of circuitry upstream of medullary control centers. We propose that severe cardiovascular decompensation, downstream of exacerbated brain damage, contributes to global hypoperfusion and death.

We also observed signs of end-organ damage, including elevation in serum levels of the liver enzyme ALT in injured *Tgfr2* cKO mice (relative to both injury-control and genotype-control groups; **Extended Data Fig. 10g**, reproduced **below**), and increased levels of cCasp3⁺ apoptotic cells in injured but not uninjured *Tgfr2* cKO livers (**Extended Data Fig. 10h,i**, reproduced **below**). These data support exacerbated liver hepatocyte damage in *Tgfr2* cKO mice after tMCAO injury, likely downstream of cardiovascular hypoperfusion.

Together, these comprehensive new data provide a compelling mechanism for elevated mortality in the *Tgfr2* cKO mice after tMCAO injury, involving exacerbated post-stroke brain edema and downstream cardiovascular decompensation and death.

“...It is more likely that the edema formation is related to off-target effects on astrocytes and their water channels.”

We agree with the reviewer that our findings of decreased oligodendrocytes and neuronal degeneration/death are potentially downstream of elevated brain edema, although direct

relationships between fibroblasts and neurons/oligodendrocytes have also been reported in stroke⁴² and EAE⁴³. We appreciate the reviewer's questions regarding the link between known roles for fibroblasts and exacerbated post-injury CNS edema. Based on lineage tracing and transcriptional studies, astrocytes are not directly affected by our genetic manipulations using Col1a2^{creER}. Rather, if astrocytic water channels are impacted, it is likely due to elevated inflammatory or damage signals in the lesional milieu, which themselves are downstream of direct effects of fibroblasts. Astrocyte-mediated edema has been associated with an upregulation of astrocytic *Aqp4*, with *Aqp4* knockout mice showing decreased edema (hemispheric enlargement) and increased survival after MCAO⁴¹. In contrast, *Tgfr2* cKO astrocytes did not show upregulated *Aqp4* (which was mildly decreased; **Fig. R2.2.1, below**). Nonetheless, we agree that astrocytes (and their vascular/fluid flux regulatory functions) are likely to participate in the downstream impacts of CNS fibroblasts, and this is explicitly discussed in the revised manuscript.

Consistent with our experimental findings, direct roles for fibroblasts in modulating vascular integrity after CNS injury have been reported. In a model of intraparenchymal hemorrhage, *Col1a1*-lineage⁺ fibroblasts were required for repairing BBB damage³⁷. Fibroblast ablation in this model led to elevated IgG staining and tracer leakage at subacute timepoints with an associated increase in FJC⁺ neurons. Future studies will be required to determine whether a similar TIMP2-dependent mechanism exists after tMCAO. Interestingly, pharmacological perturbation of fibroblasts *reduced* vasogenic edema at a similar time point after experimental stroke^{44,45}, potentially suggesting a complex relationship between fibroblasts and vascular leakage. Our data indicate that injury-associated CNS fibroblasts modulate vascular leakage, although this role could involve inflammatory, damage-associated, or other signals that act in part via astrocyte-dependent neurovascular functions.

1. Critical: I could not find any information regarding the treatment of controls in relation to tamoxifen-induced gene deletions. Tamoxifen has significant effects on T-cell recruitment (PMID: 19689284, 19689284). Since tamoxifen was administered continuously throughout the post-lesion stages, it could contribute to some of the observed phenotypes if the controls were not treated in a similar manner. Please include this information in the methods and relevant figure legends.

We have clarified our methods and figure legends to detail that experiments involving tamoxifen-induced gene or cell deletion used control mice treated with equivalent levels of tamoxifen. For a small number of exceptions, we used a combination of tamoxifen-treated controls and vehicle-injected or uninjected controls (due to breeding constraints). These exceptions have been noted in our methods and legends, but none of these exceptions involved analysis of lesional T cell recruitment.

2. Line 64-65 "... recent studies have highlighted the roles of CNS stromal cells, including both mural cells and fibroblasts, in injury9–13 and disease14–18. Goritz et al., 2011, Science PMID: 21737741 AND Holl et al., 2024, Nat Neurosci. PMID:

38849523 should be cited here as well.

We have added these citations to the manuscript and thank the reviewer for their suggestions.

3. As in Ext. Data Fig. 1q authors are required to quantify the recombination efficiency of the *Atp13a5-CreER* line under uninjured conditions. What is the percentage of pericytes labeled with tdTomato over the total number of pericytes at rest in the experiments conducted for this study?

We have now quantified the recombination efficiency in resting *Atp13a5^{creER}; Rosa26^{TdT}* mice (**Extended Data Fig. 1v,w**, reproduced below). We found that this mouse line is very sensitive under our experimental conditions, recombining ~90% of pericytes ($NG2^+$, $PDGFR\beta^+$), in line with the higher end of the previously published efficiency range for this allele⁴⁶.

4. Cell composition (blood vessel and fibroblasts numbers) and lesion size are not comparable in *Atp13a5creER; R26TdT* (Ext. Data Fig. 1u) and *Col1a2creER; R26TdT* mice (Ext. Data Fig. 1t). Replace image in Ext. Data Fig. 1u with a lesion of similar size and cellular composition. Also provide quantitative data showing the contribution of *Atp13a5*-lineage traced pericytes to the total number of lesion fibroblasts in comparison to the contribution of *Col1a2*-lineage traced fibroblasts after dMCAO.

We have added new lesional images with more comparable dMCAO lesional morphology (**Extended Data Fig. 1t,u**, reproduced below). We have also added new data quantifying lesional lineage-traced pericytes and fibroblasts. We found that fibroblasts – but not pericytes – expand within lesions after dMCAO injury (**Extended Data Fig. 1x**, reproduced below).

5. The following text from the rebuttal letter needs to be stated in the discussion:

“These results could be interpreted to contrast with the reported contribution of a *Glast*+ pericyte population after spinal cord injury (referenced above). However, our own sequencing data (Fig. R2.3, below) reveal substantial expression of *Glast* (i.e., *Slc1a3*) within PT lesional fibroblasts at all time points, confirming previous findings^{19,21}. Subsequent analysis from the cited group found that 98.3% of fibroblasts (lineage-traced with a *Col1a1creER* allele) expressed *Slc1a3*, and that these fibroblasts significantly contributed to the stromal scar after spinal cord injury¹², potentially consistent with a *Slc1a3*-lineage+ fibroblast origin for stromal cells after PT injury.”

We have updated our discussion to highlight this important point.

6. In figure 3d, laminin signal seems to localize to basement membrane of blood vessels and does not clearly overlap with a subset of lesion fibroblasts. This may be due to the use of a pan-laminin antibody and not a Laminin alpha 1 specific antibody. Recommended to revise the imaging with a *Lama1* specific antibody and/or ISH probe.

We agree and have replaced these images with new data using a Laminin $\alpha 1$ -specific monoclonal antibody. While published as a marker of laminin $\alpha 1$ ⁴⁷, the original laminin antibody used in the referenced figure was indeed generated using laminin as an immunogen. The new Laminin $\alpha 1$ -specific antibody highlights LAMA1 staining within the lineage-traced fibroblast pool (*Col1a2^{creER}*; *R26^{TdT+}*) at the lesional-parenchymal interface, with significantly less staining lining the basement membrane of vasculature within the interior lesion (**Fig. 3d**, reproduced **below**). We have also updated all other relevant figures and graphs accordingly. Specifically, we have incorporated this new antibody into our fibroblast subset imaging at 7dpi (**Extended Data Fig. 6c**), our quantification of fibroblast subsets at 7 and 21dpi (**Extended Data Fig. 6d**), and our quantification of subset marker staining within the *Cthrc1*-lineage⁺ pool (**Extended Data Fig. 6k**). Using this antibody, we confirmed the presence of a LAMA1⁺ boundary between tMCAO lesions and brain parenchyma (**Extended Data Fig. 6l**, reproduced **below**), which we were previously unable to separate from lesion-wide LAMA vascular staining. Finally, we have added quantification of LAMA1 staining in control and *Tgfbr2* cKO mice at 21dpi (**Extended Data Fig. 9p**, reproduced **below**); this analysis showed reduced pial-like fibroblasts in *Tgfbr2* cKO mice, consistent with our snRNAseq data (**Fig. 4t**).

7. In Ext data fig.1v legend, change left and right by top and bottom.

We thank the reviewer for noting this error. It has been corrected.

8. Fig. R2.12: Tamoxifen pre-treatment is insufficient to prevent fibroblast expansion. Include these data in the article.

We have updated our figures and manuscript accordingly to emphasize this point (see **Extended Data Fig. 8h,i**).

Referee #3 (Remarks to the Author):

Summary: I have read through this revised manuscript and the authors' rebuttal comments and I would like to congratulate the authors on an outstanding, rich and exciting body of work. I have no further comments.

We thank the reviewer for this generous comment and for their time, effort, and valuable feedback during this review process.

References:

1. Pullicino, P. M. *et al.* Mass effect and death from severe acute stroke. *Neurology* **49**, 1090–1095 (1997).
2. McKeown, M. E. *et al.* Midline shift greater than 3mm independently predicts outcome after ischemic stroke. *Neurocrit Care* **36**, 46–51 (2022).
3. Ong, C. J. *et al.* Enhanced Detection of Edema in Malignant Anterior Circulation Stroke (EDEMA) Score. *Stroke* **48**, 1969–1972 (2017).
4. Hacke, W. *et al.* 'Malignant' Middle Cerebral Artery Territory Infarction: Clinical Course and Prognostic Signs. *Archives of Neurology* **53**, 309–315 (1996).
5. Yoo, A. J. *et al.* Validating Imaging Biomarkers of Cerebral Edema in Patients with Severe Ischemic Stroke. *J Stroke Cerebrovasc Dis* **22**, 742–749 (2013).
6. Battey, T. W. K. *et al.* Brain edema predicts outcome after non-lacunar ischemic stroke. *Stroke* **45**, 3643–3648 (2014).
7. Thorén, M. *et al.* Predictors for Cerebral Edema in Acute Ischemic Stroke Treated With Intravenous Thrombolysis. *Stroke* **48**, 2464–2471 (2017).
8. Veltkamp, R. *et al.* Experimental ischaemic stroke induces transient cardiac atrophy and dysfunction. *J Cachexia Sarcopenia Muscle* **10**, 54–62 (2019).
9. Vornholz, L. *et al.* Acute Heart Failure After Reperfused Ischemic Stroke: Association With Systemic and Cardiac Inflammatory Responses. *Front. Physiol.* **12**, (2021).
10. Ritter, M. A. *et al.* Heart rate monitoring on the stroke unit. What does heart beat tell about prognosis? An observational study. *BMC Neurol* **11**, 47 (2011).
11. Vemmos, K. N. *et al.* U-shaped relationship between mortality and admission blood pressure in patients with acute stroke. *J Intern Med* **255**, 257–265 (2004).
12. Ouyang, M. *et al.* Low blood pressure and adverse outcomes in acute stroke: HeadPoST study explanations. *J Hypertens* **39**, 273–279 (2021).

13. Wiltschko, A. B. *et al.* Revealing the structure of pharmacobehavioral space through motion sequencing. *Nat Neurosci* **23**, 1433–1443 (2020).
14. Wiltschko, A. B. *et al.* Mapping Sub-Second Structure in Mouse Behavior. *Neuron* **88**, 1121–1135 (2015).
15. Mroz, N. M. *et al.* Type 1 lymphocytes and interferon- γ accumulate in the thalamus and restrict seizure susceptibility after traumatic brain injury. 2024.12.28.630606 Preprint at <https://doi.org/10.1101/2024.12.28.630606> (2024).
16. Filiano, A. J. *et al.* Unexpected role of interferon- γ in regulating neuronal connectivity and social behaviour. *Nature* **535**, 425–429 (2016).
17. Döhne, N., Falck, A., Janach, G. M. S., Byvaltcev, E. & Strauss, U. Interferon- γ augments GABA release in the developing neocortex via nitric oxide synthase/soluble guanylate cyclase and constrains network activity. *Front Cell Neurosci* **16**, 913299 (2022).
18. Alves de Lima, K. *et al.* Meningeal $\gamma\delta$ T cells regulate anxiety-like behavior via IL-17a signaling in neurons. *Nat Immunol* **21**, 1421–1429 (2020).
19. Janach, G. M. S. *et al.* Interferon- γ enhances neocortical synaptic inhibition by promoting membrane association and phosphorylation of GABAA receptors in a protein kinase C-dependent manner. *Brain, Behavior, and Immunity* **101**, 153–164 (2022).
20. Janach, G. M. S. *et al.* Interferon- γ acutely augments inhibition of neocortical layer 5 pyramidal neurons. *J Neuroinflammation* **17**, 69 (2020).
21. Hobson, B. D. *et al.* Conserved and cell type-specific transcriptional responses to IFN- γ in the ventral midbrain. *Brain Behav Immun* **111**, 277–291 (2023).
22. Ruan, J. & Yao, Y. Behavioral tests in rodent models of stroke. *Brain Hemorrhages* **1**, 171–184 (2020).
23. Balkaya, M., Kröber, J. M., Rex, A. & Endres, M. Assessing post-stroke behavior in mouse models of focal ischemia. *J Cereb Blood Flow Metab* **33**, 330–338 (2013).

24. McKinsey, G. L. *et al.* A new genetic strategy for targeting microglia in development and disease. *eLife* **9**, e54590 (2020).
25. Pertuy, F. *et al.* Broader expression of the mouse platelet factor 4-cre transgene beyond the megakaryocyte lineage. *J Thromb Haemost* **13**, 115–125 (2015).
26. Tiedt, R., Schomber, T., Hao-Shen, H. & Skoda, R. C. Pf4-Cre transgenic mice allow the generation of lineage-restricted gene knockouts for studying megakaryocyte and platelet function in vivo. *Blood* **109**, 1503–1506 (2007).
27. Van Hove, H. *et al.* A single-cell atlas of mouse brain macrophages reveals unique transcriptional identities shaped by ontogeny and tissue environment. *Nat Neurosci* **22**, 1021–1035 (2019).
28. Drieu, A. *et al.* Parenchymal border macrophages regulate the flow dynamics of the cerebrospinal fluid. *Nature* (2022) doi:10.1038/s41586-022-05397-3.
29. Zhao, B. *et al.* Endothelial CYP2J2 overexpression restores the BRB via METTL3-mediated ANXA1 upregulation. *FASEB J* **36**, e22619 (2022).
30. Wei, X. *et al.* MSCs act as biopatches for blood-retinal barrier preservation to enhance functional recovery after retinal I/R. *Molecular Therapy Nucleic Acids* **36**, (2025).
31. Ruano-Salguero, J. S. & Lee, K. H. Antibody transcytosis across brain endothelial-like cells occurs nonspecifically and independent of FcRn. *Sci Rep* **10**, 3685 (2020).
32. Georgakopoulou, T., van der Wijk, A.-E., van Bavel, E. & Bakker, E. N. T. P. Perivascular clearance of blood proteins after blood-brain barrier disruption in a rat model of microinfarcts. *Microvascular Research* **148**, 104515 (2023).
33. Freeze, W. M. *et al.* Blood-brain barrier leakage and microvascular lesions in cerebral amyloid angiopathy. *Stroke* **50**, 328–335 (2019).
34. Michalak, Z. *et al.* IgG Leakage May Contribute to Neuronal Dysfunction in Drug-Refractory Epilepsies With Blood-Brain Barrier Disruption. *Journal of Neuropathology & Experimental Neurology* **71**, 826–838 (2012).

35. Okada, T., Suzuki, H., Travis, Z. D. & Zhang, J. H. The Stroke-Induced Blood-Brain Barrier Disruption: Current Progress of Inspection Technique, Mechanism, and Therapeutic Target. *Curr Neuropharmacol* **18**, 1187–1212 (2020).
36. Kassner, A. & Merali, Z. Assessment of Blood–Brain Barrier Disruption in Stroke. *Stroke* **46**, 3310–3315 (2015).
37. Xu, L. *et al.* Fibroblasts repair blood-brain barrier damage and hemorrhagic brain injury via TIMP2. *Cell Reports* **41**, (2022).
38. Schroeder, H. W. & Cavacini, L. Structure and Function of Immunoglobulins. *J Allergy Clin Immunol* **125**, S41–S52 (2010).
39. Michinaga, S. & Koyama, Y. Pathogenesis of Brain Edema and Investigation into Anti-Edema Drugs. *Int J Mol Sci* **16**, 9949–9975 (2015).
40. Gu, Y. *et al.* Cerebral edema after ischemic stroke: Pathophysiology and underlying mechanisms. *Front Neurosci* **16**, 988283 (2022).
41. Manley, G. T. *et al.* Aquaporin-4 deletion in mice reduces brain edema after acute water intoxication and ischemic stroke. *Nat Med* **6**, 159–163 (2000).
42. Wen, J. *et al.* Fibrotic scar formation after cerebral ischemic stroke: targeting the Sonic hedgehog signaling pathway for scar reduction. *Neural Regeneration Research* 10.4103/NRR.NRR doi:10.4103/NRR.NRR-D-24-00999.
43. Dorrier, C. E. *et al.* CNS fibroblasts form a fibrotic scar in response to immune cell infiltration. *Nature Neuroscience* **24**, 234–244 (2021).
44. Protzmann, J. *et al.* PDGFR α inhibition reduces myofibroblast expansion in the fibrotic rim and enhances recovery after ischemic stroke. *J Clin Invest* **135**, (2025).
45. Su, E. J. *et al.* Activation of PDGF-CC by tissue plasminogen activator impairs blood-brain barrier integrity during ischemic stroke. *Nature Medicine* **14**, 731–737 (2008).
46. Guo, X. *et al.* Atp13a5 Marker Reveals Pericyte Specification in the Mouse Central Nervous System. *J Neurosci* **44**, e0727242024 (2024).

47. Xue, X. *et al.* Heterogeneous fibroblasts contribute to fibrotic scar formation after spinal cord injury in mice and monkeys. *Nat Commun* **15**, 6321 (2024).

Reviewer Response Summary: We are grateful to the reviewers for their thoughtful feedback on our revised manuscript, entitled “Dynamic fibroblast-immune interactions shape recovery after brain injury”.

Referee #1 (Remarks to the Author):

The authors have continued to improve the paper by changes on the text and adding new data on cardiovascular parameters and organ damage in the mice with tMCAO, and transcriptomic effects of IFN γ on brain cells.

The pathological significance of the new data showing a 150 micron midline shift is questionable (fig. 5c,d). It is unclear how such small hemispheric deviation can cause the hemodynamic collapse indicated by the measurement of the cardiovascular parameters, as well as end-organ damage. It is argued that the shift may have caused compression of brainstem autonomic centers, akin to those occurring in brain herniation-related coma in humans, but no evidence is provided that the small displacement reported is capable of such profound mechanical effects and damage on the brainstem.

Therefore, while the new data show a plausible cause underlying the mortality, i.e., hemodynamic collapse, the mechanisms of such profound and presumably lethal cardiovascular changes remain unclear. Statements that the cardiovascular decompensation and death is due to “exacerbated post-stroke brain edema” do not seem justified by the data. Overall, it remains puzzling why these devastating effects are observed only in the late stages of injury.

We thank the reviewer for their positive evaluation of our manuscript. We agree that while midline shift may compress brainstem autonomic centers, we have not identified precise brain regions impacted in this model, and that the precise mechanism linking tMCAO stroke to cardiovascular decompensation and death remains incompletely explained. We have updated our manuscript and conclusions to reflect these outstanding questions.

Referee #2 (Remarks to the Author):

The authors have addressed all my comments.

We thank the reviewer for their time and for their insightful comments and suggestions throughout this process.